

# The influence of reservoir traits on carbon dioxide emissions in the Belo Monte hydropower complex, Xingu River, Amazon – Brazil

Kleiton R. Araújo[1*], Henrique O. Sawakuchi[2-3], Dailson J. Bertassoli Jr.[4], André O. Sawakuchi[1,4], Karina D. da Silva[1,5], Thiago V. Bernardi[1,5], Nicholas D. Ward[6-7], Tatiana S. Pereira[1,5].

[1]Programa de Pós Graduação em Biodiversidade e Conservação, Universidade Federal do Pará, Altamira, 68372 – 040, Brazil,

[2]Centro de Energia Nuclear na Agricultura, Universidade de São Paulo, Piracicaba, Brazil,

[3]Department of Ecology and Environmental Science, Umeå University, Umeå, SE-901 87, Sweden,

[4]Instituto de Geociências, Universidade de São Paulo, São Paulo, Brazil,

[5]Faculdade de Ciências Biológicas, Universidade Federal do Pará, Altamira, 68372 – 040, Brazil,

[6]Marine Sciences Laboratory, Pacific Northwest National Laboratory, Sequim, Washington, 98382, USA,

[7]School of Oceanography, University of Washington, Seattle, Washington, 98195-5351, USA.

*Correspondence to: Kleiton R. Araújo (kleitonrabelo@rocketmail.com)

Keywords: run-of-the-river reservoir; greenhouse gas emission; tropical river damming.


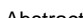

Abstract

River damming alters local hydrology, which influences aspects of the carbon cycle such as carbon dioxide ($CO_2$) production and emissions. Run-of-the-river reservoirs (RORs) are designed to have a smaller flooded area than traditional storage reservoirs, maintaining a river flow similar to natural seasonal water level fluctuation. These features may potentially reduce the impacts of the dam on the natural carbon cycle. However, little information on the influence of RORs on the carbon cycle is available, especially for the Amazon or other large tropical rivers. The Belo Monte hydropower complex is a large ROR located in the Xingu River, a clearwater tributary of the Amazon River. It is composed of two reservoirs; the Xingu Reservoir (XR) with ROR characteristics and the Intermediate Reservoir (IR) with storage reservoir traits. Here we evaluate spatiotemporal variation of surface water $CO_2$ partial pressure ($pCO_2$), $CO_2$ fluxes ($FCO_2$), and gas exchange coefficients ($k_{600}$) during the first two years after the impoundment of the Xingu River. Seasonal changes in the water level had a significant influence on $pCO_2$ with the highest average values observed during high water. The $FCO_2$ was more variable, although correlated with $pCO_2$, throughout the two first years of river impoundment. Spatial heterogeneity was observed for $pCO_2$ during both seasons while $FCO_2$ showed significant spatial heterogeneity only during the high water period. High water $FCO_2$ and $pCO_2$ values were on the same order of magnitude as emissions measured in Amazonian clearwater rivers unaffected by impoundment, but low water values were an order of magnitude higher than previous observations in clearwater rivers with natural flowing waters. Finally, we observed variability in $CO_2$ fluxes related to the type of environment (i.e., river channel, downstream the dams, outside reservoirs and flooded areas), among reservoirs and the land use of flooded areas after impoundment of the Belo Monte hydropower complex. For example, $CO_2$ emissions were 15% and 90% higher for the IR compared to XR during high and low water season, respectively, indicating that storage reservoirs may be larger sources of $CO_2$ to the atmosphere compared to RORs. Since many reservoirs are still planned to be constructed in the Amazon and throughout the world, it is critical to evaluate the implications of reservoir traits on $CO_2$ fluxes over their entire life cycle in order to generate energy that has lower emissions per KW.





## 1 Introduction

Rivers and streams are no longer considered passive pipes where terrestrial organic matter (OM) travels unchanged from land to sea (Cole et al., 2007). The OM transported by inland waters may be converted to
carbon dioxide ($CO_2$) or methane ($CH_4$) and escape to the atmosphere as gaseous emissions (Battin et al., 2009; Ward et al., 2013). Inland waters cover an area of approximately 624,000 km² (about 0.47% of the Earth's land surface) and emit about 1.8 to 3.8 Pg C  annually (Abril et al. 2014; Drake et al., 2018; Raymond et al., 2013; Sawakuchi et al., 2017). Despite the relatively small area covered by inland waters, their carbon emissions offset the oceans carbon sink ($1.42 \pm 0.53$ Pg C $y^{-1}$) (Landchützer et al., 2014).

Channel impoundment promotes several changes on river properties such as water discharge, organic and inorganic sediment input, water temperature, water turbulence and wind shear (St. Louis et al., 2000). These changes alter the microbial community structure and biogeochemical processes in the water column and riverbed sediments, with consequent impacts on the dissolved carbon load, production, and eventual release to the atmosphere as $CO_2$ (Battin et al., 2008). The intense decomposition of OM contained in flooded soils, in addition
to autochthonous OM deposited in the reservoir may lead to an increase of the $CO_2$ production, leading to higher $CO_2$ outgassing, particularly during the first years of channel impoundment (Guérin et al., 2006). Reduced water turbidity and elevated light penetration depth due to the increase in water residence time, on the other hand, may counterbalance those emissions due to higher $CO_2$ uptake by primary producers (Duarte and Prairie, 2005), or alternatively stimulate OM decomposition via photo-oxidation and/or microbial priming effects (Ward et al., 2016).

In order to minimize some of the impacts usually associated with hydropower dams, run-of-the-river (ROR) hydroelectric systems have smaller reservoirs and they operate with seasonal variations in water levels (Csiki and Rhoads, 2010; Egré and Milewski, 2002). The Belo Monte hydroelectric complex in the lower Xingu River operates as a ROR and it is the largest hydropower plant in the Amazon. It ranks third in the world in terms of installed capacity (11,233 MW), but with high variation in energy production throughout the year due to the high
seasonality of the water discharge of the Xingu River (Brasil, 2009c). The Xingu River, as other Amazonian clearwater rivers, presents relatively high emissions of $CO_2$ in natural conditions (pre–impoundment) compared to blackwater or whitewater Amazonian rivers (Sawakuchi et al., 2017). As such, alterations in the natural carbon cycling in this environment may result in direct and significant impacts on the regional carbon budget.

Little information is available for greenhouse gas (GHG) emissions from RORs and the available estimates were
mainly obtained through modeling for tropical reservoirs or measurements in small temperate reservoirs (DelSontro et al., 2016; Faria et al., 2015). Hundreds of new hydroelectric reservoirs are currently under construction or planned to be built in the tropical South America, Africa, and Asia, and many of them may be ROR reservoirs (Winemiller et al., 2016). Also, most of the GHG emissions estimates available in the literature are from old reservoirs (Barros et al, 2011; Deemer et al., 2016) and newly flooded reservoirs are still poorly
studied. Given that $CO_2$ emissions during the first years of impoundment are critical to determine the overall carbon balance of a ROR reservoir and to evaluate the carbon intensity of hydroelectricity produced from tropical rivers, assessments of GHG emissions in the early stage of flooding are especially important. As such, this study aims to evaluate the spatial and temporal variation of $CO_2$ partial pressure ($pCO_2$) and $CO_2$ emission and its relevance for GHG fluxes during the first two years after development of the reservoirs of the Belo Monte
Hydropower complex in eastern Amazon.



2 Material and methods

2.1 Study area

The Xingu River is the second largest clearwater tributary of the Amazon River. It drains an area of 504,000 km$^2$
80   and flows from central Brazil (15°S) to the lower Amazon River in eastern Amazonia (3°S) (Latrubesse et al.,
2005; Brasil, 2009a). It is classified as a clear water river, characterized by neutral to slightly alkaline pH, and low
concentration of suspended sediment, with high light penetration (Sioli, 1984). The climate of the region has high
seasonality. The rainy period usually starts in December extending until May, with rainfall peaking in March and
April (Inmet, 2017). The dry season occurs from June to November, with the driest months being September and
85   October (Fig.1). The average monthly rainfall and temperature were 188 ± 145 mm and 27.5 ± 1.01 °C,
respectively (10 year average from 2004 to 2014) (Inmet, 2017). According to the rainfall regime, river discharge
is marked by strong seasonality with the low water season occurring from September to November, and the high
water season from March to May. The historic average discharge of the Xingu River for the period from 2004 to
2014 was 1,408 ± 513 m³ s$^{-1}$ for the low water season and 18,983 ± 9,228 m³ s$^{-1}$ in the high water season (Fig.1).
The dominant land cover in the middle and lower Xingu watershed is tropical rainforest, although agriculture and
deforested areas occur mainly in the south and southwest of the basin and close to Altamira, the largest city near
the Belo Monte hydropower complex (Brasil, 2009a).

The studied area ranges from the lower Iriri River, the largest tributary of the Xingu River, to downstream of the
sector known as "*Volta Grande do Xingu*" (Xingu Great Bend), nearby the Vitória do Xingu Municipality (Fig. 2).
The Belo Monte hydroelectric complex is classified as a ROR reservoir by Eletrobrás (Brazilian energy company)
due the reduced reservoir size, lack of water level control in the main power house, short water residence time
and seasonal variation of water discharge (Brasil, 2009b; 2009c). However, the Belo Monte hydroelectric
complex is a combination of two reservoirs, one in the river channel that can be considered as a ROR reservoir
(although it has spillways, the diminished flooded area still resembles the natural river channel) and another one
that operates as an storage reservoir fed with water diverted from the first reservoir. Reservoirs residence time
(RT) data was based on Eletrobrás (2009b) and calculations on Faria et al. (2015) supplement material.

The upstream reservoir known as the "Xingu reservoir" (XR) is formed by the impoundment of the Xingu River
channel by the Pimental dam (Fig.2), which hosts 6 turbines and floodgates that maintain the water flow
upstream. The Xingu reservoir mainly inundated seasonally flooded forest (*igapó*) in islands and at the channel
margins, however the relatively reduced flooded area kept the boundaries of the river channel similar to its
natural state in the high water season (Fig.2). In addition XR is a reservoir with low RT (0.33 days). Considering
these characteristics, the Xingu reservoir will be denominated a ROR.

On ROR projects it is common to construct an associated storage reservoir to optimize the energy production
(Egré and Milewski, 2002). The waters of the XR are diverted to feed the second reservoir naturally disconnected
from the Xingu River channel, called the "Intermediate reservoir" (IR) (Brasil, 2009b; 2009c). The IR flooded
large areas of pasture and upland non-flooded forest ("*terra firme* forest"). It is impounded by the Belo Monte
dam, with a powerhouse that hosts 18 turbines comprising 97% of the installed capacity of the Belo Monte
hydropower complex (Fig.2). Different from the Pimental dam there is no water flow regulation system in the Belo
Monte dam. In comparison to XR, IR has longer RT (1.57 days). Since its characteristics resemble storage
reservoirs, the IR will be denominated a storage reservoir.





Both dams have similar intake depth at about 15-20 m (above the hypolimnion) and occupy together an area of 516 km². Approximately 267 km$^2$ of the reservoirs area correspond to flooded lands and the remaining area is the original river channel during the high water season (Brasil, 2014). The water from the intermediate reservoir returns to the Xingu river channel after flowing around 34 km over flooded lands (Fig.2) (Brasil, 2009b; 2009c).

With the deviation of the Xingu River the sector between the two dams, including part of the Xingu Great Bend, had significantly decreased its water discharge. The expected installed power capacity of both reservoirs is 11,233 MW, which is equivalent to 25.4 km²/MW (Brasil, 2009b; 2009c).

2.2 Carbon dioxide partial pressure (pCO$_2$) and CO$_2$ fluxes (FCO$_2$) from the water

The gas sampling survey (Fig.2 and Table 1) occurred during the high water level season in April 2016, May

2017 and during the low water level season in September 2017. Due to technical challenges, pCO$_2$ data was only collected during 2017 and 2017 FCO$_2$ samplings were made with different equipment (details below). In order to cover zones with different flooded substrates and hydrologic characteristics, the sampling sites included the original river channel within the XR, flooded lands (forest and pasture) of both reservoirs, upstream and downstream river channel zones outside the influence of the XR and IR, including sites in the lower Iriri River

(Fig.2). The sampling sites to measure pCO$_2$ were separated into three classes (Table 1) according to water depth: (I) near bottom: 0.5-1.0 m above the sediment interface; (II) 60%: at 60% of total water depth; (III) surface: up to 0.3 m of water depth. Four classes were created for FCO$_2$ estimation, with the purpose of evaluating spatial heterogeneity: (I) outside reservoirs: sites located on the channels of the Xingu and Iriri rivers outside reservoir areas, in sectors upstream and further downstream of the reservoirs; (II) main channel: Xingu River main channel

within the reservoir area (XR); (III) flooded areas: lands of pasture and upland forest formerly non-flooded during the high water level season and seasonally-flooded forested islands permanently inundated by the reservoirs; (IV) downstream of the dams: sites immediately downstream of the dams that receive the water discharge from turbines of the XR and IR dams. Sampling sites near the confluence of the Xingu and Iriri Rivers (sites P1 and P3, Table 1) were used as reference sites for areas without direct influence of the reservoirs. The sites further

downstream of the dams (P20 and P21) were characterized to investigate the influence of the reservoirs on the downstream FCO$_2$ (Table 1).

During the year of 2017, high and low water level seasons, values of pCO$_2$ in the water column were obtained using the headspace equilibration method accordingly to Hesslein et al. (1991). Each site was sampled in triplicates at surface, 60% and near bottom water depths. Polycarbonate bottles of 1 L were overflowed up to

three times their volume with water drawn by a submersible pump. Rubber stoppers sealed each bottle and then 60 mL of atmospheric air was injected simultaneously to the withdrawal of the same volume of water creating the headspace. The bottles were shaken for three minutes to equilibrate the gas in the water and headspace air. Water was then re-injected simultaneously to the collection of the headspace air. Atmospheric air samples (60 ml) were also collected and used for corrections considering the concentration of CO$_2$ in the atmosphere near the

water surface of each sampling site. All gas samples were transferred to evacuated glass vials capped with butyl rubber stoppers and sealed with aluminum crimps for laboratory analyses. The pCO$_2$ data was determined using a Picarro$^®$ G2201-i cavity ring-down spectroscopy (CRDS) and calculations were based on Wiesenburg and Guinasso (1979).

Carbon dioxide fluxes from the water to the atmosphere were calculated using an infrared gas analyzer (IRGA)

LI-COR$^®$ Li820 during 2016 and 2017 high water seasons. The FCO$_2$ measurements were done using a 7.7 L floating chamber coupled to the IRGA. The analyzer captures the change in CO$_2$ concentration inside the





chamber by constant recirculation driven by a micro-pump with an air flow of 150 mL min$^{-1}$. For each site, three consecutive deployments were made for five minutes each from a drifting boat to avoid extra turbulence. During the 2017 low water season, $CO_2$ mini-loggers (Bastviken et al., 2015) placed inside 6 L floating chambers were

used to measure $CO_2$ fluxes. Sensors were placed inside the 2 chambers and deployed simultaneously during 20-30 minutes with a logging time of 30 seconds. $CO_2$ fluxes from water to atmosphere were done according to the eq. (1) (Frankignoulle et al. 1998):

$$FCO_2 = \left(\frac{\delta pCO_2}{\delta t}\right)\left(\frac{V}{RT_\kappa A}\right),  \tag{1}$$

The $CO_2$ flux ($FCO_2$) in mol $CO_2$ m$^{-2}$ s$^{-1}$ is given by the changes in pCO$_2$ inside the chamber during the

deployment time ($\delta pCO_2/\delta t$, µatm s$^{-1}$), taking into account the chamber volume ($V$, m³), the universal gas constant ($R$, atm m$^3$ mol$^{-1}$ K$^{-1}$), water temperature ($T$, K) and the area covered by the chamber ($A$, m$^2$). Measurements were discarded when the R$^2$ of the linear relation between pCO$_2$ and time ($\delta pCO_2/\delta t$) were lower than 0.90 (R$^2$ < 0.90) or the same sampling site had negative $FCO_2$ values with surface pCO$_2$ higher than 380 ppm.

2.3 Gas transfer velocity ($k_{600}$)

The air-water gas transfer coefficient $k$ (cm h$^{-1}$) of $CO_2$ was estimated based on the flux measurements in association with the surface water concentration by eq. (2):

$$k = \frac{V}{A.\alpha}\ln\left(\frac{pCO_{2w}-pCO_{2i}}{pCO_{2w}-pCO_{2f}}\right)/(t_f - ti),  \tag{2}$$

Where $V$ and $A$ are the chamber volume (cm³) and area (cm²), $\alpha$ is the Ostwald solubility coefficient

(dimensionless), $t$ is the time (h), and the subscripts $w$, $i$ and $f$ refers to the partial pressure in the surface water, and initial and final time inside the chamber. Ostwald solubility coefficient was calculated from K0 as described by Wanninkhof (2009). To an adequate comparing the k values were normalized into $k_{600}$ – values following the eq. (3) and (4) (Alin et al., 2011; Jähne et al., 1987; Wanninkhof, 1992):

$$k_{600} = k_T\left(\frac{600}{Sc_T}\right)^{-0.5},  \tag{3}$$

Where $k_T$ is the measured $k$ value at in situ temperature (T), $Sc_T$ is the Schmidt number calculated from temperature and 600 is the Schmidt number for temperature of 20° C. The Schmidt number is calculated as a temperature (T) function:

$$Sc_T = 1911.1 - 118.11\,T + 3.4527\,T^2 - 0.041320\,T^3,  \tag{4}$$

Gas transfer velocity calculation is related only to 2017 period due methodological issues.

2.4 Physical-chemical characteristics of the water column

Water temperature, water depth, pH, dissolved oxygen (DO) and conductivity measurements were carried out using a multiparameter probe (EXO2®, YSI). These measurements were done following the same water depth classes applied to pCO$_2$ (surface, 60% and near bottom), during the 2016 and 2017 samplings. Technical challenges prevented measurement of these parameters, other than wind speed, air and water temperatures





during the 2017 low water sampling. Additionally, air temperature and wind speed were measured at the same
time of chamber deployments with assistance of a handheld meteorological station (Kestrel$^®$ 5500), 2 m above
the water surface.

### 2.5. Statistical analysis

Statistical analyses were performed in order to evaluate the spatial and seasonal variation of $FCO_2$, $pCO_2$ and
$k_{600}$ variables as well as to check the correlation among $CO_2$ variables and water column variables. Normality and
heterogeneity of variance was not achieved by Shapiro-Wilks and Bartlett tests, respectively. Thus, non-
parametric and multivariate statistical tests were used. The seasonal and spatial variability of $FCO_2$, $pCO_2$ and
$k_{600}$ were tested by PERMANOVA analysis (Anderson, 2001) using the Euclidian index as distance method. The
$FCO_2$ statistics were assessed separately by season due to the different sampling methods. To evaluate the
correlation between $FCO_2$ *versus* $pCO_2$, $FCO_2$ *versus* wind speed, $k_{600}$ *versus* wind speed and $pCO_2$ *versus*
physical-chemical variables (pH, DO and water temperature) a Spearman correlation test was performed (Zar,
2010). Lastly, the $pCO_2$ averages among the reservoirs were tested by a T-test to independent samples, since
the assumptions were achieved (Zar, 2010). All statistical analyses were performed in R (R Development Team
Core, 2016) using the Vegan package (Oksanen et al., 2017) and Statistica (Statsoft 8.0) using 5% (0.05) as
critical alpha for significance.

### 3 Results

#### 3.1 Temporal and spatial variability in $pCO_2$ and $FCO_2$

Our overall $pCO_2$ results (1163 ± 660 µatm) presented a significant variation between seasons ($F_{2:57}$= 0.76, R²=
0.01, p < 0.05). The $pCO_2$ averages by season ranged from 976 ± 633 to 1,391 ± 630 µatm for low and high
water seasons, respectively (Figure 3). $pCO_2$ seasonal variability was also evident within the river channel
(averaging 789 ± 140 and 675 ± 510 µatm for high and low water, respectively) and flooded areas (averaging
1,774 ± 532 and 1,438 ± 682, for high and low water, respectively). Significant overall $pCO_2$ variability was also
observed among environments ($F_{2:57}$= 14.46, R²= 0.37, p < 0.05), with highest $pCO_2$ values observed during high
water in flooded areas and lowest values occurring in the river channel during low water (Fig.3).

The $pCO_2$ spatial variability was persistent when seasons were tested separately ($F_{2:24}$= 12.00, R²= 0.60, p <
0.05 for high water season and $F_{2:26}$= 6.35, R²= 0.36, p < 0.05 for low water season). Highest average $pCO_2$ by
environment was observed during the high water season downstream of the dams (1,879 ± 551 µatm) while the
river channel showed the lowest (789 ± 140 µatm). In contrast, during the low water season the highest average
$pCO_2$ values were found in flooded areas (1,438 ± 682 µatm). However, outside reservoir areas had the lowest
$pCO_2$ (637 ± 320 µatm) during low water (Fig.3).

Water depth also influenced $pCO_2$, considering data from both seasons ($F_{2:57}$= 4.58, R²= 0.07, p < 0.05). Higher
$pCO_2$ was registered in near bottom zones in relation to the water surface, averaging 1,269 ± 689 and 998 ± 613
µatm, respectively. The highest and lowest $pCO_2$ averages were observed near the bottom and 60 % water
depths in zones of flooded areas and river channel (averaging 2,838 ± 83.19 and 281 ± 143 µatm, respectively)
(Table 2).

Significant fluctuation of $pCO_2$ can be observed according to water season and type of sampling site, as
described previously, with these variations influencing $FCO_2$ values. As such, $pCO_2$ was positively correlated to



$FCO_2$ during both high and low water (r= 0.80; p < 0.05 and r= 0.68; p < 0.05, respectively) (Fig.3). The average $FCO_2$ for all sites during 2016 and 2017 high water seasons was 1.38 $\pm$ 1.12 µmol $CO_2$ $m^{-2}$ $s^{-1}$, without significant

variation between years ($F_{1:28}$= 0.09, $R^2$= 0.01, p > 0.05). Due to the lack of variation between high water $FCO_2$ in both years, these data were treated as a single data set for the later calculations. As previously mentioned, the different sensors used to measure the $FCO_2$ in the low water season of 2017 could not be cross-calibrated with the IRGA used for the flux measurements during the two high water sampling campaigns, therefore we choose to evaluate the spatial variation separately, without a seasonal comparison for $FCO_2$.

The highest and lowest $FCO_2$ values were observed during the low water season (12.00 ± 3.21 µmol $CO_2$ $m^{-2}$ $s^{-1}$ and -0.52 µmol $CO_2$ $m^{-2}$ $s^{-1}$) (Fig.3). Significant $FCO_2$ variation between environments sampled during high water was observed ($F_{3:28}$= 7.94, $R^2$= 0.43, p < 0.05), otherwise the low water season had relatively homogeneous $FCO_2$ values ($F_{3:17}$= 2.67, $R^2$= 0.14, p > 0.05) (Fig.4). The highest and lowest average $FCO_2$ occurred during the high water season in sectors downstream of the dams (2.89 $\pm$ 1.74 µmol $CO_2$ $m^{-2}$ $s^{-1}$) and on flooded areas (0.84

$\pm$ 0.42 µmol $CO_2$ $m^{-2}$ $s^{-1}$, both reservoirs), respectively. Negative fluxes of $CO_2$ were observed during the low water season in the river channel exclusively in areas outside the reservoirs (Table 2) (Fig.4)

Similarly to $pCO_2$, the fluxes in the river channel were higher in the high water season, while on flooded areas it was lower during this season (Table 2).

3.2 Run-of-the-River and Storage reservoirs

A pronounced decrease in average $pCO_2$ values was observed from high water to low water season in the XR (1,244 ± 698 and 839 ± 646 µatm, respectively), while the opposite occurred on the IR (1,676 ± 323 and 1,797 ± 354 µatm, respectively). $FCO_2$ followed the same pattern in both reservoirs, with a decrease in average values in XR from high water (0.94 ± 0.41 µmol $CO_2$ $m^{-2}$ $s^{-1}$) to low water seasons (0.69 ± 0.28 µmol $CO_2$ $m^{-2}$ $s^{-1}$) and an $FCO_2$ increase in the IR from high water (1.08 ± 0.62) to low water seasons (7.32 ± 4.06 µmol $CO_2$ $m^{-2}$ $s^{-1}$).

Spatial analyses compared the distribution of $pCO_2$, $FCO_2$, $k_{600}$ and wind velocities within and between both reservoirs. For this analysis, flooded areas and river channel on XR were evaluated together as well as downstream the dams and outside reservoirs. $pCO_2$ showed no significant difference between the XR and the IR ($F_{2:57}$= 0.76, $R^2$= 0.01, p > 0.05), even when seasons were evaluated separately ($F_{1:24}$= 1.12, $R^2$= 0.01, p > 0.05 and $F_{2:32}$= 0.99, $R^2$= 0.03, p > 0.05 to high and low water, respectively). However, when both reservoirs were

compared, XR presented $pCO_2$ on average 721 µatm lower (T-value= -3.31, df= 39, p < 0.05). Standing vegetation type in XR flooded areas influenced $pCO_2$. Different $pCO_2$ values were observed for pasture (1,161 and 708 ± 260 µatm, for high and low water, respectively), upland forest (1,750 ± 157 and 874 ± 153 µatm, for high and low water, respectively) and seasonally flooded forest (1,189 ± 38 and 880 ± 46 µatm, to high and low water respectively). As expected, $FCO_2$ also had similar response to the flooded vegetation type (0.86 ± 0.52 and

0.73 µmol $CO_2$ $m^{-2}$ $s^{-1}$ for pasture, 1.01 ± 0.29 and 0.34 ± 0.06 µmol $CO_2$ $m^{-2}$ $s^{-1}$ for upland forest and 0.72 ± 0.37 and 0.62 ± 0.24 µmol $CO_2$ $m^{-2}$ $s^{-1}$ for seasonally flooded forest during high and low water, respectively).

As observed for $pCO_2$, there was no effect of reservoir type on $FCO_2$ variability during high water conditions ($F_{1:28}$= 0.32, $R^2$= 0.01, p > 0.05). In contrast, during low water conditions, $FCO_2$ varied significantly ($F_{1:17}$= 34.07, $R^2$= 0.61, p < 0.05), specifically in the IR, which had the highest average $FCO_2$ (0.69 ± 0.28 and 7.32 ± 4.06 µmol

$CO_2$ $m^{-2}$ $s^{-1}$ for XR and IR, respectively). Fluxes of $CO_2$ in outside reservoir zones averaged 0.48 ± 0.61 µmol $CO_2$ $m^{-2}$ $s^{-1}$ and had a gradient pattern downstream. The furthest sites downstream of IR (90 and 25 km downstream





of the Belo Monte dam, respectively) had decreased average $pCO_2$ and $FCO_2$ values in relation to downstream of the dams' outflow (Table 2).

### 3.2 Gas transfer velocity ($k_{600}$)

Gas transfer velocities were calculated for each chamber during the high and low water season of 2017. We were not able to estimate $k$ for the 2016 high water season due to lack of $pCO_2$ measurements and as described for $FCO_2$ data, on $k_{600}$ the spatial analysis also was made separately to each season due to different sensors used for flux measurements. The average $k_{600}$ was $17.8 \pm 10.2$ and $34.1 \pm 24.0$ cm h$^{-1}$ for high and low water seasons, respectively, without significant heterogeneity on the distribution through environments ($F_{2:9}= 2.41$, $R^2= 0.46$, $p >$

$0.05$ and $F_{2:12}= 0.16$, $R^2= 0.03$, $p > 0.05$, respectively). However, $k_{600}$ had a strong pattern of correlation with wind velocities ($r= 0.73$; $p < 0.05$) during the high water season, although this pattern was not observed during the low water season ($r= 0.53$; $p > 0.05$).

Wind speeds ranged from 0.7 to 4.8 m s$^{-1}$, considering measurements for all sites and periods. However, no variation was observed between sampled seasons ($F_{1:37}= 0.89$, $R^2= 0.01$, $p > 0.05$). In contrast, there was

significant variability of wind speeds among environments ($F_{2:37}= 6.13$, $R^2= 0.23$, $p < 0.05$) (Fig.5). Highest average wind speed was observed on the river channel while downstream of the dams had the lowest value ($3.21 \pm 0.89$ and $1.66 \pm 0.88$ m s$^{-1}$, respectively) (Table 3).

### 3.3 Physical-chemical characteristics

The air temperatures at the studied sites varied between 27.5 and 33.8 °C during sampling in both seasons, with

the maximum temperatures registered during the low water period. The superficial water temperature ranged from 29.2 to 32.7 °C, with maximum temperature registered during the high water period. The lowest and highest average pH values were observed in waters of flooded areas and river channel (Table 3). The water column was relatively well-oxygenated in all studied environments, reaching highest average DO concentration ($7.28 \pm 0.73$ mg L$^{-1}$) in the outside reservoirs and lowest concentration in flooded areas ($5.44 \pm 2.00$ mg L$^{-1}$) (Table 3).

Conductivity varied from 18.40 to 38.30 µS cm$^{-1}$ in the studied environments, with the highest average value ($31.60 \pm 8.63$ µS cm$^{-1}$) in flooded areas and lowest ($29.30 \pm 4.85$ µS cm$^{-1}$) downstream of the dams (Table 3).

In aquatic environments, $CO_2$ concentration is expected to correlate with physical-chemical characteristics of the water column. In the study sites, $pCO_2$ is negatively and strongly correlated with pH ($r= -0.75$; $p < 0.05$) and DO ($r= -0.88$; $p < 0.05$). Correlation between $pCO_2$ and water temperature was absent ($p > 0.05$) while $FCO_2$ was

positively correlated with wind speed ($r= 0.53$; $p < 0.05$) (Figure 5) (Table 3).

### 4 Discussion

It has been shown that the amount of $CO_2$ in the water column and $CO_2$ emissions from Amazonian rivers to the atmosphere varies significantly among seasons, with higher fluxes generally observed during the high water season (Alin et al., 2011; Rasera et al., 2013; Richey et al., 2002; Sawakuchi et al., 2017). The increase in $pCO_2$

during the high water season can be related with the large input of terrestrial organic and inorganic carbon into the rivers by lateral surface and subsurface flow of water (Raymond and Saiers, 2010, Ward et al., 2017). In the study area, $pCO_2$ in areas flooded by reservoirs of the Belo Monte hydropower complex had higher average values than in the river channel even with the decrease from the high water to the low water season. Specifically





on the IR, there was an increase in $pCO_2$ during the low water season. Decaying vegetation and the carbon stored in the flooded soils may be the major and constant source of OM in flooded areas of hydropower reservoirs that stimulates heterotrophic activity and maintains a high and more stable $CO_2$ production during initial years of impoundment (Guérin et al., 2008). Consequently, $pCO_2$ in flooded areas of both XR and IR had little variation between the high and low water season due to constant renewal of OM and weak $CO_2$ sinks.

Another factor that may influence the seasonal variability in $pCO_2$ observed in the Xingu River is nutrient levels. In Amazonian lakes nitrogen and phosphorous normally have higher concentrations during the high water season, but become more available in the low water season due to the different vertical mixing pattern of each season (Tundisi et al.,1984). Higher concentrations of these nutrients would favor primary productivity (Farjalla et al., 2006; Vidal et al., 2015), taking up part of the $CO_2$ available in the water column. However, this algal biomass is readily consumed by heterotrophs (Qin et al., 2013), and in some cases stimulates the breakdown of terrestrially-derived OM (Ward et al., 2016). Thus, increased primary production may not always result in decreased $pCO_2$, per se.

The vertical heterogeneity of $pCO_2$ in the water column of the study site was registered by data acquired at three depth classes (surface, 60% and near bottom). The higher $pCO_2$ values observed in the near bottom can be attributed to an input of $CO_2$ from groundwater (Cole et al., 2007; Marotta et al., 2011) and from heterotrophic respiration in the sediments associated with a smaller uptake of $CO_2$ by photosynthetic activity by algae and macrophytes in the deeper layers of the water column than at the surface (Hansen and Blackburn, 1992). These processes can also vary according to different environmental conditions like water depth, allochthonous inputs and reservoir dimensions (Cardoso et al., 2013; Roland et al., 2010; Pacheco et al., 2015). Higher $pCO_2$ in the bottom waters was also related with lower dissolved oxygen and high pH, since heterotrophic respiration use $O_2$ and the increase in $pCO_2$ would lead to pH decrease due to the conversion of $CO_2$ into carbonic acid ($H_2CO_3$) (Duarte and Prairie, 2005; Frankignoulle et al., 1996; Neal et al., 1998; Wang et al., 2011).

Since $pCO_2$ and $FCO_2$ are intimately correlated, the pattern observed on $pCO_2$ extends to $FCO_2$. The high water $FCO_2$ and $pCO_2$ in the Belo Monte reservoirs area (Table 2) were in the same order of magnitude of emissions measured in Amazonian clearwater rivers unaffected by impoundment, including the Tapajós River, which has hydrologic conditions similar to the Xingu River (Alin et al., 2011; Rasera et al., 2013; Sawakuchi et al., 2017). Our low water average $pCO_2$ and $FCO_2$ values, however, had an opposite pattern in relation to literature values and were one order of magnitude higher (Table 2) than observed in clearwater rivers with natural flowing waters. Even when analyzed separately, the average $FCO_2$ values observed for the Belo Monte reservoirs (XR and IR) overcomes these natural emissions.

Large spatial heterogeneity was noticed in both Belo Monte reservoirs during high water season. The XR had main channel $pCO_2$ decreasing as $FCO_2$ increased and the contrary occurred in flooded areas, including the IR. The main carbon source of the XR is likely the standing vegetation (flooded forests and pasture), where higher $pCO_2$ for flooded areas was registered. Environmental conditions such as flooded vegetation play an important role in the $CO_2$ production, since it is the main source of OM for $CO_2$ production and may help to create gradients of reservoir $CO_2$ emissions (Roland et al., 2010; Teodoru et al., 2011). However, although with vegetation clearing, the most expressive emissions recorded in the study area are from the IR, where arboreal vegetation was suppressed before flooding, indicating that the OM in the flooded soil is a major source of carbon that is consumed and converted into $CO_2$. The diversity of environmental conditions in both Belo Monte reservoirs, like



vegetal suppression in IR, minor size of XR and consequently lesser flooded area, variation on hydrodynamic forces and water depth had significant influence on $FCO_2$ and $pCO_2$ values as occurs in other hydropower reservoirs (Cardoso et al. 2013; Teodoru et al., 2011, Räsänen et al., 2018; Roland et al., 2010).

About 42% of the XR area is the original channel of the Xingu River. However, the water velocity under reservoir conditions is slower than before since the water flow is regulated by Pimental dam spillways. $FCO_2$ measured upstream of the reservoir during the high water season, in a sector where the channel is flowing under natural
conditions (Iriri river sites), was significantly higher than in the XR sector (Table 2). The increase of light penetration and nutrient supply combined with the decrease in water flow may favor primary productivity (Thomaz and Bini, 2007) as occurs in most hydropower reservoirs (McCartney, 2009; Ran et al., 2015; Zheng et al., 2011). Consequently, $CO_2$ concentrations in the water column may decrease, especially on upper layers, in response to the increased photosynthetic uptake of $CO_2$ during lower rainfall periods (Amaral et al., 2018). During the low
water season, $pCO_2$ and $FCO_2$ drastically decreased, especially in the river channel. Contrary to high water, $CO_2$ fluxes were homogeneous and relatively low in most of the environments (with exception to IR) linked to elevated photosynthetic activity (Table 2). In addition, the $CO_2$ undersaturation in relation to atmosphere and negative fluxes is attributed to elevated primary productivity due to the high light penetrance, low suspended sediment concentration, which is similar to previously observations in Amazonian floodplain lakes and other clearwater
rivers during the low water season (Amaral et al., 2018, Rasera et al 2013, Gagne-Maynard et al., 2017). The occurrence of negative $CO_2$ fluxes was observed only in outside reservoirs sectors, on furthest studied site downstream IR.

The IR is located on an upland forest area that received vegetation clearing before flooding. Even after the forest removal, the upper soil layer may have kept  a high concentration of plant-derived material and it is still an
important source of OM that may take several years to decay (Abril et al., 2005; Campo and Sancholuz, 1998; Guérin et al., 2006). This condition explains the higher average $pCO_2$ in IR than in XR, with the former area also having higher average $FCO_2$ values, possibly linked to wider open areas and a longer fetch for wind to create surface turbulence (Amaral et al., 2018; Paranaiba, et al. 2017). The XR has substrates with relatively reduced carbon storage because almost half of the area is represented by the original river channel dominated by bedrock
or sandy substrates and flooding forest islands formed by sand and mud deposition, would not store much carbon (Sawakuchi et al., 2015).

Some differences emerge from the downstream emissions of the IR (5.26 $\pm$ 1.94 µmol $CO_2$ m$^{-2}$ s$^{-1}$ and 2,122 $\pm$ 106 µatm, to high water $FCO_2$ and $pCO_2$ respectively) and XR (2.85 $\pm$ 1.12 µmol $CO_2$ m$^{-2}$ s$^{-1}$ and 1,241 $\pm$ 48.24 µatm, to high water $FCO_2$ and $pCO_2$ respectively). In comparison to $CO_2$ emission of other downstream tropical
storage reservoir, the $FCO_2$ of Petit Saut reservoir in French Guiana was almost double (10.49 $\pm$ 3.94 µmol $CO_2$ m$^{-2}$ s$^{-1}$) (Guérin et al. 2006) of our highest downstream $FCO_2$ in IR during high water season. The $FCO_2$ in downstream zones of Belo Monte and Petit Saut reservoirs show $CO_2$ supersaturation similar to the reservoir waters. Although Petit Saut has a smaller dam and reservoir compared to the Belo Monte reservoirs, the elevated $pCO_2$ drove the downstream emissions. Both Belo Monte (this study) and Petit Saut (Guérin et al. 2006)
reservoirs have waters most $CO_2$ supersaturated during the high water season. These $CO_2$ rich waters have potential to increase downstream emissions with the passage through the dam turbines as well as $FCO_2$ and $pCO_2$ may be altered for several km downstream the dam (Abril et al., 2005; Guérin et al., 2006; Kemenes et al., 2011; 2016).



In this study, the furthest downstream sites situated 90 and 25 km downstream the XR and IR, respectively, presented average $pCO_2$ and fluxes lower than the sites in the river channel upstream the XR. Our upstream XR sites also had higher $FCO_2$ than observed in undisturbed sectors of other large clearwater rivers in the Amazon (Table 2). The site downstream IR is within the river extent that could still be affected by the reservoir (Kemenes et al., 2016). However, the site further downstream the XR should be unaffected by the reservoir due to the long distance (90 km) and the fact that it receives waters from the Volta Grande region, where the bedrock riverbed

with large rapids and waterfalls quickly degas the dissolved $CO_2$ coming from the upstream reservoir. Thus, this site could potentially be considered as a reference for natural $FCO_2$ values

The decrease in $pCO_2$ and $FCO_2$ persisted in areas downstream of the Belo Monte reservoirs as indicated by measurements performed during the high water and low water seasons. The higher $FCO_2$ and $pCO_2$ in the areas downstream of the dams of both reservoirs compared to river channel during high water indicates some turbine

activity. However, this pattern is inconclusive at least for the low water season as the Belo Monte dam turbines were under installation and not fully operational. Therefore, downstream of both dams assumed river channel traits in relation to fluxes and $CO_2$ water saturation in this season.

Although no significant variation of $k_{600}$ was observed between the reservoirs of the Belo Monte hydropower complex, the average values for high and low water were similar to observed values in other rivers of the Amazon

Basin (Araguaia, Javaés, Tapajós and Teles Pires) (Alin et al. 2011; Rasera et al. 2013). We observed that in the XR reservoir area, $FCO_2$ values were higher in the main channel, where in addition to the constant water flow due to the ROR type reservoir, it also had a greater open area for wave formation in comparison with the sheltered flooded areas. This is consistent with the strong positive correlation observed between wind speed and $FCO_2$ here and in other large rivers where a vast water surface interacts with wind along its fetch, promoting the

formation of waves that enhances water turbulence, $k_{600}$ and $FCO_2$ (Abril et al., 2005; Paranaíba et al. 2017; Rasera et al., 2013; Raymond and Cole, 2001; Vachon et al., 2013).

Among the traits that differs for each hydropower dam, the operation design may play an important role for carbon fluxes in the reservoir and its downstream sector. In comparison to an older Amazonian reservoir with storage design, the Tucuruí reservoir built in 1984 in the clearwater Tocantins River presented $CO_2$ fluxes of 3.61

± 1.62 µmol $CO_2$ m$^{-2}$ s$^{-1}$ (Lima et al., 2002), which is higher than fluxes observed in the XR, but it is three times lower than the highest flux registered for the IR in this study. The Tucuruí hydropower dam lacked vegetal suppression and has a large reservoir and hypolimnetical waters (Fearnside, 2002; Kemenes et al., 2016). Nevertheless, the Tucuruí $CO_2$ fluxes were measured 18 years after reservoir filling and estimates for storage reservoirs point out that 10 years after impounding, OM from flooded soils still contributes to carbon fluxes to the

atmosphere (Abril et al., 2005; Guérin et al., 2008). Additionally, it must be considered that the Belo Monte hydropower complex had partial vegetation removal in some areas of the XR that potentially reduced observed $CO_2$ fluxes. Even with vegetal suppression, the IR emission is higher than other Amazonian dams, especially during low water conditions. Environmental and seasonal traits are important factors that influence $CO_2$ emissions, therefore it is important that reservoir type and dam design to maintain these conditions similar to

natural river traits, in order to mitigate reservoir emissions.

5 Conclusions

Here we observed variability in $CO_2$ fluxes related to the type of environment and land use of flooded areas after impoundment of the Belo Monte hydropower complex. $CO_2$ emissions were 15 and 90% higher for the IR



compared to XR during high and low water season, respectively, indicating that storage reservoirs may be larger sources of $CO_2$ to the atmosphere compared to RORs. The Belo Monte hydropower complex had average $CO_2$ emission similar to Amazonian clearwater rivers without impounding in only one season (high water) and considerably lower than other tropical reservoirs. However, IR fluxes exceeded emissions measured in storage reservoirs of other tropical rivers. The IR presented the highest fluxes of the study even with vegetation removal.

Although vegetation removal is perhaps one of the most effective approaches for reducing greenhouse gas emissions from hydropower reservoirs, we show that tropical reservoirs can present significant emissions even after vegetation suppression. Additional monitoring of $CO_2$ flux is needed to evaluate the Belo Monte hydroelectric complex emission with both dams working at full capacity, including measurements in the Volta Grande do Xingu region downstream the reservoir (XR) in the Xingu River channel. This is necessary to obtain robust and reliable assessment of carbon emissions related with the electricity produced by the Belo Monte

hydropower complex over its entire lifecycle.

Author contribution

Kleiton R. Araújo collected and analyzed the data, as prepared the manuscript with contribution of all co-authors. Henrique O. Sawakuchi designed the study, cooperated in the field sampling and supported with guidance on data analysis. Dailson J. Bertassoli Jr. also collected the data and conducted the laboratorial analysis. André O.

Sawakuchi attained the grant award, contributed to setting up the field equipment, measuring infrastructure and field sampling. Kleiton R. Araújo. Karina D. Silva and Thiago V. Bernardi conducted the statistical analysis. Nicholas D. Ward and Tatiana S. Pereira contributed with technical advice and guidance throughout the project implementation and paper writing stages.

Competing interests

We declare that we have no conflict of interests.

Acknowledgements

This study has funding by Fundação de Amparo à Pesquisa do Estado de São Paulo (FAPESP, grant 16/02656-9) and from Coordenação de Aperfeiçoamento de Pessoal de Nível Superior (CAPES) as master scholarship for Kleiton R. Araújo. We are grateful to Marcelo G. P. de Camargo, Hildegard de H. Silva, Victor A. T. Alem, Agna

L. B. Figueiredo, and Thomas K. Akabame for the field sampling and laboratorial support. André O. Sawakuchi is supported by Conselho Nacional de Desenvolvimento Científico e Tecnológico (CNPq, grant 304727/2017-2).






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

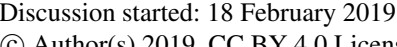

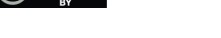

Figure captions

Fig.1: Average river discharge (in m³ s$^{-1}$) of the Xingu River (left Y axis) and precipitation (in mm month$^{-1}$) (right Y axis) at Altamira from 2004 to 2014. Bars indicate monthly standard deviation. Data is from ANA (2017) and Inmet (2017).

Fig.2: Sampling sites upstream (Iriri river), within and downstream of the reservoirs and the location of the two
dams (white bars) in the Xingu river. Land cover data is based on vegetation characterization from Almeida et al. (2016), here non-forested area groups pasture, deforested, secondary vegetation and urban areas.

Fig.3: Boxplots showing the spatial and temporal variability of $pCO_2$ and $FCO_2$. Whiskers indicate standard deviation, boxes are maximum and minimum values and the middle points are mean values. High water $FCO_2$ (2016 and 2017 campaigns) and $pCO_2$ from all depths values were averaged to characterize the environmental
category. Temporal variation may be observed by the overall seasonal variation to $pCO_2$ and $FCO_2$ during high (A) and low water (B), likewise the spatial distribution to both $pCO_2$ (C) and $FCO_2$ (D) on each season and according environment.

Fig.4: Spatial and temporal variation of the $FCO_2$ values (µmol $CO_2$ m$^{-2}$ d$^{-1}$) in the reservoirs (XR and IR) of the Belo Monte hydropower complex during high water (A) and low water (B) seasons. High water season data
include both averages of measurements carried out during the years of 2016 and 2017. Colors and circles sizes indicate type and intensity of $CO_2$ fluxes.

Fig.5: Scatterplots between $FCO_2$ (A) and $k_{600}$ (B) as a function of wind speed. Values from figure 5 (A) include high and low water seasons. Figure 5 (B) comprises only high water values for statistical correlation (Spearman correlation).









Figures

Fig.1

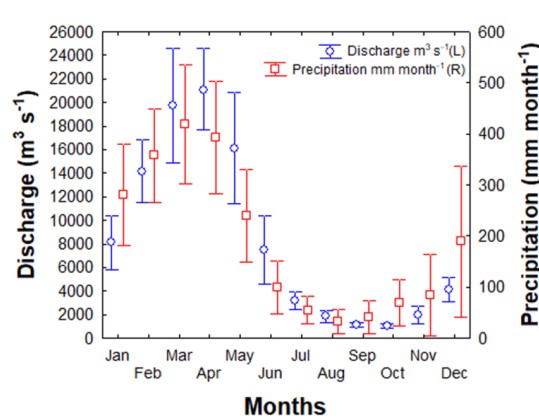

Fig.2

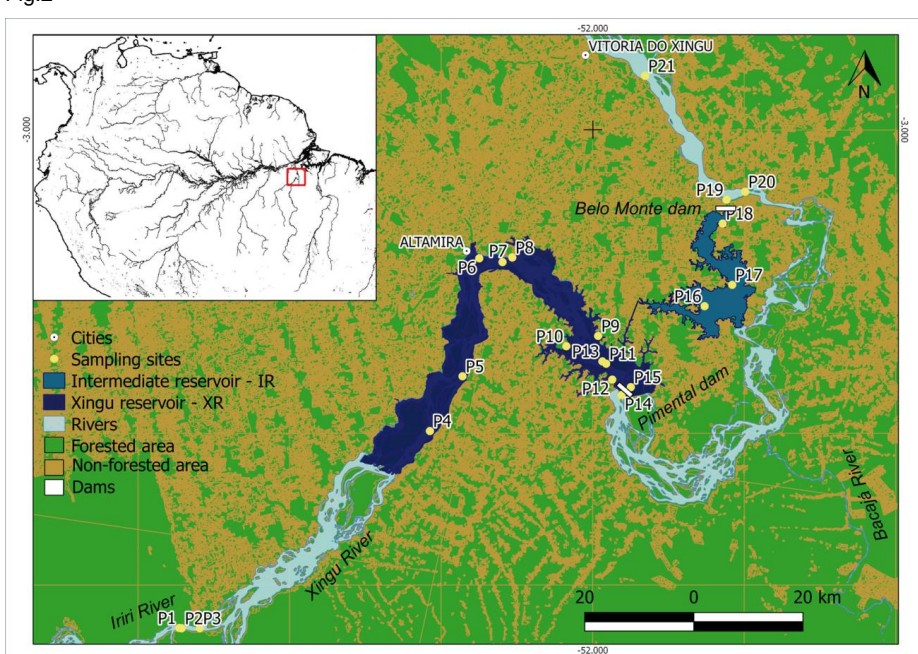






Fig.3

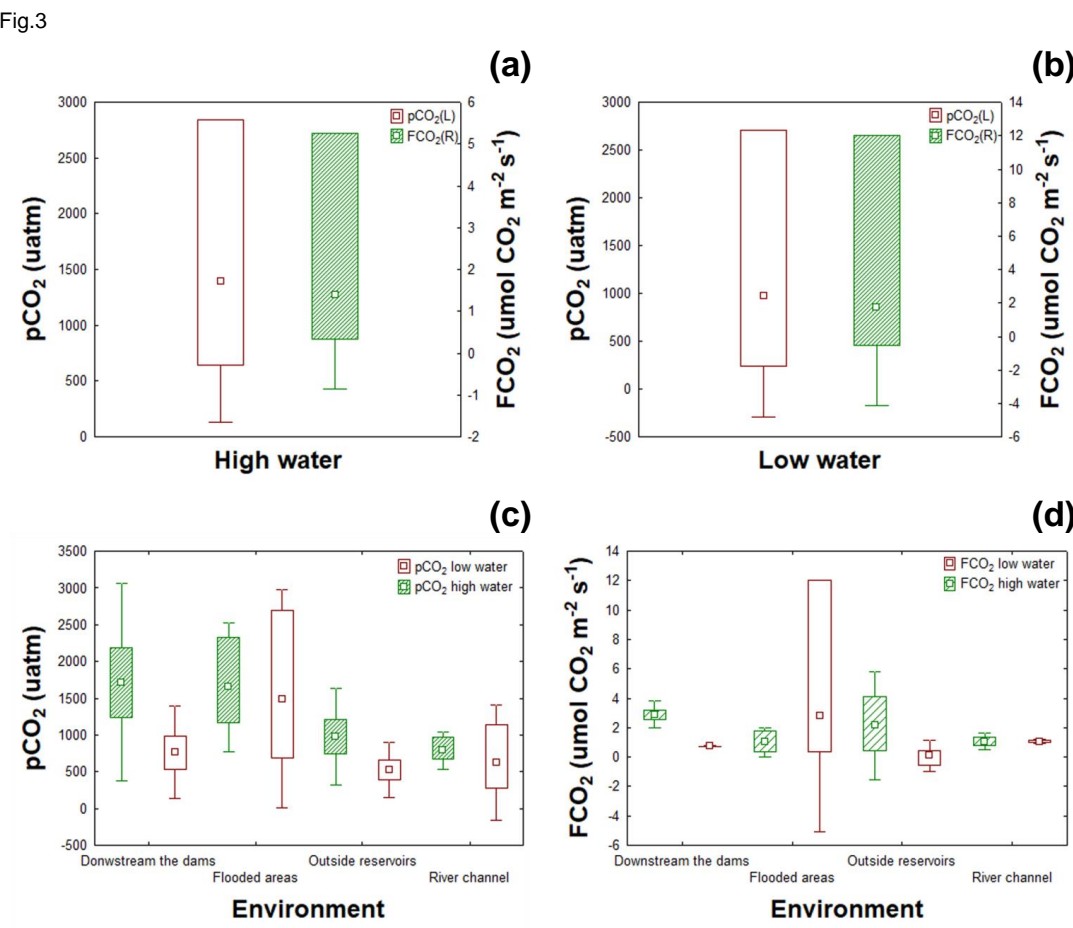








Fig.4

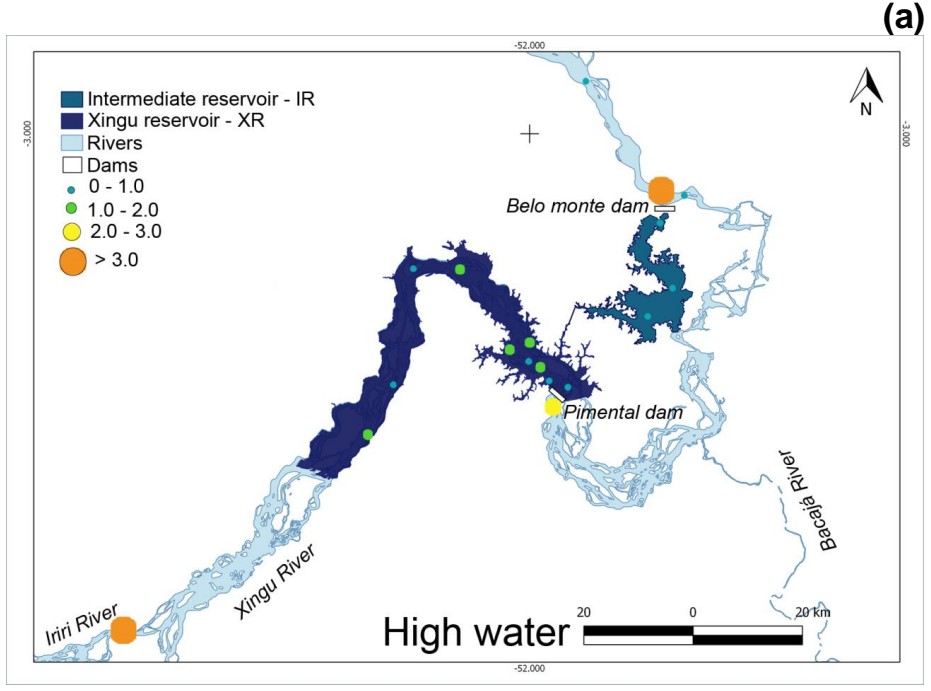

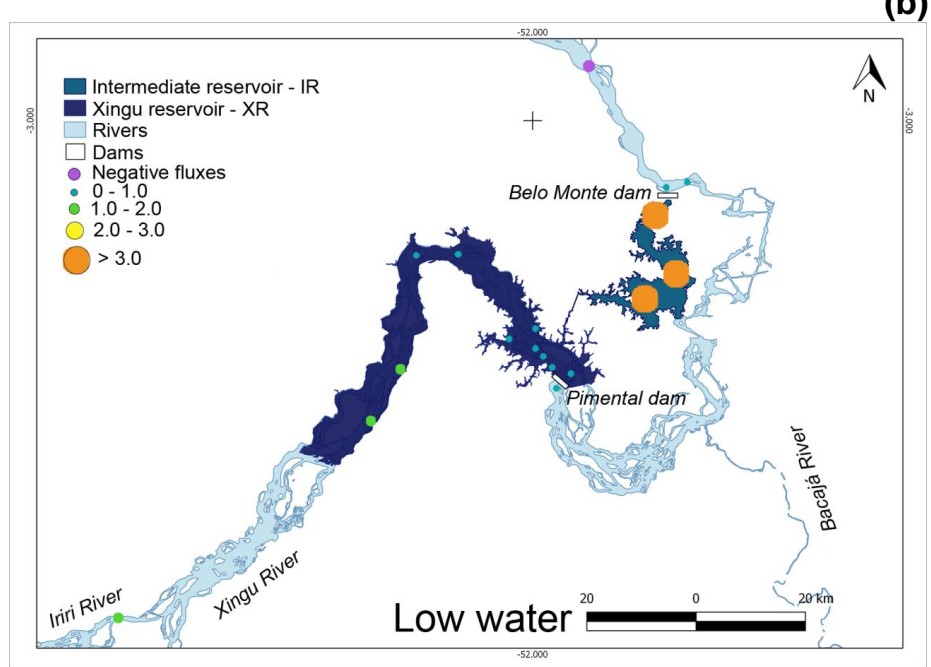





Fig.5:

**(a)**

**(b)**

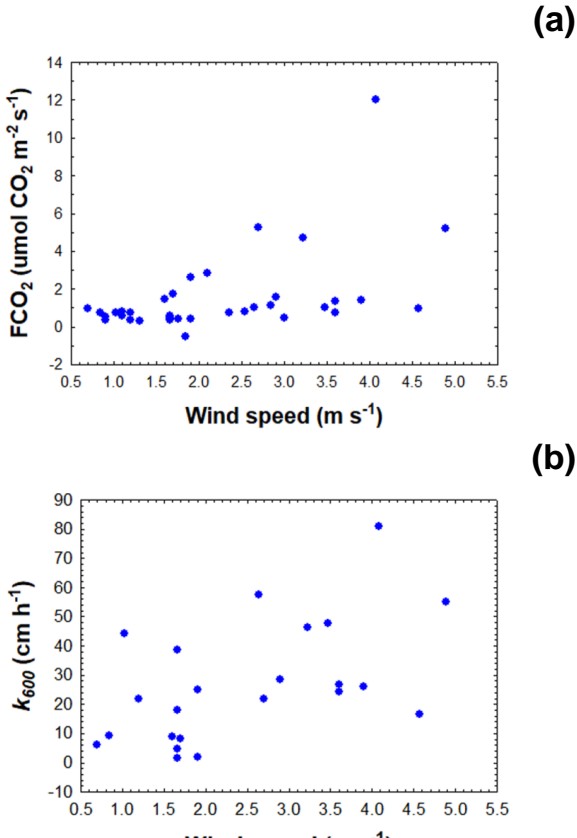









Table captions

Table 1: Locations of sampling sites in the Xingu and Iriri Rivers and reservoirs (XR and IR) of the Belo Monte hydropower complex. Sites were classified according to pre and post-flooded vegetation types, water depth and sampling season (H1: high water of 2016, H2: high water of 2017 and L: low water of 2017).

Table 2: Summary of $FCO_2$, $pCO_2$, gas transfer velocities ($k_{600}$) averages and literature values. High water season averages to $FCO_2$ comprehends both sampling years, since no significant variation was detected. $FCO_2$, $pCO_2$ and $k_{600}$ referential values ($FCO_2$ Lit, $pCO_2$ Lit and $k_{600}$ Lit, respectively) were averaged from the Amazonian clear water rivers Tapajós (Alin et al. 2011 and Sawakuchi et al. 2017), Araguaia, Javaés and Teles Pires (Rasera et al. 2013) in the correspondent season when available.

Table 3: Overall average values for physical-chemical conditions comprising the three depth classes (surface, 60% and near bottom) sampled during the high water seasons of 2016 and 2017, with exception to Temp (water temperature) and WS (wind speed), which corresponds to both high and low water. The variables pH (hydrogen potential), DO (dissolved oxygen), Cond (conductivity), Temp and WS are presented according to environment.








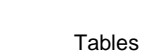
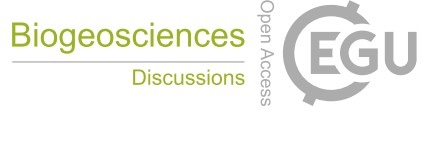

Tables

Table 1

| Site | Longitude | Latitude | Pre-flooding environment | Season | Depth (m) |
|------|-----------|----------|--------------------------|--------|-----------|
| P1 | -3.82115 | -52.682559 | River channel | H1 | ND |
| P2 | -3.82168 | -52.678553 | River channel | L | 13.0 |
| P3 | -3.82153 | -52.678599 | River channel | L | 8.0 |
| P4 | -3.49656 | -52.268961 | River channel | H2, L | 8.1 |
| P5 | -3.40623 | -52.215154 | River channel | H2, L | 7.5 |
| P6 | -3.21182 | -52.187488 | Seasonally flooded forested island | H1, H2, L | 3.0 |
| P7 | -3.21801 | -52.149169 | River channel | H1, H2, L | 20.5 |
| P8 | -3.21045 | -52.133034 | Pasture* | H1, H2, L | 0.35 |
| P9 | -3.33965 | -51.991423 | Upland forest* | H1, H2, L | 6.1 |
| P10 | -3.35664 | -52.043752 | Tributary, reservoir | H2, L | 5.1 |
| P11 | -3.38557 | -51.978184 | River channel | H1, H2, L | 19.3 |
| P12 | -3.41172 | -51.968102 | Pasture* | H1, H2, L | 6.0 |
| P13 | -3.38170 | -51.984364 | Seasonally flooded* forest | H2, L | 7.4 |
| P14 | -3.38557 | -51.978184 | River channel | H1, H2, L | 2.5 |
| P15 | -3.42413 | -51.937447 | Seasonally flooded forested island | H1, H2, L | 11.0 |
| P16 | -3.29069 | -51.815787 | Upland forest | H2, L | 20.4 |
| P17 | -3.44253 | -51.954685 | Upland forest | H2, L | 6.2 |
| P18 | -3.15452 | -51.785845 | Upland forest | H2, L | 58.3 |
| P19 | -3.11501 | -51.779624 | River channel | H1, H2, L | 6.2 |
| P20 | -3.10197 | -51.748847 | River channel | H2, L | 2.6 |
| P21 | -2.91097 | -51.913989 | River channel | H1, H2, L | 9.0 |

*ND - No data collected.*

*\*vegetation not removed prior to reservoirs filling.*








Table 2

| Environment | Reservoir | Sampling season | $FCO_2$ ($\mu mol\ CO_2\ m^{-2}\ s^{-1}$) | $pCO_2$ ($\mu atm$) Surface | $pCO_2$ 60% | $pCO_2$ Near bottom | $k_{600}$ ($cm\ h^{-1}$) | $FCO_2$ Lit ($\mu mol\ CO_2\ m^{-2}\ s^{-1}$) High water | $FCO_2$ Lit Low water | $pCO_2$ Lit ($\mu atm$) High water | $pCO_2$ Lit Low water | $k_{600}$ Lit ($cm\ h^{-1}$) High water | $k_{600}$ Lit Low water | References |
|---|---|---|---|---|---|---|---|---|---|---|---|---|---|---|
| Upstream | OR | High | 4.10 ± 2.16 | ND | ND | ND | ND | | | | | | | Alin et al. 2011 |
| | | Low | 1.06 | 501 ± 71.32 | ND | 766 ± 138 | 47.94 | | | | | | | |
| River channel | XR | High | 1.27 ± 0.31 | 771 ± 56.20 | ND | 808 ± 205 | 26.58 ± 2.10 | ND | 0.75 ± 0.41 | ND | 643 ± 172 | ND | 16.87 ± 10.36 | |
| | | Low | 0.89 ± 0.33 | 612 ± 161 | 281 ± 143 | 871 ± 783 | 30.70 ± 24.64 | | | | | | | |
| Flooded areas | XR | High | 0.78 ± 0.38 | 1,674 ± 17.80 | 1,647 ± 333 | 2,838 ± 83.19 | 8.91 ± 3.22 | 2.6 ± 1.12 | -0.06 ± 0.15 | 1,646 ± 663 | 377 ± 154 | 11.70 ± 5.45 | 5.17 ± 3.39 | Rasera et al. 2013 |
| | | Low | 0.47 ± 0.12 | 1,330 ± 1,210 | 807 ± 103 | 1,498 ± 203 | 15.07 ± 20.49 | | | | | | | |
| Flooded areas | IR | High | 1.08 ± 0.62 | 1,556 ± 375 | 1,876 ± 37.48 | 1,696 ± 455 | 7.13 ± 1.59 | 2.3 ± 0.41 | 0.4 ± 0.18 | 2,620 ± 810 | 724 ± 334 | 8.22 ± 3.80 | 5.05 ± 0.77 | |
| | | Low | 7.32 ± 4.07 | 1,526 ± 263 | ND | 2,069 ± 152 | 60.80 ± 18.02 | | | | | | | |
| Downstream the dams | OR | High | 2.89 ± 1.74 | 2,122 ± 106 | 1,729 ± 689 | 2,257 ± 42.23 | 21.86 ± 11.01 | 1.92 ± 0.96 | 0.4 ± 0.15 | 1,799 ± 753 | 1,037 ± 635 | 12.20 ± 4.35 | 7.0 ± 6.64 | Sawakuchi et al. 2017 |
| | | Low | 0.75 ± 0.01 | 663 ± 372 | ND | 861 ± 257 | 26.90 ± 24.69 | | | | | | | |
| Further downstream | OR | High | 1.55 ± 1.08 | 969 ± 341 | ND | 998 ± 316 | 13.61 ± 16.33 | 1.75 | 0.76 | 450 | 449 | ND | 16.03 | |
| | | Low | -0.07 ± 0.62 | 409 ± 137 | ND | 650 ± 239 | 34.86 ± 18.49 | | | | | | | |
| **Overall average** | | **High** | **1.38 ± 1.12** | **1,193 ± 520** | **1,618 ± 525** | **1,372 ± 755** | **15.61 ± 8.36** | | | | | | | |
| | | **Low** | **1.74 ± 2.94** | **877 ± 651** | **676 ± 276** | **1,191 ± 654** | **34.39 ± 17.74** | | | | | | | |

*IR – Intermediate reservoir.*
*ND - No data available.*
*OR - outside reservoir.*
*XR - Xingu reservoir.*



Table 3

| Environment | pH | DO (mg L$^{-1}$) | Cond (µS cm$^{-1}$) | Temp (°C) | WS (m s$^{-1}$) |
|---|---|---|---|---|---|
| Downstream of dams | 6.62 ± 0.18 | 5.87 ± 1.39 | 29.30 ± 4.85 | 29.52 ± 0.09 | 1.66 ± 0.88 |
| Flooded areas | 6.60 ± 0.26 | 5.44 ± 2.00 | 31.60 ± 8.63 | 29.85 ± 0.66 | 1.96 ± 1.13 |
| Outside Reservoirs | 6.75 ± 0.24 | 7.28 ± 0.73 | 30.59 ± 6.87 | 29.72 ± 0.36 | 2.06 ± 0.84 |
| River channel | 6.81 ± 0.21 | 6.92 ± 0.26 | 29.86 ± 5.30 | 29.44 ± 0.62 | 3.21 ± 0.89 |