# Peer review of "Carbon dioxide (CO2) concentrations and emission in the newly constructed Belo Monte hydropower complex in the Xingu River, Amazonia"

_Biogeosciences, 2019_

## Referee Comment (RC1) · Anonymous Referee #1 · 8 Apr 2019

This paper is about the CO2 concentration and emissions from a newly created hydroelectric reservoir complex in the Amazon area. Given that particularly Amazonian reservoirs have been pointed out as high emitters of greenhouse gases, and since emissions typically are higher the first years after flooding, this study is certainly valuable and interesting. In particular since the new reservoir is a run-of-the-river type, which is supposed to result in lower emissions than storage reservoirs.

The study seems to be well-conducted, based on standard methods. However, the presentation severely lacks focus and clarity. I will give in the following a few ideas on how the paper can be improved, but I really want to urge the senior authors of this

paper to support and help the first author, who is apparently a MSc student and writes his/her first paper (it says in the Acknowledgements). It also takes a thorough revision of English language use and style.

What makes this study interesting is that it studies the Belo Monte hydroelectric complex, a all-new installation in the Amazon (it's not even up at full capacity yet), the biggest in the Amazon so far, and one of the biggest in the world, and one that was heavily disputed and criticized. This is not mentioned at all in the paper! I could imagine that the story could be built around the case of this new and huge installation. New reservoirs typically have elevated emissions, but here apparently biomass was removed before flooding, at least partially. Is this visible in the data? One of the reservoirs is run-of-the-river, does it really have lower emission than the storage reservoir? These questions could be formulated as hypotheses, addressed with the data (i.e. figures should illustrate data in a way that relates to these hypotheses), and then explicitly answered in the Discussion. This would give the study a much-needed 'read thread'.

It will take a thorough rewriting of the manuscript before it may become acceptable, but since it seems to be good data from a understudied site of high interest, I think in the end this could become a valuable addition to Biogeosciences.

Detailed comments: Title: the influence of reservoir traits is not explored to any greater depth. Which traits? I'd suggest to change the title accordingly, maybe "CO2 concentrations and emission in the newly constructed Belo Monte hydropower complex in the Xingu River, Amazonia".

L41. The inland water area number seems wrong. See Verpoorter et al. 2014 GRL.

L42. Only the Raymond study gives a global estimate, the other citations are regional-scale.

L45-54. There's a lot of detail here that is not addressed by this study and could be removed here, e.g. microbial community structure or priming.

L70. While emission are typically high, the lifetime emission of a reservoir is proabably rather a function of the long-term emission level, and the short initial emission pulse may have less influence.

Study Area: This must mention that the installation is new, and it must describe in how far and where vegetation was removed before flooding, and when the flooding took place.

L106 and 114. The water retention times are very short, even for the storage reservoir it's only 1.5 days. Are these numbers correct? If so, these reservoirs, given their size, must be characterized by quite strong water flow, and thus the gas exchange velocity is probably hardly related to wind speed, but rather to water speed.

L112. 97% of the capacity are at the Belo Monte dam, so the ROR dam only produces 3% of the energy even though it contains one third of the number of turbines?

L116. Where is the hypolimnion typically starting? Did you do any depth profiles of T and/or DO? If so, please show and report! If not, please cite a study that states that the thermocline is typically at >20 m.

Section 2.2. It would be more easy to understand if you first described your sampling campaigns, and then tell about any gaps.

L144. Why was 60% of water depth chosen? Seems arbitrary. Also, it would be good to know the actual depth at these sites. A raw data table should be submitted alongside with the paper.

L150. How good was the evacuation? In my experience, it's very difficult to get a good vaccum, but probably 10% or more atmosphere will remain, which may dilute or contaminate your samples. Was this checked?

L154. Start this paragraph with saying "Diffusive $CO_2$ emission was measured with floating chambers". Also, please give the dimensions, shape and type (transparent / opaque) of the chamber.

L161. I guess you mean logging frequency, not time.

L168. Atmospheric pCO2 of 380 ppm seems like an outdated value, or are these your own measurements in air?

L184. This sentence seems unnecessary.

L191. A station is stationary. You probably mean a handheld meter or device?

2.5. Statistics. I did not know Permanova, so this should be better explained. Is it a parametric method? Because it is stated that the data did not follow normal distribution. However, later in this paragraph, you mention some data were normally distributed and used t-test; this is confusing. Also, in the entire paper, report the actual p values, not just if p is lower of higher than 0.05.

Results. In general, this section describes many findings and patterns, but it does so in a quite unstructured way, and is therefore difficult to follow. I really think it would help this paper if only the results were presented that are relevant to the hypotheses or research questions. Also, the language describing the patterns should be improved. For example, it needs to explained what numbers are given (e.g. L208, is this the mean $\pm$ standard deviation, or something else?), and comparisons between two groups describe a difference and not a variation (L208). Also, increase and decrease (e.g. L245 and L249) refer to a change over time and thus some form of time series data, while this study has data for two discrete sampling occasions, and thus can only speak about differences. It should also always be very clear what exactly was compared. For example, in L213, it was unclear what was tested here, the variability in pCO2 within and environment, or between environments?

Again concerning statistics, it is unclear to me how a comparison between two groups can render a R2 value, but maybe that's a part of the Permanova, and should in that case be better explained in the Methods.

L215. Here you speak about spatial variability, but do you mean differences of means

between different environemnts, or the variability of measurements within one environment type?

L219. "Outside reservoir areas" is not a very illuminating term. Could choose another name?

L224. 281 $\mu$atm at 60% depth, how much is that in meters? And how can deep water be undersaturated in oxygen? Typically it is oversaturated. Or was this above a macrophyte bed?

L231. Here it says the the data from the two seasons were pooled, but L237-241, the seasonal data are discussed separately. This is confusing.

L246. The seasonal difference in IR was very small, certainly not a "pronounced difference". Interestingly, FCO2 was very different between seasons in spite of similar pCO2, which indicates a strong variability in k. Was this the case?

L250. What kind of spatial analyses? Comparison of the means for different environments?

L251. "evaluated together", is this warranted? Were these two groups similar?

L256. "Pasture" is a new and undefined category.

L262. What's the measure of variability? It seems that in this study, you mostly compared means, but if you want to address the variability, you maybe want to look at relative standard deviations, interquartile ranges or something similar. If you want to stick to comparing means between environments, please formulate this explicitly in the text.

L263. Varied significantly between what?

L266. The 90 km downstream site is so far away it's not even on the map. I wonder in how far it is relevant to this study at all, or could safely be omitted.

L270-273. Go straight to the results instead of first describing what was not done.

L275. The relationship between k600 and wind speed is very weak. At any wind speed, k can vary with a factor of 2-4. This is quite often the case, and maybe even expected in such system where water moves fast, and thus water turbulence is quite independent of wind speed.

All in all, the Results give many compairosn, What about making matrix tables where you can give test statistics for each comparison?

3.3. Did you ever measure depth profiles? Would be very interesting to show these data, to asses if really the turbine intake is in the epilimnion, and to assess the potential outgassing through turbine passage.

L292. This is not one of your results.

L296. The Discussion should start with your most important finding, not with citing other studies.

L303. This seems to be an important finding. Could you make a figure that illustrates this finding, to make it visible and convincing?

L309-326. This discussion is very hypothetical and not much related to your data.

L327. Not really. In your own data, there is an example of differences in k producing very different emission fluxes in spite of similar pCO2 (see my comment above).

L328-334. This may be the main message of this paper. It would be good if you produced a Figure that illustrates this finding.

L340-341. The Methods need to describe explicitly which areas were flooded with intact biomass, or after biomass harvesting.

L350. Could you actually observe increased water clarity in your data / samplings? If not, this discussion is not helpful to explain your data.
L355. pCO2 were only lower during low water compared to high water in the downstream and dam categories. For flooded and river channel, they were similar (Fig.3). So it is not warranted to speak about a "drastic decrease".

L375-383. Could the difference between Belo Monte and Petit Saut be explained by different water intake depths? Do you have water profile data?

L391. It seems not warranted to assume that any site or time point should serve as a "reference" for river pCO2, since it varies in time and space.

L395. What is meant by "turbine activity"?

L398-406. I think you could further explore the patterns in k, e.g. between environments, and between reservoirs. Were the values in these resservoirs rather similar to other reservoirs or lakes, or rather to rivers?

L403. There is no strong positive correlation between wind speed and FCO2 in your data. Fig 5 shows weak relationships, at best.

L423-425. This sounds like the main result of this study. Make a figure to show and highlight this result, and discuss it in terms of reservoir properties and operation type.

Figure 3. In panels c and d, I would suggest you order the environments in flow direction. That is, upstream first then XR environments, then IR environemnts, then downstream. If it gets too crowded, make two separate panels for high and low water. And the same for pCO2 and FCO2 and k600, i.e. you may end up in 6 panels instead of 2. Together with panels a and b, it would be 8 panels.

Figure 4. When seeing this figure, I wonder how much of this spatial variability is driven by differences in pCO2, and how much by differences in k.

Table 2. What are the values, mean ± standard deviation? How many measurements are behind each of these averages? Could you introduce a column with "n"? The k values are high and resemble rather riverine systems than lakes or reservoirs, I guess

an effect of the fast water flow. The comparison with literature values would be better and more visible in a graph than in a table.

———————————————

---

## Referee Comment (RC2) · Anonymous Referee #2 · 17 Apr 2019

Review of Araujo et al. This manuscript describes the results from a 2-yr study during high and low water seasons on the Belo Monte hydropower complex that consists of two main reservoirs, one of which is defined as a run-of-river and the other as storage. The authors aimed to contrast the impact of these two reservoir types on the $CO_2$ dynamics of the entire complex. Additionally, they contrasted $CO_2$ dynamics across various flooded environments within the complex. The manuscript has some nice data but is predominantly descriptive. Regardless, data in tropical reservoirs is currently necessary and it is interesting to contrast these two types of system. Not to mention the huge dispute over this massive Amazonian project. I have many suggestions for how to improve this manuscript before this paper is ready for publication.

[Figure]

General comments

1. Be careful with the word 'traits' in the title. It implies features that do not vary in time. Is that the focus here? Do you mean ROR vs storage, plus flooded landscapes? That would be okay then. But if that was the case then I did not get the impression enough from your discussion that that was your focus. You need to bring out your main points much more. Try focusing the research questions or objectives more narrowly. This will help you throughout the entire publication.

2. Language overall needs improvement. Too many commas used. Too many sentences that are confusing (many are mentioned in specific comments below).

3. Abstract needs more quantitative results in it

4. Introduction does not discuss the importance of this particular reservoir more.

5. Methods – description of how reservoirs are connected is not clear. In the map figure there appears to be a channel connecting them too. Please improve the description of how the reservoirs interact, including flow directions, which should be on your Figure 2, and individual surface areas.

6. I find section 3.1 of the results very confusing to read and absorb fully. There are a lot of numbers that are perhaps not necessary and very distracting from understanding what you are trying to describe. I would suggest a schematic to help describe the temporal (high vs low water) variability you see that also includes the spatial variability (across environments). You can use weighted markers for the various fluxes and concentrations that correspond to high and low values, if not the real values.

7. Figure 2 – needs arrows for direction of flow.

8. Figure 3 – You can make these 4 plots into just 2 in the following manner: put the white boxplots from (a) and (b) that are pCO2 in the beginning of (c) labeleded 'High water' and 'Low water', and the gray boxplots that are for FCO2 in the beginning of (d) with the same labels. Also, are the environments in c and d labeled in the proper

order – from one are to another? Or does it not work like that because of the reservoir geomorphology? Either way, I would put downstream the dams on the right side since most people read left to right and you naturally think downstream to the right.

9. Figure 4 – you need units listed for the values; direction of flow arrows would be good; and mention in caption that (a) includes 2 years of data while (b) only has one year (and list which years).

10. Figure 5 – you mention these figures in terms of stats but there are no lines on it and no equations or states in the figure caption.

11. The discussion seems like a bunch of descriptive paragraphs thrown together. It is lacking some cohesive red line to follow and it is hard to locate your main points. Perhaps you can start to fix this by using subsections. Looks like you broke it down into the following: Seasonal variability; Vertical heterogeneity; FCO2; Spatial variability; Comparison to other reservoirs; k600; Operation. These are all just descriptions of data in reality. You want to discuss the most interesting findings of your study and then compare them with other studies. Figure out your few most important findings and try to arrange the discussion around those first. You also measured the system right after flooding, which is when emissions should be highest. This needs to be addressed in your conclusions.

Specific comments

Line 16-17 – did you measure clearwater rivers yourself ? if not, then either change or delete this sentence because it makes it sound like you.

Line 41 – You mention that 'inland waters' have an area of '624,000 km2' and cite who with regards to this number? This number is very small compared to the 2.5 – 5 million km2 range that actually exists for all inland waters surface area coverage. I think you mean to cite only rivers surface area with your 0.624 million km2 value so you need to be specific when you say 'inland waters' and you need a specific reference for this

river surface area number. But then you cite the 1.8 – 3.8 Pg values, presumably from Drake et al. 2018 and those values are for all inland waters specifically. If you want to discuss inland waters surface area coverage total then you need to use either Downing et al. 2006, Verpoorter et al. 2014 or Messager et al. 2016 or Feng et al. 2016.

Line 45 – clean up language (e.g., don't need 'water' so many times)

Line 50 – should be: 'to the autochthonous respiration of OM deposited'

Line 54 – should explain more how the stimulation of OM decomposition via those two processes actually effects $CO_2$ – similar to how you did in the first half of the sentence saying higher $CO_2$ uptake

Line 66 – I believe it was actually DelSontro et al. 2010 and not 2016

Line 69-70 – Start a new sentence with 'Newly flooded reservoirs..' and then give examples/references of the few poorly studied reservoirs.

Line 73 – should be 'variability' and not 'variation

Line 73 – give the abbreviation for fluxes here '(FCO2)' that you will use the rest of the paper, and delete 'and its relevance for GHG fluxes'

Line 75 – end this sentence with '..complex in eastern Amazon, a tropical region poised to gain XXX more hydropower projects in the coming decades (REF).' This puts your work into a bigger perspective at the end of your intro

Line 83 – the 1984 study is quite old…. Is there nothing newer?

Line 98-100 – in this sentence give the names of the two reservoirs after you mention them.

Line 101 – give more details about these calculates from Faria et al. 2015

Line 104 – once you have given the XR abbreviation for Xingu Reservoir then use it for the rest of the paper, and do you mean 'as islands' instead of 'in islands'?

Line 107 – 'classified' instead of 'denominated' – and this paragraph should contain the surface area of these reservoirs already

Line 115 – the residence time of the IR reservoir is still ridiculously short (1.57 days). How do you call that a storage reservoir? Still want to know the surface area of these reservoirs already

Line 116 – should give maximum depths of the reservoirs

Line 117 – why did you give the total surface area of the 2 reservoirs together? You should provide values for the two different reservoirs. If this is difficult because of the difference between rainy and dry season then state this but still give approximate values for the individual reservoirs since you are evaluating them separately.

Line 121 – what is the 25.4 km2/MW? Why should I care about this value? Give some explanation behind your reporting of this value (or don't report it).

Line 131-132 – I really do not understand your description of water depth sampling. You classified the sampling sites based on their maximum depths? Where did you measure in the water column? If a site was 10 m deep, did you sample at 3 depths? Did you sample 0.3 m, 6 m, and 9 m? Be more explicit with your description here. Why did you pick 60% of max total depth for sampling?

Line 136 – state that the flooded areas sampled were in both reservoirs if that is the case Line 143 – 'according' not 'accordingly'

Line 148 – what did you collect the headspace air in?

Line 150 – how were the gas samples transferred? Via needle and syringe because the vials were pre-capped, I presume.

Line 154-156 – combine these two sentences into one

Line 158 – if you made measurements from a drifting boat in a river, I presume you drifted quite a bit. Did you consider this drifting distance in your measurements of flux?

This is an important point. How far did you drift? You need more details regarding this sampling approach.

Line 161 – 'calculated' instead of 'done' and delete 'the eq. (1)'

Line 168 – use 'erroneous' instead of 'same sampling site'

Line 171 and eq. 2 – you say that k was based on the flux measurements but I do not see them in equation 2. I guess it is somehow in the partial pressure measurements since some are in the chamber but I think this needs a better explanation. You didn't find k using FCO2, but rather using the concentrations in the chamber? That is how I perceive this equation.

Line 176 – need 'respectively' at the end of the sentence

Line 177 – grammar is poor here

Line 184 – give a bit more detail here about how the gas transfer velocities were not calculated from 2016 data. I am guessing it is because the other loggers did not allow it somehow, but I don't see why you couldn't perform the calculations using concentrations from those loggers too.

Line 187-188 – I do not understand why or how these measurements were made according to the water depth classes. Do you just mean depths? And did you do this at each sampling site?

Line 199 – what does 'assessed separately by season' mean?

Line 208 – you should restate here specifically that you are comparing high and low water from 2017 only.

Line 208 – replace 'presented a significant variation' with 'varied significantly'

Line 221 – it gets confusing a bit when you go between comparing seasons to looking at the whole dataset so be specific when you can. For example, I would add 'From the

overall dataset,' before 'Higher pCO2 was registered..'

Line 223 – I am confused by this sentence and what is respective to each other. Rewrite this one.

Line 228 – Because you only had pCO2 data for 2017 then I guess you couldn't find a correlation between pCO2 and FCO2 in the 2016 data, correct? You need to specific again here and state that the correlation was only for the one method.

Line 232-234 – does it really matter if the two sensors were not cross calibrated in terms of absolute concentrations if it is just the slope of the increase of concentration over time that you need for flux calculations? If it is merely slope then you should be able to estimate and then compare the rates of flux, no?

Line 235-237 – how is it that that the low water season had the highest and lowest FCO2 values but was also homogeneous? This is very confusing.

Line 242-243 – this sentence is kind of just hanging here by itself. Shouldn't it belong somewhere in a paragraph.

Line 244 – I would rename this section a bit more specific to what you are doing: 'pCO2 and FCO2 in ROR versus storage reservoir'

Line 245-246 – if you consider the standard deviation of your measurements then I would say the differences are not so significant between seasons as they then overlap, especially for IR

Line 249 – the difference in IR is much more significant than XR. I would point that out here.

Line 250-252 – I don't understand what you mean here. You did a spatial analysis but lumped all spatially different environments together? I think you mean to say that you compared the total emission from XR to the total emission of IR despite the emitting environment. Is that right?

[Figure]
Line 252-255 – I don't understand how you see no sigficant difference between pCO2 of XR and IR but then suddenly find that XR had pCO2 721 uatm lower. And lower than what? I guess IR. These few sentences are very confusing.

Line 256 – You cannot just present an idea like 'Standing vegetation type in XR flooded areas influenced pCO2' without explaining the data that led you to that conclusion.

Line 264 – use 'especially' instead of 'specifically'

Line 266 – what is a 'gradient pattern downstream'??

Line 272 – again with this 'separately to each season' – I still do not understand what this means. You have to come up with a better way of describing this.

Line 274 – use 'without significant spatial heterogeneity across environments'

Line 275 – use 'k600 strongly correlated with wind. . .' and does this relate to Fig 5b? Should you reference this?

Line 280 – there is not environmental breakdown in the data in Figure 5

Line 287 – so you have water column data? Where is this data?

Line 303 – decrease in what?

Line 344 – what is 'vegetal suppression'? I figured out that it is when you remove vegetation prior to flooding but is this the correct term for this? It sounds very strange.

Line 344-345 – this sentence is too long with poor grammar

Line 354-356 – combine those sentences

Line 356 – how many of the environments? Do you mean all except IR? This is confusing. If it is just IR that is the exception then you need to state it as 'all except IR'

Line 357-358 – negative fluxes can be replaced with 'observed CO2 uptake'

Line 358 – 'light penetration and low suspended sediment'

Line 363-365 – you already spoke about this earlier. Try not to be redundant

Line 370 – need 'which' before 'would'

Line 372-373 – I don't think you need these values here in the discussion.

Line 387 – can you give a site number for the 'site downstream IR'?

Line 391 – I don't think this true and I don't think you need this sentence about a reference for natural FCO2 values

Line 397-398 – do you mean that the downstream sites resembled river channel sites in terms of pCO2 and FCO2 values? Don't use 'traits' to describe this. Traits more refers to features that don't vary.

Line 408-409 – are you saying that the old reservoir you are using for comparison is Tucurui? The grammar here is confusing.

Line 412 – what do you mean by hypolimentical waters? It should be 'hypolimnetic' by the way. But this just means bottom waters with an implication of stratification, but what specifically do you want to express here?

Line 419 – bad grammar in last sentence

---

## Author Comment (AC1) · 24 May 2019

Dear referee 1, We would like to thank you for the detailed comments, they were very constructive. Please find our answers below after each referee comment.

Referee 1: This paper is about the $CO_2$ concentration and emissions from a newly created hydroelectric reservoir complex in the Amazon area. Given that particularly Amazonian reservoirs have been pointed out as high emitters of greenhouse gases, and since emissions typically are higher the first years after flooding, this study is certainly valuable and interesting. In particular since the new reservoir is a run-of-the-river type, which is supposed to result in lower emissions than storage reservoirs. The study

seems to be well-conducted, based on standard methods. However, the presentation severely lacks focus and clarity. I will give in the following a few idea on how the paper can be improved, but I really want to urge the senior authors of this paper to support and help the first author, who is apparently a MSc student and writes his/her first paper (it says in the Acknowledgements). It also takes a thorough revision of English language use and style.

Response: Thank you, we hope to help clarify the role of run-of-the-river dams on CO2 emissions, particularly in the Amazon. We have modified the manuscript based on your suggestions and all authors have carefully reviewed the manuscript for style and clarity.

Referee 1:What makes this study interesting is that it studies the Belo Monte hydroelectric complex, a all-new installation in the Amazon (it's not even up at full capacity yet), the biggest in the Amazon so far, and one of the biggest in the world, and one that was heavily disputed and criticized. This is not mentioned at all in the paper! I could imagine that the story could be built around the case of this new and huge installation. New reservoirs typically have elevated emissions, but here apparently biomass was removed before flooding, at least partially. Is this visible in the data? One of the reservoirs is run-of-the-river, does it really have lower emission than the storage reservoir? These questions could be formulated as hypotheses, addressed with the data (i.e. figures should illustrate data in a way that relates to these hypotheses), and then explicitly answered in the Discussion. This would give the study a much-needed 'read thread'.

Response: We agree that we did not convey the controversy surrounding the Belo Monte hydropower operations in the original manuscript. We have now added a brief discussion of this topic in the Introduction. The Belo Monte hydroelectric complex is the largest hydroelectric in power capacity (11,233 MW) in the Amazon, but not the largest regarding area of reservoir. Among all the new constructed or planned dams in the Amazon, Belo Monte is in fact the most efficient in terms of energy production per

km2 of reservoir (see Faria et al 2015). Despite complete forest removal, plant-derived material still remained in the Intermediate Reservoir (IR). In the Xingu Reservoir (XR), forest removal were done only in some large islands. However, 42 % of the area of this reservoir represents the previous river channel where the riverbed consisted of bedrock and sand. Therefore, lower emissions were expected for the XR in comparison with the IR. Nevertheless, our results show higher $CO_2$ fluxes in the IR only during the low water season. A possible explanation for the lack of difference in $CO_2$ fluxes among the reservoirs at the high water season could be related to the shorter residence time, the primary productivity not necessarily heterogeneous and an influence of algal bloom in the IR. This was perhaps unclear in the original manuscript, which we have improved in the revised manuscript, including the addition of the hypotheses and answers as suggested. Thank you for this useful feedback.

Referee 1: It will take a thorough rewriting of the manuscript before it may become acceptable, but since it seems to be good data from a understudied site of high interest, I think in the end this could become a valuable addition to Biogeosciences.

Response: Thank you for your comments, they've helped to shape a stronger manuscript. We have worked hard to improve the manuscript quality and hope to contribute to the knowledge around tropical run-of-the-river reservoirs.

Detailed comments: Title: the influence of reservoir traits is not explored to any greater depth. Which traits? I'd suggest to change the title accordingly, maybe "CO2 concentrations and emission in the newly constructed Belo Monte hydropower complex in the Xingu River, Amazonia".

Response: We intended to use the word "traits" in the title to describe our comparison of storage and run-of-the-river reservoir types. However, we agree that this title was a bit unclear. We have directed the hypothesis and discussion to better explore and clarify our title.

L41. The inland water area number seems wrong. See Verpoorter et al. 2014 GRL

[Figure]

Response: The inland water area value number is related only to rivers and streams (i.e., not including lakes and wetlands) based on Raymond et al. 2013. We updated that information with the lake surface area estimated by Downing et al. 2006 and Verpoorter et al.2014. In addition, fluxes information was corrected and influx data was added based on Drake et al. 2018.

L42. Only the Raymond study gives a global estimate, the other citations are regional scale.

Response: The other citations were removed from this sentence to adopt only the global estimate of Raymond et al. 2013.

L45-54. There's a lot of detail here that is not addressed by this study and could be removed here, e.g. microbial community structure or priming.

Response: The goal of this paragraph was to address the factors involved in $CO_2$ production. We agree that this section was perhaps too detailed. We have modified this section to be more brief and concise in the revised manuscript.

L70. While emission are typically high, the lifetime emission of a reservoir is probably rather a function of the long-term emission level, and the short initial emission pulse may have less influence.

Response: We have modified this sentence to point out that emissions during the initial years are typically highest and the most uncertain, but that sustained long-term emission rates are likely important with respect to the overall carbon balance of the system over its lifetime.

Study Area: This must mention that the installation is new, and it must describe in how far and where vegetation was removed before flooding, and when the flooding took place.

Response: Thank you, we have made this change.

L106 and 114. The water retention times are very short, even for the storage reservoir it's only 1.5 days. Are these numbers correct? If so, these reservoirs, given their size, must be characterized by quite strong water flow, and thus the gas exchange velocity is probably hardly related to wind speed, but rather to water speed.

Response: Our residence time (RT) calculations were based on Faria et al. (2015) and environmental impact study (EIA) published by Norte Energia (Eletrobrás, 2009b). Since RT was underestimated we made new calculations based on Water Agency of Brazil (ANA) discharge data at the Altamira station. The corrected RT was 20.2 and 3.4 days for IR and XR, respectively, using the average discharge historic series to whole year. We have corrected the RT in the manuscript.

L112. 97% of the capacity are at the Belo Monte dam, so the ROR dam only produces 3% of the energy even though it contains one third of the number of turbines?

Response: The difference among both dams is not only in size and turbine number. The turbine model also differs between dams, which influences the generating power. The main power house is equipped with 18 turbines Francis type with active unit power of 611.11 MW. As complementary powerhouse has 6 Bulb type turbines that are considerably less potent with active unit power of only 38.85 MW.

L116. Where is the hypolimnion typically starting? Did you do any depth profiles of T and/or DO? If so, please show and report! If not, please cite a study that states that the thermocline is typically at >20 m.

Response: During our samplings the water column had a well-mixed pattern without variation in DO in most of the reservoir's area. The hypolimnion was only apparent in the IR, close to the dam, where DO decreased drastically at approximately 50 m from a total depth of 58 m. However as observed on Faria et al (2015) its formation is not expected on Belo Monte Reservoirs. Therefore this sentence was altered and hypolimnion information was withdrawn. We have now included depth profiles of variables such as DO, temperature, etc. in the supplementary material.

Section 2.2. It would be more easy to understand if you first described your sampling campaigns, and then tell about any gaps.

Response: We have made this change

L144. Why was 60% of water depth chosen? Seems arbitrary. Also, it would be good to know the actual depth at these sites. A raw data table should be submitted alongside with the paper.

Response: 60% depth was chosen as a mid-depth sampling point to compare surface and bottom waters. In deeper sites the three depths (surface, 60% and near-bottom) were sampled due the variation in water velocity. Our goal was to sample depths with different organic and inorganic matter due water flow transport. We have added depth information to Table 1.

L150. How good was the evacuation? In my experience, it's very difficult to get a good vaccum, but probably 10% or more atmosphere will remain, which may dilute or contaminate your samples. Was this checked?

Response: We are confident in our sample storage methods, which our team has extensive experience with. A vacuum pump was used to create a vacuum, which was confirmed since the volume of gas pulled from the syringe into the vial was similar to the vial volume without the needing to manually depress the syringe's plunger. We have not added these details to the manuscript, as transferring gas to vials is a common method.

L154. Start this paragraph with saying "Diffusive $CO_2$ emission was measured with floating chambers". Also, please give the dimensions, shape and type (transparent / opaque) of the chamber.

Response: We have made this change. Two different chambers were used to measure gaseous $CO_2$ emissions during whole sampling campaigns. Both chambers were round and opaque covered with reflexive aluminum tape, differing only on the dimensions (high water chamber: capacity - 7.7 L; area - 0.08 m2; height – 11.7 cm/ low water chambers: capacity - 6 L; area - 0.07 m2; height – 10.5 cm).

L161. I guess you mean logging frequency, not time.

Response: Exactly, we have modified this.

L168. Atmospheric pCO2 of 380 ppm seems like an outdated value, or are these your own measurements in air?

Response: We agree that this is an outdated value and we have re-checked our database and changed the text. For the new version, measurements were discarded when the R2 of the linear relation between pCO2 and time ($\delta$pCO2/$\delta$t) were lower than 0.90 (R2 < 0.90) or in cases where we measured negative FCO2 when the surface water pCO2 was higher than the atmospheric pCO2 based on measurement done at the same site. However, this happened only two times and could be attributed to some source of CO2 contamination when placing the chamber into the water. Thus starting with a higher pCO2 than the water.

L184. This sentence seems unnecessary

Response: We have removed it.

L191. A station is stationary. You probably mean a handheld meter or device?

Response: Updated as suggested.

2.5. Statistics. I did not know Permanova, so this should be better explained. Is it a parametric method? Because it is stated that the data did not follow normal distribution. However, later in this paragraph, you mention some data were normally distributed and used t-test; this is confusing. Also, in the entire paper, report the actual p values, not just if p is lower of higher than 0.05.

Response: Agreed, the method was superficially mentioned in the manuscript. PER-MANOVA is a multivariate variance analysis to compare variability between and within

groups using permutation to obtain p-value. Due the different hypothesis tested the data set had to be adjusted and consequently altered the data distribution. In the case of T-Test, sites located on "outside reservoirs" and "downstream of the dams" were not considered and also season. Related to p value, we have now reported all the p-values accordingly to the real value obtained from the statistical test. PERMANOVA analysis was better detailed in the methods section as suggested. We have removed T-Test analysis since it is related to a descriptive result.

Results. In general, this section describes many findings and patterns, but it does so in a quite unstructured way, and is therefore difficult to follow. I really think it would help this paper if only the results were presented that are relevant to the hypotheses or research questions. Also, the language describing the patterns should be improved. For example, it needs to explained what numbers are given (e.g. L208, is this the mean $\pm$ standard deviation, or something else?), and comparisons between two groups describe a difference and not a variation (L208). Also, increase and decrease (e.g. L245 and L249) refer to a change over time and thus some form of time series data, while this study has data for two discrete sampling occasions, and thus can only speak about differences. It should also always be very clear what exactly was compared. For example, in L213, it was unclear what was tested here, the variability in pCO2 within and environment, or between environments?

Response: Thank you, your comments were very constructive especially to this section. In line L208 and throughout the whole text we presented values as mean $\pm$ 1 standard deviation and indeed we were using the term "variation" when we were meaning "difference". Some of the comparisons were unclear due the writing style and language, but in the line L213 tested the pCO2 variability between environments. We have restructured this section as suggested and paid extra attention to the language.

Again concerning statistics, it is unclear to me how a comparison between two groups can render a R2 value, but maybe that's a part of the PERMANOVA, and should in that case be better explained in the Methods.

Response: Our statistics description did not detail PERMANOVA properly in the previous version of the manuscript, as so this test became unclear to the reader. PERMANOVA analysis tests similarity using a Euclidian distance index through permutations. The R2 value is generated by permutations. As mentioned above, PERMANOVA analysis was better explained in the methods section as suggested.

L215. Here you speak about spatial variability, but do you mean differences of means between different environemnts, or the variability of measurements within one environment type?

Response: Here the test is to evaluate if the different environments (reservoirs, downstream the dam and outside the reservoir) presented different fluxes in each season. Temporal trends sometimes may mask some spatial patterns that only become visible when seasons are treated separately. Therefore, here we refer to a PERMANOVA test similar to the one mentioned on L213, comparing pCO2 between environments.

L219. "Outside reservoir areas" is not a very illuminating term. Could choose another name?

Response: We agreed and replaced it to "unaffected river channel".

L224. 281 $\mu$atm at 60% depth, how much is that in meters? And how can deep water be undersaturated in oxygen? Typically it is oversaturated. Or was this above a macrophyte bed?

Response: The total depth of this site is 7.5 m (Table 1), the sampling depth was 4 m. This sentence describes pCO2, not dissolved oxygen. The value of pCO2 equal to 281 uatm was observed in the undisturbed river channel with significant current and no macrophyte bed. This value corresponds to undersaturated pCO2 with respect to the atmosphere, and therefore likely oxygen levels above atmospheric saturation, indicating net primary production. We are not sure what you mean by the question of how deep water can be undersaturated in oxygen. It is quite common for dissolved

oxygen levels in river water to be below atmospheric in the case of net heterotrophy.

L231. Here it says the the data from the two seasons were pooled, but L237-241, the seasonal data are discussed separately. This is confusing.

Response: Thank you for this comment. Our FCO2 data is related to a time period of two years, comprehending three seasons (2016 high water, 2017 high water and 2017 low water)(L124). The data pooled are from the same season, both high water, sampled with the same equipment and they were not statistically similar (L228). High and low water were measured with different equipment due technical issues and treated separately (L154 and L158).

L246. The seasonal difference in IR was very small, certainly not a "pronounced difference". Interestingly, FCO2 was very different between seasons in spite of similar pCO2, which indicates a strong variability in k. Was this the case?

Response: Very true, related to k, no statistically significant variation was observed between seasons (L273 – L275). We have removed the word "pronounced" and updated this line as suggested.

L250. What kind of spatial analyses? Comparison of the means for different environments?

Response: PERMANOVA was used to compare simultaneously the variation of FCO2, pCO2 and k600 between both reservoirs. This analysis did not generate difference of means, but the dissimilarity within versus and between groups through distance measures.

L251. "evaluated together", is this warranted? Were these two groups similar?

Response: Good point. Yes, our results indicate that they are similar. We have checked it by changing river channel category. Only flooded areas represented the reservoir emission, nevertheless, same results were reached. In case of dissimilar data a different classifying reveals overlapped patterns, which was not the case of river channel

and flooded areas.

L256. "Pasture" is a new and undefined category.

Response: Upland forest and pasture were the main land cover in the areas flooded by the reservoirs as described in the description of the study area and measured sites (Table 1). They are not a new category and may be classified as a flooded area subgroup.

L262. What's the measure of variability? It seems that in this study, you mostly compared means, but if you want to address the variability, you maybe want to look at relative standard deviations, interquartile ranges or something similar. If you want to stick to comparing means between environments, please formulate this explicitly in the text.

Response: There was some confusion with the term from our part. Our analysis describes difference by a distance matrix that calculates the similarity within and between groups, not variation. We assume that the poor statistics section may have complicated much of the reading. We have rewritten that section and replaced "variation" by "difference" in the whole manuscript.

L263. Varied significantly between what?

Response: The FCO2 differed significantly between XR and IR reservoirs during the low water season. This sentence has been modified accordingly in the revised manuscript.

L266. The 90 km downstream site is so far away it's not even on the map. I wonder in how far it is relevant to this study at all, or could safely be omitted.

Response: Thank you for the observation. That was a mistake in the writing; the 90 km site is downstream of the Pimental dam (P20 site – Fig. 2), not Belo Monte. This site is relevant because of its location downstream of the Volta Grande do Xingu (Xingu Great Bend) region and a few kilometers upstream to where the Belo Monte dam discharge

back into the original river. This information was properly corrected.

L270-273. Go straight to the results instead of first describing what was not done.

Response: We have made this change.

L275. The relationship between k600 and wind speed is very weak. At any wind speed, k can vary with a factor of 2-4. This is quite often the case, and maybe even expected in such system where water moves fast, and thus water turbulence is quite independent of wind speed.

Response: Thank you for this comment. We agree, particularly considering the residence time of the Belo Monte system. Since there was no significant k variability, the water turbulence must be the major factor driving CO2 diffusion.

All in all, the Results give many comparisons, What about making matrix tables where you can give test statistics for each comparison?

Response: Thank you that was a great suggestion. We have added such a table.

3.3. Did you ever measure depth profiles? Would be very interesting to show these data, to asses if really the turbine intake is in the epilimnion, and to assess the potential outgassing through turbine passage.

Response: Yes, depth profiles were made for temperature, pH, O2 and conductivity. CO2 was measured at the bottom, 60% of site depth and at the surface (0.3 m) during high water campaigns. Probably, there was no hypolimnion close to the Pimental dam due the water column uniform oxygenation. Nevertheless, its intake is on the bottom, where even with high O2, the pCO2 is higher than on surface. In Belo Monte dam pCO2 follows the same pattern, although the O2 decrease drastically at approximately 50 m (as mentioned above). As so, Belo Monte intake is in the O2 rich zone. We have created a supplementary material file and the depth profiles were added to it.

L292. This is not one of your results.

Response: Removed.

L296. The Discussion should start with your most important finding, not with citing other studies.

Response: Updated as suggested.

L303. This seems to be an important finding. Could you make a figure that illustrates this finding, to make it visible and convincing?

Response: Thank you for the suggestion. However, we chose to keep this comparison on table 2.

L309-326. This discussion is very hypothetical and not much related to your data.

Response: Thank you for this comment. We have deleted this paragraph.

L327. Not really. In your own data, there is an example of differences in k producing very different emission fluxes in spite of similar pCO2 (see my comment above).

Response: We have added the clarifying statement "…although we did observe some specific examples of differences in k producing different emission fluxes even when pCO2 was similar"

L328-334. This may be the main message of this paper. It would be good if you produced a Figure that illustrates this finding.

Response: Thank you, we agree and have added such a figure.

L340-341. The Methods need to describe explicitly which areas were flooded with intact biomass, or after biomass harvesting.

Response: We agreed. This information was already in table 1, however it was not as explicit as it should. The text was updated with the information of suppression area for each reservoir.

L350. Could you actually observe increased water clarity in your data / samplings? If

not, this discussion is not helpful to explain your data.

Response: No. Our turbidity data was not reliable due to poor calibration, as such we removed it from the paper. Updated as suggested.

L355. pCO2 were only lower during low water compared to high water in the downstream and dam categories. For flooded and river channel, they were similar (Fig.3). So it is not warranted to speak about a "drastic decrease".

Response: We have deleted the word drastic from the text. The statistical test showed difference among seasons and to environment categories, which is corroborated by the lower pCO2 averages during low water both to flooded areas and river channel (as shown in table 2). This was corrected in the new version of the manuscript.

L375-383. Could the difference between Belo Monte and Petit Saut be explained by different water intake depths? Do you have water profile data?

Response: Very good question. As mentioned above we have DO depth profiles that show signs of anoxia near the bottom in the IR in the site closest to the dam. The hypolimnion is located under the input zone of Belo Monte turbines, and according to Kemenes et al. (2011 and 2016) and Abril et al. (2005), the hypolimnetic waters may increase the downstream emissions. Therefore, possibly the intake depth plays an important role in the different downstream emissions between Petit Saut and Belo Monte complex. This information was updated in the manuscript.

L391. It seems not warranted to assume that any site or time point should serve as a "reference" for river pCO2, since it varies in time and space.

Response: We have deleted this sentence.

L395. What is meant by "turbine activity"?

Response: With the term "Turbine activity" we meant periods when turbines are either working or stopped for maintenance. Since the sampling campaigns occurred during

the installation, few of the operational turbines were actually working. This was perhaps reflected by downstream fluxes during the low water period, which resembled areas without turbine influence.

L398-406. I think you could further explore the patterns in k, e.g. between environments, and between reservoirs. Were the values in these reservoirs rather similar to other reservoirs or lakes, or rather to rivers?

Response: Thank you, very good point. The XR k600 values ($22.99 \pm 8.00$ and $22.89 \pm 21.40$ cm h-1 on high and low water, respectively) were in the range of the Furnas reservoir ($19.58 \pm 2.5$ cm h-1) located in Amazonia (Paranaíba et al., 2018). To IR ($7.13 \pm 1.59$ and $60.80 \pm 18.02$ cm h-1 on high and low water, respectively) the high water k600 was similar to Javaes river ($8.22 \pm 3.80$). Therefore, the average XR k600 was rather similar to reservoir, otherwise average IR k600 rather to river. Updated as suggested.

L403. There is no strong positive correlation between wind speed and FCO2 in your data. Fig 5 shows weak relationships, at best.

Response: Thank you for the highlight, we have updated the text as suggested.

L423-425. This sounds like the main result of this study. Make a figure to show and highlight this result, and discuss it in terms of reservoir properties and operation type.

Response: We have included new categories on figure 3 according reservoirs that show this difference and the suggested points were included in the discussion.

Figure 3. In panels c and d, I would suggest you order the environments in flow direction. That is, upstream first then XR environments, then IR environemnts, then downstream. If it gets too crowded, make two separate panels for high and low water. And the same for pCO2 and FCO2 and k600, i.e. you may end up in 6 panels instead of 2. Together with panels a and b, it would be 8 panels.

Response: We have reordered and changed the categories. We added the categories in the following order: "unaffected river upstream", "XR", "IR", "downstream of the dams" and "unaffected river downstream". Panels were also separated by season.

Figure 4. When seeing this figure, I wonder how much of this spatial variability is driven by differences in pCO2, and how much by differences in k.

Response: To make the spatial variability more visible we have added one more panel related to k600 to figure 3.

Table 2. What are the values, mean ± standard deviation? How many measurements are behind each of these averages? Could you introduce a column with "n"? The k values are high and resemble rather riverine systems than lakes or reservoirs, I guess an effect of the fast water flow. The comparison with literature values would be better and more visible in a graph than in a table.

Response: Whole values are related to averages ± standard deviation, with the exception of Sawakuchi et al. 2017. During high water FCO2 was measured three times (L157), and during low water two FCO2 measurements were made simultaneously (L 160) and headspace was sampled on triplicates (L144). It is probable that the turbulence in both reservoirs is mostly related to water flow. We chose to keep literature values in table 2 and the new column was added as suggested.

Please also note the supplement to this comment:
https://www.biogeosciences-discuss.net/bg-2019-53/bg-2019-53-AC1-supplement.pdf
* * *
[Figure]

**Fig. 1.** Fig.2

[Figure]

**Fig. 2.** Fig.3

[Figure]
Interactive
comment

[Figure]

(a)

[Figure]

(b)

**Fig. 3.** Fig.4

[Figure]

**Fig. 4.** Comment L303 suggestion: pCO2 of IR reservoir between seasons. HW is related to high water and LW to low water.

**Supplement:**

**Supplementary Material**

**The influence of reservoir traits on carbon dioxide emissions in the Belo Monte hydropower complex, Xingu River, Amazon – Brazil**

Kleiton R. Araújo[1*], Henrique O. Sawakuchi[2-3], Dailson J. Bertassoli Jr.[4], André O. Sawakuchi[1,4], Karina D. da Silva[1,5], Thiago V. Bernardi[1,5], Nicholas D. Ward[6-7], Tatiana S. Pereira[1,5].

[1]Programa de Pós Graduação em Biodiversidade e Conservação, Universidade Federal do Pará, Altamira, 68372 – 040, Brazil,

[2]Centro de Energia Nuclear na Agricultura, Universidade de São Paulo, Piracicaba, Brazil,

[3]Department of Ecology and Environmental Science, Umeå University, Umeå, SE-901 87, Sweden,

[4]Instituto de Geociências, Universidade de São Paulo, São Paulo, Brazil,

[5]Faculdade de Ciências Biológicas, Universidade Federal do Pará, Altamira, 68372 – 040, Brazil,

[6]Marine Sciences Laboratory, Pacific Northwest National Laboratory, Sequim, Washington, 98382, USA,

[7]School of Oceanography, University of Washington, Seattle, Washington, 98195-5351, USA.

*Correspondence to: Kleiton R. Araújo (kleitonrabelo@rocketmail.com)

Keywords: run-of-the-river reservoir; greenhouse gas emission; tropical river damming.

**1. Sampling details**

Based on visualization in Google Earth we estimate that the maximum distance drifted may be approximately 1 km for measurements in the river channel up and downstream of the reservoirs. In sheltered areas located in bays and over islands with standing trees, where the water flow was very low, drifting was very short and caused by wind. An estimate of the drifting distance in the natural river channel and in the main channel of the Xingu Reservoir was obtained by using the average water velocity measured by the National Water Agency of Brazil at the Altamira station. We separated the historical values into before and after 2016, when the dams was completed. Therefore, representing estimates of water velocity in the natural river (between 2005 and 2016), and in the Xingu Reservoir main channel (after 2016). The average water velocities at Altamira are 0.74 and 0.24 m s-1 for before and after the dam, respectively. Assuming that there is no resistance of the boat with the water or air, drifting speed is similar to the water velocity. The total time of deployment was up to 30 minutes for the three consecutive measurements. Based on these we found that in the main channel of the Xingu Reservoir the drifting distance would be 432 m, and 1332 m for the natural river channel up and downstream the reservoirs.

**2. Depth profiles**

During high water of 2016 and 2017 we registered depth profiles for physical-chemical variables through whole sampling area. We measured water column pH, depth, dissolved oxygen (DO), conductivity and temperature using a multiparameter probe (EXO2®, YSI). The depth profiles were registered until near bottom depth (approximately 80 % of total depth) to avoid sediment interaction. Total depths are listed on table 1 of the manuscript. Depth profile tables are presented by year and flow order, respectively.

Table S1 – P1 site depth profile from 2016 campaign.

| pH | ODO mg/L | Temp °C | Cond µS/cm | Depth m |
|---|---|---|---|---|
| 6.65 | 6.43 | 29.208 | 23.5 | 0.166 |
| 6.61 | 6.43 | 29.208 | 23.5 | 0.212 |
| 6.53 | 6.43 | 29.208 | 23.5 | 0.265 |
| 6.5 | 6.43 | 29.201 | 23.5 | 0.283 |
| 6.5 | 6.43 | 29.195 | 23.5 | 0.401 |

| pH | ODO mg/L | Temp °C | Cond µS/cm | Depth m |
|---|---|---|---|---|
| 6.55 | 6.42 | 29.195 | 23.5 | 0.678 |
| 6.6 | 6.41 | 29.184 | 23.5 | 1.046 |
| 6.56 | 6.41 | 29.191 | 23.5 | 1.416 |
| 6.61 | 6.4 | 29.209 | 23.5 | 1.787 |
| 6.62 | 6.4 | 29.207 | 23.5 | 2.171 |
| 6.63 | 6.4 | 29.217 | 23.5 | 2.614 |
| 6.63 | 6.4 | 29.198 | 23.5 | 2.831 |
| 6.63 | 6.4 | 29.191 | 23.5 | 3.039 |
| 6.62 | 6.39 | 29.2 | 23.5 | 4.061 |
| 6.64 | 6.4 | 29.213 | 23.5 | 4.165 |
| 6.64 | 6.39 | 29.205 | 23.5 | 4.359 |
| 6.64 | 6.38 | 29.211 | 23.5 | 4.723 |
| 6.63 | 6.38 | 29.203 | 23.5 | 5.145 |
| 6.65 | 6.37 | 29.205 | 23.5 | 5.632 |
| 6.69 | 6.37 | 29.209 | 23.5 | 6.042 |
| 6.65 | 6.37 | 29.203 | 23.5 | 6.453 |
| 6.63 | 6.37 | 29.202 | 23.5 | 7.633 |
| 6.66 | 6.37 | 29.2 | 23.5 | 8.006 |
| 6.62 | 6.37 | 29.198 | 23.5 | 9.085 |
| 6.61 | 6.36 | 29.198 | 23.5 | 9.492 |
| 6.61 | 6.36 | 29.198 | 23.5 | 10.793 |
| 6.62 | 6.35 | 29.197 | 23.5 | 11.27 |
| 6.6 | 6.35 | 29.198 | 23.5 | 12.4 |
| 6.56 | 6.34 | 29.2 | 23.5 | 12.796 |
| 6.59 | 6.34 | 29.198 | 23.5 | 13.738 |
| 6.56 | 6.33 | 29.2 | 23.5 | 14.053 |
| 6.56 | 6.33 | 29.199 | 23.5 | 15.073 |
| 6.58 | 6.32 | 29.198 | 23.5 | 15.3 |

59

60 Table S2 – P6 site depth profile from 2016 campaign.

| pH | ODO mg/L | Temp °C | Cond µS/cm | Depth m |
|---|---|---|---|---|
| 6.09 | 4.99 | 29.761 | 21.9 | 0.397 |
| 6.04 | 3.95 | 29.568 | 22.3 | 0.759 |
| 5.97 | 3.72 | 29.495 | 22.5 | 1.075 |
| 5.97 | 3.86 | 29.422 | 22.3 | 1.373 |
| 6.01 | 3.88 | 29.426 | 22.4 | 1.494 |
| 6.05 | 3.82 | 29.438 | 22.4 | 1.453 |
| 6.11 | 3.77 | 29.449 | 22.4 | 1.383 |
| 6.16 | 3.8 | 29.448 | 22.4 | 1.377 |
| 6.15 | 3.86 | 29.435 | 22.3 | 1.382 |
| 6.15 | 3.87 | 29.44 | 22.4 | 1.378 |
| 6.14 | 3.86 | 29.445 | 22.4 | 1.359 |

| pH | | | | |
|---|---|---|---|---|
| 6.14 | 3.82 | 29.456 | 22.5 | 1.344 |
| 6.12 | 3.75 | 29.462 | 22.5 | 1.343 |

61

62    Table S3 – P7 site depth profile from 2016 campaign.

| pH | ODO mg/L | Temp °C | Cond µS/cm | Depth m |
|---|---|---|---|---|
| 6.72 | 7.04 | 29.555 | 21.6 | 0.102 |
| 6.67 | 7.03 | 29.538 | 21.6 | 0.223 |
| 6.64 | 7.02 | 29.518 | 21.6 | 1.222 |
| 6.68 | 7.02 | 29.508 | 21.6 | 1.279 |
| 6.74 | 7 | 29.473 | 21.6 | 1.411 |
| 6.73 | 7 | 29.48 | 21.6 | 1.602 |
| 6.75 | 6.99 | 29.467 | 21.6 | 1.803 |
| 6.8 | 6.99 | 29.468 | 21.6 | 1.965 |
| 6.83 | 6.99 | 29.475 | 21.6 | 2.167 |
| 6.82 | 6.99 | 29.48 | 21.6 | 2.351 |
| 6.8 | 6.99 | 29.475 | 21.6 | 2.538 |
| 6.79 | 6.98 | 29.466 | 21.6 | 2.767 |
| 6.79 | 6.99 | 29.487 | 21.6 | 3.018 |
| 6.83 | 6.98 | 29.482 | 21.6 | 3.283 |
| 6.85 | 6.99 | 29.492 | 21.6 | 3.546 |
| 6.85 | 6.98 | 29.483 | 21.6 | 3.825 |
| 6.86 | 6.98 | 29.482 | 21.6 | 4.102 |
| 6.82 | 6.98 | 29.478 | 21.6 | 4.379 |
| 6.8 | 6.97 | 29.476 | 21.6 | 4.656 |
| 6.84 | 6.97 | 29.471 | 21.6 | 4.966 |
| 6.84 | 6.97 | 29.481 | 21.6 | 5.29 |
| 6.82 | 6.97 | 29.484 | 21.6 | 5.6 |
| 6.83 | 6.97 | 29.482 | 21.6 | 5.958 |
| 6.82 | 6.96 | 29.477 | 21.6 | 7.127 |
| 6.78 | 6.96 | 29.48 | 21.6 | 7.444 |
| 6.77 | 6.95 | 29.479 | 21.6 | 8.549 |
| 6.77 | 6.95 | 29.477 | 21.6 | 8.913 |
| 6.8 | 6.94 | 29.479 | 21.6 | 9.249 |
| 6.82 | 6.94 | 29.476 | 21.6 | 10.418 |
| 6.82 | 6.94 | 29.476 | 21.6 | 10.691 |
| 6.79 | 6.93 | 29.477 | 21.6 | 10.904 |

63

64    Table S4 – P8 site depth profile from 2016 campaign.

| pH | ODO mg/L | Temp °C | Cond µS/cm | Depth m |
|---|---|---|---|---|
| 6.48 | 6.84 | 30.409 | 24.3 | 0.143 |

| pH | ODO mg/L | Temp °C | Cond µS/cm | Depth m |
|---|---|---|---|---|
| 6.48 | 6.8 | 30.408 | 24.3 | 0.286 |
| 6.51 | 6.82 | 30.415 | 24.3 | 0.296 |
| 6.54 | 6.8 | 30.415 | 24.3 | 0.327 |
| 6.57 | 6.79 | 30.418 | 24.3 | 0.345 |
| 6.58 | 6.81 | 30.422 | 24.3 | 0.341 |
| 6.59 | 6.83 | 30.416 | 24.3 | 0.331 |
| 6.6 | 6.81 | 30.417 | 24.3 | 0.33 |
| 6.61 | 6.79 | 30.417 | 24.3 | 0.32 |
| 6.63 | 6.79 | 30.416 | 24.3 | 0.319 |
| 6.66 | 6.8 | 30.413 | 24.3 | 0.32 |
| 6.7 | 6.81 | 30.42 | 24.3 | 0.324 |
| 6.7 | 6.82 | 30.419 | 24.3 | 0.318 |
| 6.7 | 6.8 | 30.416 | 24.3 | 0.308 |
| 6.7 | 6.8 | 30.424 | 24.3 | 0.303 |

65

66    Table S5 – P12 site depth profile from 2016 campaign.

| pH | ODO mg/L | Temp °C | Cond µS/cm | Depth m |
|---|---|---|---|---|
| 6.33 | 6.13 | 29.803 | 22.5 | 0.184 |
| 6.3 | 6.05 | 29.83 | 22.5 | 0.44 |
| 6.3 | 6.06 | 29.988 | 22.5 | 0.622 |
| 6.32 | 5.96 | 29.594 | 22.4 | 1.542 |
| 6.34 | 5.89 | 29.498 | 22.4 | 1.814 |
| 6.35 | 5.83 | 29.454 | 22.3 | 2.055 |
| 6.36 | 5.78 | 29.421 | 22.3 | 2.419 |
| 6.37 | 5.76 | 29.406 | 22.3 | 2.819 |
| 6.36 | 5.74 | 29.401 | 22.3 | 3.071 |
| 6.36 | 5.72 | 29.395 | 22.3 | 3.312 |
| 6.37 | 5.7 | 29.391 | 22.3 | 3.56 |
| 6.39 | 5.68 | 29.386 | 22.3 | 3.897 |
| 6.38 | 5.64 | 29.367 | 22.3 | 4.283 |
| 6.39 | 5.62 | 29.357 | 22.4 | 4.738 |
| 6.39 | 5.61 | 29.353 | 22.3 | 5.037 |
| 6.39 | 5.61 | 29.355 | 22.3 | 5.104 |
| 6.4 | 5.62 | 29.372 | 22.3 | 4.896 |
| 6.43 | 5.63 | 29.377 | 22.4 | 4.701 |
| 6.45 | 5.64 | 29.383 | 22.4 | 4.531 |

67

68    Table S6 – P15 site depth profile from 2016 campaign.

| pH | ODO mg/L | Temp °C | Cond µS/cm | Depth m |
|---|---|---|---|---|

| pH | ODO mg/L | Temp °C | Cond µS/cm | Depth m |
|---|---|---|---|---|
| 6.44 | 6.47 | 29.516 | 22.6 | 0.969 |
| 6.39 | 6.44 | 29.529 | 22.6 | 2.615 |
| 6.41 | 6.43 | 29.526 | 22.6 | 3.157 |
| 6.45 | 6.42 | 29.527 | 22.5 | 4.641 |
| 6.47 | 6.41 | 29.527 | 22.6 | 6.059 |
| 6.49 | 6.4 | 29.527 | 22.6 | 7.644 |
| 6.68 | 6.39 | 29.528 | 22.6 | 8.447 |
| 6.66 | 6.36 | 29.528 | 22.6 | 8.707 |
| 6.65 | 6.37 | 29.528 | 22.6 | 8.926 |

69

70 Table S7 – P11 site depth profile from 2016 campaign.

| pH | ODO mg/L | Temp °C | Cond µS/cm | Depth m |
|---|---|---|---|---|
| 6.3 | 6.58 | 29.444 | 17.5 | 1.43 |
| 6.3 | 6.45 | 29.451 | 22.6 | 3.677 |
| 6.34 | 6.42 | 29.458 | 22.6 | 7.086 |
| 6.39 | 6.39 | 29.455 | 22.6 | 9.411 |
| 6.46 | 6.36 | 29.433 | 22.7 | 12.642 |
| 6.53 | 6.35 | 29.423 | 22.7 | 15.238 |
| 6.54 | 6.33 | 29.422 | 22.7 | 19.197 |
| 6.55 | 6.27 | 29.413 | 23 | 20.501 |
| 6.6 | 6.26 | 29.411 | 22.8 | 20.59 |
| 6.64 | 6.25 | 29.411 | 22.8 | 20.563 |
| 6.66 | 6.3 | 29.412 | 22.7 | 20.475 |

71

72 Table S8 – P14 site depth profile from 2016 campaign.

| pH | ODO mg/L | Temp °C | Cond µS/cm | Depth m |
|---|---|---|---|---|
| 6.59 | 7.19 | 29.67 | 23.1 | 1.374 |
| 6.57 | 7.2 | 29.666 | 23 | 2.37 |
| 6.54 | 7.22 | 29.664 | 23 | 2.877 |
| 6.51 | 7.22 | 29.664 | 23 | 4.137 |
| 6.52 | 7.22 | 29.666 | 23 | 4.648 |
| 6.54 | 7.23 | 29.66 | 22.9 | 5.757 |
| 6.63 | 7.23 | 29.664 | 22.9 | 6.036 |
| 6.64 | 7.22 | 29.665 | 22.9 | 7.077 |
| 6.63 | 7.24 | 29.659 | 22.9 | 7.436 |
| 6.59 | 7.24 | 29.662 | 22.9 | 7.655 |

73

74 Table S9 – P19 site depth profile from 2016 campaign.

| pH | ODO mg/L | Temp °C | Cond µS/cm | Depth m |
|---|---|---|---|---|
| 6.63 | 6.78 | 29.647 | 19 | 0.128 |
| 6.56 | 6.47 | 29.665 | 26.3 | 0.286 |
| 6.57 | 6.42 | 29.685 | 26.4 | 0.456 |
| 6.59 | 6.46 | 29.676 | 26.3 | 0.525 |
| 6.59 | 6.1 | 29.657 | 26.5 | 0.579 |
| 6.6 | 6.74 | 29.71 | 26.2 | 0.658 |
| 6.65 | 6.87 | 29.724 | 25.7 | 0.776 |
| 6.68 | 6.34 | 29.712 | 26.2 | 0.926 |
| 6.63 | 5.74 | 29.716 | 27 | 1.076 |
| 6.57 | 5.31 | 29.681 | 27.4 | 1.235 |
| 6.55 | 5.19 | 29.627 | 27.5 | 1.447 |
| 6.54 | 5.02 | 29.625 | 27.6 | 1.722 |
| 6.53 | 4.94 | 29.644 | 27.7 | 2.024 |
| 6.53 | 4.9 | 29.63 | 27.7 | 2.37 |
| 6.55 | 4.84 | 29.608 | 27.7 | 2.721 |
| 6.56 | 4.93 | 29.592 | 27.6 | 3.096 |
| 6.55 | 4.79 | 29.591 | 27.7 | 4.253 |
| 6.52 | 4.13 | 29.601 | 28.1 | 4.597 |
| 6.5 | 3.8 | 29.524 | 28.5 | 4.82 |
| 6.5 | 4.68 | 29.572 | 28 | 5.157 |
| 6.52 | 4.5 | 29.566 | 27.9 | 5.641 |
| 6.51 | 3.74 | 29.509 | 28.3 | 6.091 |
| 6.48 | 3.64 | 29.481 | 28.6 | 6.569 |
| 6.48 | 3.45 | 29.476 | 28.7 | 7.002 |
| 6.49 | 3.91 | 29.515 | 28.3 | 8.164 |
| 6.47 | 3.61 | 29.482 | 28.6 | 8.637 |

75

76     Table S10 – P21 site depth profile from 2016 campaign.

| pH | ODO mg/L | Temp °C | Cond µS/cm | Depth m |
|---|---|---|---|---|
| 6.88 | 8.01 | 29.843 | 24.6 | 0.578 |
| 6.87 | 8 | 29.843 | 24.6 | 0.776 |
| 6.86 | 8 | 29.844 | 24.6 | 1.014 |
| 6.87 | 8 | 29.844 | 24.6 | 1.271 |
| 6.89 | 8 | 29.844 | 24.6 | 1.546 |
| 6.92 | 8 | 29.844 | 24.6 | 1.784 |
| 6.9 | 8 | 29.844 | 24.6 | 2.014 |
| 6.9 | 8 | 29.844 | 24.6 | 2.281 |
| 6.91 | 8 | 29.844 | 24.6 | 2.587 |
| 6.94 | 7.99 | 29.843 | 24.6 | 2.903 |
| 6.94 | 7.99 | 29.844 | 24.6 | 3.125 |
| 6.93 | 7.99 | 29.845 | 24.6 | 3.312 |

| 6.92 | 7.99 | 29.845 | 24.6 | 3.483 |
|------|------|--------|------|-------|
| 6.92 | 7.99 | 29.846 | 24.6 | 3.734 |
| 6.92 | 7.98 | 29.846 | 24.6 | 4.033 |
| 6.91 | 7.99 | 29.845 | 24.6 | 4.269 |
| 6.9 | 7.99 | 29.844 | 24.6 | 4.381 |
| 6.91 | 7.98 | 29.845 | 24.6 | 5.535 |
| 6.93 | 7.98 | 29.846 | 24.6 | 5.868 |
| 6.93 | 7.97 | 29.847 | 24.6 | 7.08 |
| 6.96 | 7.97 | 29.848 | 24.6 | 7.244 |
| 6.95 | 7.96 | 29.848 | 24.6 | 8.487 |
| 6.92 | 7.96 | 29.847 | 24.6 | 8.363 |
| 6.9 | 7.96 | 29.847 | 24.6 | 8.267 |

77

78    Table S11 – P4 site depth profile from 2017 campaign.

| pH | ODO mg/L | Temp °C | Cond µS/cm | Depth m |
|------|----------|---------|------------|---------|
| 7.22 | 7.15 | 29.552 | 33.6 | 0.313 |
| 7.21 | 7.15 | 29.553 | 33.5 | 0.422 |
| 7.21 | 7.14 | 29.553 | 33.5 | 0.528 |
| 7.2 | 7.14 | 29.553 | 33.5 | 0.622 |
| 7.19 | 7.14 | 29.553 | 33.5 | 0.724 |
| 7.19 | 7.13 | 29.553 | 33.5 | 0.83 |
| 7.18 | 7.13 | 29.552 | 33.5 | 0.944 |
| 7.18 | 7.13 | 29.552 | 33.5 | 1.118 |
| 7.18 | 7.13 | 29.552 | 33.5 | 1.275 |
| 7.17 | 7.13 | 29.553 | 33.5 | 1.42 |
| 7.17 | 7.13 | 29.552 | 33.5 | 1.573 |
| 7.16 | 7.12 | 29.552 | 33.4 | 1.721 |
| 7.16 | 7.12 | 29.552 | 33.4 | 1.868 |
| 7.15 | 7.12 | 29.553 | 33.5 | 2.017 |
| 7.15 | 7.12 | 29.553 | 33.5 | 2.15 |
| 7.15 | 7.12 | 29.552 | 33.5 | 2.284 |
| 7.14 | 7.12 | 29.552 | 33.4 | 2.402 |
| 7.14 | 7.12 | 29.552 | 33.4 | 2.483 |
| 7.13 | 7.12 | 29.552 | 33.4 | 2.505 |
| 7.13 | 7.12 | 29.552 | 33.4 | 2.53 |
| 7.12 | 7.12 | 29.552 | 33.4 | 2.557 |
| 7.12 | 7.12 | 29.552 | 33.4 | 2.602 |
| 7.12 | 7.12 | 29.552 | 33.4 | 2.668 |
| 7.12 | 7.11 | 29.552 | 33.5 | 2.728 |
| 7.11 | 7.11 | 29.552 | 33.5 | 2.813 |
| 7.11 | 7.11 | 29.552 | 33.4 | 2.922 |
| 7.11 | 7.11 | 29.552 | 33.4 | 3.06 |

| 7.11 | 7.11 | 29.552 | 33.4 | 3.204 |
|---|---|---|---|---|
| 7.11 | 7.11 | 29.552 | 33.4 | 3.318 |
| 7.11 | 7.11 | 29.552 | 33.4 | 3.406 |
| 7.1 | 7.11 | 29.552 | 33.4 | 3.503 |
| 7.1 | 7.11 | 29.552 | 33.4 | 3.624 |
| 7.1 | 7.11 | 29.552 | 33.4 | 3.768 |
| 7.09 | 7.11 | 29.552 | 33.4 | 3.922 |
| 7.09 | 7.1 | 29.553 | 33.4 | 4.054 |
| 7.09 | 7.1 | 29.553 | 33.4 | 4.185 |
| 7.09 | 7.1 | 29.553 | 33.4 | 4.347 |
| 7.09 | 7.1 | 29.553 | 33.4 | 4.5 |
| 7.09 | 7.1 | 29.553 | 33.4 | 4.628 |
| 7.09 | 7.1 | 29.553 | 33.4 | 4.719 |
| 7.09 | 7.1 | 29.553 | 33.4 | 4.807 |
| 7.08 | 7.1 | 29.552 | 33.4 | 4.908 |
| 7.08 | 7.1 | 29.552 | 33.4 | 5.027 |
| 7.07 | 7.1 | 29.553 | 33.4 | 5.148 |
| 7.05 | 7.1 | 29.553 | 33.4 | 5.259 |
| 7.04 | 7.1 | 29.553 | 33.4 | 5.369 |
| 7.02 | 7.1 | 29.553 | 33.4 | 5.501 |
| 7.01 | 7.09 | 29.553 | 33.4 | 5.666 |
| 6.99 | 7.09 | 29.553 | 33.4 | 5.828 |
| 6.98 | 7.09 | 29.553 | 33.4 | 5.987 |
| 6.96 | 7.09 | 29.553 | 33.4 | 6.132 |
| 6.93 | 7.09 | 29.553 | 33.4 | 6.274 |
| 6.91 | 7.09 | 29.553 | 33.4 | 6.414 |
| 6.88 | 7.09 | 29.553 | 33.4 | 6.563 |
| 6.86 | 7.09 | 29.553 | 33.4 | 6.687 |
| 6.85 | 7.09 | 29.553 | 33.4 | 6.77 |
| 6.84 | 7.09 | 29.553 | 33.4 | 6.874 |
| 6.83 | 7.09 | 29.553 | 33.4 | 6.978 |
| 6.82 | 7.09 | 29.552 | 33.4 | 7.068 |
| 6.81 | 7.08 | 29.552 | 33.3 | 7.162 |
| 6.69 | 7.08 | 29.552 | 33.3 | 7.291 |

79

80    Table S13 – P5 site depth profile from 2017 campaign.

| pH | ODO mg/L | Temp °C | Cond µS/cm | Depth m |
|---|---|---|---|---|
| 7.27 | 7.29 | 29.934 | 32.2 | 0.157 |
| 7.27 | 7.28 | 29.932 | 32.2 | 0.246 |
| 7.26 | 7.28 | 29.932 | 32.2 | 0.357 |
| 7.25 | 7.28 | 29.932 | 32.2 | 0.464 |
| 7.24 | 7.28 | 29.93 | 32.2 | 0.553 |

| | | | | |
|---|---|---|---|---|
| 7.23 | 7.28 | 29.93 | 32.3 | 0.642 |
| 7.22 | 7.28 | 29.93 | 32.3 | 0.726 |
| 7.21 | 7.28 | 29.927 | 32.3 | 0.8 |
| 7.2 | 7.28 | 29.924 | 32.3 | 0.88 |
| 7.2 | 7.28 | 29.923 | 32.3 | 0.962 |
| 7.2 | 7.28 | 29.923 | 32.3 | 1.031 |
| 7.19 | 7.28 | 29.924 | 32.4 | 1.115 |
| 7.19 | 7.28 | 29.925 | 32.4 | 1.206 |
| 7.19 | 7.28 | 29.924 | 32.5 | 1.299 |
| 7.19 | 7.28 | 29.924 | 32.5 | 1.404 |
| 7.19 | 7.27 | 29.925 | 32.5 | 1.519 |
| 7.19 | 7.27 | 29.926 | 32.6 | 1.635 |
| 7.19 | 7.27 | 29.928 | 32.6 | 1.751 |
| 7.19 | 7.27 | 29.928 | 32.7 | 1.882 |
| 7.19 | 7.27 | 29.929 | 32.7 | 2.011 |
| 7.18 | 7.27 | 29.929 | 32.7 | 2.141 |
| 7.18 | 7.27 | 29.928 | 32.8 | 2.283 |
| 7.18 | 7.27 | 29.926 | 32.8 | 2.429 |
| 7.18 | 7.27 | 29.925 | 32.9 | 2.586 |
| 7.18 | 7.27 | 29.925 | 33 | 2.745 |
| 7.18 | 7.27 | 29.924 | 33 | 2.904 |
| 7.18 | 7.26 | 29.922 | 33.1 | 3.053 |
| 7.18 | 7.26 | 29.92 | 33.1 | 3.196 |
| 7.18 | 7.26 | 29.919 | 33.1 | 3.33 |
| 7.18 | 7.26 | 29.918 | 33.1 | 3.452 |
| 7.18 | 7.26 | 29.917 | 33.2 | 3.559 |
| 7.18 | 7.26 | 29.916 | 33.2 | 3.661 |
| 7.18 | 7.26 | 29.915 | 33.2 | 3.766 |
| 7.18 | 7.26 | 29.914 | 33.3 | 3.876 |
| 7.18 | 7.26 | 29.912 | 33.3 | 3.988 |
| 7.18 | 7.25 | 29.912 | 33.3 | 4.107 |
| 7.17 | 7.25 | 29.911 | 33.3 | 4.233 |
| 7.17 | 7.25 | 29.911 | 33.3 | 4.361 |
| 7.17 | 7.25 | 29.911 | 33.4 | 4.482 |
| 7.16 | 7.25 | 29.911 | 33.4 | 4.603 |
| 7.16 | 7.25 | 29.911 | 33.4 | 4.714 |
| 7.16 | 7.25 | 29.912 | 33.4 | 4.829 |
| 7.16 | 7.25 | 29.912 | 33.4 | 4.941 |
| 7.15 | 7.25 | 29.912 | 33.4 | 5.061 |
| 7.14 | 7.24 | 29.912 | 33.4 | 5.184 |
| 7.14 | 7.24 | 29.912 | 33.4 | 5.316 |
| 7.14 | 7.24 | 29.911 | 33.5 | 5.442 |
| 7.14 | 7.24 | 29.911 | 33.5 | 5.557 |
| 7.14 | 7.24 | 29.911 | 33.5 | 5.66 |
| 7.14 | 7.24 | 29.911 | 33.5 | 5.761 |

| 7.14 | 7.24 | 29.911 | 33.5 | 5.836 |
| 7.14 | 7.24 | 29.91 | 33.5 | 5.879 |
| 7.14 | 7.24 | 29.909 | 33.5 | 5.918 |
| 7.14 | 7.24 | 29.909 | 33.5 | 5.966 |
| 7.14 | 7.24 | 29.909 | 33.5 | 6.043 |
| 7.14 | 7.23 | 29.909 | 33.5 | 6.128 |
| 7.13 | 7.23 | 29.909 | 33.5 | 6.226 |
| 7.13 | 7.23 | 29.909 | 33.6 | 6.346 |
| 7.12 | 7.23 | 29.909 | 33.6 | 7.481 |
| 7.11 | 7.23 | 29.909 | 33.6 | 7.514 |
| 7.1 | 7.23 | 29.909 | 33.6 | 7.556 |
| 7.09 | 7.23 | 29.908 | 33.6 | 7.558 |
| 7.09 | 7.23 | 29.908 | 33.6 | 7.585 |
| 7.09 | 7.23 | 29.908 | 33.6 | 7.656 |
| 7.08 | 7.22 | 29.908 | 33.6 | 7.731 |
| 7.08 | 7.22 | 29.908 | 33.6 | 7.809 |
| 7.07 | 7.22 | 29.908 | 33.6 | 7.9 |
| 7.07 | 7.22 | 29.909 | 33.6 | 7.998 |

81

82   Table S14 – P6 site depth profile from 2017 campaign.

| pH | ODO mg/L | Temp °C | Cond µS/cm | Depth m |
| --- | --- | --- | --- | --- |
| 6.93 | 7.32 | 30.83 | 33.8 | 0.264 |
| 6.94 | 7.32 | 30.811 | 33.7 | 0.316 |
| 6.94 | 7.31 | 30.764 | 33.7 | 0.388 |
| 6.93 | 7.23 | 30.686 | 33.7 | 0.476 |
| 6.91 | 7.14 | 30.578 | 33.7 | 0.541 |
| 6.88 | 7.07 | 30.505 | 33.7 | 0.576 |
| 6.85 | 7.01 | 30.421 | 33.7 | 0.609 |
| 6.67 | 6.6 | 30.323 | 33.7 | 0.661 |
| 6.63 | 6.42 | 30.21 | 33.8 | 0.728 |
| 6.6 | 6.29 | 30.104 | 33.9 | 0.797 |
| 6.58 | 5.97 | 30.052 | 34 | 0.835 |
| 6.57 | 5.94 | 30.022 | 34.1 | 0.874 |
| 6.56 | 5.91 | 30.011 | 34.1 | 0.915 |
| 6.55 | 5.86 | 30.007 | 34.2 | 0.957 |
| 6.53 | 5.53 | 29.972 | 34.2 | 1.021 |
| 6.38 | 5.25 | 29.9 | 34.4 | 1.075 |
| 6.33 | 4.53 | 29.805 | 34.6 | 1.148 |
| 6.32 | 4.35 | 29.748 | 34.8 | 1.223 |
| 6.32 | 4.26 | 29.714 | 34.9 | 1.291 |
| 6.32 | 4.16 | 29.683 | 35 | 1.383 |
| 6.33 | 4.43 | 29.634 | 34.8 | 1.485 |

| | | | | |
|---|---|---|---|---|
| 6.35 | 4.6 | 29.583 | 34.7 | 1.578 |
| 6.37 | 4.71 | 29.556 | 34.6 | 1.639 |
| 6.4 | 4.81 | 29.547 | 34.6 | 1.689 |
| 6.43 | 4.95 | 29.483 | 34.3 | 1.761 |
| 6.44 | 5.02 | 29.389 | 34.1 | 1.855 |
| 6.44 | 4.74 | 29.275 | 34.1 | 1.938 |
| 6.43 | 4.56 | 29.209 | 34.3 | 2.007 |
| 6.42 | 4.46 | 29.165 | 34.4 | 2.084 |
| 6.4 | 4.13 | 29.141 | 34.5 | 2.163 |
| 6.39 | 3.92 | 29.107 | 34.6 | 2.264 |
| 6.37 | 3.78 | 29.073 | 34.8 | 2.363 |
| 6.35 | 3.71 | 29.035 | 34.9 | 2.42 |
| 6.33 | 3.59 | 29.023 | 35 | 2.477 |
| 6.32 | 3.5 | 29.012 | 35.1 | 2.537 |
| 6.31 | 3.4 | 29 | 35.2 | 2.591 |
| 6.3 | 2.96 | 28.98 | 35.2 | 2.675 |
| 6.29 | 2.92 | 28.96 | 35.3 | 2.757 |
| 6.28 | 2.9 | 28.954 | 35.4 | 2.808 |
| 6.27 | 2.86 | 28.944 | 35.4 | 2.855 |
| 6.27 | 2.82 | 28.938 | 35.5 | 2.903 |
| 6.27 | 2.78 | 28.933 | 35.6 | 2.965 |
| 6.26 | 2.72 | 28.929 | 35.6 | 3.05 |
| 6.26 | 2.65 | 28.924 | 35.6 | 3.105 |
| 6.26 | 2.59 | 28.92 | 35.6 | 3.153 |
| 6.25 | 2.5 | 28.91 | 35.7 | 3.21 |
| 6.24 | 2.41 | 28.909 | 35.7 | 3.227 |
| 6.24 | 2.4 | 28.915 | 35.7 | 3.248 |
| 6.24 | 2.46 | 28.919 | 35.6 | 3.266 |

83

84    Table S15 – P7 site depth profile from 2017 campaign.

| pH | ODO mg/L | Temp °C | Cond µS/cm | Depth m |
|---|---|---|---|---|
| 6.91 | 6.95 | 29.479 | 32.6 | 0.316 |
| 6.91 | 6.94 | 29.479 | 32.6 | 0.456 |
| 6.91 | 6.94 | 29.487 | 32.6 | 0.598 |
| 6.91 | 6.95 | 29.522 | 32.5 | 0.735 |
| 6.92 | 6.96 | 29.525 | 32.5 | 0.871 |
| 6.92 | 6.95 | 29.499 | 32.5 | 1.004 |
| 6.92 | 6.94 | 29.478 | 32.5 | 1.132 |
| 6.92 | 6.95 | 29.541 | 32.5 | 1.246 |
| 6.93 | 7 | 29.666 | 32.4 | 1.346 |
| 6.95 | 7.07 | 29.749 | 32.3 | 1.454 |
| 6.98 | 7.12 | 29.778 | 32.3 | 1.578 |

| | | | | |
|---|---|---|---|---|
| 7 | 7.13 | 29.742 | 32.4 | 1.723 |
| 7.01 | 7.13 | 29.73 | 32.4 | 1.855 |
| 7.01 | 7.12 | 29.682 | 32.5 | 1.978 |
| 7 | 7.08 | 29.608 | 32.5 | 2.106 |
| 6.98 | 7.03 | 29.538 | 32.6 | 2.245 |
| 6.96 | 6.98 | 29.519 | 32.7 | 3.119 |
| 6.95 | 6.97 | 29.515 | 32.7 | 3.214 |
| 6.94 | 6.96 | 29.513 | 32.7 | 3.308 |
| 6.94 | 6.95 | 29.513 | 32.7 | 3.413 |
| 6.94 | 6.95 | 29.516 | 32.8 | 3.504 |
| 6.94 | 6.95 | 29.514 | 32.8 | 3.588 |
| 6.94 | 6.95 | 29.509 | 32.9 | 3.669 |
| 6.93 | 6.94 | 29.501 | 32.9 | 3.745 |
| 6.93 | 6.94 | 29.494 | 32.9 | 3.819 |
| 6.93 | 6.93 | 29.485 | 33 | 3.962 |
| 6.93 | 6.92 | 29.484 | 33 | 4.109 |
| 6.93 | 6.92 | 29.489 | 33.1 | 4.253 |
| 6.93 | 6.92 | 29.496 | 33.1 | 4.395 |
| 6.93 | 6.93 | 29.504 | 33.1 | 4.533 |
| 6.93 | 6.93 | 29.506 | 33.2 | 4.686 |
| 6.93 | 6.93 | 29.501 | 33.2 | 4.842 |
| 6.92 | 6.93 | 29.503 | 33.2 | 5.791 |
| 6.92 | 6.93 | 29.511 | 33.2 | 5.896 |
| 6.92 | 6.93 | 29.513 | 33.3 | 5.905 |
| 6.92 | 6.93 | 29.509 | 33.3 | 5.923 |
| 6.92 | 6.93 | 29.499 | 33.3 | 5.962 |
| 6.92 | 6.92 | 29.488 | 33.4 | 6.016 |
| 6.91 | 6.91 | 29.479 | 33.4 | 6.102 |
| 6.91 | 6.9 | 29.477 | 33.4 | 7.035 |
| 6.9 | 6.9 | 29.471 | 33.5 | 7.176 |
| 6.9 | 6.89 | 29.463 | 33.5 | 7.263 |
| 6.9 | 6.88 | 29.46 | 33.5 | 7.343 |
| 6.9 | 6.88 | 29.464 | 33.5 | 7.417 |
| 6.9 | 6.89 | 29.474 | 33.5 | 7.492 |
| 6.9 | 6.89 | 29.484 | 33.6 | 7.572 |
| 6.91 | 6.9 | 29.496 | 33.6 | 7.656 |
| 6.91 | 6.91 | 29.517 | 33.6 | 7.766 |
| 6.92 | 6.93 | 29.541 | 33.6 | 8.693 |
| 6.92 | 6.94 | 29.554 | 33.6 | 8.874 |
| 6.93 | 6.94 | 29.553 | 33.6 | 9.041 |
| 6.93 | 6.94 | 29.545 | 33.6 | 9.229 |
| 6.93 | 6.94 | 29.541 | 33.6 | 9.409 |
| 6.93 | 6.93 | 29.538 | 33.7 | 10.304 |
| 6.93 | 6.93 | 29.537 | 33.7 | 10.524 |
| 6.92 | 6.93 | 29.538 | 33.7 | 10.648 |

| 6.92 | 6.93 | 29.534 | 33.7 | 10.753 |
|------|------|--------|------|--------|
| 6.92 | 6.92 | 29.524 | 33.7 | 10.858 |
| 6.91 | 6.91 | 29.513 | 33.7 | 10.971 |
| 6.91 | 6.9 | 29.499 | 33.7 | 11.079 |
| 6.9 | 6.89 | 29.482 | 33.8 | 11.899 |
| 6.89 | 6.87 | 29.471 | 33.8 | 12.004 |
| 6.89 | 6.86 | 29.458 | 33.8 | 12.133 |
| 6.88 | 6.85 | 29.455 | 33.8 | 12.223 |
| 6.88 | 6.85 | 29.462 | 33.8 | 12.324 |
| 6.88 | 6.86 | 29.47 | 33.8 | 12.403 |
| 6.88 | 6.86 | 29.474 | 33.8 | 12.483 |
| 6.88 | 6.86 | 29.477 | 33.8 | 12.566 |
| 6.88 | 6.86 | 29.483 | 33.8 | 12.654 |
| 6.88 | 6.87 | 29.493 | 33.8 | 12.81 |
| 6.88 | 6.87 | 29.504 | 33.8 | 12.963 |
| 6.88 | 6.88 | 29.51 | 33.8 | 13.114 |
| 6.88 | 6.88 | 29.511 | 33.8 | 14.017 |
| 6.89 | 6.88 | 29.514 | 33.8 | 14.163 |
| 6.89 | 6.88 | 29.519 | 33.8 | 14.229 |
| 6.89 | 6.89 | 29.519 | 33.8 | 14.346 |
| 6.89 | 6.88 | 29.513 | 33.8 | 14.502 |
| 6.89 | 6.88 | 29.504 | 33.8 | 15.504 |
| 6.88 | 6.87 | 29.492 | 33.9 | 15.612 |
| 6.88 | 6.86 | 29.484 | 33.9 | 15.668 |
| 6.87 | 6.85 | 29.481 | 33.9 | 15.74 |
| 6.87 | 6.85 | 29.486 | 33.9 | 15.821 |
| 6.87 | 6.85 | 29.483 | 33.9 | 15.908 |
| 6.86 | 6.85 | 29.476 | 33.9 | 16.01 |
| 6.86 | 6.84 | 29.474 | 33.9 | 16.103 |
| 6.85 | 6.84 | 29.477 | 33.9 | 16.965 |
| 6.85 | 6.84 | 29.475 | 33.9 | 17.125 |
| 6.85 | 6.84 | 29.472 | 33.9 | 17.24 |
| 6.84 | 6.84 | 29.473 | 33.9 | 17.346 |
| 6.84 | 6.83 | 29.473 | 33.9 | 17.48 |
| 6.83 | 6.84 | 29.478 | 33.9 | 18.401 |
| 6.83 | 6.84 | 29.486 | 33.9 | 18.702 |
| 6.82 | 6.84 | 29.492 | 33.9 | 18.833 |

85

86    Table S16 – P8 site depth profile from 2017 campaign.

| pH | ODO mg/L | Temp °C | Cond µS/cm | Depth m |
|------|------|--------|------|--------|
| 6.92 | 7.4 | 30.755 | 34.6 | 0.137 |
| 6.91 | 7.35 | 30.679 | 34.5 | 0.244 |

| | | | | |
|---|---|---|---|---|
| 6.9 | 7.31 | 30.598 | 34.4 | 0.358 |
| 6.89 | 7.28 | 30.551 | 34.3 | 0.464 |
| 6.88 | 7.26 | 30.54 | 34.2 | 0.569 |
| 6.88 | 7.25 | 30.538 | 34.1 | 0.649 |
| 6.88 | 7.24 | 30.536 | 34 | 0.71 |
| 6.88 | 7.23 | 30.532 | 33.9 | 0.757 |
| 6.88 | 7.23 | 30.53 | 33.8 | 0.778 |
| 6.88 | 7.23 | 30.532 | 33.7 | 0.766 |
| 6.88 | 7.22 | 30.534 | 33.7 | 0.738 |
| 6.88 | 7.23 | 30.535 | 33.6 | 0.708 |

87

88   Table S17 – P9 site depth profile from 2017 campaign.

| pH | ODO mg/L | Temp °C | Cond µS/cm | Depth m |
|---|---|---|---|---|
| 6.74 | 0.46 | 29.982 | 36.4 | 0.114 |
| 6.72 | 0.45 | 29.871 | 36.1 | 0.154 |
| 6.7 | 0.45 | 29.801 | 36 | 0.185 |
| 6.69 | 0.45 | 29.783 | 35.9 | 0.188 |
| 6.68 | 0.45 | 29.829 | 35.8 | 0.183 |
| 6.68 | 0.44 | 29.923 | 35.8 | 0.178 |
| 6.69 | 0.44 | 30.029 | 35.7 | 0.197 |
| 6.69 | 0.43 | 30.046 | 35.6 | 0.226 |
| 6.69 | 0.43 | 29.971 | 35.5 | 0.266 |
| 6.69 | 0.43 | 29.883 | 35.5 | 0.315 |
| 6.69 | 0.43 | 29.858 | 35.4 | 0.37 |
| 6.68 | 0.43 | 29.882 | 35.4 | 0.436 |
| 6.68 | 0.43 | 29.909 | 35.3 | 0.514 |
| 6.68 | 0.43 | 29.928 | 35.3 | 0.587 |
| 6.68 | 0.42 | 29.909 | 35.2 | 0.653 |
| 6.67 | 0.42 | 29.813 | 35.1 | 0.723 |
| 6.67 | 0.42 | 29.684 | 35 | 0.794 |
| 6.66 | 0.42 | 29.579 | 34.9 | 0.864 |
| 6.65 | 0.43 | 29.513 | 34.8 | 0.938 |
| 6.64 | 0.43 | 29.477 | 34.8 | 1.004 |
| 6.64 | 0.42 | 29.462 | 34.7 | 1.067 |
| 6.64 | 0.42 | 29.463 | 34.7 | 1.124 |
| 6.63 | 0.42 | 29.465 | 34.6 | 1.163 |
| 6.63 | 0.42 | 29.468 | 34.6 | 1.186 |
| 6.62 | 0.42 | 29.469 | 34.5 | 1.205 |
| 6.62 | 0.42 | 29.468 | 34.5 | 1.227 |
| 6.62 | 0.42 | 29.463 | 34.5 | 1.246 |
| 6.62 | 0.42 | 29.455 | 34.5 | 1.262 |
| 6.62 | 0.41 | 29.449 | 34.4 | 1.272 |

| | | | | |
|------|------|--------|------|-------|
| 6.63 | 0.41 | 29.448 | 34.4 | 1.285 |
| 6.63 | 0.41 | 29.449 | 34.4 | 1.313 |
| 6.63 | 0.41 | 29.448 | 34.4 | 1.348 |
| 6.63 | 0.41 | 29.447 | 34.4 | 1.372 |
| 6.63 | 0.41 | 29.447 | 34.3 | 1.396 |
| 6.62 | 0.41 | 29.446 | 34.3 | 1.423 |
| 6.62 | 0.41 | 29.443 | 34.3 | 1.453 |
| 6.62 | 0.41 | 29.442 | 34.3 | 1.49 |
| 6.62 | 0.4  | 29.445 | 34.3 | 1.532 |
| 6.62 | 0.4  | 29.445 | 34.3 | 1.58 |
| 6.62 | 0.4  | 29.439 | 34.3 | 1.641 |
| 6.62 | 0.4  | 29.43  | 34.3 | 1.708 |
| 6.62 | 0.4  | 29.422 | 34.3 | 1.777 |
| 6.62 | 0.4  | 29.415 | 34.3 | 1.858 |
| 6.62 | 0.4  | 29.41  | 34.2 | 1.938 |
| 6.62 | 0.4  | 29.404 | 34.2 | 2.025 |
| 6.61 | 0.4  | 29.395 | 34.2 | 2.116 |
| 6.61 | 0.4  | 29.389 | 34.2 | 2.209 |
| 6.61 | 0.4  | 29.388 | 34.2 | 2.302 |
| 6.61 | 0.4  | 29.386 | 34.2 | 2.398 |
| 6.61 | 0.4  | 29.384 | 34.2 | 2.494 |
| 6.61 | 0.4  | 29.379 | 34.2 | 2.594 |
| 6.61 | 0.4  | 29.376 | 34.2 | 2.69 |
| 6.61 | 0.4  | 29.371 | 34.2 | 2.79 |
| 6.6  | 0.4  | 29.365 | 34.2 | 2.899 |
| 6.6  | 0.4  | 29.364 | 34.2 | 3.008 |
| 6.6  | 0.41 | 29.365 | 34.2 | 3.126 |
| 6.6  | 0.41 | 29.367 | 34.2 | 3.239 |
| 6.6  | 0.41 | 29.368 | 34.1 | 3.354 |
| 6.6  | 0.41 | 29.368 | 34.1 | 3.472 |
| 6.6  | 0.41 | 29.368 | 34.1 | 3.588 |
| 6.6  | 0.41 | 29.369 | 34.1 | 3.698 |
| 6.6  | 0.41 | 29.369 | 34.1 | 3.797 |
| 6.6  | 0.41 | 29.368 | 34.1 | 3.887 |
| 6.6  | 0.41 | 29.368 | 34.1 | 3.979 |
| 6.6  | 0.41 | 29.367 | 34.1 | 4.082 |
| 6.6  | 0.41 | 29.366 | 34.1 | 4.183 |
| 6.6  | 0.41 | 29.365 | 34.1 | 4.286 |
| 6.6  | 0.41 | 29.364 | 34.1 | 4.393 |
| 6.59 | 0.41 | 29.363 | 34.1 | 4.51 |
| 6.59 | 0.41 | 29.362 | 34.1 | 4.62 |
| 6.59 | 0.41 | 29.362 | 34.2 | 4.731 |
| 6.59 | 0.41 | 29.361 | 34.2 | 4.827 |
| 6.59 | 0.41 | 29.36  | 34.2 | 4.906 |
| 6.59 | 0.41 | 29.36  | 34.2 | 4.973 |

| | | | | |
|---|---|---|---|---|
| 6.59 | 0.41 | 29.359 | 34.2 | 5.026 |
| 6.59 | 0.41 | 29.357 | 34.2 | 5.056 |
| 6.59 | 0.41 | 29.357 | 34.2 | 5.076 |
| 6.59 | 0.41 | 29.356 | 34.2 | 5.096 |
| 6.59 | 0.41 | 29.356 | 34.2 | 5.12 |
| 6.59 | 0.4 | 29.355 | 34.2 | 5.161 |
| 6.59 | 0.41 | 29.347 | 34.2 | 5.258 |
| 6.58 | 0.41 | 29.339 | 34.2 | 5.366 |
| 6.58 | 0.41 | 29.333 | 34.2 | 5.49 |
| 6.57 | 0.41 | 29.331 | 34.2 | 5.625 |
| 6.57 | 0.41 | 29.329 | 34.2 | 5.774 |
| 6.57 | 0.41 | 29.329 | 34.2 | 5.923 |

89

90    Table S18 – P10 site depth profile from 2017 campaign.

| pH | ODO mg/L | Temp °C | Cond µS/cm | Depth m |
|---|---|---|---|---|
| 6.74 | 6.68 | 30.726 | 36.9 | 0.164 |
| 6.74 | 6.69 | 30.717 | 36.9 | 0.245 |
| 6.74 | 6.7 | 30.699 | 36.8 | 0.34 |
| 6.74 | 6.71 | 30.672 | 36.8 | 0.422 |
| 6.75 | 6.72 | 30.653 | 36.8 | 0.497 |
| 6.76 | 6.73 | 30.631 | 36.8 | 0.577 |
| 6.76 | 6.74 | 30.606 | 36.9 | 0.667 |
| 6.77 | 6.75 | 30.591 | 36.9 | 0.761 |
| 6.78 | 6.76 | 30.568 | 36.8 | 0.853 |
| 6.79 | 6.77 | 30.541 | 36.8 | 0.942 |
| 6.8 | 6.78 | 30.507 | 36.8 | 1.029 |
| 6.8 | 6.79 | 30.474 | 36.7 | 1.107 |
| 6.81 | 6.78 | 30.441 | 36.6 | 1.19 |
| 6.8 | 6.73 | 30.408 | 36.6 | 1.281 |
| 6.79 | 6.29 | 30.348 | 36.7 | 1.374 |
| 6.78 | 6.1 | 30.256 | 36.8 | 1.471 |
| 6.66 | 5.69 | 30.148 | 36.7 | 1.565 |
| 6.6 | 5.6 | 30.066 | 36.6 | 1.663 |
| 6.58 | 5.54 | 30.014 | 36.6 | 1.76 |
| 6.57 | 5.45 | 30 | 36.5 | 1.858 |
| 6.56 | 5.38 | 29.988 | 36.5 | 1.959 |
| 6.56 | 5.32 | 29.976 | 36.5 | 2.059 |
| 6.55 | 5.29 | 29.968 | 36.5 | 2.159 |
| 6.54 | 5.27 | 29.963 | 36.5 | 2.256 |
| 6.54 | 5.26 | 29.957 | 36.4 | 2.352 |
| 6.53 | 5.25 | 29.952 | 36.4 | 2.451 |
| 6.53 | 5.24 | 29.945 | 36.4 | 2.548 |

| pH | ODO mg/L | Temp °C | Cond µS/cm | Depth m |
|---|---|---|---|---|
| 6.53 | 5.23 | 29.936 | 36.4 | 2.647 |
| 6.52 | 5.18 | 29.921 | 36.4 | 2.757 |
| 6.51 | 5.11 | 29.905 | 36.4 | 2.872 |
| 6.5 | 5.05 | 29.893 | 36.4 | 3.006 |
| 6.48 | 5 | 29.886 | 36.5 | 3.146 |
| 6.47 | 4.94 | 29.879 | 36.5 | 3.289 |
| 6.47 | 4.86 | 29.873 | 36.5 | 3.417 |
| 6.46 | 4.79 | 29.871 | 36.6 | 3.528 |
| 6.45 | 4.74 | 29.871 | 36.6 | 3.621 |
| 6.44 | 4.71 | 29.871 | 36.6 | 3.69 |
| 6.44 | 4.68 | 29.87 | 36.6 | 3.736 |
| 6.44 | 4.66 | 29.866 | 36.6 | 3.758 |
| 6.43 | 4.64 | 29.863 | 36.7 | 3.765 |
| 6.43 | 4.62 | 29.863 | 36.7 | 3.775 |
| 6.43 | 4.58 | 29.862 | 36.7 | 3.778 |
| 6.42 | 4.55 | 29.859 | 36.7 | 3.793 |
| 6.41 | 4.51 | 29.853 | 36.8 | 3.818 |
| 6.41 | 4.45 | 29.844 | 36.8 | 3.861 |
| 6.39 | 4.07 | 29.829 | 37 | 3.924 |
| 6.26 | 3.82 | 29.802 | 37.3 | 4.831 |

91

92   Table S19 – P11 site depth profile from 2017 campaign.

| pH | ODO mg/L | Temp °C | Cond µS/cm | Depth m |
|---|---|---|---|---|
| 6.79 | 6.81 | 29.955 | 32.7 | 1.063 |
| 6.8 | 6.77 | 29.945 | 32.7 | 1.248 |
| 6.81 | 6.76 | 29.945 | 32.8 | 1.334 |
| 6.83 | 6.75 | 29.947 | 32.9 | 1.462 |
| 6.84 | 6.75 | 29.944 | 33 | 1.592 |
| 6.85 | 6.74 | 29.938 | 33.1 | 1.699 |
| 6.86 | 6.74 | 29.939 | 33.1 | 1.787 |
| 6.88 | 6.74 | 29.945 | 33.2 | 1.862 |
| 6.88 | 6.74 | 29.949 | 33.2 | 2.718 |
| 6.89 | 6.74 | 29.95 | 33.3 | 2.816 |
| 6.89 | 6.74 | 29.949 | 33.3 | 2.865 |
| 6.9 | 6.73 | 29.95 | 33.4 | 2.933 |
| 6.9 | 6.73 | 29.95 | 33.4 | 3.019 |
| 6.9 | 6.74 | 29.952 | 33.5 | 3.096 |
| 6.9 | 6.74 | 29.955 | 33.5 | 3.178 |
| 6.91 | 6.73 | 29.957 | 33.5 | 3.256 |
| 6.91 | 6.73 | 29.955 | 33.6 | 4.092 |
| 6.91 | 6.73 | 29.955 | 33.6 | 4.161 |
| 6.91 | 6.73 | 29.95 | 33.6 | 4.252 |

| 6.91 | 6.73 | 29.942 | 33.6 | 4.341 |
| 6.91 | 6.72 | 29.934 | 33.6 | 4.423 |
| 6.91 | 6.72 | 29.935 | 33.7 | 4.508 |
| 6.91 | 6.72 | 29.938 | 33.7 | 4.592 |
| 6.92 | 6.72 | 29.939 | 33.7 | 5.364 |
| 6.92 | 6.72 | 29.936 | 33.7 | 5.467 |
| 6.92 | 6.72 | 29.932 | 33.7 | 5.517 |
| 6.92 | 6.72 | 29.932 | 33.8 | 5.578 |
| 6.92 | 6.72 | 29.934 | 33.8 | 5.669 |
| 6.91 | 6.73 | 29.936 | 33.8 | 5.769 |
| 6.91 | 6.73 | 29.935 | 33.8 | 5.861 |
| 6.91 | 6.73 | 29.934 | 33.8 | 5.939 |
| 6.91 | 6.73 | 29.931 | 33.8 | 6.011 |
| 6.92 | 6.73 | 29.929 | 33.8 | 6.169 |
| 6.92 | 6.73 | 29.928 | 33.8 | 6.325 |
| 6.92 | 6.73 | 29.928 | 33.8 | 6.471 |
| 6.92 | 6.73 | 29.928 | 33.8 | 6.587 |
| 6.92 | 6.73 | 29.928 | 33.9 | 6.693 |
| 6.91 | 6.73 | 29.927 | 33.9 | 6.799 |
| 6.91 | 6.73 | 29.929 | 33.8 | 6.92 |
| 6.92 | 6.73 | 29.93 | 33.9 | 7.041 |
| 6.92 | 6.73 | 29.931 | 33.9 | 7.153 |
| 6.92 | 6.74 | 29.932 | 33.9 | 7.267 |
| 6.92 | 6.74 | 29.932 | 33.9 | 7.39 |
| 6.92 | 6.74 | 29.931 | 33.9 | 7.536 |
| 6.91 | 6.74 | 29.93 | 33.9 | 8.263 |
| 6.91 | 6.74 | 29.931 | 33.9 | 8.431 |
| 6.91 | 6.74 | 29.933 | 33.9 | 8.58 |
| 6.91 | 6.74 | 29.934 | 33.9 | 8.697 |
| 6.92 | 6.74 | 29.934 | 33.9 | 8.815 |
| 6.92 | 6.74 | 29.932 | 33.9 | 8.922 |
| 6.92 | 6.74 | 29.929 | 33.9 | 9.034 |
| 6.92 | 6.74 | 29.925 | 33.9 | 9.92 |
| 6.92 | 6.74 | 29.925 | 33.9 | 9.999 |
| 6.92 | 6.74 | 29.925 | 33.9 | 10.004 |
| 6.91 | 6.74 | 29.926 | 33.9 | 9.96 |
| 6.91 | 6.74 | 29.927 | 33.9 | 10.078 |
| 6.91 | 6.74 | 29.927 | 33.9 | 10.918 |
| 6.92 | 6.74 | 29.928 | 33.9 | 11.088 |
| 6.92 | 6.74 | 29.927 | 33.9 | 11.335 |
| 6.92 | 6.74 | 29.926 | 33.8 | 11.414 |
| 6.91 | 6.74 | 29.925 | 33.8 | 11.512 |
| 6.91 | 6.74 | 29.925 | 33.9 | 11.602 |
| 6.91 | 6.74 | 29.925 | 33.9 | 11.709 |
| 6.91 | 6.74 | 29.925 | 33.9 | 12.506 |

| | | | | |
|---|---|---|---|---|
| 6.91 | 6.73 | 29.926 | 33.8 | 12.738 |
| 6.91 | 6.73 | 29.925 | 33.8 | 12.784 |
| 6.91 | 6.74 | 29.925 | 33.8 | 12.891 |
| 6.91 | 6.74 | 29.924 | 33.8 | 13.004 |
| 6.91 | 6.73 | 29.924 | 33.8 | 13.112 |
| 6.91 | 6.73 | 29.924 | 33.8 | 13.216 |
| 6.91 | 6.73 | 29.925 | 33.8 | 14.001 |
| 6.91 | 6.73 | 29.926 | 33.8 | 14.137 |
| 6.91 | 6.73 | 29.926 | 33.8 | 14.259 |
| 6.91 | 6.73 | 29.926 | 33.8 | 14.39 |
| 6.9 | 6.73 | 29.926 | 33.8 | 14.458 |
| 6.9 | 6.73 | 29.925 | 33.8 | 15.245 |
| 6.89 | 6.73 | 29.924 | 33.8 | 15.463 |
| 6.89 | 6.73 | 29.923 | 33.8 | 15.606 |
| 6.88 | 6.72 | 29.923 | 33.8 | 15.74 |
| 6.87 | 6.72 | 29.923 | 33.8 | 15.932 |
| 6.85 | 6.72 | 29.924 | 33.8 | 16.843 |
| 6.83 | 6.72 | 29.924 | 33.8 | 17.246 |
| 6.73 | 6.72 | 29.924 | 33.8 | 17.329 |
| 6.72 | 6.71 | 29.924 | 33.8 | 17.501 |
| 6.72 | 6.71 | 29.924 | 33.8 | 18.764 |
| 6.71 | 6.71 | 29.924 | 33.8 | 19.109 |
| 6.71 | 6.71 | 29.924 | 33.8 | 19.325 |

93

94    Table S20 – P13 site depth profile from 2017 campaign.

| pH | ODO mg/L | Temp °C | Cond µS/cm | Depth m |
|---|---|---|---|---|
| 6.9 | 6.88 | 30.218 | 34.5 | 0.142 |
| 6.91 | 6.88 | 30.22 | 34.5 | 0.202 |
| 6.91 | 6.88 | 30.219 | 34.5 | 0.265 |
| 6.91 | 6.88 | 30.2 | 34.5 | 0.335 |
| 6.91 | 6.86 | 30.165 | 34.4 | 0.404 |
| 6.91 | 6.83 | 30.125 | 34.3 | 0.488 |
| 6.9 | 6.81 | 30.099 | 34.2 | 0.567 |
| 6.89 | 6.79 | 30.084 | 34.1 | 0.629 |
| 6.88 | 6.78 | 30.08 | 34 | 0.695 |
| 6.87 | 6.77 | 30.074 | 33.9 | 0.771 |
| 6.86 | 6.76 | 30.063 | 33.8 | 0.845 |
| 6.86 | 6.75 | 30.044 | 33.7 | 1.737 |
| 6.86 | 6.73 | 30.009 | 33.6 | 1.942 |
| 6.85 | 6.7 | 29.969 | 33.5 | 2.06 |
| 6.85 | 6.66 | 29.937 | 33.4 | 2.173 |
| 6.84 | 6.62 | 29.919 | 33.3 | 2.274 |

| | | | | |
|---|---|---|---|---|
| 6.84 | 6.59 | 29.91 | 33.3 | 2.35 |
| 6.83 | 6.56 | 29.904 | 33.3 | 2.411 |
| 6.82 | 6.54 | 29.899 | 33.3 | 2.467 |
| 6.82 | 6.53 | 29.891 | 33.2 | 2.527 |
| 6.81 | 6.51 | 29.884 | 33.2 | 2.676 |
| 6.8 | 6.49 | 29.88 | 33.2 | 2.821 |
| 6.8 | 6.48 | 29.878 | 33.2 | 2.959 |
| 6.8 | 6.48 | 29.877 | 33.2 | 3.1 |
| 6.8 | 6.47 | 29.876 | 33.2 | 3.254 |
| 6.8 | 6.46 | 29.876 | 33.2 | 3.419 |
| 6.8 | 6.46 | 29.875 | 33.2 | 3.588 |
| 6.8 | 6.45 | 29.875 | 33.3 | 3.753 |
| 6.8 | 6.45 | 29.874 | 33.3 | 3.913 |
| 6.8 | 6.45 | 29.873 | 33.3 | 4.056 |
| 6.8 | 6.44 | 29.872 | 33.3 | 4.196 |
| 6.8 | 6.44 | 29.872 | 33.3 | 4.343 |
| 6.8 | 6.43 | 29.871 | 33.4 | 5.221 |
| 6.8 | 6.43 | 29.871 | 33.4 | 5.326 |
| 6.79 | 6.42 | 29.87 | 33.5 | 5.479 |
| 6.79 | 6.4 | 29.867 | 33.5 | 5.563 |
| 6.78 | 6.39 | 29.865 | 33.5 | 5.617 |
| 6.78 | 6.39 | 29.864 | 33.6 | 5.679 |
| 6.77 | 6.38 | 29.862 | 33.6 | 5.782 |
| 6.77 | 6.37 | 29.859 | 33.7 | 5.88 |
| 6.77 | 6.36 | 29.857 | 33.7 | 5.967 |
| 6.77 | 6.35 | 29.856 | 33.8 | 6.117 |
| 6.77 | 6.35 | 29.855 | 33.8 | 6.264 |

95

96    Table S21 – P12 site depth profile from 2017 campaign.

| pH | ODO mg/L | Temp °C | Cond µS/cm | Depth m |
|---|---|---|---|---|
| 7.02 | 7.03 | 30.424 | 34.5 | 0.317 |
| 7.01 | 7.03 | 30.419 | 34.4 | 0.467 |
| 7 | 7.03 | 30.393 | 34.5 | 0.631 |
| 6.99 | 7.03 | 30.358 | 34.5 | 0.809 |
| 6.98 | 7.02 | 30.335 | 34.4 | 0.953 |
| 6.97 | 7.02 | 30.328 | 34.4 | 1.063 |
| 6.96 | 7.01 | 30.318 | 34.4 | 1.169 |
| 6.96 | 7.01 | 30.291 | 34.4 | 1.257 |
| 6.96 | 7 | 30.251 | 34.5 | 1.341 |
| 6.95 | 6.97 | 30.21 | 34.4 | 1.43 |
| 6.94 | 6.94 | 30.168 | 34.4 | 1.522 |
| 6.93 | 6.9 | 30.123 | 34.5 | 1.613 |

| pH | ODO mg/L | Temp °C | Cond µS/cm | Depth m |
|---|---|---|---|---|
| 6.92 | 6.86 | 30.081 | 34.5 | 1.709 |
| 6.9 | 6.82 | 30.051 | 34.5 | 1.807 |
| 6.89 | 6.77 | 30.021 | 34.5 | 1.899 |
| 6.87 | 6.71 | 29.988 | 34.5 | 1.994 |
| 6.86 | 6.65 | 29.96 | 34.5 | 2.086 |
| 6.85 | 6.61 | 29.949 | 34.6 | 2.176 |
| 6.84 | 6.58 | 29.94 | 34.6 | 2.267 |
| 6.83 | 6.56 | 29.932 | 34.7 | 2.358 |
| 6.83 | 6.54 | 29.924 | 34.7 | 2.45 |
| 6.82 | 6.52 | 29.918 | 34.7 | 2.546 |
| 6.82 | 6.5 | 29.904 | 34.8 | 2.65 |
| 6.81 | 6.46 | 29.889 | 34.8 | 2.761 |
| 6.81 | 6.41 | 29.874 | 34.8 | 2.873 |
| 6.8 | 6.37 | 29.867 | 34.9 | 2.99 |
| 6.79 | 6.34 | 29.861 | 34.9 | 3.098 |
| 6.78 | 6.32 | 29.86 | 34.9 | 3.195 |
| 6.77 | 6.31 | 29.856 | 35 | 3.309 |
| 6.76 | 6.29 | 29.849 | 35 | 3.448 |
| 6.76 | 6.26 | 29.841 | 35 | 3.585 |
| 6.75 | 6.23 | 29.834 | 35.1 | 4.462 |
| 6.74 | 6.2 | 29.827 | 35.1 | 4.713 |
| 6.74 | 6.17 | 29.822 | 35.1 | 4.77 |
| 6.73 | 6.14 | 29.819 | 35.2 | 4.856 |
| 6.73 | 6.12 | 29.817 | 35.2 | 4.938 |
| 6.72 | 6.11 | 29.814 | 35.2 | 5.014 |
| 6.72 | 6.09 | 29.811 | 35.3 | 5.099 |
| 6.72 | 6.07 | 29.81 | 35.3 | 5.182 |
| 6.72 | 6.06 | 29.81 | 35.3 | 5.25 |
| 6.72 | 6.05 | 29.81 | 35.3 | 5.357 |
| 6.71 | 6.04 | 29.808 | 35.3 | 5.474 |
| 6.71 | 6.03 | 29.806 | 35.3 | 5.583 |
| 6.71 | 6.01 | 29.803 | 35.4 | 5.665 |
| 6.7 | 5.99 | 29.8 | 35.4 | 5.729 |
| 6.7 | 5.97 | 29.795 | 35.4 | 5.778 |
| 6.69 | 5.93 | 29.788 | 35.4 | 5.83 |
| 6.68 | 5.86 | 29.778 | 35.5 | 5.885 |
| 6.66 | 5.75 | 29.769 | 35.6 | 5.951 |

97

98    Table S22 – P15 site depth profile from 2017 campaign.

| pH | ODO mg/L | Temp °C | Cond µS/cm | Depth m |
|---|---|---|---|---|
| 7.09 | 7.03 | 29.972 | 34.9 | 1.12 |
| 7.07 | 6.99 | 29.929 | 34.7 | 1.235 |

| | | | | |
|---|---|---|---|---|
| 7.05 | 6.96 | 29.907 | 34.5 | 1.4 |
| 7.04 | 6.94 | 29.898 | 34.4 | 1.552 |
| 7.03 | 6.92 | 29.891 | 34.3 | 1.675 |
| 7.02 | 6.9 | 29.887 | 34.2 | 1.776 |
| 7 | 6.89 | 29.884 | 34.1 | 1.9 |
| 7 | 6.89 | 29.88 | 34 | 2.11 |
| 6.99 | 6.88 | 29.874 | 33.9 | 2.314 |
| 6.98 | 6.88 | 29.869 | 33.8 | 2.497 |
| 6.98 | 6.87 | 29.866 | 33.8 | 2.642 |
| 6.98 | 6.86 | 29.863 | 33.8 | 2.775 |
| 6.98 | 6.86 | 29.863 | 33.7 | 2.911 |
| 6.98 | 6.86 | 29.863 | 33.7 | 3.047 |
| 6.97 | 6.86 | 29.866 | 33.7 | 3.171 |
| 6.97 | 6.86 | 29.864 | 33.7 | 3.277 |
| 6.97 | 6.85 | 29.857 | 33.7 | 3.378 |
| 6.97 | 6.85 | 29.849 | 33.6 | 3.479 |
| 6.96 | 6.83 | 29.846 | 33.6 | 3.598 |
| 6.96 | 6.83 | 29.844 | 33.6 | 3.714 |
| 6.95 | 6.82 | 29.843 | 33.7 | 3.827 |
| 6.95 | 6.82 | 29.84 | 33.7 | 3.955 |
| 6.95 | 6.81 | 29.836 | 33.7 | 4.087 |
| 6.95 | 6.81 | 29.831 | 33.7 | 4.208 |
| 6.95 | 6.8 | 29.823 | 33.8 | 4.331 |
| 6.94 | 6.79 | 29.816 | 33.8 | 4.441 |
| 6.94 | 6.78 | 29.813 | 33.8 | 4.54 |
| 6.94 | 6.78 | 29.814 | 33.8 | 4.636 |
| 6.93 | 6.77 | 29.815 | 33.9 | 4.714 |
| 6.93 | 6.77 | 29.815 | 33.9 | 4.782 |
| 6.93 | 6.77 | 29.814 | 33.9 | 4.873 |
| 6.93 | 6.77 | 29.813 | 33.9 | 4.97 |
| 6.93 | 6.77 | 29.811 | 34 | 5.075 |
| 6.93 | 6.77 | 29.809 | 34 | 5.192 |
| 6.92 | 6.76 | 29.807 | 34 | 5.329 |
| 6.92 | 6.76 | 29.808 | 34.1 | 5.483 |
| 6.92 | 6.75 | 29.808 | 34.1 | 5.66 |
| 6.92 | 6.75 | 29.809 | 34.1 | 5.818 |
| 6.92 | 6.75 | 29.81 | 34.1 | 5.976 |
| 6.92 | 6.76 | 29.81 | 34.2 | 6.139 |
| 6.92 | 6.76 | 29.81 | 34.2 | 6.306 |
| 6.92 | 6.76 | 29.81 | 34.2 | 6.461 |
| 6.92 | 6.76 | 29.809 | 34.2 | 6.617 |
| 6.92 | 6.76 | 29.809 | 34.2 | 6.753 |
| 6.92 | 6.76 | 29.808 | 34.3 | 6.888 |
| 6.92 | 6.76 | 29.807 | 34.3 | 7.023 |
| 6.92 | 6.75 | 29.806 | 34.3 | 7.153 |

| 6.92 | 6.75 | 29.806 | 34.3 | 7.266 |
|------|------|--------|------|-------|
| 6.92 | 6.76 | 29.806 | 34.3 | 7.383 |
| 6.92 | 6.75 | 29.806 | 34.3 | 7.493 |
| 6.92 | 6.75 | 29.807 | 34.3 | 7.62 |
| 6.92 | 6.75 | 29.806 | 34.3 | 7.76 |
| 6.92 | 6.74 | 29.805 | 34.3 | 8.611 |
| 6.92 | 6.74 | 29.805 | 34.4 | 8.702 |
| 6.92 | 6.73 | 29.806 | 34.4 | 8.751 |
| 6.92 | 6.73 | 29.806 | 34.4 | 8.833 |
| 6.92 | 6.73 | 29.805 | 34.4 | 8.898 |
| 6.92 | 6.73 | 29.806 | 34.4 | 8.966 |
| 6.91 | 6.73 | 29.806 | 34.4 | 9.038 |
| 6.91 | 6.73 | 29.805 | 34.4 | 9.117 |
| 6.91 | 6.73 | 29.806 | 34.5 | 9.235 |
| 6.91 | 6.72 | 29.806 | 34.5 | 9.396 |
| 6.9 | 6.72 | 29.806 | 34.4 | 9.554 |
| 6.9 | 6.72 | 29.806 | 34.5 | 9.719 |
| 6.9 | 6.72 | 29.806 | 34.5 | 10.579 |
| 6.9 | 6.71 | 29.806 | 34.5 | 10.639 |
| 6.91 | 6.71 | 29.806 | 34.5 | 10.75 |
| 6.91 | 6.71 | 29.806 | 34.5 | 10.854 |
| 6.91 | 6.7 | 29.807 | 34.5 | 10.932 |
| 6.9 | 6.7 | 29.807 | 34.5 | 10.99 |

99

100    Table S23 – P14 site depth profile from 2017 campaign.

| pH | ODO mg/L | Temp °C | Cond µS/cm | Depth m |
|------|------|--------|------|-------|
| 6.84 | 6.67 | 29.657 | 33.1 | 0.448 |
| 6.85 | 6.67 | 29.657 | 33.1 | 0.51 |
| 6.85 | 6.67 | 29.656 | 33.1 | 0.568 |
| 6.85 | 6.67 | 29.656 | 33.1 | 0.602 |
| 6.85 | 6.67 | 29.657 | 33.1 | 0.632 |
| 6.85 | 6.67 | 29.656 | 33.1 | 0.66 |
| 6.84 | 6.67 | 29.656 | 33.1 | 0.672 |
| 6.84 | 6.67 | 29.655 | 33.2 | 0.687 |
| 6.84 | 6.67 | 29.655 | 33.2 | 0.713 |
| 6.85 | 6.67 | 29.655 | 33.2 | 0.749 |
| 6.85 | 6.67 | 29.654 | 33.2 | 0.788 |
| 6.85 | 6.67 | 29.654 | 33.3 | 0.853 |
| 6.84 | 6.67 | 29.653 | 33.3 | 0.929 |
| 6.84 | 6.67 | 29.654 | 33.3 | 0.997 |
| 6.84 | 6.67 | 29.654 | 33.3 | 1.056 |
| 6.84 | 6.67 | 29.654 | 33.4 | 1.109 |

| | | | | |
|---|---|---|---|---|
| 6.84 | 6.67 | 29.654 | 33.4 | 1.157 |
| 6.84 | 6.67 | 29.653 | 33.4 | 1.189 |
| 6.84 | 6.67 | 29.654 | 33.5 | 1.22 |
| 6.84 | 6.67 | 29.653 | 33.5 | 1.229 |
| 6.83 | 6.67 | 29.653 | 33.6 | 1.227 |
| 6.83 | 6.67 | 29.654 | 33.6 | 1.236 |
| 6.83 | 6.67 | 29.654 | 33.7 | 1.264 |
| 6.83 | 6.67 | 29.654 | 33.7 | 1.325 |
| 6.83 | 6.67 | 29.655 | 33.7 | 1.378 |
| 6.83 | 6.67 | 29.655 | 33.8 | 1.433 |
| 6.83 | 6.67 | 29.655 | 33.8 | 1.481 |
| 6.83 | 6.67 | 29.655 | 33.9 | 1.543 |
| 6.83 | 6.67 | 29.655 | 33.9 | 1.617 |
| 6.83 | 6.67 | 29.656 | 33.9 | 1.696 |
| 6.83 | 6.67 | 29.656 | 34 | 1.771 |
| 6.83 | 6.67 | 29.657 | 34 | 1.842 |
| 6.83 | 6.67 | 29.657 | 34.1 | 1.918 |
| 6.83 | 6.67 | 29.657 | 34.1 | 2.002 |
| 6.83 | 6.67 | 29.656 | 34.1 | 2.861 |
| 6.83 | 6.67 | 29.656 | 34.2 | 2.924 |
| 6.83 | 6.66 | 29.656 | 34.2 | 2.922 |
| 6.83 | 6.66 | 29.655 | 34.2 | 2.852 |
| 6.83 | 6.66 | 29.655 | 34.3 | 2.777 |
| 6.83 | 6.66 | 29.654 | 34.3 | 2.729 |

101

102    Table S24 – P16 site depth profile from 2017 campaign.

| pH | ODO mg/L | Temp °C | Cond µS/cm | Depth m |
|---|---|---|---|---|
| 6.8 | 6.65 | 30.465 | 35.4 | 0.337 |
| 6.8 | 6.64 | 30.437 | 35.2 | 0.396 |
| 6.8 | 6.63 | 30.383 | 35 | 0.447 |
| 6.81 | 6.62 | 30.332 | 34.8 | 0.493 |
| 6.81 | 6.61 | 30.329 | 34.7 | 0.538 |
| 6.8 | 6.61 | 30.349 | 34.6 | 0.587 |
| 6.8 | 6.6 | 30.326 | 34.5 | 0.642 |
| 6.8 | 6.59 | 30.249 | 34.4 | 0.702 |
| 6.8 | 6.58 | 30.158 | 34.2 | 0.771 |
| 6.8 | 6.57 | 30.09 | 34.1 | 0.839 |
| 6.8 | 6.55 | 30.04 | 34 | 0.905 |
| 6.79 | 6.53 | 30.012 | 34 | 0.977 |
| 6.78 | 6.5 | 29.985 | 34 | 1.052 |
| 6.77 | 6.49 | 29.963 | 33.9 | 1.137 |
| 6.77 | 6.48 | 29.94 | 33.9 | 1.24 |

| | | | | |
|------|------|--------|------|-------|
| 6.76 | 6.47 | 29.919 | 33.9 | 1.349 |
| 6.76 | 6.45 | 29.888 | 33.9 | 1.473 |
| 6.76 | 6.43 | 29.865 | 33.8 | 1.59 |
| 6.75 | 6.42 | 29.851 | 33.8 | 1.702 |
| 6.74 | 6.41 | 29.843 | 33.8 | 1.811 |
| 6.74 | 6.4 | 29.837 | 33.8 | 1.925 |
| 6.73 | 6.4 | 29.834 | 33.8 | 2.032 |
| 6.73 | 6.39 | 29.828 | 33.9 | 2.133 |
| 6.73 | 6.39 | 29.823 | 33.9 | 2.231 |
| 6.74 | 6.38 | 29.822 | 33.9 | 2.328 |
| 6.74 | 6.38 | 29.82 | 33.9 | 2.421 |
| 6.74 | 6.38 | 29.82 | 33.9 | 2.524 |
| 6.75 | 6.38 | 29.824 | 34 | 2.636 |
| 6.75 | 6.38 | 29.83 | 34 | 2.74 |
| 6.74 | 6.38 | 29.834 | 34 | 2.838 |
| 6.74 | 6.38 | 29.839 | 34 | 2.942 |
| 6.74 | 6.38 | 29.841 | 34 | 3.048 |
| 6.74 | 6.38 | 29.834 | 34.1 | 3.158 |
| 6.74 | 6.38 | 29.818 | 34.1 | 3.269 |
| 6.74 | 6.37 | 29.805 | 34.1 | 3.373 |
| 6.74 | 6.36 | 29.802 | 34.2 | 3.48 |
| 6.74 | 6.36 | 29.806 | 34.2 | 3.588 |
| 6.74 | 6.36 | 29.812 | 34.2 | 3.699 |
| 6.73 | 6.36 | 29.809 | 34.2 | 3.822 |
| 6.73 | 6.35 | 29.793 | 34.2 | 3.94 |
| 6.73 | 6.35 | 29.775 | 34.2 | 4.056 |
| 6.72 | 6.34 | 29.756 | 34.2 | 4.184 |
| 6.72 | 6.33 | 29.739 | 34.2 | 4.305 |
| 6.72 | 6.32 | 29.723 | 34.3 | 4.427 |
| 6.73 | 6.31 | 29.699 | 34.3 | 4.545 |
| 6.73 | 6.31 | 29.683 | 34.3 | 4.657 |
| 6.73 | 6.29 | 29.675 | 34.3 | 4.761 |
| 6.72 | 6.28 | 29.674 | 34.3 | 4.866 |
| 6.72 | 6.26 | 29.674 | 34.3 | 4.968 |
| 6.72 | 6.25 | 29.673 | 34.3 | 5.057 |
| 6.72 | 6.24 | 29.67 | 34.4 | 5.148 |
| 6.72 | 6.23 | 29.662 | 34.4 | 5.229 |
| 6.72 | 6.23 | 29.657 | 34.4 | 5.309 |
| 6.72 | 6.22 | 29.656 | 34.4 | 5.373 |
| 6.72 | 6.21 | 29.659 | 34.4 | 5.432 |
| 6.72 | 6.21 | 29.661 | 34.4 | 5.484 |
| 6.72 | 6.21 | 29.666 | 34.4 | 5.548 |
| 6.72 | 6.21 | 29.67 | 34.4 | 5.621 |
| 6.72 | 6.21 | 29.672 | 34.4 | 5.697 |
| 6.72 | 6.21 | 29.667 | 34.4 | 5.771 |

| | | | | |
|---|---|---|---|---|
| 6.72 | 6.21 | 29.661 | 34.4 | 5.854 |
| 6.72 | 6.2 | 29.657 | 34.4 | 5.939 |
| 6.72 | 6.2 | 29.657 | 34.4 | 6.034 |
| 6.71 | 6.19 | 29.658 | 34.4 | 6.126 |
| 6.71 | 6.19 | 29.66 | 34.4 | 6.161 |
| 6.7 | 6.19 | 29.662 | 34.4 | 6.177 |
| 6.7 | 6.21 | 29.662 | 34.4 | 6.223 |
| 6.69 | 6.21 | 29.659 | 34.4 | 6.284 |
| 6.69 | 6.19 | 29.659 | 34.4 | 6.356 |
| 6.69 | 6.16 | 29.657 | 34.5 | 6.432 |
| 6.7 | 6.15 | 29.654 | 34.5 | 6.508 |
| 6.7 | 6.14 | 29.648 | 34.5 | 6.597 |
| 6.7 | 6.14 | 29.642 | 34.5 | 6.752 |
| 6.7 | 6.14 | 29.635 | 34.5 | 6.877 |
| 6.7 | 6.15 | 29.632 | 34.5 | 6.982 |
| 6.7 | 6.15 | 29.632 | 34.5 | 7.067 |
| 6.7 | 6.15 | 29.632 | 34.5 | 7.149 |
| 6.7 | 6.14 | 29.632 | 34.5 | 7.237 |
| 6.69 | 6.14 | 29.632 | 34.5 | 7.318 |
| 6.69 | 6.15 | 29.631 | 34.5 | 7.394 |
| 6.69 | 6.15 | 29.63 | 34.5 | 7.472 |
| 6.69 | 6.15 | 29.63 | 34.5 | 7.552 |
| 6.69 | 6.16 | 29.63 | 34.5 | 7.636 |
| 6.69 | 6.16 | 29.63 | 34.5 | 7.739 |
| 6.69 | 6.16 | 29.629 | 34.5 | 7.848 |
| 6.69 | 6.16 | 29.628 | 34.5 | 7.972 |
| 6.69 | 6.16 | 29.627 | 34.5 | 8.118 |
| 6.69 | 6.17 | 29.627 | 34.5 | 8.275 |
| 6.69 | 6.17 | 29.628 | 34.5 | 8.437 |
| 6.7 | 6.17 | 29.628 | 34.5 | 8.611 |
| 6.7 | 6.18 | 29.628 | 34.4 | 8.781 |
| 6.71 | 6.18 | 29.628 | 34.5 | 8.93 |
| 6.71 | 6.19 | 29.627 | 34.5 | 9.058 |
| 6.71 | 6.19 | 29.626 | 34.5 | 9.163 |
| 6.72 | 6.2 | 29.625 | 34.4 | 9.25 |
| 6.72 | 6.2 | 29.625 | 34.4 | 9.316 |
| 6.72 | 6.2 | 29.625 | 34.5 | 9.365 |
| 6.72 | 6.2 | 29.626 | 34.5 | 9.421 |
| 6.71 | 6.2 | 29.626 | 34.4 | 9.48 |
| 6.71 | 6.2 | 29.625 | 34.4 | 9.546 |
| 6.71 | 6.2 | 29.624 | 34.4 | 9.626 |
| 6.71 | 6.2 | 29.623 | 34.4 | 9.726 |
| 6.71 | 6.2 | 29.624 | 34.4 | 9.806 |
| 6.71 | 6.2 | 29.624 | 34.4 | 9.908 |
| 6.71 | 6.2 | 29.623 | 34.5 | 10.01 |

| | | | | |
|------|------|--------|------|--------|
| 6.71 | 6.21 | 29.622 | 34.4 | 10.11 |
| 6.71 | 6.21 | 29.622 | 34.4 | 10.218 |
| 6.71 | 6.21 | 29.622 | 34.4 | 10.317 |
| 6.71 | 6.21 | 29.622 | 34.5 | 10.402 |
| 6.72 | 6.21 | 29.622 | 34.5 | 10.486 |
| 6.72 | 6.21 | 29.622 | 34.4 | 10.593 |
| 6.71 | 6.2  | 29.621 | 34.5 | 10.712 |
| 6.71 | 6.2  | 29.621 | 34.5 | 10.842 |
| 6.71 | 6.2  | 29.62  | 34.5 | 10.974 |
| 6.71 | 6.2  | 29.619 | 34.4 | 11.839 |
| 6.71 | 6.21 | 29.619 | 34.4 | 11.939 |
| 6.71 | 6.21 | 29.618 | 34.5 | 11.905 |
| 6.71 | 6.21 | 29.616 | 34.5 | 11.956 |
| 6.71 | 6.21 | 29.615 | 34.5 | 12.009 |
| 6.71 | 6.21 | 29.614 | 34.5 | 12.065 |
| 6.71 | 6.21 | 29.613 | 34.4 | 12.134 |
| 6.71 | 6.21 | 29.611 | 34.4 | 12.196 |
| 6.7  | 6.2  | 29.607 | 34.5 | 12.257 |
| 6.7  | 6.18 | 29.604 | 34.5 | 12.358 |
| 6.69 | 6.16 | 29.601 | 34.5 | 12.463 |
| 6.69 | 6.14 | 29.601 | 34.5 | 12.565 |
| 6.68 | 6.13 | 29.598 | 34.5 | 12.663 |
| 6.68 | 6.11 | 29.595 | 34.5 | 12.762 |
| 6.67 | 6.08 | 29.586 | 34.6 | 12.862 |
| 6.66 | 6.01 | 29.57  | 34.6 | 12.976 |
| 6.64 | 5.88 | 29.555 | 34.7 | 13.063 |
| 6.62 | 5.75 | 29.546 | 34.8 | 13.183 |
| 6.6  | 5.67 | 29.545 | 34.8 | 13.306 |
| 6.58 | 5.64 | 29.545 | 34.8 | 13.441 |
| 6.57 | 5.63 | 29.545 | 34.8 | 13.582 |
| 6.57 | 5.63 | 29.55  | 34.8 | 13.724 |
| 6.57 | 5.65 | 29.555 | 34.8 | 13.832 |
| 6.57 | 5.69 | 29.56  | 34.7 | 13.957 |
| 6.57 | 5.69 | 29.555 | 34.7 | 14.106 |
| 6.56 | 5.62 | 29.54  | 34.8 | 14.247 |
| 6.55 | 5.52 | 29.526 | 34.9 | 14.391 |
| 6.54 | 5.46 | 29.52  | 34.9 | 14.54 |
| 6.52 | 5.43 | 29.516 | 34.9 | 15.477 |
| 6.51 | 5.39 | 29.507 | 34.9 | 15.554 |
| 6.5  | 5.33 | 29.504 | 34.9 | 15.655 |
| 6.49 | 5.29 | 29.501 | 34.9 | 15.748 |
| 6.48 | 5.27 | 29.498 | 34.9 | 15.838 |
| 6.48 | 5.24 | 29.494 | 35   | 15.94 |
| 6.47 | 5.23 | 29.491 | 35   | 16.032 |
| 6.47 | 5.21 | 29.491 | 35   | 16.117 |

| | | | | |
|---|---|---|---|---|
| 6.47 | 5.2 | 29.491 | 34.9 | 16.23 |
| 6.47 | 5.2 | 29.493 | 34.9 | 16.404 |
| 6.47 | 5.2 | 29.491 | 35 | 16.549 |
| 6.46 | 5.19 | 29.486 | 35 | 16.683 |
| 6.46 | 5.17 | 29.477 | 35 | 16.815 |
| 6.46 | 5.15 | 29.473 | 35 | 16.925 |
| 6.45 | 5.13 | 29.473 | 35 | 17.051 |
| 6.45 | 5.13 | 29.469 | 35 | 17.186 |
| 6.45 | 5.11 | 29.46 | 35.1 | 17.324 |
| 6.44 | 5.07 | 29.448 | 35 | 17.947 |
| 6.43 | 5.04 | 29.44 | 35 | 18.037 |
| 6.43 | 5.01 | 29.436 | 35 | 18.148 |
| 6.43 | 5 | 29.429 | 35.1 | 18.281 |
| 6.43 | 4.97 | 29.41 | 35.1 | 18.399 |
| 6.42 | 4.88 | 29.376 | 35.2 | 18.513 |
| 6.4 | 4.53 | 29.34 | 35.2 | 19.16 |

103

104    Table S25 – P17 site depth profile from 2017 campaign.

| pH | ODO mg/L | Temp °C | Cond µS/cm | Depth m |
|---|---|---|---|---|
| 6.94 | 7.09 | 30.432 | 36.6 | 0.215 |
| 6.94 | 6.99 | 30.047 | 36.3 | 0.345 |
| 6.92 | 6.91 | 29.839 | 36 | 0.496 |
| 6.89 | 6.85 | 29.729 | 35.9 | 0.626 |
| 6.86 | 6.71 | 29.696 | 35.8 | 0.759 |
| 6.83 | 6.63 | 29.672 | 35.7 | 0.893 |
| 6.8 | 6.55 | 29.647 | 35.6 | 1.049 |
| 6.78 | 6.5 | 29.624 | 35.5 | 1.21 |
| 6.76 | 6.45 | 29.607 | 35.4 | 1.362 |
| 6.75 | 6.41 | 29.593 | 35.4 | 1.506 |
| 6.74 | 6.38 | 29.584 | 35.3 | 1.65 |
| 6.74 | 6.36 | 29.577 | 35.2 | 1.798 |
| 6.73 | 6.35 | 29.57 | 35.2 | 1.969 |
| 6.73 | 6.33 | 29.564 | 35.2 | 2.109 |
| 6.72 | 6.31 | 29.56 | 35.1 | 2.23 |
| 6.72 | 6.3 | 29.557 | 35.1 | 2.341 |
| 6.71 | 6.28 | 29.553 | 35 | 2.438 |
| 6.7 | 6.27 | 29.548 | 35 | 2.534 |
| 6.7 | 6.25 | 29.545 | 35 | 2.632 |
| 6.7 | 6.24 | 29.542 | 34.9 | 2.725 |
| 6.7 | 6.22 | 29.539 | 34.9 | 2.803 |
| 6.7 | 6.21 | 29.536 | 34.9 | 2.894 |
| 6.69 | 6.2 | 29.533 | 34.8 | 2.995 |

| | | | | |
|---|---|---|---|---|
| 6.69 | 6.19 | 29.531 | 34.8 | 3.094 |
| 6.69 | 6.18 | 29.529 | 34.8 | 3.197 |
| 6.69 | 6.17 | 29.527 | 34.8 | 3.292 |
| 6.69 | 6.16 | 29.525 | 34.7 | 3.38 |
| 6.69 | 6.16 | 29.522 | 34.7 | 3.47 |
| 6.69 | 6.14 | 29.503 | 34.7 | 3.561 |
| 6.68 | 6.1 | 29.486 | 34.7 | 3.645 |
| 6.68 | 6.05 | 29.464 | 34.7 | 3.729 |
| 6.67 | 5.99 | 29.444 | 34.7 | 3.816 |
| 6.66 | 5.94 | 29.431 | 34.7 | 3.9 |
| 6.66 | 5.89 | 29.42 | 34.7 | 3.988 |
| 6.65 | 5.86 | 29.415 | 34.7 | 4.086 |
| 6.65 | 5.84 | 29.423 | 34.7 | 4.184 |
| 6.64 | 5.83 | 29.434 | 34.7 | 4.287 |
| 6.64 | 5.83 | 29.427 | 34.7 | 4.389 |
| 6.63 | 5.8 | 29.412 | 34.7 | 4.489 |
| 6.62 | 5.76 | 29.403 | 34.7 | 4.584 |
| 6.61 | 5.74 | 29.397 | 34.7 | 4.678 |
| 6.61 | 5.72 | 29.389 | 34.7 | 4.765 |
| 6.61 | 5.7 | 29.384 | 34.7 | 4.846 |
| 6.61 | 5.69 | 29.384 | 34.7 | 4.927 |
| 6.61 | 5.69 | 29.386 | 34.7 | 5.002 |
| 6.61 | 5.69 | 29.388 | 34.6 | 5.081 |
| 6.62 | 5.69 | 29.386 | 34.7 | 5.156 |
| 6.62 | 5.69 | 29.382 | 34.6 | 5.229 |
| 6.62 | 5.69 | 29.378 | 34.6 | 5.304 |
| 6.62 | 5.68 | 29.377 | 34.6 | 5.384 |
| 6.62 | 5.67 | 29.375 | 34.6 | 5.464 |
| 6.61 | 5.67 | 29.375 | 34.6 | 5.545 |
| 6.61 | 5.66 | 29.373 | 34.6 | 5.626 |
| 6.61 | 5.66 | 29.373 | 34.6 | 5.706 |
| 6.61 | 5.66 | 29.372 | 34.6 | 5.788 |
| 6.61 | 5.65 | 29.372 | 34.6 | 5.871 |

105

106    Table S26 – P18 site depth profile from 2017 campaign.

| pH | ODO mg/L | Temp °C | Cond µS/cm | Depth m |
|---|---|---|---|---|
| 6.7 | 6.26 | 29.64 | 34.2 | 0.978 |
| 6.68 | 6.21 | 29.581 | 34.1 | 0.966 |
| 6.68 | 6.16 | 29.534 | 34.1 | 0.967 |
| 6.67 | 6.14 | 29.527 | 34.1 | 0.964 |
| 6.66 | 6.12 | 29.515 | 34.1 | 1.084 |
| 6.65 | 6.08 | 29.49 | 34 | 1.196 |

| | | | | |
|---|---|---|---|---|
| 6.64 | 6.01 | 29.454 | 34 | 2.034 |
| 6.63 | 5.94 | 29.426 | 33.9 | 2.033 |
| 6.61 | 5.89 | 29.41 | 33.8 | 2.139 |
| 6.61 | 5.86 | 29.41 | 33.8 | 2.29 |
| 6.6 | 5.86 | 29.414 | 33.7 | 2.361 |
| 6.6 | 5.85 | 29.415 | 33.6 | 2.425 |
| 6.6 | 5.85 | 29.413 | 33.5 | 2.531 |
| 6.6 | 5.84 | 29.41 | 33.4 | 2.607 |
| 6.6 | 5.84 | 29.408 | 33.4 | 3.492 |
| 6.6 | 5.83 | 29.406 | 33.3 | 3.612 |
| 6.6 | 5.82 | 29.407 | 33.2 | 3.612 |
| 6.6 | 5.81 | 29.409 | 33.1 | 3.611 |
| 6.59 | 5.81 | 29.409 | 33.1 | 3.767 |
| 6.59 | 5.8 | 29.406 | 33.1 | 3.862 |
| 6.6 | 5.8 | 29.401 | 33.1 | 3.921 |
| 6.6 | 5.79 | 29.401 | 33 | 3.983 |
| 6.59 | 5.79 | 29.402 | 33 | 4.071 |
| 6.6 | 5.79 | 29.404 | 33 | 4.197 |
| 6.6 | 5.79 | 29.406 | 33 | 4.333 |
| 6.59 | 5.79 | 29.408 | 33 | 5.268 |
| 6.59 | 5.78 | 29.409 | 32.9 | 5.263 |
| 6.6 | 5.78 | 29.41 | 32.9 | 5.264 |
| 6.6 | 5.78 | 29.409 | 33 | 5.325 |
| 6.6 | 5.78 | 29.409 | 32.9 | 5.43 |
| 6.6 | 5.78 | 29.412 | 32.9 | 5.486 |
| 6.6 | 5.79 | 29.423 | 33 | 5.521 |
| 6.6 | 5.79 | 29.435 | 33 | 5.594 |
| 6.6 | 5.79 | 29.442 | 33 | 5.706 |
| 6.61 | 5.8 | 29.445 | 33 | 5.838 |
| 6.61 | 5.8 | 29.446 | 33.1 | 5.975 |
| 6.6 | 5.8 | 29.447 | 33.1 | 6.098 |
| 6.6 | 5.8 | 29.448 | 33.1 | 6.227 |
| 6.6 | 5.81 | 29.45 | 33.1 | 6.357 |
| 6.6 | 5.81 | 29.448 | 33.2 | 7.172 |
| 6.59 | 5.8 | 29.441 | 33.2 | 7.171 |
| 6.58 | 5.8 | 29.422 | 33.2 | 7.165 |
| 6.58 | 5.78 | 29.399 | 33.2 | 7.18 |
| 6.56 | 5.75 | 29.367 | 33.3 | 7.329 |
| 6.54 | 5.7 | 29.341 | 33.3 | 7.418 |
| 6.52 | 5.66 | 29.329 | 33.3 | 7.475 |
| 6.49 | 5.64 | 29.326 | 33.4 | 8.308 |
| 6.48 | 5.63 | 29.323 | 33.4 | 8.301 |
| 6.47 | 5.62 | 29.323 | 33.4 | 8.303 |
| 6.46 | 5.61 | 29.322 | 33.5 | 8.408 |
| 6.45 | 5.6 | 29.321 | 33.5 | 8.524 |

| | | | | |
|------|------|--------|------|--------|
| 6.45 | 5.6  | 29.32  | 33.5 | 8.593  |
| 6.45 | 5.59 | 29.32  | 33.6 | 9.39   |
| 6.44 | 5.59 | 29.319 | 33.6 | 9.396  |
| 6.43 | 5.58 | 29.317 | 33.7 | 9.396  |
| 6.42 | 5.58 | 29.316 | 33.7 | 9.4    |
| 6.41 | 5.57 | 29.315 | 33.8 | 9.472  |
| 6.4  | 5.57 | 29.314 | 33.8 | 9.584  |
| 6.39 | 5.56 | 29.313 | 33.8 | 9.653  |
| 6.38 | 5.56 | 29.312 | 33.8 | 9.702  |
| 6.37 | 5.56 | 29.312 | 33.9 | 9.752  |
| 6.36 | 5.56 | 29.312 | 33.9 | 9.884  |
| 6.36 | 5.56 | 29.312 | 33.9 | 9.999  |
| 6.36 | 5.55 | 29.312 | 33.9 | 10.143 |
| 6.35 | 5.55 | 29.312 | 33.9 | 10.89  |
| 6.34 | 5.55 | 29.311 | 34   | 10.899 |
| 6.34 | 5.55 | 29.31  | 34   | 10.893 |
| 6.33 | 5.55 | 29.31  | 34   | 10.891 |
| 6.33 | 5.54 | 29.309 | 34   | 10.918 |
| 6.33 | 5.54 | 29.309 | 34   | 11.016 |
| 6.32 | 5.54 | 29.308 | 34   | 11.081 |
| 6.32 | 5.54 | 29.308 | 34.1 | 11.128 |
| 6.32 | 5.53 | 29.307 | 34.1 | 11.174 |
| 6.32 | 5.53 | 29.304 | 34.1 | 12.023 |
| 6.31 | 5.52 | 29.3   | 34.1 | 12.019 |
| 6.29 | 5.51 | 29.297 | 34.2 | 12.018 |
| 6.28 | 5.5  | 29.296 | 34.2 | 12.016 |
| 6.27 | 5.49 | 29.296 | 34.2 | 12.014 |
| 6.28 | 5.49 | 29.298 | 34.2 | 12.013 |
| 6.29 | 5.49 | 29.298 | 34.2 | 12.011 |
| 6.29 | 5.49 | 29.293 | 34.2 | 12.072 |
| 6.28 | 5.48 | 29.289 | 34.2 | 12.141 |
| 6.26 | 5.47 | 29.286 | 34.2 | 12.216 |
| 6.24 | 5.47 | 29.288 | 34.3 | 12.296 |
| 6.23 | 5.47 | 29.287 | 34.3 | 12.373 |
| 6.24 | 5.46 | 29.284 | 34.3 | 12.47  |
| 6.23 | 5.45 | 29.281 | 34.2 | 12.611 |
| 6.22 | 5.44 | 29.277 | 34.2 | 12.735 |
| 6.22 | 5.43 | 29.277 | 34.3 | 12.8   |
| 6.21 | 5.43 | 29.277 | 34.3 | 12.865 |
| 6.22 | 5.42 | 29.278 | 34.3 | 12.927 |
| 6.23 | 5.42 | 29.279 | 34.3 | 13.038 |
| 6.22 | 5.42 | 29.28  | 34.3 | 13.176 |
| 6.22 | 5.43 | 29.281 | 34.3 | 13.265 |
| 6.22 | 5.43 | 29.281 | 34.3 | 13.341 |
| 6.22 | 5.43 | 29.28  | 34.3 | 13.418 |

| | | | | |
|------|------|--------|------|--------|
| 6.23 | 5.42 | 29.279 | 34.3 | 13.498 |
| 6.25 | 5.42 | 29.278 | 34.3 | 13.576 |
| 6.26 | 5.42 | 29.277 | 34.3 | 13.702 |
| 6.25 | 5.41 | 29.278 | 34.3 | 13.79 |
| 6.24 | 5.41 | 29.279 | 34.3 | 13.878 |
| 6.23 | 5.41 | 29.28 | 34.3 | 13.968 |
| 6.23 | 5.42 | 29.281 | 34.3 | 14.054 |
| 6.24 | 5.42 | 29.281 | 34.3 | 14.14 |
| 6.26 | 5.42 | 29.28 | 34.3 | 14.226 |
| 6.27 | 5.42 | 29.279 | 34.3 | 14.311 |
| 6.27 | 5.41 | 29.278 | 34.3 | 14.386 |
| 6.26 | 5.41 | 29.277 | 34.3 | 14.463 |
| 6.25 | 5.4 | 29.275 | 34.3 | 14.538 |
| 6.23 | 5.4 | 29.271 | 34.3 | 15.477 |
| 6.22 | 5.39 | 29.269 | 34.3 | 15.475 |
| 6.22 | 5.38 | 29.268 | 34.3 | 15.476 |
| 6.22 | 5.37 | 29.267 | 34.3 | 15.478 |
| 6.23 | 5.37 | 29.267 | 34.3 | 15.482 |
| 6.24 | 5.36 | 29.267 | 34.3 | 15.486 |
| 6.26 | 5.36 | 29.267 | 34.3 | 15.501 |
| 6.26 | 5.36 | 29.269 | 34.3 | 15.584 |
| 6.26 | 5.37 | 29.27 | 34.3 | 15.682 |
| 6.26 | 5.37 | 29.271 | 34.3 | 15.787 |
| 6.25 | 5.38 | 29.27 | 34.3 | 16.864 |
| 6.24 | 5.37 | 29.268 | 34.3 | 16.896 |
| 6.24 | 5.36 | 29.267 | 34.3 | 16.899 |
| 6.25 | 5.36 | 29.266 | 34.3 | 16.899 |
| 6.24 | 5.36 | 29.264 | 34.4 | 16.999 |
| 6.23 | 5.35 | 29.261 | 34.3 | 17.143 |
| 6.23 | 5.34 | 29.26 | 34.3 | 17.236 |
| 6.22 | 5.33 | 29.259 | 34.3 | 18.29 |
| 6.21 | 5.33 | 29.26 | 34.3 | 18.433 |
| 6.2 | 5.32 | 29.26 | 34.3 | 18.433 |
| 6.21 | 5.32 | 29.261 | 34.3 | 18.434 |
| 6.2 | 5.32 | 29.261 | 34.3 | 18.539 |
| 6.2 | 5.32 | 29.26 | 34.3 | 18.685 |
| 6.2 | 5.32 | 29.255 | 34.3 | 18.775 |
| 6.2 | 5.31 | 29.251 | 34.4 | 18.831 |
| 6.2 | 5.3 | 29.247 | 34.4 | 18.871 |
| 6.2 | 5.29 | 29.247 | 34.3 | 20.133 |
| 6.2 | 5.28 | 29.246 | 34.3 | 20.136 |
| 6.19 | 5.27 | 29.244 | 34.3 | 20.148 |
| 6.19 | 5.27 | 29.245 | 34.4 | 20.159 |
| 6.18 | 5.27 | 29.244 | 34.4 | 21.108 |
| 6.17 | 5.26 | 29.236 | 34.3 | 21.121 |

| | | | | |
|---|---|---|---|---|
| 6.17 | 5.24 | 29.226 | 34.3 | 21.126 |
| 6.16 | 5.22 | 29.223 | 34.4 | 21.159 |
| 6.15 | 5.22 | 29.229 | 34.4 | 21.352 |
| 6.15 | 5.22 | 29.236 | 34.3 | 21.465 |
| 6.15 | 5.22 | 29.241 | 34.3 | 21.535 |
| 6.16 | 5.23 | 29.243 | 34.4 | 21.584 |
| 6.17 | 5.23 | 29.243 | 34.4 | 21.626 |
| 6.18 | 5.23 | 29.243 | 34.4 | 22.544 |
| 6.18 | 5.24 | 29.242 | 34.4 | 22.276 |
| 6.19 | 5.24 | 29.241 | 34.3 | 22.274 |
| 6.18 | 5.24 | 29.241 | 34.3 | 22.43 |
| 6.17 | 5.24 | 29.237 | 34.3 | 22.557 |
| 6.17 | 5.23 | 29.235 | 34.3 | 22.626 |
| 6.17 | 5.23 | 29.235 | 34.3 | 22.683 |
| 6.16 | 5.22 | 29.235 | 34.3 | 23.534 |
| 6.16 | 5.22 | 29.231 | 34.3 | 23.52 |
| 6.16 | 5.21 | 29.228 | 34.3 | 23.52 |
| 6.16 | 5.21 | 29.226 | 34.3 | 23.521 |
| 6.16 | 5.2 | 29.223 | 34.4 | 23.525 |
| 6.18 | 5.19 | 29.221 | 34.4 | 23.531 |
| 6.19 | 5.19 | 29.22 | 34.4 | 23.534 |
| 6.2 | 5.18 | 29.219 | 34.3 | 24.404 |
| 6.19 | 5.18 | 29.215 | 34.3 | 24.386 |
| 6.18 | 5.17 | 29.208 | 34.3 | 24.399 |
| 6.17 | 5.15 | 29.202 | 34.3 | 24.399 |
| 6.16 | 5.14 | 29.193 | 34.3 | 24.435 |
| 6.15 | 5.09 | 29.173 | 34.3 | 24.571 |
| 6.14 | 5.04 | 29.153 | 34.3 | 24.651 |
| 6.14 | 4.99 | 29.152 | 34.3 | 24.704 |
| 6.14 | 4.97 | 29.159 | 34.3 | 24.742 |
| 6.15 | 4.97 | 29.168 | 34.3 | 24.818 |
| 6.15 | 4.98 | 29.167 | 34.3 | 25.583 |
| 6.15 | 4.99 | 29.16 | 34.3 | 25.592 |
| 6.15 | 4.98 | 29.153 | 34.3 | 25.591 |
| 6.14 | 4.97 | 29.153 | 34.3 | 25.591 |
| 6.13 | 4.97 | 29.154 | 34.3 | 25.678 |
| 6.13 | 4.97 | 29.156 | 34.3 | 25.791 |
| 6.13 | 4.97 | 29.157 | 34.3 | 25.866 |
| 6.13 | 4.97 | 29.157 | 34.3 | 26.841 |
| 6.11 | 4.97 | 29.155 | 34.3 | 26.87 |
| 6.11 | 4.97 | 29.153 | 34.3 | 26.877 |
| 6.11 | 4.96 | 29.152 | 34.3 | 26.879 |
| 6.1 | 4.96 | 29.145 | 34.3 | 27.007 |
| 6.09 | 4.94 | 29.134 | 34.3 | 27.124 |
| 6.09 | 4.91 | 29.126 | 34.2 | 27.193 |

| 6.1 | 4.89 | 29.129 | 34.2 | 27.242 |
|---|---|---|---|---|
| 6.11 | 4.88 | 29.138 | 34.2 | 27.279 |
| 6.11 | 4.89 | 29.139 | 34.2 | 28.141 |
| 6.11 | 4.9 | 29.133 | 34.2 | 28.143 |
| 6.12 | 4.89 | 29.129 | 34.2 | 28.142 |
| 6.11 | 4.88 | 29.129 | 34.2 | 28.269 |
| 6.11 | 4.87 | 29.128 | 34.2 | 28.402 |
| 6.11 | 4.87 | 29.128 | 34.2 | 28.479 |
| 6.1 | 4.87 | 29.127 | 34.2 | 29.464 |
| 6.1 | 4.86 | 29.126 | 34.2 | 29.465 |
| 6.1 | 4.86 | 29.125 | 34.2 | 29.458 |
| 6.11 | 4.86 | 29.126 | 34.2 | 29.457 |
| 6.1 | 4.86 | 29.126 | 34.2 | 29.606 |
| 6.11 | 4.86 | 29.126 | 34.2 | 29.696 |
| 6.12 | 4.86 | 29.127 | 34.2 | 30.576 |
| 6.11 | 4.86 | 29.127 | 34.2 | 30.67 |
| 6.1 | 4.86 | 29.126 | 34.2 | 30.666 |
| 6.11 | 4.86 | 29.126 | 34.2 | 30.678 |
| 6.1 | 4.85 | 29.124 | 34.2 | 30.868 |
| 6.1 | 4.85 | 29.125 | 34.2 | 30.97 |
| 6.11 | 4.85 | 29.126 | 34.2 | 31.035 |
| 6.1 | 4.85 | 29.128 | 34.2 | 31.929 |
| 6.1 | 4.86 | 29.129 | 34.2 | 31.922 |
| 6.11 | 4.86 | 29.131 | 34.2 | 31.907 |
| 6.11 | 4.87 | 29.132 | 34.2 | 31.968 |
| 6.11 | 4.87 | 29.133 | 34.2 | 32.097 |
| 6.12 | 4.87 | 29.133 | 34.2 | 32.175 |
| 6.13 | 4.87 | 29.134 | 34.2 | 32.223 |
| 6.13 | 4.88 | 29.135 | 34.2 | 33.223 |
| 6.13 | 4.88 | 29.135 | 34.2 | 33.251 |
| 6.13 | 4.88 | 29.134 | 34.2 | 33.254 |
| 6.13 | 4.87 | 29.129 | 34.2 | 33.253 |
| 6.13 | 4.85 | 29.122 | 34.2 | 33.256 |
| 6.14 | 4.84 | 29.119 | 34.2 | 33.262 |
| 6.15 | 4.83 | 29.123 | 34.2 | 33.269 |
| 6.15 | 4.84 | 29.129 | 34.2 | 34.022 |
| 6.15 | 4.85 | 29.133 | 34.2 | 34.034 |
| 6.15 | 4.87 | 29.134 | 34.2 | 34.045 |
| 6.15 | 4.87 | 29.133 | 34.2 | 34.094 |
| 6.14 | 4.86 | 29.129 | 34.2 | 34.263 |
| 6.14 | 4.85 | 29.125 | 34.2 | 34.373 |
| 6.14 | 4.84 | 29.124 | 34.2 | 34.448 |
| 6.14 | 4.83 | 29.123 | 34.2 | 35.556 |
| 6.13 | 4.83 | 29.12 | 34.2 | 35.61 |
| 6.13 | 4.82 | 29.115 | 34.2 | 35.606 |

| | | | | |
|------|------|--------|------|--------|
| 6.14 | 4.8  | 29.113 | 34.2 | 35.607 |
| 6.14 | 4.8  | 29.114 | 34.2 | 35.611 |
| 6.14 | 4.8  | 29.116 | 34.2 | 36.535 |
| 6.14 | 4.8  | 29.118 | 34.2 | 36.541 |
| 6.14 | 4.8  | 29.118 | 34.2 | 36.555 |
| 6.14 | 4.8  | 29.117 | 34.2 | 36.564 |
| 6.14 | 4.8  | 29.118 | 34.2 | 37.415 |
| 6.14 | 4.81 | 29.119 | 34.2 | 37.423 |
| 6.14 | 4.81 | 29.122 | 34.2 | 37.425 |
| 6.15 | 4.81 | 29.122 | 34.2 | 37.457 |
| 6.14 | 4.81 | 29.12  | 34.2 | 38.396 |
| 6.14 | 4.81 | 29.117 | 34.2 | 38.392 |
| 6.15 | 4.8  | 29.115 | 34.2 | 38.395 |
| 6.14 | 4.79 | 29.114 | 34.2 | 39.22  |
| 6.14 | 4.78 | 29.112 | 34.2 | 39.212 |
| 6.14 | 4.78 | 29.112 | 34.2 | 39.216 |
| 6.13 | 4.78 | 29.112 | 34.2 | 40.042 |
| 6.12 | 4.77 | 29.113 | 34.2 | 40.043 |
| 6.13 | 4.77 | 29.113 | 34.2 | 40.045 |
| 6.13 | 4.78 | 29.114 | 34.2 | 40.048 |
| 6.13 | 4.78 | 29.114 | 34.2 | 40.836 |
| 6.13 | 4.78 | 29.114 | 34.2 | 40.841 |
| 6.14 | 4.78 | 29.114 | 34.2 | 40.842 |
| 6.14 | 4.78 | 29.114 | 34.2 | 40.845 |
| 6.15 | 4.78 | 29.114 | 34.2 | 40.848 |
| 6.17 | 4.78 | 29.114 | 34.2 | 40.851 |
| 6.17 | 4.78 | 29.113 | 34.2 | 41.627 |
| 6.17 | 4.77 | 29.113 | 34.2 | 41.635 |
| 6.17 | 4.77 | 29.113 | 34.2 | 41.632 |
| 6.17 | 4.77 | 29.113 | 34.2 | 42.365 |
| 6.16 | 4.77 | 29.112 | 34.2 | 42.365 |
| 6.16 | 4.77 | 29.111 | 34.2 | 42.363 |
| 6.15 | 4.76 | 29.11  | 34.2 | 43.138 |
| 6.14 | 4.76 | 29.11  | 34.2 | 43.129 |
| 6.14 | 4.76 | 29.111 | 34.2 | 43.119 |
| 6.14 | 4.76 | 29.112 | 34.2 | 43.109 |
| 6.14 | 4.76 | 29.112 | 34.2 | 43.835 |
| 6.15 | 4.76 | 29.112 | 34.2 | 43.826 |
| 6.15 | 4.76 | 29.112 | 34.2 | 43.817 |
| 6.16 | 4.76 | 29.112 | 34.2 | 43.814 |
| 6.15 | 4.76 | 29.109 | 34.2 | 44.56  |
| 6.15 | 4.75 | 29.106 | 34.2 | 44.544 |
| 6.15 | 4.75 | 29.105 | 34.2 | 44.545 |
| 6.14 | 4.74 | 29.105 | 34.2 | 45.354 |
| 6.14 | 4.74 | 29.105 | 34.2 | 45.403 |

| | | | | |
|------|------|--------|------|--------|
| 6.14 | 4.74 | 29.104 | 34.2 | 45.409 |
| 6.13 | 4.74 | 29.103 | 34.2 | 45.453 |
| 6.12 | 4.73 | 29.099 | 34.2 | 45.609 |
| 6.13 | 4.72 | 29.097 | 34.2 | 45.696 |
| 6.13 | 4.72 | 29.096 | 34.2 | 46.471 |
| 6.13 | 4.71 | 29.096 | 34.2 | 46.486 |
| 6.13 | 4.71 | 29.096 | 34.2 | 46.483 |
| 6.13 | 4.71 | 29.096 | 34.2 | 46.487 |
| 6.13 | 4.71 | 29.096 | 34.2 | 46.623 |
| 6.14 | 4.71 | 29.096 | 34.2 | 46.699 |
| 6.14 | 4.71 | 29.096 | 34.2 | 46.746 |
| 6.14 | 4.71 | 29.096 | 34.2 | 46.85  |
| 6.14 | 4.7  | 29.096 | 34.2 | 46.932 |
| 6.15 | 4.7  | 29.096 | 34.2 | 47.055 |
| 6.15 | 4.7  | 29.096 | 34.2 | 47.17  |
| 6.15 | 4.7  | 29.096 | 34.2 | 47.979 |
| 6.16 | 4.7  | 29.095 | 34.2 | 47.981 |
| 6.16 | 4.7  | 29.096 | 34.2 | 48.07  |
| 6.16 | 4.7  | 29.096 | 34.2 | 48.261 |
| 6.16 | 4.7  | 29.096 | 34.2 | 48.357 |
| 6.16 | 4.7  | 29.097 | 34.2 | 48.414 |
| 6.15 | 4.7  | 29.096 | 34.2 | 49.271 |
| 6.16 | 4.7  | 29.096 | 34.2 | 49.255 |
| 6.16 | 4.7  | 29.096 | 34.2 | 49.241 |
| 6.16 | 4.7  | 29.096 | 34.2 | 49.385 |
| 6.16 | 4.7  | 29.096 | 34.2 | 49.518 |
| 6.17 | 4.7  | 29.095 | 34.2 | 49.584 |
| 6.17 | 4.7  | 29.095 | 34.2 | 50.497 |
| 6.17 | 4.69 | 29.095 | 34.2 | 50.478 |
| 6.17 | 4.69 | 29.094 | 34.2 | 50.479 |
| 6.17 | 4.69 | 29.094 | 34.2 | 50.607 |
| 6.16 | 4.69 | 29.093 | 34.2 | 50.745 |
| 6.16 | 4.68 | 29.093 | 34.1 | 50.818 |
| 6.16 | 4.68 | 29.093 | 34.2 | 50.872 |
| 6.16 | 4.68 | 29.094 | 34.2 | 51.757 |
| 6.16 | 4.67 | 29.093 | 34.2 | 51.77  |
| 6.16 | 4.67 | 29.092 | 34.2 | 51.802 |
| 6.16 | 4.66 | 29.092 | 34.2 | 51.821 |
| 6.17 | 4.66 | 29.092 | 34.2 | 51.831 |
| 6.18 | 4.66 | 29.093 | 34.2 | 51.838 |
| 6.18 | 4.66 | 29.093 | 34.2 | 52.623 |
| 6.18 | 4.66 | 29.092 | 34.2 | 52.573 |
| 6.19 | 4.66 | 29.092 | 34.2 | 52.578 |
| 6.19 | 4.66 | 29.092 | 34.2 | 52.58  |
| 6.19 | 4.66 | 29.092 | 34.2 | 52.581 |

| | | | | |
|------|------|--------|------|--------|
| 6.2 | 4.66 | 29.092 | 34.2 | 52.585 |
| 6.2 | 4.66 | 29.093 | 34.2 | 52.591 |
| 6.21 | 4.66 | 29.092 | 34.2 | 52.598 |
| 6.21 | 4.66 | 29.092 | 34.2 | 52.606 |
| 6.22 | 4.66 | 29.092 | 34.2 | 52.617 |
| 6.22 | 4.65 | 29.092 | 34.2 | 52.629 |
| 6.23 | 4.65 | 29.092 | 34.2 | 52.64 |
| 6.23 | 4.65 | 29.093 | 34.2 | 52.652 |
| 6.24 | 4.65 | 29.093 | 34.2 | 52.664 |
| 6.24 | 4.65 | 29.093 | 34.2 | 52.675 |
| 6.25 | 4.66 | 29.092 | 34.2 | 52.686 |
| 6.25 | 4.66 | 29.093 | 34.2 | 52.694 |
| 6.25 | 4.66 | 29.093 | 34.2 | 52.7 |
| 6.26 | 4.66 | 29.093 | 34.2 | 52.706 |
| 6.26 | 4.66 | 29.093 | 34.2 | 52.713 |
| 6.26 | 4.66 | 29.093 | 34.2 | 52.717 |
| 6.26 | 4.66 | 29.093 | 34.2 | 52.718 |
| 6.26 | 4.66 | 29.093 | 34.1 | 52.716 |
| 6.27 | 4.66 | 29.092 | 34.1 | 52.715 |
| 6.27 | 4.66 | 29.092 | 34.2 | 53.507 |
| 6.26 | 4.65 | 29.092 | 34.2 | 53.535 |
| 6.26 | 4.65 | 29.092 | 34.2 | 53.547 |
| 6.26 | 4.65 | 29.091 | 34.2 | 53.691 |
| 6.25 | 4.65 | 29.089 | 34.2 | 53.833 |
| 6.25 | 4.64 | 29.087 | 34.2 | 53.91 |
| 6.25 | 4.63 | 29.087 | 34.2 | 53.959 |
| 6.24 | 4.61 | 29.085 | 34.2 | 54.75 |
| 6.24 | 4.59 | 29.083 | 34.2 | 54.752 |
| 6.24 | 4.57 | 29.08 | 34.2 | 54.76 |
| 6.23 | 4.56 | 29.079 | 34.2 | 54.82 |
| 6.22 | 4.54 | 29.078 | 34.2 | 55.759 |
| 6.22 | 4.52 | 29.076 | 34.2 | 55.795 |
| 6.22 | 4.51 | 29.075 | 34.2 | 55.794 |
| 6.22 | 4.5 | 29.075 | 34.2 | 55.889 |
| 6.21 | 4.48 | 29.073 | 34.3 | 56.027 |
| 6.21 | 4.46 | 29.072 | 34.3 | 56.108 |
| 6.21 | 4.44 | 29.072 | 34.3 | 56.932 |
| 6.2 | 4.42 | 29.072 | 34.3 | 56.951 |
| 6.2 | 4.4 | 29.071 | 34.3 | 56.955 |
| 6.19 | 4.39 | 29.071 | 34.3 | 57.007 |
| 6.19 | 4.38 | 29.07 | 34.4 | 57.163 |
| 6.19 | 4.37 | 29.069 | 34.4 | 57.264 |
| 6.19 | 4.33 | 29.068 | 34.6 | 57.335 |
| 6.2 | 4.25 | 29.068 | 34.7 | 57.295 |
| 6.22 | 4.22 | 29.069 | 34.7 | 57.338 |

| | | | | |
|------|------|--------|------|--------|
| 6.23 | 4.26 | 29.07 | 34.6 | 57.438 |
| 6.25 | 4.29 | 29.069 | 34.6 | 57.534 |
| 6.25 | 4.29 | 29.069 | 35 | 57.585 |
| 6.35 | 4.24 | 29.068 | 36.6 | 57.611 |
| 6.38 | 3.89 | 29.067 | 39.8 | 57.626 |
| 6.39 | 3.66 | 29.068 | 44.4 | 57.684 |
| 6.41 | 3.54 | 29.069 | 47.8 | 57.742 |
| 6.41 | 3.59 | 29.071 | 48.4 | 57.744 |
| 6.42 | 3.65 | 29.074 | 47 | 57.743 |
| 6.42 | 3.77 | 29.077 | 45.2 | 57.743 |
| 6.41 | 3.73 | 29.075 | 43.5 | 57.735 |
| 6.41 | 3.63 | 29.074 | 42.4 | 57.709 |
| 6.52 | 3.54 | 29.071 | 42.2 | 57.68 |
| 6.59 | 2.31 | 29.068 | 43.5 | 57.67 |
| 6.58 | 1.32 | 29.065 | 46.7 | 57.664 |
| 6.58 | 1.08 | 29.063 | 50.7 | 57.654 |
| 6.59 | 1.55 | 29.066 | 50.7 | 57.642 |
| 6.59 | 1.04 | 29.07 | 50.7 | 57.648 |
| 6.58 | 0.76 | 29.07 | 52.4 | 57.659 |
| 6.57 | 0.66 | 29.069 | 55.7 | 57.673 |
| 6.55 | 0.59 | 29.068 | 57.7 | 57.689 |
| 6.52 | 0.49 | 29.065 | 61.4 | 57.69 |
| 6.5 | 0.42 | 29.064 | 63.7 | 57.69 |
| 6.49 | 0.71 | 29.067 | 60.6 | 57.691 |
| 6.58 | 1.12 | 29.077 | 58.2 | 57.697 |
| 6.53 | 0.57 | 29.082 | 59.6 | 57.699 |
| 6.52 | 0.46 | 29.082 | 64.9 | 57.703 |
| 6.51 | 0.41 | 29.079 | 66.7 | 57.702 |
| 6.5 | 0.36 | 29.079 | 65.5 | 57.692 |
| 6.49 | 0.32 | 29.082 | 63.5 | 57.687 |
| 6.48 | 0.3 | 29.085 | 64 | 57.698 |
| 6.47 | 0.28 | 29.086 | 66.2 | 57.709 |
| 6.46 | 0.72 | 29.085 | 67.7 | 57.703 |
| 6.77 | 0.8 | 29.08 | 69.7 | 57.695 |
| 6.84 | 1.44 | 29.074 | 66.2 | 57.683 |
| 6.72 | 2.38 | 29.068 | 56.5 | 57.665 |
| 6.7 | 2.06 | 29.067 | 44.3 | 57.668 |
| 6.72 | 2.41 | 29.066 | 44 | 57.666 |
| 6.72 | 2.67 | 29.065 | 43.8 | 57.694 |
| 6.72 | 3.44 | 29.067 | 40 | 57.711 |
| 6.72 | 3.64 | 29.068 | 37.4 | 57.75 |
| 6.61 | 3.69 | 29.067 | 38.2 | 57.833 |
| 6.58 | 3.69 | 29.066 | 40.5 | 57.924 |

107

108    Table S27 – P19 site depth profile from 2017 campaign.

| pH | ODO mg/L | Temp °C | Cond µS/cm | Depth m |
|---|---|---|---|---|
| 6.52 | 5.2 | 29.431 | 34.5 | 0.344 |
| 6.52 | 5.2 | 29.432 | 34.5 | 0.388 |
| 6.52 | 5.19 | 29.43 | 34.4 | 0.43 |
| 6.52 | 5.19 | 29.428 | 34.3 | 0.494 |
| 6.51 | 5.19 | 29.429 | 34.2 | 0.548 |
| 6.51 | 5.18 | 29.43 | 34.2 | 0.619 |
| 6.51 | 5.18 | 29.432 | 34.1 | 0.677 |
| 6.51 | 5.18 | 29.44 | 34 | 0.752 |
| 6.51 | 5.18 | 29.45 | 33.9 | 0.837 |
| 6.51 | 5.19 | 29.459 | 33.8 | 0.914 |
| 6.51 | 5.19 | 29.458 | 33.7 | 0.982 |
| 6.51 | 5.19 | 29.458 | 33.7 | 1.045 |
| 6.51 | 5.19 | 29.458 | 33.7 | 1.106 |
| 6.51 | 5.19 | 29.459 | 33.6 | 1.171 |
| 6.51 | 5.18 | 29.457 | 33.6 | 1.246 |
| 6.51 | 5.18 | 29.457 | 33.5 | 1.322 |
| 6.51 | 5.18 | 29.452 | 33.5 | 1.39 |
| 6.51 | 5.18 | 29.446 | 33.5 | 1.475 |
| 6.51 | 5.18 | 29.437 | 33.5 | 1.575 |
| 6.5 | 5.18 | 29.434 | 33.5 | 1.659 |
| 6.5 | 5.18 | 29.432 | 33.4 | 1.755 |
| 6.49 | 5.17 | 29.432 | 33.4 | 1.832 |
| 6.5 | 5.17 | 29.433 | 33.4 | 1.923 |
| 6.5 | 5.17 | 29.437 | 33.4 | 2.004 |
| 6.5 | 5.17 | 29.44 | 33.5 | 2.08 |
| 6.5 | 5.17 | 29.44 | 33.5 | 2.147 |
| 6.5 | 5.17 | 29.435 | 33.5 | 2.222 |
| 6.5 | 5.17 | 29.43 | 33.5 | 2.289 |
| 6.51 | 5.17 | 29.428 | 33.5 | 2.352 |
| 6.51 | 5.17 | 29.432 | 33.5 | 2.433 |
| 6.51 | 5.17 | 29.433 | 33.6 | 2.507 |
| 6.5 | 5.17 | 29.429 | 33.6 | 2.589 |
| 6.5 | 5.17 | 29.42 | 33.6 | 2.682 |
| 6.49 | 5.17 | 29.414 | 33.6 | 2.778 |
| 6.49 | 5.16 | 29.412 | 33.7 | 2.873 |
| 6.49 | 5.16 | 29.414 | 33.7 | 2.961 |
| 6.5 | 5.16 | 29.415 | 33.7 | 3.042 |
| 6.5 | 5.16 | 29.415 | 33.8 | 3.12 |
| 6.51 | 5.16 | 29.417 | 33.8 | 3.201 |
| 6.51 | 5.16 | 29.419 | 33.8 | 3.293 |
| 6.5 | 5.16 | 29.417 | 33.9 | 3.408 |
| 6.5 | 5.16 | 29.414 | 33.9 | 4.269 |

| | | | | |
|---|---|---|---|---|
| 6.49 | 5.16 | 29.412 | 33.9 | 4.424 |
| 6.49 | 5.16 | 29.412 | 34 | 4.503 |
| 6.48 | 5.16 | 29.412 | 34 | 4.588 |
| 6.48 | 5.16 | 29.412 | 34 | 4.659 |
| 6.48 | 5.16 | 29.412 | 34.1 | 4.782 |
| 6.48 | 5.16 | 29.413 | 34.1 | 4.867 |
| 6.47 | 5.16 | 29.414 | 34.1 | 4.974 |
| 6.47 | 5.16 | 29.415 | 34.2 | 5.062 |
| 6.48 | 5.16 | 29.415 | 34.2 | 5.232 |
| 6.48 | 5.16 | 29.415 | 34.2 | 5.397 |
| 6.48 | 5.16 | 29.415 | 34.2 | 5.561 |
| 6.48 | 5.16 | 29.415 | 34.3 | 5.703 |
| 6.48 | 5.16 | 29.414 | 34.3 | 5.836 |
| 6.48 | 5.16 | 29.412 | 34.3 | 5.969 |
| 6.48 | 5.15 | 29.413 | 34.3 | 6.107 |
| 6.48 | 5.15 | 29.413 | 34.4 | 6.233 |
| 6.48 | 5.15 | 29.414 | 34.4 | 7.064 |
| 6.49 | 5.15 | 29.414 | 34.4 | 7.166 |
| 6.49 | 5.15 | 29.414 | 34.4 | 7.266 |
| 6.49 | 5.15 | 29.414 | 34.4 | 7.353 |
| 6.5 | 5.15 | 29.414 | 34.4 | 7.454 |
| 6.5 | 5.15 | 29.413 | 34.5 | 7.516 |
| 6.49 | 5.15 | 29.412 | 34.5 | 7.601 |
| 6.49 | 5.15 | 29.413 | 34.5 | 8.45 |
| 6.48 | 5.14 | 29.413 | 34.5 | 8.558 |
| 6.48 | 5.14 | 29.413 | 34.5 | 8.638 |
| 6.48 | 5.14 | 29.412 | 34.5 | 8.777 |
| 6.48 | 5.14 | 29.411 | 34.6 | 8.901 |
| 6.49 | 5.14 | 29.412 | 34.6 | 9.042 |
| 6.49 | 5.14 | 29.412 | 34.6 | 9.893 |

109

110    Table S28 – P20 site depth profile from 2017 campaign.

| pH | ODO mg/L | Temp °C | Cond µS/cm | Depth m |
|---|---|---|---|---|
| 7.01 | 7.98 | 30.036 | 37.5 | 0.433 |
| 7.03 | 7.98 | 30.034 | 37.5 | 0.523 |
| 7.04 | 7.98 | 30.033 | 37.6 | 0.617 |
| 7.05 | 7.98 | 30.032 | 37.6 | 0.719 |
| 7.06 | 7.98 | 30.032 | 37.6 | 0.822 |
| 7.07 | 7.98 | 30.032 | 37.6 | 0.92 |
| 7.07 | 7.99 | 30.033 | 37.6 | 1.012 |
| 7.08 | 7.99 | 30.033 | 37.6 | 1.098 |
| 7.08 | 7.99 | 30.034 | 37.5 | 1.185 |

| | | | | |
|---|---|---|---|---|
| 7.08 | 8 | 30.034 | 37.5 | 1.279 |
| 7.09 | 8 | 30.036 | 37.5 | 1.378 |
| 7.09 | 8 | 30.037 | 37.4 | 1.469 |
| 7.09 | 8 | 30.035 | 37.3 | 1.56 |
| 7.09 | 8 | 30.029 | 37.2 | 1.653 |
| 7.09 | 8 | 30.027 | 37.2 | 1.743 |
| 7.09 | 8 | 30.026 | 37.1 | 1.832 |
| 7.09 | 8 | 30.026 | 37 | 1.919 |
| 7.09 | 8 | 30.026 | 36.9 | 1.986 |
| 7.09 | 8 | 30.027 | 36.9 | 2.039 |
| 7.09 | 8 | 30.028 | 36.8 | 2.101 |
| 7.09 | 8 | 30.027 | 36.8 | 2.162 |
| 7.09 | 8.01 | 30.025 | 36.8 | 2.236 |
| 7.09 | 8.01 | 30.024 | 36.7 | 2.332 |
| 7.09 | 8 | 30.024 | 36.7 | 2.415 |
| 7.09 | 8 | 30.024 | 36.7 | 2.508 |
| 7.09 | 8 | 30.023 | 36.7 | 2.63 |
| 7.09 | 8 | 30.023 | 36.7 | 2.767 |
| 7.09 | 8 | 30.023 | 36.7 | 2.921 |
| 7.09 | 8.01 | 30.023 | 36.7 | 3.77 |
| 7.09 | 8.01 | 30.023 | 36.7 | 3.916 |
| 7.09 | 8 | 30.023 | 36.7 | 3.944 |
| 7.09 | 8 | 30.024 | 36.7 | 4.001 |
| 7.08 | 8.01 | 30.025 | 36.7 | 4.045 |
| 7.08 | 8.01 | 30.027 | 36.8 | 4.086 |
| 7.08 | 8.01 | 30.03 | 36.8 | 4.137 |
| 7.08 | 8.01 | 30.033 | 36.8 | 4.189 |
| 7.08 | 8.02 | 30.034 | 36.8 | 4.241 |
| 7.09 | 8.02 | 30.033 | 36.9 | 4.345 |
| 7.09 | 8.02 | 30.031 | 36.9 | 4.443 |
| 7.09 | 8.01 | 30.031 | 36.9 | 4.539 |
| 7.09 | 8.01 | 30.03 | 37 | 4.635 |
| 7.09 | 8.01 | 30.029 | 37 | 4.722 |
| 7.09 | 8.01 | 30.026 | 37 | 4.803 |
| 7.09 | 8.01 | 30.025 | 37.1 | 4.891 |
| 7.09 | 8.01 | 30.024 | 37.1 | 4.966 |
| 7.08 | 8.02 | 30.024 | 37.2 | 5.054 |
| 7.08 | 8.02 | 30.025 | 37.2 | 5.145 |
| 7.08 | 8.01 | 30.026 | 37.3 | 5.272 |
| 7.08 | 8.01 | 30.026 | 37.3 | 6.297 |
| 7.08 | 8.02 | 30.025 | 37.3 | 6.392 |
| 7.08 | 8.02 | 30.027 | 37.4 | 6.333 |
| 7.09 | 8.02 | 30.029 | 37.4 | 6.317 |
| 7.09 | 8.02 | 30.028 | 37.5 | 6.318 |
| 7.09 | 8.02 | 30.025 | 37.5 | 6.33 |

| | | | | |
|------|------|--------|------|--------|
| 7.09 | 8.02 | 30.024 | 37.5 | 6.354 |
| 7.09 | 8.02 | 30.023 | 37.6 | 6.382 |
| 7.1  | 8.02 | 30.025 | 37.6 | 6.415 |
| 7.1  | 8.02 | 30.025 | 37.7 | 6.457 |
| 7.09 | 8.02 | 30.025 | 37.7 | 6.494 |
| 7.09 | 8.02 | 30.025 | 37.7 | 6.51  |
| 7.1  | 8.02 | 30.026 | 37.7 | 6.509 |
| 7.1  | 8.02 | 30.027 | 37.8 | 6.498 |
| 7.1  | 8.03 | 30.025 | 37.8 | 6.523 |
| 7.1  | 8.02 | 30.023 | 37.8 | 6.594 |
| 7.09 | 8.02 | 30.021 | 37.9 | 6.68  |
| 7.09 | 8.02 | 30.022 | 37.9 | 7.487 |
| 7.09 | 8.02 | 30.023 | 37.9 | 7.491 |
| 7.09 | 8.02 | 30.022 | 37.9 | 7.581 |
| 7.09 | 8.02 | 30.021 | 37.9 | 7.714 |
| 7.09 | 8.02 | 30.02  | 38   | 7.822 |
| 7.1  | 8.01 | 30.02  | 38   | 7.893 |
| 7.1  | 8.01 | 30.02  | 38   | 7.947 |
| 7.1  | 8.01 | 30.02  | 38   | 8.004 |
| 7.1  | 8.01 | 30.02  | 38   | 8.129 |
| 7.1  | 8.01 | 30.02  | 38.1 | 8.274 |
| 7.1  | 8.01 | 30.02  | 38.1 | 8.382 |
| 7.1  | 8.01 | 30.021 | 38.1 | 8.461 |
| 7.09 | 8.01 | 30.021 | 38.1 | 8.552 |
| 7.08 | 8.01 | 30.019 | 38.1 | 8.656 |
| 7.07 | 8    | 30.016 | 38.1 | 8.777 |
| 7.06 | 8    | 30.015 | 38.1 | 8.895 |
| 7.05 | 8    | 30.014 | 38.1 | 9     |
| 7.05 | 8    | 30.013 | 38.1 | 9.11  |
| 7.05 | 8    | 30.012 | 38.1 | 9.233 |
| 7.06 | 8    | 30.011 | 38.1 | 9.369 |
| 7.07 | 7.99 | 30.012 | 38.1 | 9.496 |
| 7.07 | 7.99 | 30.011 | 38.1 | 9.632 |
| 7.06 | 7.99 | 30.011 | 38.2 | 9.753 |
| 7.05 | 7.99 | 30.012 | 38.2 | 9.872 |
| 7.04 | 8    | 30.014 | 38.2 | 10.008 |
| 7.03 | 8    | 30.015 | 38.2 | 10.144 |
| 7.02 | 8    | 30.014 | 38.2 | 10.268 |
| 7.01 | 8    | 30.014 | 38.2 | 10.383 |
| 7    | 7.99 | 30.014 | 38.2 | 10.487 |
| 6.99 | 7.99 | 30.013 | 38.2 | 10.595 |
| 6.97 | 7.99 | 30.014 | 38.2 | 10.717 |
| 6.95 | 7.99 | 30.016 | 38.2 | 10.86 |
| 6.93 | 7.99 | 30.018 | 38.2 | 11.676 |
| 6.92 | 7.99 | 30.018 | 38.2 | 11.754 |

| | | | | |
|------|------|--------|------|--------|
| 6.91 | 7.99 | 30.017 | 38.2 | 11.815 |
| 6.89 | 7.99 | 30.016 | 38.2 | 11.846 |
| 6.88 | 7.99 | 30.017 | 38.2 | 11.916 |
| 6.86 | 7.99 | 30.017 | 38.2 | 11.989 |
| 6.84 | 7.99 | 30.017 | 38.2 | 12.069 |
| 6.82 | 7.98 | 30.016 | 38.2 | 12.137 |
| 6.81 | 7.98 | 30.016 | 38.2 | 12.246 |
| 6.79 | 7.98 | 30.015 | 38.2 | 12.365 |
| 6.78 | 7.98 | 30.014 | 38.2 | 12.501 |
| 6.76 | 7.98 | 30.014 | 38.2 | 12.646 |
| 6.75 | 7.98 | 30.015 | 38.2 | 12.788 |
| 6.73 | 7.98 | 30.015 | 38.2 | 12.929 |
| 6.71 | 7.98 | 30.014 | 38.2 | 13.05 |
| 6.7 | 7.98 | 30.012 | 38.2 | 13.18 |
| 6.68 | 7.97 | 30.011 | 38.2 | 13.308 |
| 6.67 | 7.97 | 30.012 | 38.2 | 13.428 |
| 6.66 | 7.97 | 30.011 | 38.3 | 13.54 |
| 6.66 | 7.97 | 30.012 | 38.3 | 13.646 |
| 6.66 | 7.97 | 30.012 | 38.2 | 13.75 |
| 6.67 | 7.96 | 30.012 | 38.2 | 13.851 |
| 6.67 | 7.96 | 30.013 | 38.2 | 13.953 |
| 6.66 | 7.96 | 30.014 | 38.3 | 14.049 |
| 6.66 | 7.96 | 30.014 | 38.3 | 14.726 |
| 6.65 | 7.96 | 30.013 | 38.3 | 14.87 |
| 6.63 | 7.97 | 30.013 | 38.3 | 14.991 |
| 6.61 | 7.97 | 30.012 | 38.3 | 15.089 |
| 6.6 | 7.97 | 30.012 | 38.3 | 15.178 |
| 6.58 | 7.97 | 30.01 | 38.3 | 15.24 |
| 6.56 | 7.97 | 30.01 | 38.3 | 15.331 |
| 6.54 | 7.97 | 30.01 | 38.3 | 15.418 |
| 6.53 | 7.97 | 30.012 | 38.3 | 15.528 |
| 6.52 | 7.97 | 30.014 | 38.3 | 15.691 |
| 6.51 | 7.97 | 30.015 | 38.3 | 16.65 |
| 6.5 | 7.97 | 30.016 | 38.3 | 16.851 |
| 6.49 | 7.97 | 30.016 | 38.3 | 16.996 |
| 6.47 | 7.96 | 30.016 | 38.3 | 17.122 |
| 6.47 | 7.96 | 30.017 | 38.3 | 17.231 |
| 6.46 | 7.96 | 30.017 | 38.3 | 18.169 |
| 6.45 | 7.95 | 30.016 | 38.3 | 18.384 |
| 6.45 | 7.95 | 30.015 | 38.3 | 18.51 |
| 6.44 | 7.95 | 30.014 | 38.3 | 18.638 |
| 6.43 | 7.95 | 30.014 | 38.3 | 18.734 |
| 6.43 | 7.95 | 30.014 | 38.3 | 18.824 |
| 6.43 | 7.95 | 30.015 | 38.3 | 18.91 |
| 6.43 | 7.95 | 30.015 | 38.3 | 18.977 |

111

112    Table S29 – P21 site depth profile from 2017 campaign.

| pH | ODO mg/L | Temp °C | Cond µS/cm | Depth m |
|---|---|---|---|---|
| 6.83 | 6.85 | 30.073 | 36.7 | 0.117 |
| 6.82 | 6.87 | 30.074 | 36.6 | 0.161 |
| 6.81 | 6.88 | 30.082 | 36.5 | 0.207 |
| 6.82 | 6.89 | 30.08 | 36.4 | 0.246 |
| 6.81 | 6.89 | 30.073 | 36.3 | 0.269 |
| 6.81 | 6.89 | 30.071 | 36.2 | 0.296 |
| 6.81 | 6.88 | 30.066 | 36.1 | 0.329 |
| 6.81 | 6.87 | 30.053 | 36.1 | 0.362 |
| 6.81 | 6.86 | 30.036 | 36 | 0.419 |
| 6.81 | 6.86 | 30.031 | 35.9 | 0.505 |
| 6.81 | 6.86 | 30.031 | 35.9 | 0.604 |
| 6.81 | 6.86 | 30.025 | 35.8 | 0.718 |
| 6.81 | 6.86 | 30.017 | 35.8 | 0.843 |
| 6.81 | 6.86 | 30.015 | 35.8 | 0.985 |
| 6.81 | 6.86 | 30.013 | 35.7 | 1.137 |
| 6.81 | 6.86 | 30.011 | 35.7 | 1.298 |
| 6.8 | 6.86 | 30.008 | 35.7 | 1.44 |
| 6.8 | 6.85 | 30.006 | 35.7 | 1.577 |
| 6.8 | 6.85 | 30.004 | 35.7 | 1.714 |
| 6.8 | 6.85 | 30.002 | 35.7 | 1.844 |
| 6.8 | 6.85 | 30.001 | 35.7 | 1.97 |
| 6.8 | 6.84 | 30.001 | 35.7 | 2.083 |
| 6.81 | 6.84 | 30.002 | 35.7 | 2.199 |
| 6.81 | 6.84 | 30.004 | 35.8 | 2.321 |
| 6.8 | 6.84 | 30.001 | 35.8 | 2.446 |
| 6.8 | 6.84 | 29.997 | 35.8 | 3.399 |
| 6.8 | 6.84 | 29.997 | 35.8 | 3.577 |
| 6.8 | 6.84 | 29.999 | 35.9 | 3.573 |
| 6.79 | 6.84 | 30 | 35.9 | 3.625 |
| 6.79 | 6.84 | 29.999 | 35.9 | 3.699 |
| 6.78 | 6.84 | 29.998 | 35.9 | 3.772 |
| 6.78 | 6.84 | 29.996 | 36 | 3.847 |
| 6.78 | 6.84 | 29.994 | 36 | 3.929 |
| 6.78 | 6.84 | 29.993 | 36.1 | 4.026 |
| 6.78 | 6.83 | 29.992 | 36.1 | 4.735 |
| 6.78 | 6.83 | 29.992 | 36.1 | 4.741 |
| 6.78 | 6.83 | 29.992 | 36.2 | 4.753 |
| 6.78 | 6.83 | 29.992 | 36.2 | 4.776 |
| 6.79 | 6.83 | 29.997 | 36.2 | 4.824 |

| | | | | |
|---|---|---|---|---|
| 6.79 | 6.82 | 29.998 | 36.3 | 4.904 |
| 6.79 | 6.82 | 29.998 | 36.3 | 4.992 |
| 6.79 | 6.82 | 29.998 | 36.3 | 5.766 |
| 6.8 | 6.81 | 30 | 36.4 | 5.884 |
| 6.8 | 6.81 | 29.998 | 36.4 | 5.855 |
| 6.8 | 6.81 | 29.994 | 36.5 | 5.938 |
| 6.8 | 6.81 | 29.992 | 36.5 | 6.017 |
| 6.8 | 6.81 | 29.992 | 36.5 | 6.104 |
| 6.8 | 6.81 | 29.992 | 36.5 | 6.191 |
| 6.8 | 6.81 | 29.991 | 36.6 | 6.276 |
| 6.8 | 6.8 | 29.991 | 36.6 | 6.351 |
| 6.8 | 6.8 | 29.991 | 36.6 | 6.485 |
| 6.8 | 6.8 | 29.99 | 36.7 | 6.622 |
| 6.8 | 6.8 | 29.99 | 36.7 | 6.737 |
| 6.79 | 6.8 | 29.99 | 36.7 | 6.877 |
| 6.78 | 6.8 | 29.99 | 36.7 | 7.778 |
| 6.77 | 6.8 | 29.99 | 36.8 | 7.857 |
| 6.77 | 6.8 | 29.99 | 36.8 | 7.942 |
| 6.77 | 6.8 | 29.99 | 36.8 | 8.071 |
| 6.77 | 6.8 | 29.99 | 36.8 | 8.191 |
| 6.77 | 6.79 | 29.99 | 36.8 | 8.255 |
| 6.77 | 6.8 | 29.99 | 36.8 | 8.313 |
| 6.77 | 6.8 | 29.99 | 36.8 | 8.334 |
| 6.77 | 6.81 | 29.99 | 36.9 | 8.341 |
| 6.77 | 6.82 | 29.99 | 36.9 | 8.377 |
| 6.78 | 6.83 | 29.991 | 36.9 | 8.397 |
| 6.77 | 6.83 | 29.99 | 36.9 | 8.38 |
| 6.77 | 6.83 | 29.99 | 36.9 | 8.349 |
| 6.77 | 6.83 | 29.99 | 36.9 | 8.317 |
| 6.76 | 6.82 | 29.99 | 36.9 | 8.279 |
| 6.76 | 6.81 | 29.989 | 36.9 | 8.271 |
| 6.76 | 6.8 | 29.989 | 36.9 | 8.299 |
| 6.75 | 6.79 | 29.989 | 36.9 | 8.363 |

113

114

115

116

117

118

119    **3. $k_{600}$ correlation scatterplots**

120

[Figure]

121

122    Fig.1: Scatterplots between $FCO_2$ (A) and $k_{600}$ (B) as a function of wind speed. Values from
123    figure 5 (A) include high and low water seasons. Figure 5 (B) comprises only high water values
124    for statistical correlation (Spearman correlation). Rho values are located on each image left
125    superior side.

---

## Author Comment (AC2) · 24 May 2019

Dear referee 2, We would like to thank you for the enlightening comments and suggestions. Please find our answers bellow after each referee comment.

Referee 2: Review of Araujo et al. This manuscript describes the results from a 2-yr study during high and low water seasons on the Belo Monte hydropower complex that consists of two main reservoirs, one of which is defined as a run-of-river and the other as storage. The authors aimed to contrast the impact of these two reservoir types on the CO2 dynamics of the entire complex. Additionally, they contrasted CO2 dynamics across various flooded environments within the complex. The manuscript has some

nice data but is predominantly descriptive. Regardless, data in tropical reservoirs is currently necessary and it is interesting to contrast these two types of system. Not to mention the huge dispute over this massive Amazonian project. I have many suggestions for how to improve this manuscript before this paper is ready for publication.

Response: We appreciate your suggestions, which have greatly enhanced the manuscript and its potential impact.

General comments 1. Be careful with the word 'traits' in the title. It implies features that do not vary in time. Is that the focus here? Do you mean ROR vs storage, plus flooded landscapes? That would be okay then. But if that was the case then I did not get the impression enough from your discussion that that was your focus. You need to bring out your main points much more. Try focusing the research questions or objectives more narrowly. This will help you throughout the entire publication.

Response:Thank you and that was the point. Basically our goal was to define the group of characteristics that classify each reservoir as 'ROR' or 'storage'. However flooded areas are defined as one of the environments that compose the reservoirs. Therefore, our point was to relate 'trait' only with reservoir type. We agreed with your suggestions related to text structure and they were taken into account to better describe the influence of "traits" in our study.

2. Language overall needs improvement. Too many commas used. Too many sentences that are confusing (many are mentioned in specific comments below).

Response: Updated as suggested.

3. Abstract needs more quantitative results in it

Response: Thank you, this section was revised.

4. Introduction does not discuss the importance of this particular reservoir more.

Response: Agreed, the introduction was too general, especially regarding specific information about the Belo Monte complex. More details concerning Belo Monte and its controversy were added to the text as suggested by both reviewers.

5. Methods – description of how reservoirs are connected is not clear. In the map figure there appears to be a channel connecting them too. Please improve the description of how the reservoirs interact, including flow directions, which should be on your Figure 2, and individual surface areas.

Response: At the left margin of the Xingu river channel the water flow is adducted through a 28 km channel to feed the main power house (Eletrobrás, 2009). This channel links both reservoirs, since XR is located in the river channel and regulates the IR water flow. The channel description and reservoir interaction was clarified. Also, Figure 2 was updated as suggested.

6. I find section 3.1 of the results very confusing to read and absorb fully. There are a lot of numbers that are perhaps not necessary and very distracting from understanding what you are trying to describe. I would suggest a schematic to help describe the temporal (high vs low water) variability you see that also includes the spatial variability (across environments). You can use weighted markers for the various fluxes and concentrations that correspond to high and low values, if not the real values.

Response: Thank you, we appreciate this suggestion. The results section had some data not essential for addressing our hypothesis. This section was revised for conciseness and clarity. After these changes the text became clearer and we believe that a schematic figure is not needed.

7. Figure 2 – needs arrows for direction of flow.

Response: Done.

8. Figure 3 – You can make these 4 plots into just 2 in the following manner: put the white boxplots from (a) and (b) that are pCO2 in the beginning of (c) labeled 'High water' and 'Low water', and the gray boxplots that are for FCO2 in the beginning of

(d) with the same labels. Also, are the environments in c and d labeled in the proper order – from one are to another? Or does it not work like that because of the reservoir geomorphology? Either way, I would put downstream the dams on the right side since most people read left to right and you naturally think downstream to the right.

Response: Thank you for the interesting suggestion. The environments were previously organized in alphabetical order on the plots. However, we agree that it will be easier for the reader to follow the downstream orientation. Therefore, it was corrected to flow order. We added more plots to this image according referee 1 suggestion, with season separately and an additional variable (k600). The categories were also changed to "unaffected river upstream", "XR", "IR", "downstream the dams" and "unaffected river downstream".

9. Figure 4 – you need units listed for the values; direction of flow arrows would be good; and mention in caption that (a) includes 2 years of data while (b) only has one year (and list which years).

Response: Updated as suggested.

10. Figure 5 – you mention these figures in terms of stats but there are no lines on it and no equations or states in the figure caption.

Response: Both figures are related to k600 and FCO2 correlation pattern with wind data, representing the interaction of these variables. Spearman correlation is ranked and do not have mathematical model or equation, as so Rho values were added to each image. In addition since figure 3 was updated with k600 panels, as suggested by referee 1, this figure will be included in the supplement material.

11. The discussion seems like a bunch of descriptive paragraphs thrown together. It is lacking some cohesive red line to follow and it is hard to locate your main points. Perhaps you can start to fix this by using subsections. Looks like you broke it down into the following: Seasonal variability; Vertical heterogeneity; FCO2; Spatial variability;

Comparison to other reservoirs; k600; Operation. These are all just descriptions of data in reality. You want to discuss the most interesting findings of your study and then compare them with other studies. Figure out your few most important findings and try to arrange the discussion around those first. You also measured the system right after flooding, which is when emissions should be highest. This needs to be addressed in your conclusions.

Response: The discussion was rearranged and divided into subsections that we believe are now more connected with our main findings as nicely suggested. We also gave some attention to the text in order to clarify our hypothesis. In addition, we highlighted in the manuscript that measurements were made during the first years after impounding.

Specific comments

Line 16-17 – did you measure clearwater rivers yourself ? if not, then either change or delete this sentence because it makes it sound like you.

Response: Our measurements were done only on the Xingu river; therefore, we altered the sentence to clarify this issue.

Line 41 – You mention that 'inland waters' have an area of '624,000 km2' and cite who with regards to this number? This number is very small compared to the 2.5 – 5 million km2 range that actually exists for all inland waters surface area coverage. I think you mean to cite only rivers surface area with your 0.624 million km2 value so you need to be specific when you say 'inland waters' and you need a specific reference for this river surface area number. But then you cite the 1.8 – 3.8 Pg values, presumably from Drake et al. 2018 and those values are for all inland waters specifically. If you want to discuss inland waters surface area coverage total then you need to use either Downing et al. 2006, Verpoorter et al. 2014 or Messager et al. 2016 or Feng et al. 2016.

Response: Thank you, valid comment. The area number was related to Raymond et

al. 2013, and we only refer to rivers and streams. However, in this sentence our goal was really to discuss inland waters as a whole and on a global scale. The inland waters area estimate was corrected according to Downing et al. 2006 and Veerpoorter et al. 2014 and emissions based on Raymond et al. 2013, which estimate also describes inland water emission at global scale and Drake et al. 2018.

Line 45 – clean up language (e.g., don't need 'water' so many times

Response: Done.

Line 50 – should be: 'to the autochthonous respiration of OM deposited'

Response: Thank you, we have updated this sentence as suggested.

Line 54 – should explain more how the stimulation of OM decomposition via those two processes actually effects CO2 – similar to how you did in the first half of the sentence saying higher CO2 uptake

Response: Agreed, those processes were added to the text and briefly explained as suggested.

Line 66 – I believe it was actually DelSontro et al. 2010 and not 2016

Response: Absolutely, thank you. This citation was corrected.

Line 69-70 – Start a new sentence with 'Newly flooded reservoirs...' and then give examples/references of the few poorly studied reservoirs.

Response: We altered this sentence and added the reservoirs of Petit Saut an Eastmain-1 as examples, from Abril et al. 2005 and Teodoru et al. 2012, respectively.

Line 73 – should be 'variability' and not 'variation Response: Updated as suggested.

Line 73 – give the abbreviation for fluxes here '(FCO2)' that you will use the rest of the paper, and delete 'and its relevance for GHG fluxes'

Response: Thank you, properly corrected.

Line 75 – end this sentence with '..complex in eastern Amazon, a tropical region poised to gain XXX more hydropower projects in the coming decades (REF).' This puts your work into a bigger perspective at the end of your intro.

Response: Thank you, this sentence was included as suggested.

Line 83 – the 1984 study is quite old... Is there nothing newer?

Response: It is related to a classical study that classifies Amazonian rivers according to some physical chemical characteristics. Although relatively old, it is still largely used for Amazonian river classification.

Line 98-100 – in this sentence give the names of the two reservoirs after you mention them.

Response: This sentence was updated as suggested.

Line 101 – give more details about these calculates from Faria et al. 2015

Response: The residence time was calculated by the equation RT= V/ Q, where RT is the residence time in seconds, V is the reservoir volume in m3 and Q is the volumetric discharge in m3/s. To convert RT in days the value was divided by the number of seconds in a day. We altered this sentence and added this information in the text.

Line 104 – once you have given the XR abbreviation for Xingu Reservoir then use it for the rest of the paper, and do you mean 'as islands' instead of 'in islands'?

Response: Actually we meant to use 'on islands' and that was corrected. We altered also other Xingu Reservoir usages through the text to XR abbreviation.

Line 107 – 'classified' instead of 'denominated' – and this paragraph should contain the surface area of these reservoirs already

Response: Reservoirs surface areas were added to this paragraph and the sentence was corrected as suggested.

Line 115 – the residence time of the IR reservoir is still ridiculously short (1.57 days). How do you call that a storage reservoir? Still want to know the surface area of these reservoirs already

Response: There was an error in our RT calculations due to the discharge data that we used. The previous RT values were based on an environmental impact study (EIA) that estimated the highest discharge values of each reservoir. We performed new calculations using the average historic discharge series from Water Agency of Brazil database. The corrected RT of 20.2 (IR) and 3.4 days (XR) were updated in the manuscript. In addition, we added the surface areas of the IR (154 km$^2$) and XR (342 km$^2$, including the 249 km$^2$ originally occupied by the river channel).

Line 116 – should give maximum depths of the reservoirs

Response: Thank you. Maximum depths were added to this sentence.

Line 117 – why did you give the total surface area of the 2 reservoirs together? You should provide values for the two different reservoirs. If this is difficult because of the difference between rainy and dry season then state this but still give approximate values for the individual reservoirs since you are evaluating them separately.

Response: Both reservoirs areas were added as suggested. In addition, river channel area was also included.

Line 121 – what is the 25.4 km2/MW? Why should I care about this value? Give some explanation behind your reporting of this value (or don't report it).

Response: Removed.

Line 131-132 – I really do not understand your description of water depth sampling. You classified the sampling sites based on their maximum depths? Where did you measure in the water column? If a site was 10 m deep, did you sample at 3 depths? Did you sample 0.3 m, 6 m, and 9 m? Be more explicit with your description here. Why did you pick 60% of max total depth for sampling?

Response: The sampling depth method was as you describe. Our 60% depth was a mid-depth sampling point to compare to surface and bottom waters. The three depths were sampled only in deeper sites where higher water velocity variation occurs. Since water flow and topography drives pressure gradients on sediment interface that affect particulate matter transport (Huettel et al. 1996), our goal was to sample depths with organic and inorganic matter differences. In shallower sites only 60 % category was sampled. We revised this sentence to clarify the text.

Line 136 – state that the flooded areas sampled were in both reservoirs if that is the case.

Response: Thank you, updated as suggested.

Line 143 – 'according' not 'accordingly'

Response: We corrected this word.

Line 148 – what did you collect the headspace air in?

Response: The air samples were collected using 60 ml syringes. We have updated this information in the sentence.

Line 150 – how were the gas samples transferred? Via needle and syringe because the vials were pre-capped, I presume.

Response: Exactly, gas samples were transferred into evacuated vials via needle and syringe. Vials were pre-capped with the butyl rubber stoppers and sealed with aluminum crimps. The vials were evacuated immediately before transferring samples. We updated this sentence in the manuscript.

Line 154-156 – combine these two sentences into one

Response: Thank you, we have made this change.

Line 158 – if you made measurements from a drifting boat in a river, I presume you

drifted quite a bit. Did you consider this drifting distance in your measurements of flux? This is an important point. How far did you drift? You need more details regarding this sampling approach.

Response: Drifting distance was not measured during deployments. Based on visualization in Google Earth we estimate that the maximum distance drifted may be approximately 1 km for measurements in the river channel up and downstream of the reservoirs. In sheltered areas located in bays and over islands with standing trees, where the water flow was very low, drifting was very short and caused by wind. An estimate of the drifting distance in the natural river channel and in the main channel of the Xingu Reservoir was obtained by using the average water velocity measured by the National Water Agency of Brazil at the Altamira station. We separated the historical values into before and after 2016, when the dams was completed. Therefore, representing estimates of water velocity in the natural river (between 2005 and 2016), and in the Xingu Reservoir main channel (after 2016). The average water velocities at Altamira are 0.74 and 0.24 m s-1 for before and after the dam, respectively. Assuming that there is no resistance of the boat with the water or air, drifting speed is similar to the water velocity. The total time of deployment was up to 30 minutes for the three consecutive measurements. Based on these we found that in the main channel of the Xingu Reservoir the drifting distance would be 432 m, and 1332 m for the natural river channel up and downstream the reservoirs. These details were added to supplementary material.

Line 161 – 'calculated' instead of 'done' and delete 'the eq. (1)'

Response: Thank you, done.

Line 168 – use 'erroneous' instead of 'same sampling site'

Response: Thank you, updated as suggested.

Line 171 and eq. 2 – you say that k was based on the flux measurements but I do not

see them in equation 2. I guess it is somehow in the partial pressure measurements since some are in the chamber but I think this needs a better explanation. You didn't find k using FCO2, but rather using the concentrations in the chamber? That is how I perceive this equation.

Response: Thank you, that is correct. The calculations were not made with fluxes, but with the CO2 partial pressures inside the chamber. We corrected this sentence in the manuscript.

Line 176 – need 'respectively' at the end of the sentence

Response: We have altered this sentence.

Line 177 – grammar is poor here

Response: Thank you, sentence rewritten.

Line 184 – give a bit more detail here about how the gas transfer velocities were not calculated from 2016 data. I am guessing it is because the other loggers did not allow it somehow, but I don't see why you couldn't perform the calculations using concentrations from those loggers too.

Response: The gas transfer velocities were not calculated from 2016 data due lack of headspace sampling in this campaign. The only season without k600 was 2016 high water, the calculations were made using loggers concentrations too. This sentence was removed to a more brief and concise manuscript.

Line 187-188 – I do not understand why or how these measurements were made according to the water depth classes. Do you just mean depths? And did you do this at each sampling site?

Response: We made depth profiles at each sampling site. To standardize the data with pCO2 the values were selected from near bottom (0.5-1.0 m above the sediment interface), 60% (at 60% of total water depth) and surface (up to 0.3 m of water depth).

Line 199 – what does 'assessed separately by season' mean?

Response: Thank you for the observation. That is related to the data statistically tested individually, since there was no inter-calibration among the different sampling method on each season.

Line 208 – you should restate here specifically that you are comparing high and low water from 2017 only.

Response: We have added this statement as suggested.

Line 208 – replace 'presented a significant variation' with 'varied significantly'

Response: Thank you, we have altered as suggested.

Line 221 – it gets confusing a bit when you go between comparing seasons to looking at the whole dataset so be specific when you can. For example, I would add 'From the overall dataset,' before 'Higher pCO2 was registered..'

Response: Thank you, we have reevaluated this section to clarify the manuscript. This sentence was altered as suggested.

Line 223 – I am confused by this sentence and what is respective to each other. Rewrite this one.

Response: This sentence was rewritten as suggested.

Line 228 – Because you only had pCO2 data for 2017 then I guess you couldn't find a correlation between pCO2 and FCO2 in the 2016 data, correct? You need to specific again here and state that the correlation was only for the one method.

Response: Actually there was correlation between pCO2 and FCO2. The data corresponds to 2017 samplings and was evaluated separately by season. Since that it was not clear this sentence was rewritten.

Line 232-234 – does it really matter if the two sensors were not cross calibrated in

terms of absolute concentrations if it is just the slope of the increase of concentration over time that you need for flux calculations? If it is merely slope then you should be able to estimate and then compare the rates of flux, no?

Response: We agree but are being cautious considering there are not published results comparing the two systems.

Line 235-237 – how is it that that the low water season had the highest and lowest FCO2 values but was also homogeneous? This is very confusing.

Response: The homogeneity in the FCO2 occurred when both reservoirs were evaluated together, however when each reservoir is considered separately the fluxes differed. Therefore, the pattern observed in low water season is driven by the reservoirs characteristics, not the spatial heterogeneity. In the low water season the IR reached its highest FCO2 due the capacity to accumulate organic matter in the substrate and plant-derived material left from vegetation clearing. Otherwise XR decreased its FCO2 both to a probable raise in photosynthetic activity as lower retention of organic matter due rocky and sandy substrates that domain most of the reservoir that is composed by the natural river channel.

Line 242-243 – this sentence is kind of just hanging here by itself. Shouldn't it belong somewhere in a paragraph.

Response: We have removed this sentence.

Line 244 – I would rename this section a bit more specific to what you are doing: 'pCO2 and FCO2 in ROR versus storage reservoir'

Response: Thank you, we have altered the section name as suggested.

Line 245-246 – if you consider the standard deviation of your measurements then I would say the differences are not so significant between seasons as they then overlap, especially for IR

Response: Sites that maintained the high pCO2 even on low water season due the constant organic matter source may have caused some overlapping on XR standard deviation. Otherwise it was not general since river channel and outside reservoirs pCO2 drastically decreased. The seasonal difference consequently was most apparent and especially driven by XR, however our point was to highlight that pCO2 in IR has not decreased.

Line 249 – the difference in IR is much more significant than XR. I would point that out here.

Response: Thank you, we have added this alteration to the manuscript.

Line 250-252 – I don't understand what you mean here. You did a spatial analysis but lumped all spatially different environments together? I think you mean to say that you compared the total emission from XR to the total emission of IR despite the emitting environment. Is that right?

Response: Thank you, this was exactly what we meant. We have better addressed this in the text.

Line 252-255 – I don't understand how you see no significant difference between pCO2 of XR and IR but then suddenly find that XR had pCO2 721 uatm lower. And lower than what? I guess IR. These few sentences are very confusing.

Response: When XR and IR pCO2 from whole 2017 periods was evaluated, separately from other variables (i.e. depth, unaffected river channel and season), a difference of 721 $\mu$atm was observed. However if depth, unaffected river channel, and season are considered, no significant difference is observed in the pCO2. More details were added in the text to make it clearer and we have removed T-Test analysis since it is related to a descriptive result.

Line 256 – You cannot just present an idea like 'Standing vegetation type in XR flooded areas influenced pCO2' without explaining the data that led you to that conclusion.

Response: Thank you. We have altered this sentence and better detailed the data.

Line 264 – use 'especially' instead of 'specifically' Response: Done.

Line 266 – what is a 'gradient pattern downstream'??

Response: We refer to the pattern of both pCO2 and FCO2 that are higher directly downstream the dam and decreases on the sites most distant from the reservoir. We have altered this sentence and added a most suitable term.

Line 272 – again with this 'separately to each season' – I still do not understand what this means. You have to come up with a better way of describing this.

Response: Since the FCO2 data was sampled with different equipment, different datasets were created according to the sampling season to run the statistical analysis. This sentence was removed following referee 1 suggestion.

Line 274 – use 'without significant spatial heterogeneity across environments'

Response: Thank you, we have modified this sentence as suggested.

Line 275 – use 'k600 strongly correlated with wind...' and does this relate to Fig 5b? Should you reference this?

Response: Yes, this sentence has relation with fig 5b. We altered the text and added such reference.

Line 280 – there is not environmental breakdown in the data in Figure 5

Response: Thank you, this reference was removed.

Line 287 – so you have water column data? Where is this data?

Response: We made depth profiles to pH, O2, conductivity and temperature on 2016 and 1027 high water. A supplement material was created and those data were added to it.

Line 303 – decrease in what?

Response: The pCO2 decreased due the transition from high to low water probably due to raise on primary production as Rasera et al. 2013 observed on clearwater rivers. We have rewritten this sentence.

Line 344 – what is 'vegetal suppression'? I figured out that it is when you remove vegetation prior to flooding but is this the correct term for this? It sounds very strange.

Response: Vegetation clearing is the most adequate term. This was altered through whole text.

Line 344-345 – this sentence is too long with poor grammar

Response: Thank you, this sentence was rewritten.

Line 354-356 – combine those sentences

Response: Done.

Line 356 – how many of the environments? Do you mean all except IR? This is confusing. If it is just IR that is the exception then you need to state it as 'all except IR'

Response: Exactly, only IR had raise on FCO2 and pCO2 on low water. The sentence was altered as suggested.

Line 357-358 – negative fluxes can be replaced with 'observed CO2 uptake'

Response: Thank you, done.

Line 358 – 'light penetration and low suspended sediment'

Response: Thank you, updated as suggested.

Line 363-365 – you already spoke about this earlier. Try not to be redundant

Response: We have altered this sentence detailing the influence of vegetation prior flooding on FCO2.

Line 370 – need 'which' before 'would'

Response: Done.

Line 372-373 – I don't think you need these values here in the discussion.

Response: We agreed, altered as suggested.

Line 387 – can you give a site number for the 'site downstream IR'?

Response: Absolutely, this site is P21. This information was updated in the manuscript.

Line 391 – I don't think this true and I don't think you need this sentence about a reference for natural FCO2 values

Response: Agreed. We have removed this sentence.

Line 397-398 – do you mean that the downstream sites resembled river channel sites in terms of pCO2 and FCO2 values? Don't use 'traits' to describe this. Traits more refers to features that don't vary. Yes, that was what we meant in this sentence. The word 'traits' was removed and the sentence rewritten.

Line 408-409 – are you saying that the old reservoir you are using for comparison is Tucurui? The grammar here is confusing.

Response: Exactly, Tucuruí reservoir was compared to both XR and IR. To clarify this sentence it was rewritten.

Line 412 – what do you mean by hypolimentical waters? It should be 'hypolimnetic' by the way. But this just means bottom waters with an implication of stratification, but what specifically do you want to express here?

Response: We have removed this sentence. Line 419 – bad grammar in last sentence

Response: Thank you, we revised that sentence.

Please also note the supplement to this comment:

https://www.biogeosciences-discuss.net/bg-2019-53/bg-2019-53-AC2-supplement.pdf

[Figure]

**Fig. 1.** Fig.2

[Figure]

**Fig. 2.** Fig.3

[Figure]

[Figure]

**Fig. 3.** Fig.4

[Figure]

**Supplement:**

**Supplementary Material**

**The influence of reservoir traits on carbon dioxide emissions in the Belo Monte hydropower complex, Xingu River, Amazon – Brazil**

Kleiton R. Araújo[1*], Henrique O. Sawakuchi[2-3], Dailson J. Bertassoli Jr.[4], André O. Sawakuchi[1,4], Karina D. da Silva[1,5], Thiago V. Bernardi[1,5], Nicholas D. Ward[6-7], Tatiana S. Pereira[1,5].

[1]Programa de Pós Graduação em Biodiversidade e Conservação, Universidade Federal do Pará, Altamira, 68372 – 040, Brazil,

[2]Centro de Energia Nuclear na Agricultura, Universidade de São Paulo, Piracicaba, Brazil,

[3]Department of Ecology and Environmental Science, Umeå University, Umeå, SE-901 87, Sweden,

[4]Instituto de Geociências, Universidade de São Paulo, São Paulo, Brazil,

[5]Faculdade de Ciências Biológicas, Universidade Federal do Pará, Altamira, 68372 – 040, Brazil,

[6]Marine Sciences Laboratory, Pacific Northwest National Laboratory, Sequim, Washington, 98382, USA,

[7]School of Oceanography, University of Washington, Seattle, Washington, 98195-5351, USA.

*Correspondence to: Kleiton R. Araújo (kleitonrabelo@rocketmail.com)

Keywords: run-of-the-river reservoir; greenhouse gas emission; tropical river damming.

**1. Sampling details**

Based on visualization in Google Earth we estimate that the maximum distance drifted may be approximately 1 km for measurements in the river channel up and downstream of the reservoirs. In sheltered areas located in bays and over islands with standing trees, where the water flow was very low, drifting was very short and caused by wind. An estimate of the drifting distance in the natural river channel and in the main channel of the Xingu Reservoir was obtained by using the average water velocity measured by the National Water Agency of Brazil at the Altamira station. We separated the historical values into before and after 2016, when the dams was completed. Therefore, representing estimates of water velocity in the natural river (between 2005 and 2016), and in the Xingu Reservoir main channel (after 2016). The average water velocities at Altamira are 0.74 and 0.24 m s-1 for before and after the dam, respectively. Assuming that there is no resistance of the boat with the water or air, drifting speed is similar to the water velocity. The total time of deployment was up to 30 minutes for the three consecutive measurements. Based on these we found that in the main channel of the Xingu Reservoir the drifting distance would be 432 m, and 1332 m for the natural river channel up and downstream the reservoirs.

**2. Depth profiles**

During high water of 2016 and 2017 we registered depth profiles for physical-chemical variables through whole sampling area. We measured water column pH, depth, dissolved oxygen (DO), conductivity and temperature using a multiparameter probe (EXO2®, YSI). The depth profiles were registered until near bottom depth (approximately 80 % of total depth) to avoid sediment interaction. Total depths are listed on table 1 of the manuscript. Depth profile tables are presented by year and flow order, respectively.

Table S1 – P1 site depth profile from 2016 campaign.

| pH | ODO mg/L | Temp °C | Cond µS/cm | Depth m |
|---|---|---|---|---|
| 6.65 | 6.43 | 29.208 | 23.5 | 0.166 |
| 6.61 | 6.43 | 29.208 | 23.5 | 0.212 |
| 6.53 | 6.43 | 29.208 | 23.5 | 0.265 |
| 6.5 | 6.43 | 29.201 | 23.5 | 0.283 |
| 6.5 | 6.43 | 29.195 | 23.5 | 0.401 |

| pH | ODO mg/L | Temp °C | Cond µS/cm | Depth m |
|---|---|---|---|---|
| 6.55 | 6.42 | 29.195 | 23.5 | 0.678 |
| 6.6 | 6.41 | 29.184 | 23.5 | 1.046 |
| 6.56 | 6.41 | 29.191 | 23.5 | 1.416 |
| 6.61 | 6.4 | 29.209 | 23.5 | 1.787 |
| 6.62 | 6.4 | 29.207 | 23.5 | 2.171 |
| 6.63 | 6.4 | 29.217 | 23.5 | 2.614 |
| 6.63 | 6.4 | 29.198 | 23.5 | 2.831 |
| 6.63 | 6.4 | 29.191 | 23.5 | 3.039 |
| 6.62 | 6.39 | 29.2 | 23.5 | 4.061 |
| 6.64 | 6.4 | 29.213 | 23.5 | 4.165 |
| 6.64 | 6.39 | 29.205 | 23.5 | 4.359 |
| 6.64 | 6.38 | 29.211 | 23.5 | 4.723 |
| 6.63 | 6.38 | 29.203 | 23.5 | 5.145 |
| 6.65 | 6.37 | 29.205 | 23.5 | 5.632 |
| 6.69 | 6.37 | 29.209 | 23.5 | 6.042 |
| 6.65 | 6.37 | 29.203 | 23.5 | 6.453 |
| 6.63 | 6.37 | 29.202 | 23.5 | 7.633 |
| 6.66 | 6.37 | 29.2 | 23.5 | 8.006 |
| 6.62 | 6.37 | 29.198 | 23.5 | 9.085 |
| 6.61 | 6.36 | 29.198 | 23.5 | 9.492 |
| 6.61 | 6.36 | 29.198 | 23.5 | 10.793 |
| 6.62 | 6.35 | 29.197 | 23.5 | 11.27 |
| 6.6 | 6.35 | 29.198 | 23.5 | 12.4 |
| 6.56 | 6.34 | 29.2 | 23.5 | 12.796 |
| 6.59 | 6.34 | 29.198 | 23.5 | 13.738 |
| 6.56 | 6.33 | 29.2 | 23.5 | 14.053 |
| 6.56 | 6.33 | 29.199 | 23.5 | 15.073 |
| 6.58 | 6.32 | 29.198 | 23.5 | 15.3 |

59

60 Table S2 – P6 site depth profile from 2016 campaign.

| pH | ODO mg/L | Temp °C | Cond µS/cm | Depth m |
|---|---|---|---|---|
| 6.09 | 4.99 | 29.761 | 21.9 | 0.397 |
| 6.04 | 3.95 | 29.568 | 22.3 | 0.759 |
| 5.97 | 3.72 | 29.495 | 22.5 | 1.075 |
| 5.97 | 3.86 | 29.422 | 22.3 | 1.373 |
| 6.01 | 3.88 | 29.426 | 22.4 | 1.494 |
| 6.05 | 3.82 | 29.438 | 22.4 | 1.453 |
| 6.11 | 3.77 | 29.449 | 22.4 | 1.383 |
| 6.16 | 3.8 | 29.448 | 22.4 | 1.377 |
| 6.15 | 3.86 | 29.435 | 22.3 | 1.382 |
| 6.15 | 3.87 | 29.44 | 22.4 | 1.378 |
| 6.14 | 3.86 | 29.445 | 22.4 | 1.359 |

| pH | | | | |
|---|---|---|---|---|
| 6.14 | 3.82 | 29.456 | 22.5 | 1.344 |
| 6.12 | 3.75 | 29.462 | 22.5 | 1.343 |

61

62    Table S3 – P7 site depth profile from 2016 campaign.

| pH | ODO mg/L | Temp °C | Cond µS/cm | Depth m |
|---|---|---|---|---|
| 6.72 | 7.04 | 29.555 | 21.6 | 0.102 |
| 6.67 | 7.03 | 29.538 | 21.6 | 0.223 |
| 6.64 | 7.02 | 29.518 | 21.6 | 1.222 |
| 6.68 | 7.02 | 29.508 | 21.6 | 1.279 |
| 6.74 | 7 | 29.473 | 21.6 | 1.411 |
| 6.73 | 7 | 29.48 | 21.6 | 1.602 |
| 6.75 | 6.99 | 29.467 | 21.6 | 1.803 |
| 6.8 | 6.99 | 29.468 | 21.6 | 1.965 |
| 6.83 | 6.99 | 29.475 | 21.6 | 2.167 |
| 6.82 | 6.99 | 29.48 | 21.6 | 2.351 |
| 6.8 | 6.99 | 29.475 | 21.6 | 2.538 |
| 6.79 | 6.98 | 29.466 | 21.6 | 2.767 |
| 6.79 | 6.99 | 29.487 | 21.6 | 3.018 |
| 6.83 | 6.98 | 29.482 | 21.6 | 3.283 |
| 6.85 | 6.99 | 29.492 | 21.6 | 3.546 |
| 6.85 | 6.98 | 29.483 | 21.6 | 3.825 |
| 6.86 | 6.98 | 29.482 | 21.6 | 4.102 |
| 6.82 | 6.98 | 29.478 | 21.6 | 4.379 |
| 6.8 | 6.97 | 29.476 | 21.6 | 4.656 |
| 6.84 | 6.97 | 29.471 | 21.6 | 4.966 |
| 6.84 | 6.97 | 29.481 | 21.6 | 5.29 |
| 6.82 | 6.97 | 29.484 | 21.6 | 5.6 |
| 6.83 | 6.97 | 29.482 | 21.6 | 5.958 |
| 6.82 | 6.96 | 29.477 | 21.6 | 7.127 |
| 6.78 | 6.96 | 29.48 | 21.6 | 7.444 |
| 6.77 | 6.95 | 29.479 | 21.6 | 8.549 |
| 6.77 | 6.95 | 29.477 | 21.6 | 8.913 |
| 6.8 | 6.94 | 29.479 | 21.6 | 9.249 |
| 6.82 | 6.94 | 29.476 | 21.6 | 10.418 |
| 6.82 | 6.94 | 29.476 | 21.6 | 10.691 |
| 6.79 | 6.93 | 29.477 | 21.6 | 10.904 |

63

64    Table S4 – P8 site depth profile from 2016 campaign.

| pH | ODO mg/L | Temp °C | Cond µS/cm | Depth m |
|---|---|---|---|---|
| 6.48 | 6.84 | 30.409 | 24.3 | 0.143 |

| pH | ODO mg/L | Temp °C | Cond µS/cm | Depth m |
|---|---|---|---|---|
| 6.48 | 6.8 | 30.408 | 24.3 | 0.286 |
| 6.51 | 6.82 | 30.415 | 24.3 | 0.296 |
| 6.54 | 6.8 | 30.415 | 24.3 | 0.327 |
| 6.57 | 6.79 | 30.418 | 24.3 | 0.345 |
| 6.58 | 6.81 | 30.422 | 24.3 | 0.341 |
| 6.59 | 6.83 | 30.416 | 24.3 | 0.331 |
| 6.6 | 6.81 | 30.417 | 24.3 | 0.33 |
| 6.61 | 6.79 | 30.417 | 24.3 | 0.32 |
| 6.63 | 6.79 | 30.416 | 24.3 | 0.319 |
| 6.66 | 6.8 | 30.413 | 24.3 | 0.32 |
| 6.7 | 6.81 | 30.42 | 24.3 | 0.324 |
| 6.7 | 6.82 | 30.419 | 24.3 | 0.318 |
| 6.7 | 6.8 | 30.416 | 24.3 | 0.308 |
| 6.7 | 6.8 | 30.424 | 24.3 | 0.303 |

65

66    Table S5 – P12 site depth profile from 2016 campaign.

| pH | ODO mg/L | Temp °C | Cond µS/cm | Depth m |
|---|---|---|---|---|
| 6.33 | 6.13 | 29.803 | 22.5 | 0.184 |
| 6.3 | 6.05 | 29.83 | 22.5 | 0.44 |
| 6.3 | 6.06 | 29.988 | 22.5 | 0.622 |
| 6.32 | 5.96 | 29.594 | 22.4 | 1.542 |
| 6.34 | 5.89 | 29.498 | 22.4 | 1.814 |
| 6.35 | 5.83 | 29.454 | 22.3 | 2.055 |
| 6.36 | 5.78 | 29.421 | 22.3 | 2.419 |
| 6.37 | 5.76 | 29.406 | 22.3 | 2.819 |
| 6.36 | 5.74 | 29.401 | 22.3 | 3.071 |
| 6.36 | 5.72 | 29.395 | 22.3 | 3.312 |
| 6.37 | 5.7 | 29.391 | 22.3 | 3.56 |
| 6.39 | 5.68 | 29.386 | 22.3 | 3.897 |
| 6.38 | 5.64 | 29.367 | 22.3 | 4.283 |
| 6.39 | 5.62 | 29.357 | 22.4 | 4.738 |
| 6.39 | 5.61 | 29.353 | 22.3 | 5.037 |
| 6.39 | 5.61 | 29.355 | 22.3 | 5.104 |
| 6.4 | 5.62 | 29.372 | 22.3 | 4.896 |
| 6.43 | 5.63 | 29.377 | 22.4 | 4.701 |
| 6.45 | 5.64 | 29.383 | 22.4 | 4.531 |

67

68    Table S6 – P15 site depth profile from 2016 campaign.

| pH | ODO mg/L | Temp °C | Cond µS/cm | Depth m |
|---|---|---|---|---|

| pH | ODO mg/L | Temp °C | Cond µS/cm | Depth m |
|---|---|---|---|---|
| 6.44 | 6.47 | 29.516 | 22.6 | 0.969 |
| 6.39 | 6.44 | 29.529 | 22.6 | 2.615 |
| 6.41 | 6.43 | 29.526 | 22.6 | 3.157 |
| 6.45 | 6.42 | 29.527 | 22.5 | 4.641 |
| 6.47 | 6.41 | 29.527 | 22.6 | 6.059 |
| 6.49 | 6.4 | 29.527 | 22.6 | 7.644 |
| 6.68 | 6.39 | 29.528 | 22.6 | 8.447 |
| 6.66 | 6.36 | 29.528 | 22.6 | 8.707 |
| 6.65 | 6.37 | 29.528 | 22.6 | 8.926 |

69

70 Table S7 – P11 site depth profile from 2016 campaign.

| pH | ODO mg/L | Temp °C | Cond µS/cm | Depth m |
|---|---|---|---|---|
| 6.3 | 6.58 | 29.444 | 17.5 | 1.43 |
| 6.3 | 6.45 | 29.451 | 22.6 | 3.677 |
| 6.34 | 6.42 | 29.458 | 22.6 | 7.086 |
| 6.39 | 6.39 | 29.455 | 22.6 | 9.411 |
| 6.46 | 6.36 | 29.433 | 22.7 | 12.642 |
| 6.53 | 6.35 | 29.423 | 22.7 | 15.238 |
| 6.54 | 6.33 | 29.422 | 22.7 | 19.197 |
| 6.55 | 6.27 | 29.413 | 23 | 20.501 |
| 6.6 | 6.26 | 29.411 | 22.8 | 20.59 |
| 6.64 | 6.25 | 29.411 | 22.8 | 20.563 |
| 6.66 | 6.3 | 29.412 | 22.7 | 20.475 |

71

72 Table S8 – P14 site depth profile from 2016 campaign.

| pH | ODO mg/L | Temp °C | Cond µS/cm | Depth m |
|---|---|---|---|---|
| 6.59 | 7.19 | 29.67 | 23.1 | 1.374 |
| 6.57 | 7.2 | 29.666 | 23 | 2.37 |
| 6.54 | 7.22 | 29.664 | 23 | 2.877 |
| 6.51 | 7.22 | 29.664 | 23 | 4.137 |
| 6.52 | 7.22 | 29.666 | 23 | 4.648 |
| 6.54 | 7.23 | 29.66 | 22.9 | 5.757 |
| 6.63 | 7.23 | 29.664 | 22.9 | 6.036 |
| 6.64 | 7.22 | 29.665 | 22.9 | 7.077 |
| 6.63 | 7.24 | 29.659 | 22.9 | 7.436 |
| 6.59 | 7.24 | 29.662 | 22.9 | 7.655 |

73

74 Table S9 – P19 site depth profile from 2016 campaign.

| pH | ODO mg/L | Temp °C | Cond µS/cm | Depth m |
|---|---|---|---|---|
| 6.63 | 6.78 | 29.647 | 19 | 0.128 |
| 6.56 | 6.47 | 29.665 | 26.3 | 0.286 |
| 6.57 | 6.42 | 29.685 | 26.4 | 0.456 |
| 6.59 | 6.46 | 29.676 | 26.3 | 0.525 |
| 6.59 | 6.1 | 29.657 | 26.5 | 0.579 |
| 6.6 | 6.74 | 29.71 | 26.2 | 0.658 |
| 6.65 | 6.87 | 29.724 | 25.7 | 0.776 |
| 6.68 | 6.34 | 29.712 | 26.2 | 0.926 |
| 6.63 | 5.74 | 29.716 | 27 | 1.076 |
| 6.57 | 5.31 | 29.681 | 27.4 | 1.235 |
| 6.55 | 5.19 | 29.627 | 27.5 | 1.447 |
| 6.54 | 5.02 | 29.625 | 27.6 | 1.722 |
| 6.53 | 4.94 | 29.644 | 27.7 | 2.024 |
| 6.53 | 4.9 | 29.63 | 27.7 | 2.37 |
| 6.55 | 4.84 | 29.608 | 27.7 | 2.721 |
| 6.56 | 4.93 | 29.592 | 27.6 | 3.096 |
| 6.55 | 4.79 | 29.591 | 27.7 | 4.253 |
| 6.52 | 4.13 | 29.601 | 28.1 | 4.597 |
| 6.5 | 3.8 | 29.524 | 28.5 | 4.82 |
| 6.5 | 4.68 | 29.572 | 28 | 5.157 |
| 6.52 | 4.5 | 29.566 | 27.9 | 5.641 |
| 6.51 | 3.74 | 29.509 | 28.3 | 6.091 |
| 6.48 | 3.64 | 29.481 | 28.6 | 6.569 |
| 6.48 | 3.45 | 29.476 | 28.7 | 7.002 |
| 6.49 | 3.91 | 29.515 | 28.3 | 8.164 |
| 6.47 | 3.61 | 29.482 | 28.6 | 8.637 |

75

76     Table S10 – P21 site depth profile from 2016 campaign.

| pH | ODO mg/L | Temp °C | Cond µS/cm | Depth m |
|---|---|---|---|---|
| 6.88 | 8.01 | 29.843 | 24.6 | 0.578 |
| 6.87 | 8 | 29.843 | 24.6 | 0.776 |
| 6.86 | 8 | 29.844 | 24.6 | 1.014 |
| 6.87 | 8 | 29.844 | 24.6 | 1.271 |
| 6.89 | 8 | 29.844 | 24.6 | 1.546 |
| 6.92 | 8 | 29.844 | 24.6 | 1.784 |
| 6.9 | 8 | 29.844 | 24.6 | 2.014 |
| 6.9 | 8 | 29.844 | 24.6 | 2.281 |
| 6.91 | 8 | 29.844 | 24.6 | 2.587 |
| 6.94 | 7.99 | 29.843 | 24.6 | 2.903 |
| 6.94 | 7.99 | 29.844 | 24.6 | 3.125 |
| 6.93 | 7.99 | 29.845 | 24.6 | 3.312 |

| 6.92 | 7.99 | 29.845 | 24.6 | 3.483 |
|------|------|--------|------|-------|
| 6.92 | 7.99 | 29.846 | 24.6 | 3.734 |
| 6.92 | 7.98 | 29.846 | 24.6 | 4.033 |
| 6.91 | 7.99 | 29.845 | 24.6 | 4.269 |
| 6.9 | 7.99 | 29.844 | 24.6 | 4.381 |
| 6.91 | 7.98 | 29.845 | 24.6 | 5.535 |
| 6.93 | 7.98 | 29.846 | 24.6 | 5.868 |
| 6.93 | 7.97 | 29.847 | 24.6 | 7.08 |
| 6.96 | 7.97 | 29.848 | 24.6 | 7.244 |
| 6.95 | 7.96 | 29.848 | 24.6 | 8.487 |
| 6.92 | 7.96 | 29.847 | 24.6 | 8.363 |
| 6.9 | 7.96 | 29.847 | 24.6 | 8.267 |

77

78    Table S11 – P4 site depth profile from 2017 campaign.

| pH | ODO mg/L | Temp °C | Cond µS/cm | Depth m |
|------|----------|---------|------------|---------|
| 7.22 | 7.15 | 29.552 | 33.6 | 0.313 |
| 7.21 | 7.15 | 29.553 | 33.5 | 0.422 |
| 7.21 | 7.14 | 29.553 | 33.5 | 0.528 |
| 7.2 | 7.14 | 29.553 | 33.5 | 0.622 |
| 7.19 | 7.14 | 29.553 | 33.5 | 0.724 |
| 7.19 | 7.13 | 29.553 | 33.5 | 0.83 |
| 7.18 | 7.13 | 29.552 | 33.5 | 0.944 |
| 7.18 | 7.13 | 29.552 | 33.5 | 1.118 |
| 7.18 | 7.13 | 29.552 | 33.5 | 1.275 |
| 7.17 | 7.13 | 29.553 | 33.5 | 1.42 |
| 7.17 | 7.13 | 29.552 | 33.5 | 1.573 |
| 7.16 | 7.12 | 29.552 | 33.4 | 1.721 |
| 7.16 | 7.12 | 29.552 | 33.4 | 1.868 |
| 7.15 | 7.12 | 29.553 | 33.5 | 2.017 |
| 7.15 | 7.12 | 29.553 | 33.5 | 2.15 |
| 7.15 | 7.12 | 29.552 | 33.5 | 2.284 |
| 7.14 | 7.12 | 29.552 | 33.4 | 2.402 |
| 7.14 | 7.12 | 29.552 | 33.4 | 2.483 |
| 7.13 | 7.12 | 29.552 | 33.4 | 2.505 |
| 7.13 | 7.12 | 29.552 | 33.4 | 2.53 |
| 7.12 | 7.12 | 29.552 | 33.4 | 2.557 |
| 7.12 | 7.12 | 29.552 | 33.4 | 2.602 |
| 7.12 | 7.12 | 29.552 | 33.4 | 2.668 |
| 7.12 | 7.11 | 29.552 | 33.5 | 2.728 |
| 7.11 | 7.11 | 29.552 | 33.5 | 2.813 |
| 7.11 | 7.11 | 29.552 | 33.4 | 2.922 |
| 7.11 | 7.11 | 29.552 | 33.4 | 3.06 |

| 7.11 | 7.11 | 29.552 | 33.4 | 3.204 |
|---|---|---|---|---|
| 7.11 | 7.11 | 29.552 | 33.4 | 3.318 |
| 7.11 | 7.11 | 29.552 | 33.4 | 3.406 |
| 7.1 | 7.11 | 29.552 | 33.4 | 3.503 |
| 7.1 | 7.11 | 29.552 | 33.4 | 3.624 |
| 7.1 | 7.11 | 29.552 | 33.4 | 3.768 |
| 7.09 | 7.11 | 29.552 | 33.4 | 3.922 |
| 7.09 | 7.1 | 29.553 | 33.4 | 4.054 |
| 7.09 | 7.1 | 29.553 | 33.4 | 4.185 |
| 7.09 | 7.1 | 29.553 | 33.4 | 4.347 |
| 7.09 | 7.1 | 29.553 | 33.4 | 4.5 |
| 7.09 | 7.1 | 29.553 | 33.4 | 4.628 |
| 7.09 | 7.1 | 29.553 | 33.4 | 4.719 |
| 7.09 | 7.1 | 29.553 | 33.4 | 4.807 |
| 7.08 | 7.1 | 29.552 | 33.4 | 4.908 |
| 7.08 | 7.1 | 29.552 | 33.4 | 5.027 |
| 7.07 | 7.1 | 29.553 | 33.4 | 5.148 |
| 7.05 | 7.1 | 29.553 | 33.4 | 5.259 |
| 7.04 | 7.1 | 29.553 | 33.4 | 5.369 |
| 7.02 | 7.1 | 29.553 | 33.4 | 5.501 |
| 7.01 | 7.09 | 29.553 | 33.4 | 5.666 |
| 6.99 | 7.09 | 29.553 | 33.4 | 5.828 |
| 6.98 | 7.09 | 29.553 | 33.4 | 5.987 |
| 6.96 | 7.09 | 29.553 | 33.4 | 6.132 |
| 6.93 | 7.09 | 29.553 | 33.4 | 6.274 |
| 6.91 | 7.09 | 29.553 | 33.4 | 6.414 |
| 6.88 | 7.09 | 29.553 | 33.4 | 6.563 |
| 6.86 | 7.09 | 29.553 | 33.4 | 6.687 |
| 6.85 | 7.09 | 29.553 | 33.4 | 6.77 |
| 6.84 | 7.09 | 29.553 | 33.4 | 6.874 |
| 6.83 | 7.09 | 29.553 | 33.4 | 6.978 |
| 6.82 | 7.09 | 29.552 | 33.4 | 7.068 |
| 6.81 | 7.08 | 29.552 | 33.3 | 7.162 |
| 6.69 | 7.08 | 29.552 | 33.3 | 7.291 |

79

80    Table S13 – P5 site depth profile from 2017 campaign.

| pH | ODO mg/L | Temp °C | Cond µS/cm | Depth m |
|---|---|---|---|---|
| 7.27 | 7.29 | 29.934 | 32.2 | 0.157 |
| 7.27 | 7.28 | 29.932 | 32.2 | 0.246 |
| 7.26 | 7.28 | 29.932 | 32.2 | 0.357 |
| 7.25 | 7.28 | 29.932 | 32.2 | 0.464 |
| 7.24 | 7.28 | 29.93 | 32.2 | 0.553 |

| | | | | |
|---|---|---|---|---|
| 7.23 | 7.28 | 29.93 | 32.3 | 0.642 |
| 7.22 | 7.28 | 29.93 | 32.3 | 0.726 |
| 7.21 | 7.28 | 29.927 | 32.3 | 0.8 |
| 7.2 | 7.28 | 29.924 | 32.3 | 0.88 |
| 7.2 | 7.28 | 29.923 | 32.3 | 0.962 |
| 7.2 | 7.28 | 29.923 | 32.3 | 1.031 |
| 7.19 | 7.28 | 29.924 | 32.4 | 1.115 |
| 7.19 | 7.28 | 29.925 | 32.4 | 1.206 |
| 7.19 | 7.28 | 29.924 | 32.5 | 1.299 |
| 7.19 | 7.28 | 29.924 | 32.5 | 1.404 |
| 7.19 | 7.27 | 29.925 | 32.5 | 1.519 |
| 7.19 | 7.27 | 29.926 | 32.6 | 1.635 |
| 7.19 | 7.27 | 29.928 | 32.6 | 1.751 |
| 7.19 | 7.27 | 29.928 | 32.7 | 1.882 |
| 7.19 | 7.27 | 29.929 | 32.7 | 2.011 |
| 7.18 | 7.27 | 29.929 | 32.7 | 2.141 |
| 7.18 | 7.27 | 29.928 | 32.8 | 2.283 |
| 7.18 | 7.27 | 29.926 | 32.8 | 2.429 |
| 7.18 | 7.27 | 29.925 | 32.9 | 2.586 |
| 7.18 | 7.27 | 29.925 | 33 | 2.745 |
| 7.18 | 7.27 | 29.924 | 33 | 2.904 |
| 7.18 | 7.26 | 29.922 | 33.1 | 3.053 |
| 7.18 | 7.26 | 29.92 | 33.1 | 3.196 |
| 7.18 | 7.26 | 29.919 | 33.1 | 3.33 |
| 7.18 | 7.26 | 29.918 | 33.1 | 3.452 |
| 7.18 | 7.26 | 29.917 | 33.2 | 3.559 |
| 7.18 | 7.26 | 29.916 | 33.2 | 3.661 |
| 7.18 | 7.26 | 29.915 | 33.2 | 3.766 |
| 7.18 | 7.26 | 29.914 | 33.3 | 3.876 |
| 7.18 | 7.26 | 29.912 | 33.3 | 3.988 |
| 7.18 | 7.25 | 29.912 | 33.3 | 4.107 |
| 7.17 | 7.25 | 29.911 | 33.3 | 4.233 |
| 7.17 | 7.25 | 29.911 | 33.3 | 4.361 |
| 7.17 | 7.25 | 29.911 | 33.4 | 4.482 |
| 7.16 | 7.25 | 29.911 | 33.4 | 4.603 |
| 7.16 | 7.25 | 29.911 | 33.4 | 4.714 |
| 7.16 | 7.25 | 29.912 | 33.4 | 4.829 |
| 7.16 | 7.25 | 29.912 | 33.4 | 4.941 |
| 7.15 | 7.25 | 29.912 | 33.4 | 5.061 |
| 7.14 | 7.24 | 29.912 | 33.4 | 5.184 |
| 7.14 | 7.24 | 29.912 | 33.4 | 5.316 |
| 7.14 | 7.24 | 29.911 | 33.5 | 5.442 |
| 7.14 | 7.24 | 29.911 | 33.5 | 5.557 |
| 7.14 | 7.24 | 29.911 | 33.5 | 5.66 |
| 7.14 | 7.24 | 29.911 | 33.5 | 5.761 |

| 7.14 | 7.24 | 29.911 | 33.5 | 5.836 |
| 7.14 | 7.24 | 29.91 | 33.5 | 5.879 |
| 7.14 | 7.24 | 29.909 | 33.5 | 5.918 |
| 7.14 | 7.24 | 29.909 | 33.5 | 5.966 |
| 7.14 | 7.24 | 29.909 | 33.5 | 6.043 |
| 7.14 | 7.23 | 29.909 | 33.5 | 6.128 |
| 7.13 | 7.23 | 29.909 | 33.5 | 6.226 |
| 7.13 | 7.23 | 29.909 | 33.6 | 6.346 |
| 7.12 | 7.23 | 29.909 | 33.6 | 7.481 |
| 7.11 | 7.23 | 29.909 | 33.6 | 7.514 |
| 7.1 | 7.23 | 29.909 | 33.6 | 7.556 |
| 7.09 | 7.23 | 29.908 | 33.6 | 7.558 |
| 7.09 | 7.23 | 29.908 | 33.6 | 7.585 |
| 7.09 | 7.23 | 29.908 | 33.6 | 7.656 |
| 7.08 | 7.22 | 29.908 | 33.6 | 7.731 |
| 7.08 | 7.22 | 29.908 | 33.6 | 7.809 |
| 7.07 | 7.22 | 29.908 | 33.6 | 7.9 |
| 7.07 | 7.22 | 29.909 | 33.6 | 7.998 |

81

82   Table S14 – P6 site depth profile from 2017 campaign.

| pH | ODO mg/L | Temp °C | Cond µS/cm | Depth m |
| --- | --- | --- | --- | --- |
| 6.93 | 7.32 | 30.83 | 33.8 | 0.264 |
| 6.94 | 7.32 | 30.811 | 33.7 | 0.316 |
| 6.94 | 7.31 | 30.764 | 33.7 | 0.388 |
| 6.93 | 7.23 | 30.686 | 33.7 | 0.476 |
| 6.91 | 7.14 | 30.578 | 33.7 | 0.541 |
| 6.88 | 7.07 | 30.505 | 33.7 | 0.576 |
| 6.85 | 7.01 | 30.421 | 33.7 | 0.609 |
| 6.67 | 6.6 | 30.323 | 33.7 | 0.661 |
| 6.63 | 6.42 | 30.21 | 33.8 | 0.728 |
| 6.6 | 6.29 | 30.104 | 33.9 | 0.797 |
| 6.58 | 5.97 | 30.052 | 34 | 0.835 |
| 6.57 | 5.94 | 30.022 | 34.1 | 0.874 |
| 6.56 | 5.91 | 30.011 | 34.1 | 0.915 |
| 6.55 | 5.86 | 30.007 | 34.2 | 0.957 |
| 6.53 | 5.53 | 29.972 | 34.2 | 1.021 |
| 6.38 | 5.25 | 29.9 | 34.4 | 1.075 |
| 6.33 | 4.53 | 29.805 | 34.6 | 1.148 |
| 6.32 | 4.35 | 29.748 | 34.8 | 1.223 |
| 6.32 | 4.26 | 29.714 | 34.9 | 1.291 |
| 6.32 | 4.16 | 29.683 | 35 | 1.383 |
| 6.33 | 4.43 | 29.634 | 34.8 | 1.485 |

| | | | | |
|---|---|---|---|---|
| 6.35 | 4.6 | 29.583 | 34.7 | 1.578 |
| 6.37 | 4.71 | 29.556 | 34.6 | 1.639 |
| 6.4 | 4.81 | 29.547 | 34.6 | 1.689 |
| 6.43 | 4.95 | 29.483 | 34.3 | 1.761 |
| 6.44 | 5.02 | 29.389 | 34.1 | 1.855 |
| 6.44 | 4.74 | 29.275 | 34.1 | 1.938 |
| 6.43 | 4.56 | 29.209 | 34.3 | 2.007 |
| 6.42 | 4.46 | 29.165 | 34.4 | 2.084 |
| 6.4 | 4.13 | 29.141 | 34.5 | 2.163 |
| 6.39 | 3.92 | 29.107 | 34.6 | 2.264 |
| 6.37 | 3.78 | 29.073 | 34.8 | 2.363 |
| 6.35 | 3.71 | 29.035 | 34.9 | 2.42 |
| 6.33 | 3.59 | 29.023 | 35 | 2.477 |
| 6.32 | 3.5 | 29.012 | 35.1 | 2.537 |
| 6.31 | 3.4 | 29 | 35.2 | 2.591 |
| 6.3 | 2.96 | 28.98 | 35.2 | 2.675 |
| 6.29 | 2.92 | 28.96 | 35.3 | 2.757 |
| 6.28 | 2.9 | 28.954 | 35.4 | 2.808 |
| 6.27 | 2.86 | 28.944 | 35.4 | 2.855 |
| 6.27 | 2.82 | 28.938 | 35.5 | 2.903 |
| 6.27 | 2.78 | 28.933 | 35.6 | 2.965 |
| 6.26 | 2.72 | 28.929 | 35.6 | 3.05 |
| 6.26 | 2.65 | 28.924 | 35.6 | 3.105 |
| 6.26 | 2.59 | 28.92 | 35.6 | 3.153 |
| 6.25 | 2.5 | 28.91 | 35.7 | 3.21 |
| 6.24 | 2.41 | 28.909 | 35.7 | 3.227 |
| 6.24 | 2.4 | 28.915 | 35.7 | 3.248 |
| 6.24 | 2.46 | 28.919 | 35.6 | 3.266 |

83

84    Table S15 – P7 site depth profile from 2017 campaign.

| pH | ODO mg/L | Temp °C | Cond µS/cm | Depth m |
|---|---|---|---|---|
| 6.91 | 6.95 | 29.479 | 32.6 | 0.316 |
| 6.91 | 6.94 | 29.479 | 32.6 | 0.456 |
| 6.91 | 6.94 | 29.487 | 32.6 | 0.598 |
| 6.91 | 6.95 | 29.522 | 32.5 | 0.735 |
| 6.92 | 6.96 | 29.525 | 32.5 | 0.871 |
| 6.92 | 6.95 | 29.499 | 32.5 | 1.004 |
| 6.92 | 6.94 | 29.478 | 32.5 | 1.132 |
| 6.92 | 6.95 | 29.541 | 32.5 | 1.246 |
| 6.93 | 7 | 29.666 | 32.4 | 1.346 |
| 6.95 | 7.07 | 29.749 | 32.3 | 1.454 |
| 6.98 | 7.12 | 29.778 | 32.3 | 1.578 |

| | | | | |
|---|---|---|---|---|
| 7 | 7.13 | 29.742 | 32.4 | 1.723 |
| 7.01 | 7.13 | 29.73 | 32.4 | 1.855 |
| 7.01 | 7.12 | 29.682 | 32.5 | 1.978 |
| 7 | 7.08 | 29.608 | 32.5 | 2.106 |
| 6.98 | 7.03 | 29.538 | 32.6 | 2.245 |
| 6.96 | 6.98 | 29.519 | 32.7 | 3.119 |
| 6.95 | 6.97 | 29.515 | 32.7 | 3.214 |
| 6.94 | 6.96 | 29.513 | 32.7 | 3.308 |
| 6.94 | 6.95 | 29.513 | 32.7 | 3.413 |
| 6.94 | 6.95 | 29.516 | 32.8 | 3.504 |
| 6.94 | 6.95 | 29.514 | 32.8 | 3.588 |
| 6.94 | 6.95 | 29.509 | 32.9 | 3.669 |
| 6.93 | 6.94 | 29.501 | 32.9 | 3.745 |
| 6.93 | 6.94 | 29.494 | 32.9 | 3.819 |
| 6.93 | 6.93 | 29.485 | 33 | 3.962 |
| 6.93 | 6.92 | 29.484 | 33 | 4.109 |
| 6.93 | 6.92 | 29.489 | 33.1 | 4.253 |
| 6.93 | 6.92 | 29.496 | 33.1 | 4.395 |
| 6.93 | 6.93 | 29.504 | 33.1 | 4.533 |
| 6.93 | 6.93 | 29.506 | 33.2 | 4.686 |
| 6.93 | 6.93 | 29.501 | 33.2 | 4.842 |
| 6.92 | 6.93 | 29.503 | 33.2 | 5.791 |
| 6.92 | 6.93 | 29.511 | 33.2 | 5.896 |
| 6.92 | 6.93 | 29.513 | 33.3 | 5.905 |
| 6.92 | 6.93 | 29.509 | 33.3 | 5.923 |
| 6.92 | 6.93 | 29.499 | 33.3 | 5.962 |
| 6.92 | 6.92 | 29.488 | 33.4 | 6.016 |
| 6.91 | 6.91 | 29.479 | 33.4 | 6.102 |
| 6.91 | 6.9 | 29.477 | 33.4 | 7.035 |
| 6.9 | 6.9 | 29.471 | 33.5 | 7.176 |
| 6.9 | 6.89 | 29.463 | 33.5 | 7.263 |
| 6.9 | 6.88 | 29.46 | 33.5 | 7.343 |
| 6.9 | 6.88 | 29.464 | 33.5 | 7.417 |
| 6.9 | 6.89 | 29.474 | 33.5 | 7.492 |
| 6.9 | 6.89 | 29.484 | 33.6 | 7.572 |
| 6.91 | 6.9 | 29.496 | 33.6 | 7.656 |
| 6.91 | 6.91 | 29.517 | 33.6 | 7.766 |
| 6.92 | 6.93 | 29.541 | 33.6 | 8.693 |
| 6.92 | 6.94 | 29.554 | 33.6 | 8.874 |
| 6.93 | 6.94 | 29.553 | 33.6 | 9.041 |
| 6.93 | 6.94 | 29.545 | 33.6 | 9.229 |
| 6.93 | 6.94 | 29.541 | 33.6 | 9.409 |
| 6.93 | 6.93 | 29.538 | 33.7 | 10.304 |
| 6.93 | 6.93 | 29.537 | 33.7 | 10.524 |
| 6.92 | 6.93 | 29.538 | 33.7 | 10.648 |

| 6.92 | 6.93 | 29.534 | 33.7 | 10.753 |
|------|------|--------|------|--------|
| 6.92 | 6.92 | 29.524 | 33.7 | 10.858 |
| 6.91 | 6.91 | 29.513 | 33.7 | 10.971 |
| 6.91 | 6.9 | 29.499 | 33.7 | 11.079 |
| 6.9 | 6.89 | 29.482 | 33.8 | 11.899 |
| 6.89 | 6.87 | 29.471 | 33.8 | 12.004 |
| 6.89 | 6.86 | 29.458 | 33.8 | 12.133 |
| 6.88 | 6.85 | 29.455 | 33.8 | 12.223 |
| 6.88 | 6.85 | 29.462 | 33.8 | 12.324 |
| 6.88 | 6.86 | 29.47 | 33.8 | 12.403 |
| 6.88 | 6.86 | 29.474 | 33.8 | 12.483 |
| 6.88 | 6.86 | 29.477 | 33.8 | 12.566 |
| 6.88 | 6.86 | 29.483 | 33.8 | 12.654 |
| 6.88 | 6.87 | 29.493 | 33.8 | 12.81 |
| 6.88 | 6.87 | 29.504 | 33.8 | 12.963 |
| 6.88 | 6.88 | 29.51 | 33.8 | 13.114 |
| 6.88 | 6.88 | 29.511 | 33.8 | 14.017 |
| 6.89 | 6.88 | 29.514 | 33.8 | 14.163 |
| 6.89 | 6.88 | 29.519 | 33.8 | 14.229 |
| 6.89 | 6.89 | 29.519 | 33.8 | 14.346 |
| 6.89 | 6.88 | 29.513 | 33.8 | 14.502 |
| 6.89 | 6.88 | 29.504 | 33.8 | 15.504 |
| 6.88 | 6.87 | 29.492 | 33.9 | 15.612 |
| 6.88 | 6.86 | 29.484 | 33.9 | 15.668 |
| 6.87 | 6.85 | 29.481 | 33.9 | 15.74 |
| 6.87 | 6.85 | 29.486 | 33.9 | 15.821 |
| 6.87 | 6.85 | 29.483 | 33.9 | 15.908 |
| 6.86 | 6.85 | 29.476 | 33.9 | 16.01 |
| 6.86 | 6.84 | 29.474 | 33.9 | 16.103 |
| 6.85 | 6.84 | 29.477 | 33.9 | 16.965 |
| 6.85 | 6.84 | 29.475 | 33.9 | 17.125 |
| 6.85 | 6.84 | 29.472 | 33.9 | 17.24 |
| 6.84 | 6.84 | 29.473 | 33.9 | 17.346 |
| 6.84 | 6.83 | 29.473 | 33.9 | 17.48 |
| 6.83 | 6.84 | 29.478 | 33.9 | 18.401 |
| 6.83 | 6.84 | 29.486 | 33.9 | 18.702 |
| 6.82 | 6.84 | 29.492 | 33.9 | 18.833 |

85

86    Table S16 – P8 site depth profile from 2017 campaign.

| pH | ODO mg/L | Temp °C | Cond µS/cm | Depth m |
|------|------|--------|------|--------|
| 6.92 | 7.4 | 30.755 | 34.6 | 0.137 |
| 6.91 | 7.35 | 30.679 | 34.5 | 0.244 |

| | | | | |
|---|---|---|---|---|
| 6.9 | 7.31 | 30.598 | 34.4 | 0.358 |
| 6.89 | 7.28 | 30.551 | 34.3 | 0.464 |
| 6.88 | 7.26 | 30.54 | 34.2 | 0.569 |
| 6.88 | 7.25 | 30.538 | 34.1 | 0.649 |
| 6.88 | 7.24 | 30.536 | 34 | 0.71 |
| 6.88 | 7.23 | 30.532 | 33.9 | 0.757 |
| 6.88 | 7.23 | 30.53 | 33.8 | 0.778 |
| 6.88 | 7.23 | 30.532 | 33.7 | 0.766 |
| 6.88 | 7.22 | 30.534 | 33.7 | 0.738 |
| 6.88 | 7.23 | 30.535 | 33.6 | 0.708 |

87

88   Table S17 – P9 site depth profile from 2017 campaign.

| pH | ODO mg/L | Temp °C | Cond µS/cm | Depth m |
|---|---|---|---|---|
| 6.74 | 0.46 | 29.982 | 36.4 | 0.114 |
| 6.72 | 0.45 | 29.871 | 36.1 | 0.154 |
| 6.7 | 0.45 | 29.801 | 36 | 0.185 |
| 6.69 | 0.45 | 29.783 | 35.9 | 0.188 |
| 6.68 | 0.45 | 29.829 | 35.8 | 0.183 |
| 6.68 | 0.44 | 29.923 | 35.8 | 0.178 |
| 6.69 | 0.44 | 30.029 | 35.7 | 0.197 |
| 6.69 | 0.43 | 30.046 | 35.6 | 0.226 |
| 6.69 | 0.43 | 29.971 | 35.5 | 0.266 |
| 6.69 | 0.43 | 29.883 | 35.5 | 0.315 |
| 6.69 | 0.43 | 29.858 | 35.4 | 0.37 |
| 6.68 | 0.43 | 29.882 | 35.4 | 0.436 |
| 6.68 | 0.43 | 29.909 | 35.3 | 0.514 |
| 6.68 | 0.43 | 29.928 | 35.3 | 0.587 |
| 6.68 | 0.42 | 29.909 | 35.2 | 0.653 |
| 6.67 | 0.42 | 29.813 | 35.1 | 0.723 |
| 6.67 | 0.42 | 29.684 | 35 | 0.794 |
| 6.66 | 0.42 | 29.579 | 34.9 | 0.864 |
| 6.65 | 0.43 | 29.513 | 34.8 | 0.938 |
| 6.64 | 0.43 | 29.477 | 34.8 | 1.004 |
| 6.64 | 0.42 | 29.462 | 34.7 | 1.067 |
| 6.64 | 0.42 | 29.463 | 34.7 | 1.124 |
| 6.63 | 0.42 | 29.465 | 34.6 | 1.163 |
| 6.63 | 0.42 | 29.468 | 34.6 | 1.186 |
| 6.62 | 0.42 | 29.469 | 34.5 | 1.205 |
| 6.62 | 0.42 | 29.468 | 34.5 | 1.227 |
| 6.62 | 0.42 | 29.463 | 34.5 | 1.246 |
| 6.62 | 0.42 | 29.455 | 34.5 | 1.262 |
| 6.62 | 0.41 | 29.449 | 34.4 | 1.272 |

| | | | | |
|------|------|--------|------|-------|
| 6.63 | 0.41 | 29.448 | 34.4 | 1.285 |
| 6.63 | 0.41 | 29.449 | 34.4 | 1.313 |
| 6.63 | 0.41 | 29.448 | 34.4 | 1.348 |
| 6.63 | 0.41 | 29.447 | 34.4 | 1.372 |
| 6.63 | 0.41 | 29.447 | 34.3 | 1.396 |
| 6.62 | 0.41 | 29.446 | 34.3 | 1.423 |
| 6.62 | 0.41 | 29.443 | 34.3 | 1.453 |
| 6.62 | 0.41 | 29.442 | 34.3 | 1.49 |
| 6.62 | 0.4  | 29.445 | 34.3 | 1.532 |
| 6.62 | 0.4  | 29.445 | 34.3 | 1.58 |
| 6.62 | 0.4  | 29.439 | 34.3 | 1.641 |
| 6.62 | 0.4  | 29.43  | 34.3 | 1.708 |
| 6.62 | 0.4  | 29.422 | 34.3 | 1.777 |
| 6.62 | 0.4  | 29.415 | 34.3 | 1.858 |
| 6.62 | 0.4  | 29.41  | 34.2 | 1.938 |
| 6.62 | 0.4  | 29.404 | 34.2 | 2.025 |
| 6.61 | 0.4  | 29.395 | 34.2 | 2.116 |
| 6.61 | 0.4  | 29.389 | 34.2 | 2.209 |
| 6.61 | 0.4  | 29.388 | 34.2 | 2.302 |
| 6.61 | 0.4  | 29.386 | 34.2 | 2.398 |
| 6.61 | 0.4  | 29.384 | 34.2 | 2.494 |
| 6.61 | 0.4  | 29.379 | 34.2 | 2.594 |
| 6.61 | 0.4  | 29.376 | 34.2 | 2.69 |
| 6.61 | 0.4  | 29.371 | 34.2 | 2.79 |
| 6.6  | 0.4  | 29.365 | 34.2 | 2.899 |
| 6.6  | 0.4  | 29.364 | 34.2 | 3.008 |
| 6.6  | 0.41 | 29.365 | 34.2 | 3.126 |
| 6.6  | 0.41 | 29.367 | 34.2 | 3.239 |
| 6.6  | 0.41 | 29.368 | 34.1 | 3.354 |
| 6.6  | 0.41 | 29.368 | 34.1 | 3.472 |
| 6.6  | 0.41 | 29.368 | 34.1 | 3.588 |
| 6.6  | 0.41 | 29.369 | 34.1 | 3.698 |
| 6.6  | 0.41 | 29.369 | 34.1 | 3.797 |
| 6.6  | 0.41 | 29.368 | 34.1 | 3.887 |
| 6.6  | 0.41 | 29.368 | 34.1 | 3.979 |
| 6.6  | 0.41 | 29.367 | 34.1 | 4.082 |
| 6.6  | 0.41 | 29.366 | 34.1 | 4.183 |
| 6.6  | 0.41 | 29.365 | 34.1 | 4.286 |
| 6.6  | 0.41 | 29.364 | 34.1 | 4.393 |
| 6.59 | 0.41 | 29.363 | 34.1 | 4.51 |
| 6.59 | 0.41 | 29.362 | 34.1 | 4.62 |
| 6.59 | 0.41 | 29.362 | 34.2 | 4.731 |
| 6.59 | 0.41 | 29.361 | 34.2 | 4.827 |
| 6.59 | 0.41 | 29.36  | 34.2 | 4.906 |
| 6.59 | 0.41 | 29.36  | 34.2 | 4.973 |

| | | | | |
|---|---|---|---|---|
| 6.59 | 0.41 | 29.359 | 34.2 | 5.026 |
| 6.59 | 0.41 | 29.357 | 34.2 | 5.056 |
| 6.59 | 0.41 | 29.357 | 34.2 | 5.076 |
| 6.59 | 0.41 | 29.356 | 34.2 | 5.096 |
| 6.59 | 0.41 | 29.356 | 34.2 | 5.12 |
| 6.59 | 0.4 | 29.355 | 34.2 | 5.161 |
| 6.59 | 0.41 | 29.347 | 34.2 | 5.258 |
| 6.58 | 0.41 | 29.339 | 34.2 | 5.366 |
| 6.58 | 0.41 | 29.333 | 34.2 | 5.49 |
| 6.57 | 0.41 | 29.331 | 34.2 | 5.625 |
| 6.57 | 0.41 | 29.329 | 34.2 | 5.774 |
| 6.57 | 0.41 | 29.329 | 34.2 | 5.923 |

89

90    Table S18 – P10 site depth profile from 2017 campaign.

| pH | ODO mg/L | Temp °C | Cond µS/cm | Depth m |
|---|---|---|---|---|
| 6.74 | 6.68 | 30.726 | 36.9 | 0.164 |
| 6.74 | 6.69 | 30.717 | 36.9 | 0.245 |
| 6.74 | 6.7 | 30.699 | 36.8 | 0.34 |
| 6.74 | 6.71 | 30.672 | 36.8 | 0.422 |
| 6.75 | 6.72 | 30.653 | 36.8 | 0.497 |
| 6.76 | 6.73 | 30.631 | 36.8 | 0.577 |
| 6.76 | 6.74 | 30.606 | 36.9 | 0.667 |
| 6.77 | 6.75 | 30.591 | 36.9 | 0.761 |
| 6.78 | 6.76 | 30.568 | 36.8 | 0.853 |
| 6.79 | 6.77 | 30.541 | 36.8 | 0.942 |
| 6.8 | 6.78 | 30.507 | 36.8 | 1.029 |
| 6.8 | 6.79 | 30.474 | 36.7 | 1.107 |
| 6.81 | 6.78 | 30.441 | 36.6 | 1.19 |
| 6.8 | 6.73 | 30.408 | 36.6 | 1.281 |
| 6.79 | 6.29 | 30.348 | 36.7 | 1.374 |
| 6.78 | 6.1 | 30.256 | 36.8 | 1.471 |
| 6.66 | 5.69 | 30.148 | 36.7 | 1.565 |
| 6.6 | 5.6 | 30.066 | 36.6 | 1.663 |
| 6.58 | 5.54 | 30.014 | 36.6 | 1.76 |
| 6.57 | 5.45 | 30 | 36.5 | 1.858 |
| 6.56 | 5.38 | 29.988 | 36.5 | 1.959 |
| 6.56 | 5.32 | 29.976 | 36.5 | 2.059 |
| 6.55 | 5.29 | 29.968 | 36.5 | 2.159 |
| 6.54 | 5.27 | 29.963 | 36.5 | 2.256 |
| 6.54 | 5.26 | 29.957 | 36.4 | 2.352 |
| 6.53 | 5.25 | 29.952 | 36.4 | 2.451 |
| 6.53 | 5.24 | 29.945 | 36.4 | 2.548 |

| pH | ODO mg/L | Temp °C | Cond µS/cm | Depth m |
|---|---|---|---|---|
| 6.53 | 5.23 | 29.936 | 36.4 | 2.647 |
| 6.52 | 5.18 | 29.921 | 36.4 | 2.757 |
| 6.51 | 5.11 | 29.905 | 36.4 | 2.872 |
| 6.5 | 5.05 | 29.893 | 36.4 | 3.006 |
| 6.48 | 5 | 29.886 | 36.5 | 3.146 |
| 6.47 | 4.94 | 29.879 | 36.5 | 3.289 |
| 6.47 | 4.86 | 29.873 | 36.5 | 3.417 |
| 6.46 | 4.79 | 29.871 | 36.6 | 3.528 |
| 6.45 | 4.74 | 29.871 | 36.6 | 3.621 |
| 6.44 | 4.71 | 29.871 | 36.6 | 3.69 |
| 6.44 | 4.68 | 29.87 | 36.6 | 3.736 |
| 6.44 | 4.66 | 29.866 | 36.6 | 3.758 |
| 6.43 | 4.64 | 29.863 | 36.7 | 3.765 |
| 6.43 | 4.62 | 29.863 | 36.7 | 3.775 |
| 6.43 | 4.58 | 29.862 | 36.7 | 3.778 |
| 6.42 | 4.55 | 29.859 | 36.7 | 3.793 |
| 6.41 | 4.51 | 29.853 | 36.8 | 3.818 |
| 6.41 | 4.45 | 29.844 | 36.8 | 3.861 |
| 6.39 | 4.07 | 29.829 | 37 | 3.924 |
| 6.26 | 3.82 | 29.802 | 37.3 | 4.831 |

91

92   Table S19 – P11 site depth profile from 2017 campaign.

| pH | ODO mg/L | Temp °C | Cond µS/cm | Depth m |
|---|---|---|---|---|
| 6.79 | 6.81 | 29.955 | 32.7 | 1.063 |
| 6.8 | 6.77 | 29.945 | 32.7 | 1.248 |
| 6.81 | 6.76 | 29.945 | 32.8 | 1.334 |
| 6.83 | 6.75 | 29.947 | 32.9 | 1.462 |
| 6.84 | 6.75 | 29.944 | 33 | 1.592 |
| 6.85 | 6.74 | 29.938 | 33.1 | 1.699 |
| 6.86 | 6.74 | 29.939 | 33.1 | 1.787 |
| 6.88 | 6.74 | 29.945 | 33.2 | 1.862 |
| 6.88 | 6.74 | 29.949 | 33.2 | 2.718 |
| 6.89 | 6.74 | 29.95 | 33.3 | 2.816 |
| 6.89 | 6.74 | 29.949 | 33.3 | 2.865 |
| 6.9 | 6.73 | 29.95 | 33.4 | 2.933 |
| 6.9 | 6.73 | 29.95 | 33.4 | 3.019 |
| 6.9 | 6.74 | 29.952 | 33.5 | 3.096 |
| 6.9 | 6.74 | 29.955 | 33.5 | 3.178 |
| 6.91 | 6.73 | 29.957 | 33.5 | 3.256 |
| 6.91 | 6.73 | 29.955 | 33.6 | 4.092 |
| 6.91 | 6.73 | 29.955 | 33.6 | 4.161 |
| 6.91 | 6.73 | 29.95 | 33.6 | 4.252 |

| 6.91 | 6.73 | 29.942 | 33.6 | 4.341 |
| 6.91 | 6.72 | 29.934 | 33.6 | 4.423 |
| 6.91 | 6.72 | 29.935 | 33.7 | 4.508 |
| 6.91 | 6.72 | 29.938 | 33.7 | 4.592 |
| 6.92 | 6.72 | 29.939 | 33.7 | 5.364 |
| 6.92 | 6.72 | 29.936 | 33.7 | 5.467 |
| 6.92 | 6.72 | 29.932 | 33.7 | 5.517 |
| 6.92 | 6.72 | 29.932 | 33.8 | 5.578 |
| 6.92 | 6.72 | 29.934 | 33.8 | 5.669 |
| 6.91 | 6.73 | 29.936 | 33.8 | 5.769 |
| 6.91 | 6.73 | 29.935 | 33.8 | 5.861 |
| 6.91 | 6.73 | 29.934 | 33.8 | 5.939 |
| 6.91 | 6.73 | 29.931 | 33.8 | 6.011 |
| 6.92 | 6.73 | 29.929 | 33.8 | 6.169 |
| 6.92 | 6.73 | 29.928 | 33.8 | 6.325 |
| 6.92 | 6.73 | 29.928 | 33.8 | 6.471 |
| 6.92 | 6.73 | 29.928 | 33.8 | 6.587 |
| 6.92 | 6.73 | 29.928 | 33.9 | 6.693 |
| 6.91 | 6.73 | 29.927 | 33.9 | 6.799 |
| 6.91 | 6.73 | 29.929 | 33.8 | 6.92 |
| 6.92 | 6.73 | 29.93 | 33.9 | 7.041 |
| 6.92 | 6.73 | 29.931 | 33.9 | 7.153 |
| 6.92 | 6.74 | 29.932 | 33.9 | 7.267 |
| 6.92 | 6.74 | 29.932 | 33.9 | 7.39 |
| 6.92 | 6.74 | 29.931 | 33.9 | 7.536 |
| 6.91 | 6.74 | 29.93 | 33.9 | 8.263 |
| 6.91 | 6.74 | 29.931 | 33.9 | 8.431 |
| 6.91 | 6.74 | 29.933 | 33.9 | 8.58 |
| 6.91 | 6.74 | 29.934 | 33.9 | 8.697 |
| 6.92 | 6.74 | 29.934 | 33.9 | 8.815 |
| 6.92 | 6.74 | 29.932 | 33.9 | 8.922 |
| 6.92 | 6.74 | 29.929 | 33.9 | 9.034 |
| 6.92 | 6.74 | 29.925 | 33.9 | 9.92 |
| 6.92 | 6.74 | 29.925 | 33.9 | 9.999 |
| 6.92 | 6.74 | 29.925 | 33.9 | 10.004 |
| 6.91 | 6.74 | 29.926 | 33.9 | 9.96 |
| 6.91 | 6.74 | 29.927 | 33.9 | 10.078 |
| 6.91 | 6.74 | 29.927 | 33.9 | 10.918 |
| 6.92 | 6.74 | 29.928 | 33.9 | 11.088 |
| 6.92 | 6.74 | 29.927 | 33.9 | 11.335 |
| 6.92 | 6.74 | 29.926 | 33.8 | 11.414 |
| 6.91 | 6.74 | 29.925 | 33.8 | 11.512 |
| 6.91 | 6.74 | 29.925 | 33.9 | 11.602 |
| 6.91 | 6.74 | 29.925 | 33.9 | 11.709 |
| 6.91 | 6.74 | 29.925 | 33.9 | 12.506 |

| | | | | |
|---|---|---|---|---|
| 6.91 | 6.73 | 29.926 | 33.8 | 12.738 |
| 6.91 | 6.73 | 29.925 | 33.8 | 12.784 |
| 6.91 | 6.74 | 29.925 | 33.8 | 12.891 |
| 6.91 | 6.74 | 29.924 | 33.8 | 13.004 |
| 6.91 | 6.73 | 29.924 | 33.8 | 13.112 |
| 6.91 | 6.73 | 29.924 | 33.8 | 13.216 |
| 6.91 | 6.73 | 29.925 | 33.8 | 14.001 |
| 6.91 | 6.73 | 29.926 | 33.8 | 14.137 |
| 6.91 | 6.73 | 29.926 | 33.8 | 14.259 |
| 6.91 | 6.73 | 29.926 | 33.8 | 14.39 |
| 6.9 | 6.73 | 29.926 | 33.8 | 14.458 |
| 6.9 | 6.73 | 29.925 | 33.8 | 15.245 |
| 6.89 | 6.73 | 29.924 | 33.8 | 15.463 |
| 6.89 | 6.73 | 29.923 | 33.8 | 15.606 |
| 6.88 | 6.72 | 29.923 | 33.8 | 15.74 |
| 6.87 | 6.72 | 29.923 | 33.8 | 15.932 |
| 6.85 | 6.72 | 29.924 | 33.8 | 16.843 |
| 6.83 | 6.72 | 29.924 | 33.8 | 17.246 |
| 6.73 | 6.72 | 29.924 | 33.8 | 17.329 |
| 6.72 | 6.71 | 29.924 | 33.8 | 17.501 |
| 6.72 | 6.71 | 29.924 | 33.8 | 18.764 |
| 6.71 | 6.71 | 29.924 | 33.8 | 19.109 |
| 6.71 | 6.71 | 29.924 | 33.8 | 19.325 |

93

94    Table S20 – P13 site depth profile from 2017 campaign.

| pH | ODO mg/L | Temp °C | Cond µS/cm | Depth m |
|---|---|---|---|---|
| 6.9 | 6.88 | 30.218 | 34.5 | 0.142 |
| 6.91 | 6.88 | 30.22 | 34.5 | 0.202 |
| 6.91 | 6.88 | 30.219 | 34.5 | 0.265 |
| 6.91 | 6.88 | 30.2 | 34.5 | 0.335 |
| 6.91 | 6.86 | 30.165 | 34.4 | 0.404 |
| 6.91 | 6.83 | 30.125 | 34.3 | 0.488 |
| 6.9 | 6.81 | 30.099 | 34.2 | 0.567 |
| 6.89 | 6.79 | 30.084 | 34.1 | 0.629 |
| 6.88 | 6.78 | 30.08 | 34 | 0.695 |
| 6.87 | 6.77 | 30.074 | 33.9 | 0.771 |
| 6.86 | 6.76 | 30.063 | 33.8 | 0.845 |
| 6.86 | 6.75 | 30.044 | 33.7 | 1.737 |
| 6.86 | 6.73 | 30.009 | 33.6 | 1.942 |
| 6.85 | 6.7 | 29.969 | 33.5 | 2.06 |
| 6.85 | 6.66 | 29.937 | 33.4 | 2.173 |
| 6.84 | 6.62 | 29.919 | 33.3 | 2.274 |

| | | | | |
|---|---|---|---|---|
| 6.84 | 6.59 | 29.91 | 33.3 | 2.35 |
| 6.83 | 6.56 | 29.904 | 33.3 | 2.411 |
| 6.82 | 6.54 | 29.899 | 33.3 | 2.467 |
| 6.82 | 6.53 | 29.891 | 33.2 | 2.527 |
| 6.81 | 6.51 | 29.884 | 33.2 | 2.676 |
| 6.8 | 6.49 | 29.88 | 33.2 | 2.821 |
| 6.8 | 6.48 | 29.878 | 33.2 | 2.959 |
| 6.8 | 6.48 | 29.877 | 33.2 | 3.1 |
| 6.8 | 6.47 | 29.876 | 33.2 | 3.254 |
| 6.8 | 6.46 | 29.876 | 33.2 | 3.419 |
| 6.8 | 6.46 | 29.875 | 33.2 | 3.588 |
| 6.8 | 6.45 | 29.875 | 33.3 | 3.753 |
| 6.8 | 6.45 | 29.874 | 33.3 | 3.913 |
| 6.8 | 6.45 | 29.873 | 33.3 | 4.056 |
| 6.8 | 6.44 | 29.872 | 33.3 | 4.196 |
| 6.8 | 6.44 | 29.872 | 33.3 | 4.343 |
| 6.8 | 6.43 | 29.871 | 33.4 | 5.221 |
| 6.8 | 6.43 | 29.871 | 33.4 | 5.326 |
| 6.79 | 6.42 | 29.87 | 33.5 | 5.479 |
| 6.79 | 6.4 | 29.867 | 33.5 | 5.563 |
| 6.78 | 6.39 | 29.865 | 33.5 | 5.617 |
| 6.78 | 6.39 | 29.864 | 33.6 | 5.679 |
| 6.77 | 6.38 | 29.862 | 33.6 | 5.782 |
| 6.77 | 6.37 | 29.859 | 33.7 | 5.88 |
| 6.77 | 6.36 | 29.857 | 33.7 | 5.967 |
| 6.77 | 6.35 | 29.856 | 33.8 | 6.117 |
| 6.77 | 6.35 | 29.855 | 33.8 | 6.264 |

95

96    Table S21 – P12 site depth profile from 2017 campaign.

| pH | ODO mg/L | Temp °C | Cond µS/cm | Depth m |
|---|---|---|---|---|
| 7.02 | 7.03 | 30.424 | 34.5 | 0.317 |
| 7.01 | 7.03 | 30.419 | 34.4 | 0.467 |
| 7 | 7.03 | 30.393 | 34.5 | 0.631 |
| 6.99 | 7.03 | 30.358 | 34.5 | 0.809 |
| 6.98 | 7.02 | 30.335 | 34.4 | 0.953 |
| 6.97 | 7.02 | 30.328 | 34.4 | 1.063 |
| 6.96 | 7.01 | 30.318 | 34.4 | 1.169 |
| 6.96 | 7.01 | 30.291 | 34.4 | 1.257 |
| 6.96 | 7 | 30.251 | 34.5 | 1.341 |
| 6.95 | 6.97 | 30.21 | 34.4 | 1.43 |
| 6.94 | 6.94 | 30.168 | 34.4 | 1.522 |
| 6.93 | 6.9 | 30.123 | 34.5 | 1.613 |

| pH | ODO mg/L | Temp °C | Cond µS/cm | Depth m |
|---|---|---|---|---|
| 6.92 | 6.86 | 30.081 | 34.5 | 1.709 |
| 6.9 | 6.82 | 30.051 | 34.5 | 1.807 |
| 6.89 | 6.77 | 30.021 | 34.5 | 1.899 |
| 6.87 | 6.71 | 29.988 | 34.5 | 1.994 |
| 6.86 | 6.65 | 29.96 | 34.5 | 2.086 |
| 6.85 | 6.61 | 29.949 | 34.6 | 2.176 |
| 6.84 | 6.58 | 29.94 | 34.6 | 2.267 |
| 6.83 | 6.56 | 29.932 | 34.7 | 2.358 |
| 6.83 | 6.54 | 29.924 | 34.7 | 2.45 |
| 6.82 | 6.52 | 29.918 | 34.7 | 2.546 |
| 6.82 | 6.5 | 29.904 | 34.8 | 2.65 |
| 6.81 | 6.46 | 29.889 | 34.8 | 2.761 |
| 6.81 | 6.41 | 29.874 | 34.8 | 2.873 |
| 6.8 | 6.37 | 29.867 | 34.9 | 2.99 |
| 6.79 | 6.34 | 29.861 | 34.9 | 3.098 |
| 6.78 | 6.32 | 29.86 | 34.9 | 3.195 |
| 6.77 | 6.31 | 29.856 | 35 | 3.309 |
| 6.76 | 6.29 | 29.849 | 35 | 3.448 |
| 6.76 | 6.26 | 29.841 | 35 | 3.585 |
| 6.75 | 6.23 | 29.834 | 35.1 | 4.462 |
| 6.74 | 6.2 | 29.827 | 35.1 | 4.713 |
| 6.74 | 6.17 | 29.822 | 35.1 | 4.77 |
| 6.73 | 6.14 | 29.819 | 35.2 | 4.856 |
| 6.73 | 6.12 | 29.817 | 35.2 | 4.938 |
| 6.72 | 6.11 | 29.814 | 35.2 | 5.014 |
| 6.72 | 6.09 | 29.811 | 35.3 | 5.099 |
| 6.72 | 6.07 | 29.81 | 35.3 | 5.182 |
| 6.72 | 6.06 | 29.81 | 35.3 | 5.25 |
| 6.72 | 6.05 | 29.81 | 35.3 | 5.357 |
| 6.71 | 6.04 | 29.808 | 35.3 | 5.474 |
| 6.71 | 6.03 | 29.806 | 35.3 | 5.583 |
| 6.71 | 6.01 | 29.803 | 35.4 | 5.665 |
| 6.7 | 5.99 | 29.8 | 35.4 | 5.729 |
| 6.7 | 5.97 | 29.795 | 35.4 | 5.778 |
| 6.69 | 5.93 | 29.788 | 35.4 | 5.83 |
| 6.68 | 5.86 | 29.778 | 35.5 | 5.885 |
| 6.66 | 5.75 | 29.769 | 35.6 | 5.951 |

97

98    Table S22 – P15 site depth profile from 2017 campaign.

| pH | ODO mg/L | Temp °C | Cond µS/cm | Depth m |
|---|---|---|---|---|
| 7.09 | 7.03 | 29.972 | 34.9 | 1.12 |
| 7.07 | 6.99 | 29.929 | 34.7 | 1.235 |

| | | | | |
|---|---|---|---|---|
| 7.05 | 6.96 | 29.907 | 34.5 | 1.4 |
| 7.04 | 6.94 | 29.898 | 34.4 | 1.552 |
| 7.03 | 6.92 | 29.891 | 34.3 | 1.675 |
| 7.02 | 6.9 | 29.887 | 34.2 | 1.776 |
| 7 | 6.89 | 29.884 | 34.1 | 1.9 |
| 7 | 6.89 | 29.88 | 34 | 2.11 |
| 6.99 | 6.88 | 29.874 | 33.9 | 2.314 |
| 6.98 | 6.88 | 29.869 | 33.8 | 2.497 |
| 6.98 | 6.87 | 29.866 | 33.8 | 2.642 |
| 6.98 | 6.86 | 29.863 | 33.8 | 2.775 |
| 6.98 | 6.86 | 29.863 | 33.7 | 2.911 |
| 6.98 | 6.86 | 29.863 | 33.7 | 3.047 |
| 6.97 | 6.86 | 29.866 | 33.7 | 3.171 |
| 6.97 | 6.86 | 29.864 | 33.7 | 3.277 |
| 6.97 | 6.85 | 29.857 | 33.7 | 3.378 |
| 6.97 | 6.85 | 29.849 | 33.6 | 3.479 |
| 6.96 | 6.83 | 29.846 | 33.6 | 3.598 |
| 6.96 | 6.83 | 29.844 | 33.6 | 3.714 |
| 6.95 | 6.82 | 29.843 | 33.7 | 3.827 |
| 6.95 | 6.82 | 29.84 | 33.7 | 3.955 |
| 6.95 | 6.81 | 29.836 | 33.7 | 4.087 |
| 6.95 | 6.81 | 29.831 | 33.7 | 4.208 |
| 6.95 | 6.8 | 29.823 | 33.8 | 4.331 |
| 6.94 | 6.79 | 29.816 | 33.8 | 4.441 |
| 6.94 | 6.78 | 29.813 | 33.8 | 4.54 |
| 6.94 | 6.78 | 29.814 | 33.8 | 4.636 |
| 6.93 | 6.77 | 29.815 | 33.9 | 4.714 |
| 6.93 | 6.77 | 29.815 | 33.9 | 4.782 |
| 6.93 | 6.77 | 29.814 | 33.9 | 4.873 |
| 6.93 | 6.77 | 29.813 | 33.9 | 4.97 |
| 6.93 | 6.77 | 29.811 | 34 | 5.075 |
| 6.93 | 6.77 | 29.809 | 34 | 5.192 |
| 6.92 | 6.76 | 29.807 | 34 | 5.329 |
| 6.92 | 6.76 | 29.808 | 34.1 | 5.483 |
| 6.92 | 6.75 | 29.808 | 34.1 | 5.66 |
| 6.92 | 6.75 | 29.809 | 34.1 | 5.818 |
| 6.92 | 6.75 | 29.81 | 34.1 | 5.976 |
| 6.92 | 6.76 | 29.81 | 34.2 | 6.139 |
| 6.92 | 6.76 | 29.81 | 34.2 | 6.306 |
| 6.92 | 6.76 | 29.81 | 34.2 | 6.461 |
| 6.92 | 6.76 | 29.809 | 34.2 | 6.617 |
| 6.92 | 6.76 | 29.809 | 34.2 | 6.753 |
| 6.92 | 6.76 | 29.808 | 34.3 | 6.888 |
| 6.92 | 6.76 | 29.807 | 34.3 | 7.023 |
| 6.92 | 6.75 | 29.806 | 34.3 | 7.153 |

| 6.92 | 6.75 | 29.806 | 34.3 | 7.266 |
|------|------|--------|------|-------|
| 6.92 | 6.76 | 29.806 | 34.3 | 7.383 |
| 6.92 | 6.75 | 29.806 | 34.3 | 7.493 |
| 6.92 | 6.75 | 29.807 | 34.3 | 7.62 |
| 6.92 | 6.75 | 29.806 | 34.3 | 7.76 |
| 6.92 | 6.74 | 29.805 | 34.3 | 8.611 |
| 6.92 | 6.74 | 29.805 | 34.4 | 8.702 |
| 6.92 | 6.73 | 29.806 | 34.4 | 8.751 |
| 6.92 | 6.73 | 29.806 | 34.4 | 8.833 |
| 6.92 | 6.73 | 29.805 | 34.4 | 8.898 |
| 6.92 | 6.73 | 29.806 | 34.4 | 8.966 |
| 6.91 | 6.73 | 29.806 | 34.4 | 9.038 |
| 6.91 | 6.73 | 29.805 | 34.4 | 9.117 |
| 6.91 | 6.73 | 29.806 | 34.5 | 9.235 |
| 6.91 | 6.72 | 29.806 | 34.5 | 9.396 |
| 6.9 | 6.72 | 29.806 | 34.4 | 9.554 |
| 6.9 | 6.72 | 29.806 | 34.5 | 9.719 |
| 6.9 | 6.72 | 29.806 | 34.5 | 10.579 |
| 6.9 | 6.71 | 29.806 | 34.5 | 10.639 |
| 6.91 | 6.71 | 29.806 | 34.5 | 10.75 |
| 6.91 | 6.71 | 29.806 | 34.5 | 10.854 |
| 6.91 | 6.7 | 29.807 | 34.5 | 10.932 |
| 6.9 | 6.7 | 29.807 | 34.5 | 10.99 |

99

100    Table S23 – P14 site depth profile from 2017 campaign.

| pH | ODO mg/L | Temp °C | Cond µS/cm | Depth m |
|------|------|--------|------|-------|
| 6.84 | 6.67 | 29.657 | 33.1 | 0.448 |
| 6.85 | 6.67 | 29.657 | 33.1 | 0.51 |
| 6.85 | 6.67 | 29.656 | 33.1 | 0.568 |
| 6.85 | 6.67 | 29.656 | 33.1 | 0.602 |
| 6.85 | 6.67 | 29.657 | 33.1 | 0.632 |
| 6.85 | 6.67 | 29.656 | 33.1 | 0.66 |
| 6.84 | 6.67 | 29.656 | 33.1 | 0.672 |
| 6.84 | 6.67 | 29.655 | 33.2 | 0.687 |
| 6.84 | 6.67 | 29.655 | 33.2 | 0.713 |
| 6.85 | 6.67 | 29.655 | 33.2 | 0.749 |
| 6.85 | 6.67 | 29.654 | 33.2 | 0.788 |
| 6.85 | 6.67 | 29.654 | 33.3 | 0.853 |
| 6.84 | 6.67 | 29.653 | 33.3 | 0.929 |
| 6.84 | 6.67 | 29.654 | 33.3 | 0.997 |
| 6.84 | 6.67 | 29.654 | 33.3 | 1.056 |
| 6.84 | 6.67 | 29.654 | 33.4 | 1.109 |

| | | | | |
|---|---|---|---|---|
| 6.84 | 6.67 | 29.654 | 33.4 | 1.157 |
| 6.84 | 6.67 | 29.653 | 33.4 | 1.189 |
| 6.84 | 6.67 | 29.654 | 33.5 | 1.22 |
| 6.84 | 6.67 | 29.653 | 33.5 | 1.229 |
| 6.83 | 6.67 | 29.653 | 33.6 | 1.227 |
| 6.83 | 6.67 | 29.654 | 33.6 | 1.236 |
| 6.83 | 6.67 | 29.654 | 33.7 | 1.264 |
| 6.83 | 6.67 | 29.654 | 33.7 | 1.325 |
| 6.83 | 6.67 | 29.655 | 33.7 | 1.378 |
| 6.83 | 6.67 | 29.655 | 33.8 | 1.433 |
| 6.83 | 6.67 | 29.655 | 33.8 | 1.481 |
| 6.83 | 6.67 | 29.655 | 33.9 | 1.543 |
| 6.83 | 6.67 | 29.655 | 33.9 | 1.617 |
| 6.83 | 6.67 | 29.656 | 33.9 | 1.696 |
| 6.83 | 6.67 | 29.656 | 34 | 1.771 |
| 6.83 | 6.67 | 29.657 | 34 | 1.842 |
| 6.83 | 6.67 | 29.657 | 34.1 | 1.918 |
| 6.83 | 6.67 | 29.657 | 34.1 | 2.002 |
| 6.83 | 6.67 | 29.656 | 34.1 | 2.861 |
| 6.83 | 6.67 | 29.656 | 34.2 | 2.924 |
| 6.83 | 6.66 | 29.656 | 34.2 | 2.922 |
| 6.83 | 6.66 | 29.655 | 34.2 | 2.852 |
| 6.83 | 6.66 | 29.655 | 34.3 | 2.777 |
| 6.83 | 6.66 | 29.654 | 34.3 | 2.729 |

101

102    Table S24 – P16 site depth profile from 2017 campaign.

| pH | ODO mg/L | Temp °C | Cond µS/cm | Depth m |
|---|---|---|---|---|
| 6.8 | 6.65 | 30.465 | 35.4 | 0.337 |
| 6.8 | 6.64 | 30.437 | 35.2 | 0.396 |
| 6.8 | 6.63 | 30.383 | 35 | 0.447 |
| 6.81 | 6.62 | 30.332 | 34.8 | 0.493 |
| 6.81 | 6.61 | 30.329 | 34.7 | 0.538 |
| 6.8 | 6.61 | 30.349 | 34.6 | 0.587 |
| 6.8 | 6.6 | 30.326 | 34.5 | 0.642 |
| 6.8 | 6.59 | 30.249 | 34.4 | 0.702 |
| 6.8 | 6.58 | 30.158 | 34.2 | 0.771 |
| 6.8 | 6.57 | 30.09 | 34.1 | 0.839 |
| 6.8 | 6.55 | 30.04 | 34 | 0.905 |
| 6.79 | 6.53 | 30.012 | 34 | 0.977 |
| 6.78 | 6.5 | 29.985 | 34 | 1.052 |
| 6.77 | 6.49 | 29.963 | 33.9 | 1.137 |
| 6.77 | 6.48 | 29.94 | 33.9 | 1.24 |

| | | | | |
|------|------|--------|------|-------|
| 6.76 | 6.47 | 29.919 | 33.9 | 1.349 |
| 6.76 | 6.45 | 29.888 | 33.9 | 1.473 |
| 6.76 | 6.43 | 29.865 | 33.8 | 1.59 |
| 6.75 | 6.42 | 29.851 | 33.8 | 1.702 |
| 6.74 | 6.41 | 29.843 | 33.8 | 1.811 |
| 6.74 | 6.4 | 29.837 | 33.8 | 1.925 |
| 6.73 | 6.4 | 29.834 | 33.8 | 2.032 |
| 6.73 | 6.39 | 29.828 | 33.9 | 2.133 |
| 6.73 | 6.39 | 29.823 | 33.9 | 2.231 |
| 6.74 | 6.38 | 29.822 | 33.9 | 2.328 |
| 6.74 | 6.38 | 29.82 | 33.9 | 2.421 |
| 6.74 | 6.38 | 29.82 | 33.9 | 2.524 |
| 6.75 | 6.38 | 29.824 | 34 | 2.636 |
| 6.75 | 6.38 | 29.83 | 34 | 2.74 |
| 6.74 | 6.38 | 29.834 | 34 | 2.838 |
| 6.74 | 6.38 | 29.839 | 34 | 2.942 |
| 6.74 | 6.38 | 29.841 | 34 | 3.048 |
| 6.74 | 6.38 | 29.834 | 34.1 | 3.158 |
| 6.74 | 6.38 | 29.818 | 34.1 | 3.269 |
| 6.74 | 6.37 | 29.805 | 34.1 | 3.373 |
| 6.74 | 6.36 | 29.802 | 34.2 | 3.48 |
| 6.74 | 6.36 | 29.806 | 34.2 | 3.588 |
| 6.74 | 6.36 | 29.812 | 34.2 | 3.699 |
| 6.73 | 6.36 | 29.809 | 34.2 | 3.822 |
| 6.73 | 6.35 | 29.793 | 34.2 | 3.94 |
| 6.73 | 6.35 | 29.775 | 34.2 | 4.056 |
| 6.72 | 6.34 | 29.756 | 34.2 | 4.184 |
| 6.72 | 6.33 | 29.739 | 34.2 | 4.305 |
| 6.72 | 6.32 | 29.723 | 34.3 | 4.427 |
| 6.73 | 6.31 | 29.699 | 34.3 | 4.545 |
| 6.73 | 6.31 | 29.683 | 34.3 | 4.657 |
| 6.73 | 6.29 | 29.675 | 34.3 | 4.761 |
| 6.72 | 6.28 | 29.674 | 34.3 | 4.866 |
| 6.72 | 6.26 | 29.674 | 34.3 | 4.968 |
| 6.72 | 6.25 | 29.673 | 34.3 | 5.057 |
| 6.72 | 6.24 | 29.67 | 34.4 | 5.148 |
| 6.72 | 6.23 | 29.662 | 34.4 | 5.229 |
| 6.72 | 6.23 | 29.657 | 34.4 | 5.309 |
| 6.72 | 6.22 | 29.656 | 34.4 | 5.373 |
| 6.72 | 6.21 | 29.659 | 34.4 | 5.432 |
| 6.72 | 6.21 | 29.661 | 34.4 | 5.484 |
| 6.72 | 6.21 | 29.666 | 34.4 | 5.548 |
| 6.72 | 6.21 | 29.67 | 34.4 | 5.621 |
| 6.72 | 6.21 | 29.672 | 34.4 | 5.697 |
| 6.72 | 6.21 | 29.667 | 34.4 | 5.771 |

| | | | | |
|---|---|---|---|---|
| 6.72 | 6.21 | 29.661 | 34.4 | 5.854 |
| 6.72 | 6.2 | 29.657 | 34.4 | 5.939 |
| 6.72 | 6.2 | 29.657 | 34.4 | 6.034 |
| 6.71 | 6.19 | 29.658 | 34.4 | 6.126 |
| 6.71 | 6.19 | 29.66 | 34.4 | 6.161 |
| 6.7 | 6.19 | 29.662 | 34.4 | 6.177 |
| 6.7 | 6.21 | 29.662 | 34.4 | 6.223 |
| 6.69 | 6.21 | 29.659 | 34.4 | 6.284 |
| 6.69 | 6.19 | 29.659 | 34.4 | 6.356 |
| 6.69 | 6.16 | 29.657 | 34.5 | 6.432 |
| 6.7 | 6.15 | 29.654 | 34.5 | 6.508 |
| 6.7 | 6.14 | 29.648 | 34.5 | 6.597 |
| 6.7 | 6.14 | 29.642 | 34.5 | 6.752 |
| 6.7 | 6.14 | 29.635 | 34.5 | 6.877 |
| 6.7 | 6.15 | 29.632 | 34.5 | 6.982 |
| 6.7 | 6.15 | 29.632 | 34.5 | 7.067 |
| 6.7 | 6.15 | 29.632 | 34.5 | 7.149 |
| 6.7 | 6.14 | 29.632 | 34.5 | 7.237 |
| 6.69 | 6.14 | 29.632 | 34.5 | 7.318 |
| 6.69 | 6.15 | 29.631 | 34.5 | 7.394 |
| 6.69 | 6.15 | 29.63 | 34.5 | 7.472 |
| 6.69 | 6.15 | 29.63 | 34.5 | 7.552 |
| 6.69 | 6.16 | 29.63 | 34.5 | 7.636 |
| 6.69 | 6.16 | 29.63 | 34.5 | 7.739 |
| 6.69 | 6.16 | 29.629 | 34.5 | 7.848 |
| 6.69 | 6.16 | 29.628 | 34.5 | 7.972 |
| 6.69 | 6.16 | 29.627 | 34.5 | 8.118 |
| 6.69 | 6.17 | 29.627 | 34.5 | 8.275 |
| 6.69 | 6.17 | 29.628 | 34.5 | 8.437 |
| 6.7 | 6.17 | 29.628 | 34.5 | 8.611 |
| 6.7 | 6.18 | 29.628 | 34.4 | 8.781 |
| 6.71 | 6.18 | 29.628 | 34.5 | 8.93 |
| 6.71 | 6.19 | 29.627 | 34.5 | 9.058 |
| 6.71 | 6.19 | 29.626 | 34.5 | 9.163 |
| 6.72 | 6.2 | 29.625 | 34.4 | 9.25 |
| 6.72 | 6.2 | 29.625 | 34.4 | 9.316 |
| 6.72 | 6.2 | 29.625 | 34.5 | 9.365 |
| 6.72 | 6.2 | 29.626 | 34.5 | 9.421 |
| 6.71 | 6.2 | 29.626 | 34.4 | 9.48 |
| 6.71 | 6.2 | 29.625 | 34.4 | 9.546 |
| 6.71 | 6.2 | 29.624 | 34.4 | 9.626 |
| 6.71 | 6.2 | 29.623 | 34.4 | 9.726 |
| 6.71 | 6.2 | 29.624 | 34.4 | 9.806 |
| 6.71 | 6.2 | 29.624 | 34.4 | 9.908 |
| 6.71 | 6.2 | 29.623 | 34.5 | 10.01 |

| | | | | |
|------|------|--------|------|--------|
| 6.71 | 6.21 | 29.622 | 34.4 | 10.11 |
| 6.71 | 6.21 | 29.622 | 34.4 | 10.218 |
| 6.71 | 6.21 | 29.622 | 34.4 | 10.317 |
| 6.71 | 6.21 | 29.622 | 34.5 | 10.402 |
| 6.72 | 6.21 | 29.622 | 34.5 | 10.486 |
| 6.72 | 6.21 | 29.622 | 34.4 | 10.593 |
| 6.71 | 6.2  | 29.621 | 34.5 | 10.712 |
| 6.71 | 6.2  | 29.621 | 34.5 | 10.842 |
| 6.71 | 6.2  | 29.62  | 34.5 | 10.974 |
| 6.71 | 6.2  | 29.619 | 34.4 | 11.839 |
| 6.71 | 6.21 | 29.619 | 34.4 | 11.939 |
| 6.71 | 6.21 | 29.618 | 34.5 | 11.905 |
| 6.71 | 6.21 | 29.616 | 34.5 | 11.956 |
| 6.71 | 6.21 | 29.615 | 34.5 | 12.009 |
| 6.71 | 6.21 | 29.614 | 34.5 | 12.065 |
| 6.71 | 6.21 | 29.613 | 34.4 | 12.134 |
| 6.71 | 6.21 | 29.611 | 34.4 | 12.196 |
| 6.7  | 6.2  | 29.607 | 34.5 | 12.257 |
| 6.7  | 6.18 | 29.604 | 34.5 | 12.358 |
| 6.69 | 6.16 | 29.601 | 34.5 | 12.463 |
| 6.69 | 6.14 | 29.601 | 34.5 | 12.565 |
| 6.68 | 6.13 | 29.598 | 34.5 | 12.663 |
| 6.68 | 6.11 | 29.595 | 34.5 | 12.762 |
| 6.67 | 6.08 | 29.586 | 34.6 | 12.862 |
| 6.66 | 6.01 | 29.57  | 34.6 | 12.976 |
| 6.64 | 5.88 | 29.555 | 34.7 | 13.063 |
| 6.62 | 5.75 | 29.546 | 34.8 | 13.183 |
| 6.6  | 5.67 | 29.545 | 34.8 | 13.306 |
| 6.58 | 5.64 | 29.545 | 34.8 | 13.441 |
| 6.57 | 5.63 | 29.545 | 34.8 | 13.582 |
| 6.57 | 5.63 | 29.55  | 34.8 | 13.724 |
| 6.57 | 5.65 | 29.555 | 34.8 | 13.832 |
| 6.57 | 5.69 | 29.56  | 34.7 | 13.957 |
| 6.57 | 5.69 | 29.555 | 34.7 | 14.106 |
| 6.56 | 5.62 | 29.54  | 34.8 | 14.247 |
| 6.55 | 5.52 | 29.526 | 34.9 | 14.391 |
| 6.54 | 5.46 | 29.52  | 34.9 | 14.54 |
| 6.52 | 5.43 | 29.516 | 34.9 | 15.477 |
| 6.51 | 5.39 | 29.507 | 34.9 | 15.554 |
| 6.5  | 5.33 | 29.504 | 34.9 | 15.655 |
| 6.49 | 5.29 | 29.501 | 34.9 | 15.748 |
| 6.48 | 5.27 | 29.498 | 34.9 | 15.838 |
| 6.48 | 5.24 | 29.494 | 35   | 15.94 |
| 6.47 | 5.23 | 29.491 | 35   | 16.032 |
| 6.47 | 5.21 | 29.491 | 35   | 16.117 |

| | | | | |
|---|---|---|---|---|
| 6.47 | 5.2 | 29.491 | 34.9 | 16.23 |
| 6.47 | 5.2 | 29.493 | 34.9 | 16.404 |
| 6.47 | 5.2 | 29.491 | 35 | 16.549 |
| 6.46 | 5.19 | 29.486 | 35 | 16.683 |
| 6.46 | 5.17 | 29.477 | 35 | 16.815 |
| 6.46 | 5.15 | 29.473 | 35 | 16.925 |
| 6.45 | 5.13 | 29.473 | 35 | 17.051 |
| 6.45 | 5.13 | 29.469 | 35 | 17.186 |
| 6.45 | 5.11 | 29.46 | 35.1 | 17.324 |
| 6.44 | 5.07 | 29.448 | 35 | 17.947 |
| 6.43 | 5.04 | 29.44 | 35 | 18.037 |
| 6.43 | 5.01 | 29.436 | 35 | 18.148 |
| 6.43 | 5 | 29.429 | 35.1 | 18.281 |
| 6.43 | 4.97 | 29.41 | 35.1 | 18.399 |
| 6.42 | 4.88 | 29.376 | 35.2 | 18.513 |
| 6.4 | 4.53 | 29.34 | 35.2 | 19.16 |

103

104    Table S25 – P17 site depth profile from 2017 campaign.

| pH | ODO mg/L | Temp °C | Cond µS/cm | Depth m |
|---|---|---|---|---|
| 6.94 | 7.09 | 30.432 | 36.6 | 0.215 |
| 6.94 | 6.99 | 30.047 | 36.3 | 0.345 |
| 6.92 | 6.91 | 29.839 | 36 | 0.496 |
| 6.89 | 6.85 | 29.729 | 35.9 | 0.626 |
| 6.86 | 6.71 | 29.696 | 35.8 | 0.759 |
| 6.83 | 6.63 | 29.672 | 35.7 | 0.893 |
| 6.8 | 6.55 | 29.647 | 35.6 | 1.049 |
| 6.78 | 6.5 | 29.624 | 35.5 | 1.21 |
| 6.76 | 6.45 | 29.607 | 35.4 | 1.362 |
| 6.75 | 6.41 | 29.593 | 35.4 | 1.506 |
| 6.74 | 6.38 | 29.584 | 35.3 | 1.65 |
| 6.74 | 6.36 | 29.577 | 35.2 | 1.798 |
| 6.73 | 6.35 | 29.57 | 35.2 | 1.969 |
| 6.73 | 6.33 | 29.564 | 35.2 | 2.109 |
| 6.72 | 6.31 | 29.56 | 35.1 | 2.23 |
| 6.72 | 6.3 | 29.557 | 35.1 | 2.341 |
| 6.71 | 6.28 | 29.553 | 35 | 2.438 |
| 6.7 | 6.27 | 29.548 | 35 | 2.534 |
| 6.7 | 6.25 | 29.545 | 35 | 2.632 |
| 6.7 | 6.24 | 29.542 | 34.9 | 2.725 |
| 6.7 | 6.22 | 29.539 | 34.9 | 2.803 |
| 6.7 | 6.21 | 29.536 | 34.9 | 2.894 |
| 6.69 | 6.2 | 29.533 | 34.8 | 2.995 |

| | | | | |
|---|---|---|---|---|
| 6.69 | 6.19 | 29.531 | 34.8 | 3.094 |
| 6.69 | 6.18 | 29.529 | 34.8 | 3.197 |
| 6.69 | 6.17 | 29.527 | 34.8 | 3.292 |
| 6.69 | 6.16 | 29.525 | 34.7 | 3.38 |
| 6.69 | 6.16 | 29.522 | 34.7 | 3.47 |
| 6.69 | 6.14 | 29.503 | 34.7 | 3.561 |
| 6.68 | 6.1 | 29.486 | 34.7 | 3.645 |
| 6.68 | 6.05 | 29.464 | 34.7 | 3.729 |
| 6.67 | 5.99 | 29.444 | 34.7 | 3.816 |
| 6.66 | 5.94 | 29.431 | 34.7 | 3.9 |
| 6.66 | 5.89 | 29.42 | 34.7 | 3.988 |
| 6.65 | 5.86 | 29.415 | 34.7 | 4.086 |
| 6.65 | 5.84 | 29.423 | 34.7 | 4.184 |
| 6.64 | 5.83 | 29.434 | 34.7 | 4.287 |
| 6.64 | 5.83 | 29.427 | 34.7 | 4.389 |
| 6.63 | 5.8 | 29.412 | 34.7 | 4.489 |
| 6.62 | 5.76 | 29.403 | 34.7 | 4.584 |
| 6.61 | 5.74 | 29.397 | 34.7 | 4.678 |
| 6.61 | 5.72 | 29.389 | 34.7 | 4.765 |
| 6.61 | 5.7 | 29.384 | 34.7 | 4.846 |
| 6.61 | 5.69 | 29.384 | 34.7 | 4.927 |
| 6.61 | 5.69 | 29.386 | 34.7 | 5.002 |
| 6.61 | 5.69 | 29.388 | 34.6 | 5.081 |
| 6.62 | 5.69 | 29.386 | 34.7 | 5.156 |
| 6.62 | 5.69 | 29.382 | 34.6 | 5.229 |
| 6.62 | 5.69 | 29.378 | 34.6 | 5.304 |
| 6.62 | 5.68 | 29.377 | 34.6 | 5.384 |
| 6.62 | 5.67 | 29.375 | 34.6 | 5.464 |
| 6.61 | 5.67 | 29.375 | 34.6 | 5.545 |
| 6.61 | 5.66 | 29.373 | 34.6 | 5.626 |
| 6.61 | 5.66 | 29.373 | 34.6 | 5.706 |
| 6.61 | 5.66 | 29.372 | 34.6 | 5.788 |
| 6.61 | 5.65 | 29.372 | 34.6 | 5.871 |

105

106    Table S26 – P18 site depth profile from 2017 campaign.

| pH | ODO mg/L | Temp °C | Cond µS/cm | Depth m |
|---|---|---|---|---|
| 6.7 | 6.26 | 29.64 | 34.2 | 0.978 |
| 6.68 | 6.21 | 29.581 | 34.1 | 0.966 |
| 6.68 | 6.16 | 29.534 | 34.1 | 0.967 |
| 6.67 | 6.14 | 29.527 | 34.1 | 0.964 |
| 6.66 | 6.12 | 29.515 | 34.1 | 1.084 |
| 6.65 | 6.08 | 29.49 | 34 | 1.196 |

| | | | | |
|---|---|---|---|---|
| 6.64 | 6.01 | 29.454 | 34 | 2.034 |
| 6.63 | 5.94 | 29.426 | 33.9 | 2.033 |
| 6.61 | 5.89 | 29.41 | 33.8 | 2.139 |
| 6.61 | 5.86 | 29.41 | 33.8 | 2.29 |
| 6.6 | 5.86 | 29.414 | 33.7 | 2.361 |
| 6.6 | 5.85 | 29.415 | 33.6 | 2.425 |
| 6.6 | 5.85 | 29.413 | 33.5 | 2.531 |
| 6.6 | 5.84 | 29.41 | 33.4 | 2.607 |
| 6.6 | 5.84 | 29.408 | 33.4 | 3.492 |
| 6.6 | 5.83 | 29.406 | 33.3 | 3.612 |
| 6.6 | 5.82 | 29.407 | 33.2 | 3.612 |
| 6.6 | 5.81 | 29.409 | 33.1 | 3.611 |
| 6.59 | 5.81 | 29.409 | 33.1 | 3.767 |
| 6.59 | 5.8 | 29.406 | 33.1 | 3.862 |
| 6.6 | 5.8 | 29.401 | 33.1 | 3.921 |
| 6.6 | 5.79 | 29.401 | 33 | 3.983 |
| 6.59 | 5.79 | 29.402 | 33 | 4.071 |
| 6.6 | 5.79 | 29.404 | 33 | 4.197 |
| 6.6 | 5.79 | 29.406 | 33 | 4.333 |
| 6.59 | 5.79 | 29.408 | 33 | 5.268 |
| 6.59 | 5.78 | 29.409 | 32.9 | 5.263 |
| 6.6 | 5.78 | 29.41 | 32.9 | 5.264 |
| 6.6 | 5.78 | 29.409 | 33 | 5.325 |
| 6.6 | 5.78 | 29.409 | 32.9 | 5.43 |
| 6.6 | 5.78 | 29.412 | 32.9 | 5.486 |
| 6.6 | 5.79 | 29.423 | 33 | 5.521 |
| 6.6 | 5.79 | 29.435 | 33 | 5.594 |
| 6.6 | 5.79 | 29.442 | 33 | 5.706 |
| 6.61 | 5.8 | 29.445 | 33 | 5.838 |
| 6.61 | 5.8 | 29.446 | 33.1 | 5.975 |
| 6.6 | 5.8 | 29.447 | 33.1 | 6.098 |
| 6.6 | 5.8 | 29.448 | 33.1 | 6.227 |
| 6.6 | 5.81 | 29.45 | 33.1 | 6.357 |
| 6.6 | 5.81 | 29.448 | 33.2 | 7.172 |
| 6.59 | 5.8 | 29.441 | 33.2 | 7.171 |
| 6.58 | 5.8 | 29.422 | 33.2 | 7.165 |
| 6.58 | 5.78 | 29.399 | 33.2 | 7.18 |
| 6.56 | 5.75 | 29.367 | 33.3 | 7.329 |
| 6.54 | 5.7 | 29.341 | 33.3 | 7.418 |
| 6.52 | 5.66 | 29.329 | 33.3 | 7.475 |
| 6.49 | 5.64 | 29.326 | 33.4 | 8.308 |
| 6.48 | 5.63 | 29.323 | 33.4 | 8.301 |
| 6.47 | 5.62 | 29.323 | 33.4 | 8.303 |
| 6.46 | 5.61 | 29.322 | 33.5 | 8.408 |
| 6.45 | 5.6 | 29.321 | 33.5 | 8.524 |

| | | | | |
|------|------|--------|------|--------|
| 6.45 | 5.6  | 29.32  | 33.5 | 8.593  |
| 6.45 | 5.59 | 29.32  | 33.6 | 9.39   |
| 6.44 | 5.59 | 29.319 | 33.6 | 9.396  |
| 6.43 | 5.58 | 29.317 | 33.7 | 9.396  |
| 6.42 | 5.58 | 29.316 | 33.7 | 9.4    |
| 6.41 | 5.57 | 29.315 | 33.8 | 9.472  |
| 6.4  | 5.57 | 29.314 | 33.8 | 9.584  |
| 6.39 | 5.56 | 29.313 | 33.8 | 9.653  |
| 6.38 | 5.56 | 29.312 | 33.8 | 9.702  |
| 6.37 | 5.56 | 29.312 | 33.9 | 9.752  |
| 6.36 | 5.56 | 29.312 | 33.9 | 9.884  |
| 6.36 | 5.56 | 29.312 | 33.9 | 9.999  |
| 6.36 | 5.55 | 29.312 | 33.9 | 10.143 |
| 6.35 | 5.55 | 29.312 | 33.9 | 10.89  |
| 6.34 | 5.55 | 29.311 | 34   | 10.899 |
| 6.34 | 5.55 | 29.31  | 34   | 10.893 |
| 6.33 | 5.55 | 29.31  | 34   | 10.891 |
| 6.33 | 5.54 | 29.309 | 34   | 10.918 |
| 6.33 | 5.54 | 29.309 | 34   | 11.016 |
| 6.32 | 5.54 | 29.308 | 34   | 11.081 |
| 6.32 | 5.54 | 29.308 | 34.1 | 11.128 |
| 6.32 | 5.53 | 29.307 | 34.1 | 11.174 |
| 6.32 | 5.53 | 29.304 | 34.1 | 12.023 |
| 6.31 | 5.52 | 29.3   | 34.1 | 12.019 |
| 6.29 | 5.51 | 29.297 | 34.2 | 12.018 |
| 6.28 | 5.5  | 29.296 | 34.2 | 12.016 |
| 6.27 | 5.49 | 29.296 | 34.2 | 12.014 |
| 6.28 | 5.49 | 29.298 | 34.2 | 12.013 |
| 6.29 | 5.49 | 29.298 | 34.2 | 12.011 |
| 6.29 | 5.49 | 29.293 | 34.2 | 12.072 |
| 6.28 | 5.48 | 29.289 | 34.2 | 12.141 |
| 6.26 | 5.47 | 29.286 | 34.2 | 12.216 |
| 6.24 | 5.47 | 29.288 | 34.3 | 12.296 |
| 6.23 | 5.47 | 29.287 | 34.3 | 12.373 |
| 6.24 | 5.46 | 29.284 | 34.3 | 12.47  |
| 6.23 | 5.45 | 29.281 | 34.2 | 12.611 |
| 6.22 | 5.44 | 29.277 | 34.2 | 12.735 |
| 6.22 | 5.43 | 29.277 | 34.3 | 12.8   |
| 6.21 | 5.43 | 29.277 | 34.3 | 12.865 |
| 6.22 | 5.42 | 29.278 | 34.3 | 12.927 |
| 6.23 | 5.42 | 29.279 | 34.3 | 13.038 |
| 6.22 | 5.42 | 29.28  | 34.3 | 13.176 |
| 6.22 | 5.43 | 29.281 | 34.3 | 13.265 |
| 6.22 | 5.43 | 29.281 | 34.3 | 13.341 |
| 6.22 | 5.43 | 29.28  | 34.3 | 13.418 |

| | | | | |
|------|------|--------|------|--------|
| 6.23 | 5.42 | 29.279 | 34.3 | 13.498 |
| 6.25 | 5.42 | 29.278 | 34.3 | 13.576 |
| 6.26 | 5.42 | 29.277 | 34.3 | 13.702 |
| 6.25 | 5.41 | 29.278 | 34.3 | 13.79 |
| 6.24 | 5.41 | 29.279 | 34.3 | 13.878 |
| 6.23 | 5.41 | 29.28 | 34.3 | 13.968 |
| 6.23 | 5.42 | 29.281 | 34.3 | 14.054 |
| 6.24 | 5.42 | 29.281 | 34.3 | 14.14 |
| 6.26 | 5.42 | 29.28 | 34.3 | 14.226 |
| 6.27 | 5.42 | 29.279 | 34.3 | 14.311 |
| 6.27 | 5.41 | 29.278 | 34.3 | 14.386 |
| 6.26 | 5.41 | 29.277 | 34.3 | 14.463 |
| 6.25 | 5.4 | 29.275 | 34.3 | 14.538 |
| 6.23 | 5.4 | 29.271 | 34.3 | 15.477 |
| 6.22 | 5.39 | 29.269 | 34.3 | 15.475 |
| 6.22 | 5.38 | 29.268 | 34.3 | 15.476 |
| 6.22 | 5.37 | 29.267 | 34.3 | 15.478 |
| 6.23 | 5.37 | 29.267 | 34.3 | 15.482 |
| 6.24 | 5.36 | 29.267 | 34.3 | 15.486 |
| 6.26 | 5.36 | 29.267 | 34.3 | 15.501 |
| 6.26 | 5.36 | 29.269 | 34.3 | 15.584 |
| 6.26 | 5.37 | 29.27 | 34.3 | 15.682 |
| 6.26 | 5.37 | 29.271 | 34.3 | 15.787 |
| 6.25 | 5.38 | 29.27 | 34.3 | 16.864 |
| 6.24 | 5.37 | 29.268 | 34.3 | 16.896 |
| 6.24 | 5.36 | 29.267 | 34.3 | 16.899 |
| 6.25 | 5.36 | 29.266 | 34.3 | 16.899 |
| 6.24 | 5.36 | 29.264 | 34.4 | 16.999 |
| 6.23 | 5.35 | 29.261 | 34.3 | 17.143 |
| 6.23 | 5.34 | 29.26 | 34.3 | 17.236 |
| 6.22 | 5.33 | 29.259 | 34.3 | 18.29 |
| 6.21 | 5.33 | 29.26 | 34.3 | 18.433 |
| 6.2 | 5.32 | 29.26 | 34.3 | 18.433 |
| 6.21 | 5.32 | 29.261 | 34.3 | 18.434 |
| 6.2 | 5.32 | 29.261 | 34.3 | 18.539 |
| 6.2 | 5.32 | 29.26 | 34.3 | 18.685 |
| 6.2 | 5.32 | 29.255 | 34.3 | 18.775 |
| 6.2 | 5.31 | 29.251 | 34.4 | 18.831 |
| 6.2 | 5.3 | 29.247 | 34.4 | 18.871 |
| 6.2 | 5.29 | 29.247 | 34.3 | 20.133 |
| 6.2 | 5.28 | 29.246 | 34.3 | 20.136 |
| 6.19 | 5.27 | 29.244 | 34.3 | 20.148 |
| 6.19 | 5.27 | 29.245 | 34.4 | 20.159 |
| 6.18 | 5.27 | 29.244 | 34.4 | 21.108 |
| 6.17 | 5.26 | 29.236 | 34.3 | 21.121 |

| | | | | |
|---|---|---|---|---|
| 6.17 | 5.24 | 29.226 | 34.3 | 21.126 |
| 6.16 | 5.22 | 29.223 | 34.4 | 21.159 |
| 6.15 | 5.22 | 29.229 | 34.4 | 21.352 |
| 6.15 | 5.22 | 29.236 | 34.3 | 21.465 |
| 6.15 | 5.22 | 29.241 | 34.3 | 21.535 |
| 6.16 | 5.23 | 29.243 | 34.4 | 21.584 |
| 6.17 | 5.23 | 29.243 | 34.4 | 21.626 |
| 6.18 | 5.23 | 29.243 | 34.4 | 22.544 |
| 6.18 | 5.24 | 29.242 | 34.4 | 22.276 |
| 6.19 | 5.24 | 29.241 | 34.3 | 22.274 |
| 6.18 | 5.24 | 29.241 | 34.3 | 22.43 |
| 6.17 | 5.24 | 29.237 | 34.3 | 22.557 |
| 6.17 | 5.23 | 29.235 | 34.3 | 22.626 |
| 6.17 | 5.23 | 29.235 | 34.3 | 22.683 |
| 6.16 | 5.22 | 29.235 | 34.3 | 23.534 |
| 6.16 | 5.22 | 29.231 | 34.3 | 23.52 |
| 6.16 | 5.21 | 29.228 | 34.3 | 23.52 |
| 6.16 | 5.21 | 29.226 | 34.3 | 23.521 |
| 6.16 | 5.2 | 29.223 | 34.4 | 23.525 |
| 6.18 | 5.19 | 29.221 | 34.4 | 23.531 |
| 6.19 | 5.19 | 29.22 | 34.4 | 23.534 |
| 6.2 | 5.18 | 29.219 | 34.3 | 24.404 |
| 6.19 | 5.18 | 29.215 | 34.3 | 24.386 |
| 6.18 | 5.17 | 29.208 | 34.3 | 24.399 |
| 6.17 | 5.15 | 29.202 | 34.3 | 24.399 |
| 6.16 | 5.14 | 29.193 | 34.3 | 24.435 |
| 6.15 | 5.09 | 29.173 | 34.3 | 24.571 |
| 6.14 | 5.04 | 29.153 | 34.3 | 24.651 |
| 6.14 | 4.99 | 29.152 | 34.3 | 24.704 |
| 6.14 | 4.97 | 29.159 | 34.3 | 24.742 |
| 6.15 | 4.97 | 29.168 | 34.3 | 24.818 |
| 6.15 | 4.98 | 29.167 | 34.3 | 25.583 |
| 6.15 | 4.99 | 29.16 | 34.3 | 25.592 |
| 6.15 | 4.98 | 29.153 | 34.3 | 25.591 |
| 6.14 | 4.97 | 29.153 | 34.3 | 25.591 |
| 6.13 | 4.97 | 29.154 | 34.3 | 25.678 |
| 6.13 | 4.97 | 29.156 | 34.3 | 25.791 |
| 6.13 | 4.97 | 29.157 | 34.3 | 25.866 |
| 6.13 | 4.97 | 29.157 | 34.3 | 26.841 |
| 6.11 | 4.97 | 29.155 | 34.3 | 26.87 |
| 6.11 | 4.97 | 29.153 | 34.3 | 26.877 |
| 6.11 | 4.96 | 29.152 | 34.3 | 26.879 |
| 6.1 | 4.96 | 29.145 | 34.3 | 27.007 |
| 6.09 | 4.94 | 29.134 | 34.3 | 27.124 |
| 6.09 | 4.91 | 29.126 | 34.2 | 27.193 |

| 6.1 | 4.89 | 29.129 | 34.2 | 27.242 |
|---|---|---|---|---|
| 6.11 | 4.88 | 29.138 | 34.2 | 27.279 |
| 6.11 | 4.89 | 29.139 | 34.2 | 28.141 |
| 6.11 | 4.9 | 29.133 | 34.2 | 28.143 |
| 6.12 | 4.89 | 29.129 | 34.2 | 28.142 |
| 6.11 | 4.88 | 29.129 | 34.2 | 28.269 |
| 6.11 | 4.87 | 29.128 | 34.2 | 28.402 |
| 6.11 | 4.87 | 29.128 | 34.2 | 28.479 |
| 6.1 | 4.87 | 29.127 | 34.2 | 29.464 |
| 6.1 | 4.86 | 29.126 | 34.2 | 29.465 |
| 6.1 | 4.86 | 29.125 | 34.2 | 29.458 |
| 6.11 | 4.86 | 29.126 | 34.2 | 29.457 |
| 6.1 | 4.86 | 29.126 | 34.2 | 29.606 |
| 6.11 | 4.86 | 29.126 | 34.2 | 29.696 |
| 6.12 | 4.86 | 29.127 | 34.2 | 30.576 |
| 6.11 | 4.86 | 29.127 | 34.2 | 30.67 |
| 6.1 | 4.86 | 29.126 | 34.2 | 30.666 |
| 6.11 | 4.86 | 29.126 | 34.2 | 30.678 |
| 6.1 | 4.85 | 29.124 | 34.2 | 30.868 |
| 6.1 | 4.85 | 29.125 | 34.2 | 30.97 |
| 6.11 | 4.85 | 29.126 | 34.2 | 31.035 |
| 6.1 | 4.85 | 29.128 | 34.2 | 31.929 |
| 6.1 | 4.86 | 29.129 | 34.2 | 31.922 |
| 6.11 | 4.86 | 29.131 | 34.2 | 31.907 |
| 6.11 | 4.87 | 29.132 | 34.2 | 31.968 |
| 6.11 | 4.87 | 29.133 | 34.2 | 32.097 |
| 6.12 | 4.87 | 29.133 | 34.2 | 32.175 |
| 6.13 | 4.87 | 29.134 | 34.2 | 32.223 |
| 6.13 | 4.88 | 29.135 | 34.2 | 33.223 |
| 6.13 | 4.88 | 29.135 | 34.2 | 33.251 |
| 6.13 | 4.88 | 29.134 | 34.2 | 33.254 |
| 6.13 | 4.87 | 29.129 | 34.2 | 33.253 |
| 6.13 | 4.85 | 29.122 | 34.2 | 33.256 |
| 6.14 | 4.84 | 29.119 | 34.2 | 33.262 |
| 6.15 | 4.83 | 29.123 | 34.2 | 33.269 |
| 6.15 | 4.84 | 29.129 | 34.2 | 34.022 |
| 6.15 | 4.85 | 29.133 | 34.2 | 34.034 |
| 6.15 | 4.87 | 29.134 | 34.2 | 34.045 |
| 6.15 | 4.87 | 29.133 | 34.2 | 34.094 |
| 6.14 | 4.86 | 29.129 | 34.2 | 34.263 |
| 6.14 | 4.85 | 29.125 | 34.2 | 34.373 |
| 6.14 | 4.84 | 29.124 | 34.2 | 34.448 |
| 6.14 | 4.83 | 29.123 | 34.2 | 35.556 |
| 6.13 | 4.83 | 29.12 | 34.2 | 35.61 |
| 6.13 | 4.82 | 29.115 | 34.2 | 35.606 |

| | | | | |
|------|------|--------|------|--------|
| 6.14 | 4.8  | 29.113 | 34.2 | 35.607 |
| 6.14 | 4.8  | 29.114 | 34.2 | 35.611 |
| 6.14 | 4.8  | 29.116 | 34.2 | 36.535 |
| 6.14 | 4.8  | 29.118 | 34.2 | 36.541 |
| 6.14 | 4.8  | 29.118 | 34.2 | 36.555 |
| 6.14 | 4.8  | 29.117 | 34.2 | 36.564 |
| 6.14 | 4.8  | 29.118 | 34.2 | 37.415 |
| 6.14 | 4.81 | 29.119 | 34.2 | 37.423 |
| 6.14 | 4.81 | 29.122 | 34.2 | 37.425 |
| 6.15 | 4.81 | 29.122 | 34.2 | 37.457 |
| 6.14 | 4.81 | 29.12  | 34.2 | 38.396 |
| 6.14 | 4.81 | 29.117 | 34.2 | 38.392 |
| 6.15 | 4.8  | 29.115 | 34.2 | 38.395 |
| 6.14 | 4.79 | 29.114 | 34.2 | 39.22  |
| 6.14 | 4.78 | 29.112 | 34.2 | 39.212 |
| 6.14 | 4.78 | 29.112 | 34.2 | 39.216 |
| 6.13 | 4.78 | 29.112 | 34.2 | 40.042 |
| 6.12 | 4.77 | 29.113 | 34.2 | 40.043 |
| 6.13 | 4.77 | 29.113 | 34.2 | 40.045 |
| 6.13 | 4.78 | 29.114 | 34.2 | 40.048 |
| 6.13 | 4.78 | 29.114 | 34.2 | 40.836 |
| 6.13 | 4.78 | 29.114 | 34.2 | 40.841 |
| 6.14 | 4.78 | 29.114 | 34.2 | 40.842 |
| 6.14 | 4.78 | 29.114 | 34.2 | 40.845 |
| 6.15 | 4.78 | 29.114 | 34.2 | 40.848 |
| 6.17 | 4.78 | 29.114 | 34.2 | 40.851 |
| 6.17 | 4.78 | 29.113 | 34.2 | 41.627 |
| 6.17 | 4.77 | 29.113 | 34.2 | 41.635 |
| 6.17 | 4.77 | 29.113 | 34.2 | 41.632 |
| 6.17 | 4.77 | 29.113 | 34.2 | 42.365 |
| 6.16 | 4.77 | 29.112 | 34.2 | 42.365 |
| 6.16 | 4.77 | 29.111 | 34.2 | 42.363 |
| 6.15 | 4.76 | 29.11  | 34.2 | 43.138 |
| 6.14 | 4.76 | 29.11  | 34.2 | 43.129 |
| 6.14 | 4.76 | 29.111 | 34.2 | 43.119 |
| 6.14 | 4.76 | 29.112 | 34.2 | 43.109 |
| 6.14 | 4.76 | 29.112 | 34.2 | 43.835 |
| 6.15 | 4.76 | 29.112 | 34.2 | 43.826 |
| 6.15 | 4.76 | 29.112 | 34.2 | 43.817 |
| 6.16 | 4.76 | 29.112 | 34.2 | 43.814 |
| 6.15 | 4.76 | 29.109 | 34.2 | 44.56  |
| 6.15 | 4.75 | 29.106 | 34.2 | 44.544 |
| 6.15 | 4.75 | 29.105 | 34.2 | 44.545 |
| 6.14 | 4.74 | 29.105 | 34.2 | 45.354 |
| 6.14 | 4.74 | 29.105 | 34.2 | 45.403 |

| | | | | |
|------|------|--------|------|--------|
| 6.14 | 4.74 | 29.104 | 34.2 | 45.409 |
| 6.13 | 4.74 | 29.103 | 34.2 | 45.453 |
| 6.12 | 4.73 | 29.099 | 34.2 | 45.609 |
| 6.13 | 4.72 | 29.097 | 34.2 | 45.696 |
| 6.13 | 4.72 | 29.096 | 34.2 | 46.471 |
| 6.13 | 4.71 | 29.096 | 34.2 | 46.486 |
| 6.13 | 4.71 | 29.096 | 34.2 | 46.483 |
| 6.13 | 4.71 | 29.096 | 34.2 | 46.487 |
| 6.13 | 4.71 | 29.096 | 34.2 | 46.623 |
| 6.14 | 4.71 | 29.096 | 34.2 | 46.699 |
| 6.14 | 4.71 | 29.096 | 34.2 | 46.746 |
| 6.14 | 4.71 | 29.096 | 34.2 | 46.85  |
| 6.14 | 4.7  | 29.096 | 34.2 | 46.932 |
| 6.15 | 4.7  | 29.096 | 34.2 | 47.055 |
| 6.15 | 4.7  | 29.096 | 34.2 | 47.17  |
| 6.15 | 4.7  | 29.096 | 34.2 | 47.979 |
| 6.16 | 4.7  | 29.095 | 34.2 | 47.981 |
| 6.16 | 4.7  | 29.096 | 34.2 | 48.07  |
| 6.16 | 4.7  | 29.096 | 34.2 | 48.261 |
| 6.16 | 4.7  | 29.096 | 34.2 | 48.357 |
| 6.16 | 4.7  | 29.097 | 34.2 | 48.414 |
| 6.15 | 4.7  | 29.096 | 34.2 | 49.271 |
| 6.16 | 4.7  | 29.096 | 34.2 | 49.255 |
| 6.16 | 4.7  | 29.096 | 34.2 | 49.241 |
| 6.16 | 4.7  | 29.096 | 34.2 | 49.385 |
| 6.16 | 4.7  | 29.096 | 34.2 | 49.518 |
| 6.17 | 4.7  | 29.095 | 34.2 | 49.584 |
| 6.17 | 4.7  | 29.095 | 34.2 | 50.497 |
| 6.17 | 4.69 | 29.095 | 34.2 | 50.478 |
| 6.17 | 4.69 | 29.094 | 34.2 | 50.479 |
| 6.17 | 4.69 | 29.094 | 34.2 | 50.607 |
| 6.16 | 4.69 | 29.093 | 34.2 | 50.745 |
| 6.16 | 4.68 | 29.093 | 34.1 | 50.818 |
| 6.16 | 4.68 | 29.093 | 34.2 | 50.872 |
| 6.16 | 4.68 | 29.094 | 34.2 | 51.757 |
| 6.16 | 4.67 | 29.093 | 34.2 | 51.77  |
| 6.16 | 4.67 | 29.092 | 34.2 | 51.802 |
| 6.16 | 4.66 | 29.092 | 34.2 | 51.821 |
| 6.17 | 4.66 | 29.092 | 34.2 | 51.831 |
| 6.18 | 4.66 | 29.093 | 34.2 | 51.838 |
| 6.18 | 4.66 | 29.093 | 34.2 | 52.623 |
| 6.18 | 4.66 | 29.092 | 34.2 | 52.573 |
| 6.19 | 4.66 | 29.092 | 34.2 | 52.578 |
| 6.19 | 4.66 | 29.092 | 34.2 | 52.58  |
| 6.19 | 4.66 | 29.092 | 34.2 | 52.581 |

| | | | | |
|------|------|--------|------|--------|
| 6.2 | 4.66 | 29.092 | 34.2 | 52.585 |
| 6.2 | 4.66 | 29.093 | 34.2 | 52.591 |
| 6.21 | 4.66 | 29.092 | 34.2 | 52.598 |
| 6.21 | 4.66 | 29.092 | 34.2 | 52.606 |
| 6.22 | 4.66 | 29.092 | 34.2 | 52.617 |
| 6.22 | 4.65 | 29.092 | 34.2 | 52.629 |
| 6.23 | 4.65 | 29.092 | 34.2 | 52.64 |
| 6.23 | 4.65 | 29.093 | 34.2 | 52.652 |
| 6.24 | 4.65 | 29.093 | 34.2 | 52.664 |
| 6.24 | 4.65 | 29.093 | 34.2 | 52.675 |
| 6.25 | 4.66 | 29.092 | 34.2 | 52.686 |
| 6.25 | 4.66 | 29.093 | 34.2 | 52.694 |
| 6.25 | 4.66 | 29.093 | 34.2 | 52.7 |
| 6.26 | 4.66 | 29.093 | 34.2 | 52.706 |
| 6.26 | 4.66 | 29.093 | 34.2 | 52.713 |
| 6.26 | 4.66 | 29.093 | 34.2 | 52.717 |
| 6.26 | 4.66 | 29.093 | 34.2 | 52.718 |
| 6.26 | 4.66 | 29.093 | 34.1 | 52.716 |
| 6.27 | 4.66 | 29.092 | 34.1 | 52.715 |
| 6.27 | 4.66 | 29.092 | 34.2 | 53.507 |
| 6.26 | 4.65 | 29.092 | 34.2 | 53.535 |
| 6.26 | 4.65 | 29.092 | 34.2 | 53.547 |
| 6.26 | 4.65 | 29.091 | 34.2 | 53.691 |
| 6.25 | 4.65 | 29.089 | 34.2 | 53.833 |
| 6.25 | 4.64 | 29.087 | 34.2 | 53.91 |
| 6.25 | 4.63 | 29.087 | 34.2 | 53.959 |
| 6.24 | 4.61 | 29.085 | 34.2 | 54.75 |
| 6.24 | 4.59 | 29.083 | 34.2 | 54.752 |
| 6.24 | 4.57 | 29.08 | 34.2 | 54.76 |
| 6.23 | 4.56 | 29.079 | 34.2 | 54.82 |
| 6.22 | 4.54 | 29.078 | 34.2 | 55.759 |
| 6.22 | 4.52 | 29.076 | 34.2 | 55.795 |
| 6.22 | 4.51 | 29.075 | 34.2 | 55.794 |
| 6.22 | 4.5 | 29.075 | 34.2 | 55.889 |
| 6.21 | 4.48 | 29.073 | 34.3 | 56.027 |
| 6.21 | 4.46 | 29.072 | 34.3 | 56.108 |
| 6.21 | 4.44 | 29.072 | 34.3 | 56.932 |
| 6.2 | 4.42 | 29.072 | 34.3 | 56.951 |
| 6.2 | 4.4 | 29.071 | 34.3 | 56.955 |
| 6.19 | 4.39 | 29.071 | 34.3 | 57.007 |
| 6.19 | 4.38 | 29.07 | 34.4 | 57.163 |
| 6.19 | 4.37 | 29.069 | 34.4 | 57.264 |
| 6.19 | 4.33 | 29.068 | 34.6 | 57.335 |
| 6.2 | 4.25 | 29.068 | 34.7 | 57.295 |
| 6.22 | 4.22 | 29.069 | 34.7 | 57.338 |

| | | | | |
|------|------|--------|------|--------|
| 6.23 | 4.26 | 29.07 | 34.6 | 57.438 |
| 6.25 | 4.29 | 29.069 | 34.6 | 57.534 |
| 6.25 | 4.29 | 29.069 | 35 | 57.585 |
| 6.35 | 4.24 | 29.068 | 36.6 | 57.611 |
| 6.38 | 3.89 | 29.067 | 39.8 | 57.626 |
| 6.39 | 3.66 | 29.068 | 44.4 | 57.684 |
| 6.41 | 3.54 | 29.069 | 47.8 | 57.742 |
| 6.41 | 3.59 | 29.071 | 48.4 | 57.744 |
| 6.42 | 3.65 | 29.074 | 47 | 57.743 |
| 6.42 | 3.77 | 29.077 | 45.2 | 57.743 |
| 6.41 | 3.73 | 29.075 | 43.5 | 57.735 |
| 6.41 | 3.63 | 29.074 | 42.4 | 57.709 |
| 6.52 | 3.54 | 29.071 | 42.2 | 57.68 |
| 6.59 | 2.31 | 29.068 | 43.5 | 57.67 |
| 6.58 | 1.32 | 29.065 | 46.7 | 57.664 |
| 6.58 | 1.08 | 29.063 | 50.7 | 57.654 |
| 6.59 | 1.55 | 29.066 | 50.7 | 57.642 |
| 6.59 | 1.04 | 29.07 | 50.7 | 57.648 |
| 6.58 | 0.76 | 29.07 | 52.4 | 57.659 |
| 6.57 | 0.66 | 29.069 | 55.7 | 57.673 |
| 6.55 | 0.59 | 29.068 | 57.7 | 57.689 |
| 6.52 | 0.49 | 29.065 | 61.4 | 57.69 |
| 6.5 | 0.42 | 29.064 | 63.7 | 57.69 |
| 6.49 | 0.71 | 29.067 | 60.6 | 57.691 |
| 6.58 | 1.12 | 29.077 | 58.2 | 57.697 |
| 6.53 | 0.57 | 29.082 | 59.6 | 57.699 |
| 6.52 | 0.46 | 29.082 | 64.9 | 57.703 |
| 6.51 | 0.41 | 29.079 | 66.7 | 57.702 |
| 6.5 | 0.36 | 29.079 | 65.5 | 57.692 |
| 6.49 | 0.32 | 29.082 | 63.5 | 57.687 |
| 6.48 | 0.3 | 29.085 | 64 | 57.698 |
| 6.47 | 0.28 | 29.086 | 66.2 | 57.709 |
| 6.46 | 0.72 | 29.085 | 67.7 | 57.703 |
| 6.77 | 0.8 | 29.08 | 69.7 | 57.695 |
| 6.84 | 1.44 | 29.074 | 66.2 | 57.683 |
| 6.72 | 2.38 | 29.068 | 56.5 | 57.665 |
| 6.7 | 2.06 | 29.067 | 44.3 | 57.668 |
| 6.72 | 2.41 | 29.066 | 44 | 57.666 |
| 6.72 | 2.67 | 29.065 | 43.8 | 57.694 |
| 6.72 | 3.44 | 29.067 | 40 | 57.711 |
| 6.72 | 3.64 | 29.068 | 37.4 | 57.75 |
| 6.61 | 3.69 | 29.067 | 38.2 | 57.833 |
| 6.58 | 3.69 | 29.066 | 40.5 | 57.924 |

107

108    Table S27 – P19 site depth profile from 2017 campaign.

| pH | ODO mg/L | Temp °C | Cond µS/cm | Depth m |
|---|---|---|---|---|
| 6.52 | 5.2 | 29.431 | 34.5 | 0.344 |
| 6.52 | 5.2 | 29.432 | 34.5 | 0.388 |
| 6.52 | 5.19 | 29.43 | 34.4 | 0.43 |
| 6.52 | 5.19 | 29.428 | 34.3 | 0.494 |
| 6.51 | 5.19 | 29.429 | 34.2 | 0.548 |
| 6.51 | 5.18 | 29.43 | 34.2 | 0.619 |
| 6.51 | 5.18 | 29.432 | 34.1 | 0.677 |
| 6.51 | 5.18 | 29.44 | 34 | 0.752 |
| 6.51 | 5.18 | 29.45 | 33.9 | 0.837 |
| 6.51 | 5.19 | 29.459 | 33.8 | 0.914 |
| 6.51 | 5.19 | 29.458 | 33.7 | 0.982 |
| 6.51 | 5.19 | 29.458 | 33.7 | 1.045 |
| 6.51 | 5.19 | 29.458 | 33.7 | 1.106 |
| 6.51 | 5.19 | 29.459 | 33.6 | 1.171 |
| 6.51 | 5.18 | 29.457 | 33.6 | 1.246 |
| 6.51 | 5.18 | 29.457 | 33.5 | 1.322 |
| 6.51 | 5.18 | 29.452 | 33.5 | 1.39 |
| 6.51 | 5.18 | 29.446 | 33.5 | 1.475 |
| 6.51 | 5.18 | 29.437 | 33.5 | 1.575 |
| 6.5 | 5.18 | 29.434 | 33.5 | 1.659 |
| 6.5 | 5.18 | 29.432 | 33.4 | 1.755 |
| 6.49 | 5.17 | 29.432 | 33.4 | 1.832 |
| 6.5 | 5.17 | 29.433 | 33.4 | 1.923 |
| 6.5 | 5.17 | 29.437 | 33.4 | 2.004 |
| 6.5 | 5.17 | 29.44 | 33.5 | 2.08 |
| 6.5 | 5.17 | 29.44 | 33.5 | 2.147 |
| 6.5 | 5.17 | 29.435 | 33.5 | 2.222 |
| 6.5 | 5.17 | 29.43 | 33.5 | 2.289 |
| 6.51 | 5.17 | 29.428 | 33.5 | 2.352 |
| 6.51 | 5.17 | 29.432 | 33.5 | 2.433 |
| 6.51 | 5.17 | 29.433 | 33.6 | 2.507 |
| 6.5 | 5.17 | 29.429 | 33.6 | 2.589 |
| 6.5 | 5.17 | 29.42 | 33.6 | 2.682 |
| 6.49 | 5.17 | 29.414 | 33.6 | 2.778 |
| 6.49 | 5.16 | 29.412 | 33.7 | 2.873 |
| 6.49 | 5.16 | 29.414 | 33.7 | 2.961 |
| 6.5 | 5.16 | 29.415 | 33.7 | 3.042 |
| 6.5 | 5.16 | 29.415 | 33.8 | 3.12 |
| 6.51 | 5.16 | 29.417 | 33.8 | 3.201 |
| 6.51 | 5.16 | 29.419 | 33.8 | 3.293 |
| 6.5 | 5.16 | 29.417 | 33.9 | 3.408 |
| 6.5 | 5.16 | 29.414 | 33.9 | 4.269 |

| | | | | |
|---|---|---|---|---|
| 6.49 | 5.16 | 29.412 | 33.9 | 4.424 |
| 6.49 | 5.16 | 29.412 | 34 | 4.503 |
| 6.48 | 5.16 | 29.412 | 34 | 4.588 |
| 6.48 | 5.16 | 29.412 | 34 | 4.659 |
| 6.48 | 5.16 | 29.412 | 34.1 | 4.782 |
| 6.48 | 5.16 | 29.413 | 34.1 | 4.867 |
| 6.47 | 5.16 | 29.414 | 34.1 | 4.974 |
| 6.47 | 5.16 | 29.415 | 34.2 | 5.062 |
| 6.48 | 5.16 | 29.415 | 34.2 | 5.232 |
| 6.48 | 5.16 | 29.415 | 34.2 | 5.397 |
| 6.48 | 5.16 | 29.415 | 34.2 | 5.561 |
| 6.48 | 5.16 | 29.415 | 34.3 | 5.703 |
| 6.48 | 5.16 | 29.414 | 34.3 | 5.836 |
| 6.48 | 5.16 | 29.412 | 34.3 | 5.969 |
| 6.48 | 5.15 | 29.413 | 34.3 | 6.107 |
| 6.48 | 5.15 | 29.413 | 34.4 | 6.233 |
| 6.48 | 5.15 | 29.414 | 34.4 | 7.064 |
| 6.49 | 5.15 | 29.414 | 34.4 | 7.166 |
| 6.49 | 5.15 | 29.414 | 34.4 | 7.266 |
| 6.49 | 5.15 | 29.414 | 34.4 | 7.353 |
| 6.5 | 5.15 | 29.414 | 34.4 | 7.454 |
| 6.5 | 5.15 | 29.413 | 34.5 | 7.516 |
| 6.49 | 5.15 | 29.412 | 34.5 | 7.601 |
| 6.49 | 5.15 | 29.413 | 34.5 | 8.45 |
| 6.48 | 5.14 | 29.413 | 34.5 | 8.558 |
| 6.48 | 5.14 | 29.413 | 34.5 | 8.638 |
| 6.48 | 5.14 | 29.412 | 34.5 | 8.777 |
| 6.48 | 5.14 | 29.411 | 34.6 | 8.901 |
| 6.49 | 5.14 | 29.412 | 34.6 | 9.042 |
| 6.49 | 5.14 | 29.412 | 34.6 | 9.893 |

109

110    Table S28 – P20 site depth profile from 2017 campaign.

| pH | ODO mg/L | Temp °C | Cond µS/cm | Depth m |
|---|---|---|---|---|
| 7.01 | 7.98 | 30.036 | 37.5 | 0.433 |
| 7.03 | 7.98 | 30.034 | 37.5 | 0.523 |
| 7.04 | 7.98 | 30.033 | 37.6 | 0.617 |
| 7.05 | 7.98 | 30.032 | 37.6 | 0.719 |
| 7.06 | 7.98 | 30.032 | 37.6 | 0.822 |
| 7.07 | 7.98 | 30.032 | 37.6 | 0.92 |
| 7.07 | 7.99 | 30.033 | 37.6 | 1.012 |
| 7.08 | 7.99 | 30.033 | 37.6 | 1.098 |
| 7.08 | 7.99 | 30.034 | 37.5 | 1.185 |

| | | | | |
|---|---|---|---|---|
| 7.08 | 8 | 30.034 | 37.5 | 1.279 |
| 7.09 | 8 | 30.036 | 37.5 | 1.378 |
| 7.09 | 8 | 30.037 | 37.4 | 1.469 |
| 7.09 | 8 | 30.035 | 37.3 | 1.56 |
| 7.09 | 8 | 30.029 | 37.2 | 1.653 |
| 7.09 | 8 | 30.027 | 37.2 | 1.743 |
| 7.09 | 8 | 30.026 | 37.1 | 1.832 |
| 7.09 | 8 | 30.026 | 37 | 1.919 |
| 7.09 | 8 | 30.026 | 36.9 | 1.986 |
| 7.09 | 8 | 30.027 | 36.9 | 2.039 |
| 7.09 | 8 | 30.028 | 36.8 | 2.101 |
| 7.09 | 8 | 30.027 | 36.8 | 2.162 |
| 7.09 | 8.01 | 30.025 | 36.8 | 2.236 |
| 7.09 | 8.01 | 30.024 | 36.7 | 2.332 |
| 7.09 | 8 | 30.024 | 36.7 | 2.415 |
| 7.09 | 8 | 30.024 | 36.7 | 2.508 |
| 7.09 | 8 | 30.023 | 36.7 | 2.63 |
| 7.09 | 8 | 30.023 | 36.7 | 2.767 |
| 7.09 | 8 | 30.023 | 36.7 | 2.921 |
| 7.09 | 8.01 | 30.023 | 36.7 | 3.77 |
| 7.09 | 8.01 | 30.023 | 36.7 | 3.916 |
| 7.09 | 8 | 30.023 | 36.7 | 3.944 |
| 7.09 | 8 | 30.024 | 36.7 | 4.001 |
| 7.08 | 8.01 | 30.025 | 36.7 | 4.045 |
| 7.08 | 8.01 | 30.027 | 36.8 | 4.086 |
| 7.08 | 8.01 | 30.03 | 36.8 | 4.137 |
| 7.08 | 8.01 | 30.033 | 36.8 | 4.189 |
| 7.08 | 8.02 | 30.034 | 36.8 | 4.241 |
| 7.09 | 8.02 | 30.033 | 36.9 | 4.345 |
| 7.09 | 8.02 | 30.031 | 36.9 | 4.443 |
| 7.09 | 8.01 | 30.031 | 36.9 | 4.539 |
| 7.09 | 8.01 | 30.03 | 37 | 4.635 |
| 7.09 | 8.01 | 30.029 | 37 | 4.722 |
| 7.09 | 8.01 | 30.026 | 37 | 4.803 |
| 7.09 | 8.01 | 30.025 | 37.1 | 4.891 |
| 7.09 | 8.01 | 30.024 | 37.1 | 4.966 |
| 7.08 | 8.02 | 30.024 | 37.2 | 5.054 |
| 7.08 | 8.02 | 30.025 | 37.2 | 5.145 |
| 7.08 | 8.01 | 30.026 | 37.3 | 5.272 |
| 7.08 | 8.01 | 30.026 | 37.3 | 6.297 |
| 7.08 | 8.02 | 30.025 | 37.3 | 6.392 |
| 7.08 | 8.02 | 30.027 | 37.4 | 6.333 |
| 7.09 | 8.02 | 30.029 | 37.4 | 6.317 |
| 7.09 | 8.02 | 30.028 | 37.5 | 6.318 |
| 7.09 | 8.02 | 30.025 | 37.5 | 6.33 |

| | | | | |
|------|------|--------|------|--------|
| 7.09 | 8.02 | 30.024 | 37.5 | 6.354 |
| 7.09 | 8.02 | 30.023 | 37.6 | 6.382 |
| 7.1  | 8.02 | 30.025 | 37.6 | 6.415 |
| 7.1  | 8.02 | 30.025 | 37.7 | 6.457 |
| 7.09 | 8.02 | 30.025 | 37.7 | 6.494 |
| 7.09 | 8.02 | 30.025 | 37.7 | 6.51  |
| 7.1  | 8.02 | 30.026 | 37.7 | 6.509 |
| 7.1  | 8.02 | 30.027 | 37.8 | 6.498 |
| 7.1  | 8.03 | 30.025 | 37.8 | 6.523 |
| 7.1  | 8.02 | 30.023 | 37.8 | 6.594 |
| 7.09 | 8.02 | 30.021 | 37.9 | 6.68  |
| 7.09 | 8.02 | 30.022 | 37.9 | 7.487 |
| 7.09 | 8.02 | 30.023 | 37.9 | 7.491 |
| 7.09 | 8.02 | 30.022 | 37.9 | 7.581 |
| 7.09 | 8.02 | 30.021 | 37.9 | 7.714 |
| 7.09 | 8.02 | 30.02  | 38   | 7.822 |
| 7.1  | 8.01 | 30.02  | 38   | 7.893 |
| 7.1  | 8.01 | 30.02  | 38   | 7.947 |
| 7.1  | 8.01 | 30.02  | 38   | 8.004 |
| 7.1  | 8.01 | 30.02  | 38   | 8.129 |
| 7.1  | 8.01 | 30.02  | 38.1 | 8.274 |
| 7.1  | 8.01 | 30.02  | 38.1 | 8.382 |
| 7.1  | 8.01 | 30.021 | 38.1 | 8.461 |
| 7.09 | 8.01 | 30.021 | 38.1 | 8.552 |
| 7.08 | 8.01 | 30.019 | 38.1 | 8.656 |
| 7.07 | 8    | 30.016 | 38.1 | 8.777 |
| 7.06 | 8    | 30.015 | 38.1 | 8.895 |
| 7.05 | 8    | 30.014 | 38.1 | 9     |
| 7.05 | 8    | 30.013 | 38.1 | 9.11  |
| 7.05 | 8    | 30.012 | 38.1 | 9.233 |
| 7.06 | 8    | 30.011 | 38.1 | 9.369 |
| 7.07 | 7.99 | 30.012 | 38.1 | 9.496 |
| 7.07 | 7.99 | 30.011 | 38.1 | 9.632 |
| 7.06 | 7.99 | 30.011 | 38.2 | 9.753 |
| 7.05 | 7.99 | 30.012 | 38.2 | 9.872 |
| 7.04 | 8    | 30.014 | 38.2 | 10.008 |
| 7.03 | 8    | 30.015 | 38.2 | 10.144 |
| 7.02 | 8    | 30.014 | 38.2 | 10.268 |
| 7.01 | 8    | 30.014 | 38.2 | 10.383 |
| 7    | 7.99 | 30.014 | 38.2 | 10.487 |
| 6.99 | 7.99 | 30.013 | 38.2 | 10.595 |
| 6.97 | 7.99 | 30.014 | 38.2 | 10.717 |
| 6.95 | 7.99 | 30.016 | 38.2 | 10.86 |
| 6.93 | 7.99 | 30.018 | 38.2 | 11.676 |
| 6.92 | 7.99 | 30.018 | 38.2 | 11.754 |

| | | | | |
|------|------|--------|------|--------|
| 6.91 | 7.99 | 30.017 | 38.2 | 11.815 |
| 6.89 | 7.99 | 30.016 | 38.2 | 11.846 |
| 6.88 | 7.99 | 30.017 | 38.2 | 11.916 |
| 6.86 | 7.99 | 30.017 | 38.2 | 11.989 |
| 6.84 | 7.99 | 30.017 | 38.2 | 12.069 |
| 6.82 | 7.98 | 30.016 | 38.2 | 12.137 |
| 6.81 | 7.98 | 30.016 | 38.2 | 12.246 |
| 6.79 | 7.98 | 30.015 | 38.2 | 12.365 |
| 6.78 | 7.98 | 30.014 | 38.2 | 12.501 |
| 6.76 | 7.98 | 30.014 | 38.2 | 12.646 |
| 6.75 | 7.98 | 30.015 | 38.2 | 12.788 |
| 6.73 | 7.98 | 30.015 | 38.2 | 12.929 |
| 6.71 | 7.98 | 30.014 | 38.2 | 13.05 |
| 6.7 | 7.98 | 30.012 | 38.2 | 13.18 |
| 6.68 | 7.97 | 30.011 | 38.2 | 13.308 |
| 6.67 | 7.97 | 30.012 | 38.2 | 13.428 |
| 6.66 | 7.97 | 30.011 | 38.3 | 13.54 |
| 6.66 | 7.97 | 30.012 | 38.3 | 13.646 |
| 6.66 | 7.97 | 30.012 | 38.2 | 13.75 |
| 6.67 | 7.96 | 30.012 | 38.2 | 13.851 |
| 6.67 | 7.96 | 30.013 | 38.2 | 13.953 |
| 6.66 | 7.96 | 30.014 | 38.3 | 14.049 |
| 6.66 | 7.96 | 30.014 | 38.3 | 14.726 |
| 6.65 | 7.96 | 30.013 | 38.3 | 14.87 |
| 6.63 | 7.97 | 30.013 | 38.3 | 14.991 |
| 6.61 | 7.97 | 30.012 | 38.3 | 15.089 |
| 6.6 | 7.97 | 30.012 | 38.3 | 15.178 |
| 6.58 | 7.97 | 30.01 | 38.3 | 15.24 |
| 6.56 | 7.97 | 30.01 | 38.3 | 15.331 |
| 6.54 | 7.97 | 30.01 | 38.3 | 15.418 |
| 6.53 | 7.97 | 30.012 | 38.3 | 15.528 |
| 6.52 | 7.97 | 30.014 | 38.3 | 15.691 |
| 6.51 | 7.97 | 30.015 | 38.3 | 16.65 |
| 6.5 | 7.97 | 30.016 | 38.3 | 16.851 |
| 6.49 | 7.97 | 30.016 | 38.3 | 16.996 |
| 6.47 | 7.96 | 30.016 | 38.3 | 17.122 |
| 6.47 | 7.96 | 30.017 | 38.3 | 17.231 |
| 6.46 | 7.96 | 30.017 | 38.3 | 18.169 |
| 6.45 | 7.95 | 30.016 | 38.3 | 18.384 |
| 6.45 | 7.95 | 30.015 | 38.3 | 18.51 |
| 6.44 | 7.95 | 30.014 | 38.3 | 18.638 |
| 6.43 | 7.95 | 30.014 | 38.3 | 18.734 |
| 6.43 | 7.95 | 30.014 | 38.3 | 18.824 |
| 6.43 | 7.95 | 30.015 | 38.3 | 18.91 |
| 6.43 | 7.95 | 30.015 | 38.3 | 18.977 |

111

112    Table S29 – P21 site depth profile from 2017 campaign.

| pH | ODO mg/L | Temp °C | Cond µS/cm | Depth m |
|---|---|---|---|---|
| 6.83 | 6.85 | 30.073 | 36.7 | 0.117 |
| 6.82 | 6.87 | 30.074 | 36.6 | 0.161 |
| 6.81 | 6.88 | 30.082 | 36.5 | 0.207 |
| 6.82 | 6.89 | 30.08 | 36.4 | 0.246 |
| 6.81 | 6.89 | 30.073 | 36.3 | 0.269 |
| 6.81 | 6.89 | 30.071 | 36.2 | 0.296 |
| 6.81 | 6.88 | 30.066 | 36.1 | 0.329 |
| 6.81 | 6.87 | 30.053 | 36.1 | 0.362 |
| 6.81 | 6.86 | 30.036 | 36 | 0.419 |
| 6.81 | 6.86 | 30.031 | 35.9 | 0.505 |
| 6.81 | 6.86 | 30.031 | 35.9 | 0.604 |
| 6.81 | 6.86 | 30.025 | 35.8 | 0.718 |
| 6.81 | 6.86 | 30.017 | 35.8 | 0.843 |
| 6.81 | 6.86 | 30.015 | 35.8 | 0.985 |
| 6.81 | 6.86 | 30.013 | 35.7 | 1.137 |
| 6.81 | 6.86 | 30.011 | 35.7 | 1.298 |
| 6.8 | 6.86 | 30.008 | 35.7 | 1.44 |
| 6.8 | 6.85 | 30.006 | 35.7 | 1.577 |
| 6.8 | 6.85 | 30.004 | 35.7 | 1.714 |
| 6.8 | 6.85 | 30.002 | 35.7 | 1.844 |
| 6.8 | 6.85 | 30.001 | 35.7 | 1.97 |
| 6.8 | 6.84 | 30.001 | 35.7 | 2.083 |
| 6.81 | 6.84 | 30.002 | 35.7 | 2.199 |
| 6.81 | 6.84 | 30.004 | 35.8 | 2.321 |
| 6.8 | 6.84 | 30.001 | 35.8 | 2.446 |
| 6.8 | 6.84 | 29.997 | 35.8 | 3.399 |
| 6.8 | 6.84 | 29.997 | 35.8 | 3.577 |
| 6.8 | 6.84 | 29.999 | 35.9 | 3.573 |
| 6.79 | 6.84 | 30 | 35.9 | 3.625 |
| 6.79 | 6.84 | 29.999 | 35.9 | 3.699 |
| 6.78 | 6.84 | 29.998 | 35.9 | 3.772 |
| 6.78 | 6.84 | 29.996 | 36 | 3.847 |
| 6.78 | 6.84 | 29.994 | 36 | 3.929 |
| 6.78 | 6.84 | 29.993 | 36.1 | 4.026 |
| 6.78 | 6.83 | 29.992 | 36.1 | 4.735 |
| 6.78 | 6.83 | 29.992 | 36.1 | 4.741 |
| 6.78 | 6.83 | 29.992 | 36.2 | 4.753 |
| 6.78 | 6.83 | 29.992 | 36.2 | 4.776 |
| 6.79 | 6.83 | 29.997 | 36.2 | 4.824 |

| | | | | |
|---|---|---|---|---|
| 6.79 | 6.82 | 29.998 | 36.3 | 4.904 |
| 6.79 | 6.82 | 29.998 | 36.3 | 4.992 |
| 6.79 | 6.82 | 29.998 | 36.3 | 5.766 |
| 6.8 | 6.81 | 30 | 36.4 | 5.884 |
| 6.8 | 6.81 | 29.998 | 36.4 | 5.855 |
| 6.8 | 6.81 | 29.994 | 36.5 | 5.938 |
| 6.8 | 6.81 | 29.992 | 36.5 | 6.017 |
| 6.8 | 6.81 | 29.992 | 36.5 | 6.104 |
| 6.8 | 6.81 | 29.992 | 36.5 | 6.191 |
| 6.8 | 6.81 | 29.991 | 36.6 | 6.276 |
| 6.8 | 6.8 | 29.991 | 36.6 | 6.351 |
| 6.8 | 6.8 | 29.991 | 36.6 | 6.485 |
| 6.8 | 6.8 | 29.99 | 36.7 | 6.622 |
| 6.8 | 6.8 | 29.99 | 36.7 | 6.737 |
| 6.79 | 6.8 | 29.99 | 36.7 | 6.877 |
| 6.78 | 6.8 | 29.99 | 36.7 | 7.778 |
| 6.77 | 6.8 | 29.99 | 36.8 | 7.857 |
| 6.77 | 6.8 | 29.99 | 36.8 | 7.942 |
| 6.77 | 6.8 | 29.99 | 36.8 | 8.071 |
| 6.77 | 6.8 | 29.99 | 36.8 | 8.191 |
| 6.77 | 6.79 | 29.99 | 36.8 | 8.255 |
| 6.77 | 6.8 | 29.99 | 36.8 | 8.313 |
| 6.77 | 6.8 | 29.99 | 36.8 | 8.334 |
| 6.77 | 6.81 | 29.99 | 36.9 | 8.341 |
| 6.77 | 6.82 | 29.99 | 36.9 | 8.377 |
| 6.78 | 6.83 | 29.991 | 36.9 | 8.397 |
| 6.77 | 6.83 | 29.99 | 36.9 | 8.38 |
| 6.77 | 6.83 | 29.99 | 36.9 | 8.349 |
| 6.77 | 6.83 | 29.99 | 36.9 | 8.317 |
| 6.76 | 6.82 | 29.99 | 36.9 | 8.279 |
| 6.76 | 6.81 | 29.989 | 36.9 | 8.271 |
| 6.76 | 6.8 | 29.989 | 36.9 | 8.299 |
| 6.75 | 6.79 | 29.989 | 36.9 | 8.363 |

113

114

115

116

117

118

119    **3. $k_{600}$ correlation scatterplots**

120

[Figure]

121

122    Fig.1: Scatterplots between $FCO_2$ (A) and $k_{600}$ (B) as a function of wind speed. Values from
123    figure 5 (A) include high and low water seasons. Figure 5 (B) comprises only high water values
124    for statistical correlation (Spearman correlation). Rho values are located on each image left
125    superior side.

---

## Author Response (AR1)

Dear editor,

We would like to thank you for the opportunity to submit a revised manuscript.

Please find our answers below after each referee comment *in italic*.

Referee 1 comments

This paper is about the $CO_2$ concentration and emissions from a newly created hydroelectric reservoir complex in the Amazon area. Given that particularly Amazonian reservoirs have been pointed out as high emitters of greenhouse gases, and since emissions typically are higher the first years after flooding, this study is certainly valuable and interesting. In particular since the new reservoir is a run-of-the-river type, which is supposed to result in lower emissions than storage reservoirs. The study seems to be well-conducted, based on standard methods. However, the presentation severely lacks focus and clarity. I will give in the following a few idea on how the paper can be improved, but I really want to urge the senior authors of this paper to support and help the first author, who is apparently a MSc student and writes his/her first paper (it says in the Acknowledgements). It also takes a thorough revision of English language use and style.

*Thank you, we hope to help clarify the role of run-of-the-river dams on $CO_2$ emissions, particularly in the Amazon. We have modified the manuscript based on your suggestions, and all authors have carefully reviewed the manuscript for style and clarity.*

What makes this study interesting is that it studies the Belo Monte hydroelectric complex, a all-new installation in the Amazon (it's not even up at full capacity yet), the biggest in the Amazon so far, and one of the biggest in the world, and one that was heavily disputed and criticized. This is not mentioned at all in the paper! I could imagine that the story could be built around the case of this new and huge installation. New reservoirs typically have elevated emissions, but here apparently biomass was removed before flooding, at least partially. Is this visible in the data? One of the reservoirs is run-of-the-river, does it really have lower emission than the storage reservoir? These questions could be formulated as hypotheses, addressed with the data (i.e. figures should illustrate data in a way that relates to these hypotheses), and then explicitly answered in the Discussion. This would give the study a much-needed 'read thread'.

*We agree that we did not convey the controversy surrounding the Belo Monte hydropower operations in the original manuscript. We have now added in the Introduction section a brief discussion about the historical debate and controversies regarding the construction of Belo Monte (L63-68). It is worth mentioning that the Belo Monte hydroelectric complex is the largest in power capacity (11,233 MW) in the Amazon, but not the largest regarding the area of the reservoir. Among all the newly constructed or planned dams in the Amazon, Belo Monte is, in fact, the most efficient in terms of energy production per $km^2$ of reservoir according to Faria et al 2015.*

*Despite complete forest removal, plant-derived material still remained in the Intermediate Reservoir (IR). In the Xingu Reservoir (XR), forest removal was done only*

*in some large islands. However, 59 % of the area of this reservoir represents the previous river channel where the riverbed consisted of bedrock and sand. Therefore, lower emissions were expected for the XR in comparison with the IR. Nevertheless, our results show higher $CO_2$ fluxes in the IR only during the low water season (Figure 3/Table 2). A possible explanation for the lack of difference in $CO_2$ fluxes among the reservoirs at the high water season could be related to the shorter residence time, vegetation clearing and organic material inputs and availability in this season (more details regarding this can be found in the discussion section (Lines 303-318 and 372-378). A more detailed discussion was added in the lines previously mentioned and percentage of river channel area was corrected (L319).*

*The study hypotheses were perhaps unclear in the original manuscript. We have improved this in the revised manuscript, including the addition of the following hypotheses (1) the two Belo Monte reservoirs have contrasting $CO_2$ partial pressure ($pCO_2$) in the water and carbon dioxide fluxes to the atmosphere ($FCO_2$); and (2) the clearing of forest vegetation significantly reduces the emissions from areas flooded by the reservoirs during the first two years after channel impoundment (L95-99). Based on referee 1 suggestions, we have proposed these hypotheses that best fits the manuscript's storyline. Also, some sentences were added to support and better address these hypotheses (L84-91). In order to clarify our objective it was also rewritten (L91-94). Thank you for this useful feedback.*

It will take a thorough rewriting of the manuscript before it may become acceptable, but since it seems to be good data from a understudied site of high interest, I think in the end this could become a valuable addition to Biogeosciences.

*Thank you for your comments, they've helped to shape a stronger manuscript. We have worked hard to improve the manuscript quality and hope to contribute to the knowledge of tropical run-of-the-river reservoirs.*

Detailed comments: Title: the influence of reservoir traits is not explored to any greater depth. Which traits? I'd suggest to change the title accordingly, maybe "$CO_2$ concentrations and emission in the newly constructed Belo Monte hydropower complex in the Xingu River, Amazonia".

*We intended to use the word "traits" in the title to describe our comparison of storage and run-of-the-river reservoir types. However, we agree that this title was a bit unclear. After a restructuring of the manuscript we decided to accept the suggested title change, which fits the scope of the manuscript well.*

L41. The inland water area number seems wrong. See Verpoorter et al. 2014 GRL

*The inland water area value number is related only to rivers and streams (i.e., not including lakes and wetlands) based on Raymond et al. 2013. We updated that information with the lake surface area estimated by Downing et al. 2006 and Verpoorter et al.2014. In addition, flux information was corrected and terrestrial carbon influx data was added based on Drake et al. 2018 (L40-44).*

L42. Only the Raymond study gives a global estimate, the other citations are regional scale.

*The other citations were removed from this sentence to adopt only the global estimate of Raymond et al. 2013 (L42).*

L45-54. There's a lot of detail here that is not addressed by this study and could be removed here, e.g. microbial community structure or priming.

*The goal of this paragraph was to address the factors involved in $CO_2$ production. We agree that this section was perhaps too detailed. However, we feel that these concepts link well with our discussion of OM availability later in the manuscript (Lines 312-318 and 380-382) as $CO_2$ sources. Thus, we have altered these sentences to improve flow with the rest of the introduction, but feel that these are important concepts to introduce (L45-57).*

L70. While emissions are typically high, the lifetime emission of a reservoir is probablyrather a function of the long-term emission level, and the short initial emission pulse
may have less influence.

*We have modified this sentence to point out that emissions during the initial years are typically highest and the most uncertain, but that sustained long-term emission rates are likely important for the overall carbon balance of the system over its lifetime (L73-83).*

Study Area: This must mention that the installation is new, and it must describe in how far and where vegetation was removed before flooding, and when the flooding took place.

*Thank you, we have made this change (L119). In addition several other alterations had to be done in this sub-section according to the new hypotheses. Reservoir description was refined and rewritten as retention time calculations (L119-143).*

L106 and 114. The water retention times are very short, even for the storage reservoir it's only 1.5 days. Are these numbers correct? If so, these reservoirs, given their size, must be characterized by quite strong water flow, and thus the gas exchange velocity is probably hardly related to wind speed, but rather to water speed.

*Our residence time (RT) calculations had a mistake that is now fixed. The corrected RT was 20.2 and 3.4 days for IR and XR, respectively. Details regarding the RT estimate and values can be found in the manuscript (L135-143).*

L112. 97% of the capacity are at the Belo Monte dam, so the ROR dam only produces 3% of the energy even though it contains one third of the number of turbines?

*The difference among both dams is not only in size and turbine number. The turbine model also differs between dams, which influences the generating power. The main powerhouse is equipped with 18 turbines Francis type with active unit power of 611.11 MW as complementary powerhouse has 6 Bulb type turbines that are considerably less potent with active unit power of only 38.85 MW.*

L116. Where is the hypolimnion typically starting? Did you do any depth profiles of T and/or DO? If so, please show and report! If not, please cite a study that states that the thermocline is typically at >20 m.

*During our samplings, the water column had a well-mixed pattern without variation in DO in most of the reservoir's area. The hypolimnion was only apparent in the IR, close to the dam, where DO decreased drastically at approximately 50 m from a total depth of 58 m. However, as observed on Faria et al (2015), its formation is not expected on Belo Monte Reservoirs. Therefore this sentence was altered, and hypolimnion information was withdrawn. We have now included depth profiles in material and methods and graphs of variables such as DO, temperature, etc. in the supplemental material (L207) (supplement material figure S1).*

Section 2.2. It would be more easy to understand if you first described your sampling campaigns, and then tell about any gaps.

*We have made this change accordingly (L187-189).*

L144. Why was 60% of water depth chosen? Seems arbitrary. Also, it would be good to know the actual depth at these sites. A raw data table should be submitted alongside with the paper.

*60% depth was chosen as a mid-depth sampling point to compare surface and bottom waters. In deeper sites, the three depths (surface, 60%, and near-bottom) were sampled due to the variation in water velocity. Our goal was to sample depths with different organic and inorganic matter abundance due to water flow transport (L162-164). We have added depth information to Table 1.*

L150. How good was the evacuation? In my experience, it's very difficult to get a good vaccum, but probably 10% or more atmosphere will remain, which may dilute or contaminate your samples. Was this checked?

*We are confident in our sample storage methods, which our team has extensive experience with. A vacuum pump was used to create a vacuum, which was confirmed since the volume of gas pulled from the syringe into the vial was similar to the vial volume without the needing to manually depress the syringe's plunger. We have not added these details to the manuscript, as transferring gas to vials is a common method.*

L154. Start this paragraph with saying "Diffusive $CO_2$ emission was measured with floating chambers". Also, please give the dimensions, shape and type (transparent / opaque) of the chamber.

*The text was changed accordingly (L173).*

*Two types of floating chambers were used in different sampling campaigns. Both types were made of opaque polypropylene and  were covered with reflective aluminum tape. They were round and their volume and area were  the 7.7 L and 0.08 $m^2$; and 6 L and 0.07 $m^2$. Information about the chambers used was added to the text (L174-175 and L178-179).*

L161. I guess you mean logging frequency, not time.

*Exactly, we have modified this (L181).*

L168. Atmospheric $pCO_2$ of 380 ppm seems like an outdated value, or are these your

own measurements in air?

*The atmospheric $pCO_2$ of 380 ppm was used based on an outdated database. The data was re-evaluated and now measurements were discarded when the $R^2$ of the linear relation between $pCO_2$ and time ($\delta pCO_2/\delta t$), measured during chamber deployment, were lower than 0.90 ($R^2 < 0.90$) or in cases where we measured negative $FCO_2$ when the surface water $pCO_2$ was higher than the atmospheric $pCO_2$ based on measurement done at the same site. However, this happened only two times and could be attributed to some source of $CO_2$ contamination when placing the chamber into the water — thus, starting with a higher $pCO_2$ than the water (L187-189).*

L184. This sentence seems unnecessary

*We have removed it.*

L191. A station is stationary. You probably mean a handheld meter or device?

*Updated as suggested (L213).*

2.5. Statistics. I did not know Permanova, so this should be better explained. Is it a parametric method? Because it is stated that the data did not follow normal distribution. However, later in this paragraph, you mention some data were normally distributed and used t-test; this is confusing. Also, in the entire paper, report the actual p values, not just if p is lower of higher than 0.05.

*Agreed, the method was superficially mentioned in the manuscript. PERMANOVA is a multivariate variance analysis to compare variability between and within groups using permutation to obtain p-value. Due to the different hypotheses tested, the data set had to be adjusted and consequently altered the data distribution. In the case of T-Test, sites located on "outside reservoirs" and "downstream of the dams" were not considered and also season. Related to p value, we have now reported all the p-values accordingly to the real value obtained from the statistical test. PERMANOVA analysis was better detailed in the methods section as suggested (L219-221). We have removed T-Test analysis since it is related to a descriptive result.*

Results. In general, this section describes many findings and patterns, but it does so in a quite unstructured way, and is therefore difficult to follow. I really think it would help this paper if only the results were presented that are relevant to the hypotheses or research questions. Also, the language describing the patterns should be improved. For example, it needs to explained what numbers are given (e.g. L208, is this the mean ± standard deviation, or something else?), and comparisons between two groups describe a difference and not a variation (L208). Also, increase and decrease (e.g. L245 and L249) refer to a change over time and thus some form of time series data, while this study has data for two discrete sampling occasions, and thus can only speak about differences. It should also always be very clear what exactly was compared. For example, in L213, it was unclear what was tested here, the variability in $pCO_2$ within and environment, or between environments?

*Thank you, your comments were very constructiveto this section. Throughout the whole text, we presented values as mean ± standard deviation and indeed we were using the term "variation" when we meant "difference". It was corrected.*

*Most of this section was re-written for a clearer understanding (Lines 230-246, 254-259, 261-272 and 274-278). Here we meant that we have tested the $pCO_2$ variability between environments.*

*Several sentences had to be moved or removed from the text resulting in a complete rewriting of the section. The over usage of average values was revaluated and most of them also removed. Statistical analyses previously presented only for surface water are now were updated to include all depths (L241-243) (Table 3).*

Again concerning statistics, it is unclear to me how a comparison between two groups can render a $R^2$ value, but maybe that's a part of the PERMANOVA, and should in that case be better explained in the Methods.

*Our statistics description did not detail PERMANOVA correctly in the previous version of the manuscript, as so this test became unclear to the reader. PERMANOVA analysis tests similarity using a Euclidian distance index through permutations. The $R^2$ value is generated by permutations. As mentioned above, PERMANOVA analysis was better explained in the methods section as suggested (L219-221).*

L215. Here you speak about spatial variability, but do you mean differences of means between different environemnts, or the variability of measurements within one environment type?

*Here the test is to evaluate if the different environments (reservoirs, downstream the dam and outside the reservoir) presented different fluxes in each season. Temporal trends sometimes may mask some spatial patterns that only become visible when seasons are treated separately. Therefore, here we refer to a PERMANOVA test similar comparing $pCO_2$ between environments.*

L219. "Outside reservoir areas" is not a very illuminating term. Could choose another name?

*We agreed and replaced it to "unaffected river channel" (L149).*

L224. 281 µatm at 60% depth, how much is that in meters? And how can deep water be undersaturated in oxygen? Typically it is oversaturated. Or was this above a macrophyte bed?

*The total depth of this site is 7.5 m (Table 1), and the sampling depth was 4 m. This sentence describes $pCO_2$, not dissolved oxygen. The value of $pCO_2$ equal to 281 $\mu$atm was observed in the undisturbed river channel without macrophyte bed. Sub-atmospheric $pCO_2$ has been previsouly observed in other large clear water rivers in the Amazon region, resulting in negative $CO_2$ fluxes (Rasera et al. 2013), indicating net primary production. We are not sure what you mean by the question of how deep water can be undersaturated in oxygen—these large clear water rivers in the Amazon are known to have low turbidity, favoring algal productivity and resulting in high dissolved oxygen levels. However, in the Amazon mainstem where the high turbidity reduce algal productivity lower levels of dissolved $O_2$ and high $pCO_2$ are observed due to net heterotrophy.*

L231. Here it says the data from the two seasons were pooled, but L237-241, the seasonal data are discussed separately. This is confusing.

*Thank you for this comment. Our $FCO_2$ data is related to a time period of two years, including three seasons (2016 high water, 2017 high water and 2017 low water)( L189-191). The data pooled are from the same season, both high water, sampled with the same equipment and they were not statistically similar(L244-246). High and low water were measured with different equipment due technical issues and treated separately (Lines 174 and 178).*

L246. The seasonal difference in IR was very small, certainly not a "pronounced difference". Interestingly, $FCO_2$ was very different between seasons in spite of similar $pCO_2$, which indicates a strong variability in k. Was this the case?

*Despite the difference in averages, we observed that the seasonal difference in the IR was not significant and we removed it from the text. Regarding k, no statistically significant variation was observed between seasons (L274-276).*

L250. What kind of spatial analyses? Comparison of the means for different environments?

*PERMANOVA was used to compare simultaneously the variation of $FCO_2$, $pCO_2$ and $k_{600}$ between both reservoirs. This analysis did not generate difference of means, but the dissimilarity within versus and between groups through distance measures.*

L251. "evaluated together", is this warranted? Were these two groups similar?

*Our results indicate that they are similar. We have checked it by changing the river channel category to unaffected river channel (wich was grouped with outside reservoir sites, downstream the dams included). Only flooded areas represented each reservoir emission. Nevertheless, the same results were reached. A different classification reveals overlapping patterns; if a significant result was reached it would point to dissimilarity among groups.*

L256. "Pasture" is a new and undefined category.

*Upland forest and pasture were the main land cover in the areas flooded by the reservoirs as described in the description of the study area and measured sites (Table 1). They are both included in the flooded area category for the previous analysis. Here we evaluate if there was a difference in $pCO_2$ and $CO_2$ flux among the type of vegetation flooded. Thus, they were considered as a flooded area subgroup.*

L262. What's the measure of variability? It seems that in this study, you mostly compared means, but if you want to address the variability, you maybe want to look at relative standard deviations, interquartile ranges or something similar. If you want to stick to comparing means between environments, please formulate this explicitly in the text.

*There was some confusion with the term from our part. Our analysis describes difference by a distance matrix that calculates the similarity within and between groups, not variation. We assume that the poor statistics section may have complicated much of the reading. We have rewritten that section and replaced "variation" by "difference" in the whole manuscript.*

L263. Varied significantly between what?

*The $FCO_2$ differed significantly between XR and IR reservoirs during the low water season. This sentence has been modified accordingly in the revised manuscript (L268-269).*

L266. The 90 km downstream site is so far away it's not even on the map. I wonder in

how far it is relevant to this study at all, or could safely be omitted.

*Thank you for the observation. That was a mistake in the writing; the 90 km site is downstream of the Pimental dam (P20 site – Fig. 2), not Belo Monte. We believe that this site is relevant because of its location downstream of the Volta Grande do Xingu (Xingu Great Bend) region and a few kilometers upstream to where the Belo Monte dam discharge back into the original river. This information was properly corrected (L257-259).*

L270-273. Go straight to the results instead of first describing what was not done.

*We have made this change (L274).*

L275. The relationship between $k_{600}$ and wind speed is very weak. At any wind speed, k can vary with a factor of 2-4. This is quite often the case, and maybe even expected in such system where water moves fast, and thus water turbulence is quite independent of wind speed.

*Thank you for this comment. We agree, particularly considering the short residence time of the Belo Monte system. Since there was no significant k variability, the water turbulence may be the major factor driving $CO_2$ diffusion. We have corrected it in the manuscript (L276-277).*

All in all, the Results give many comparisons, What about making matrix tables where you can give test statistics for each comparison?

*Thank you that was a great suggestion. We have added such a table (Table 3).*

3.3. Did you ever measure depth profiles? Would be very interesting to show these data, to asses if really the turbine intake is in the epilimnion, and to assess the potential outgassing through turbine passage.

*Yes, depth profiles were made for temperature, pH, $O_2$ and conductivity. $CO_2$ was measured at the bottom, 60% of site depth and at the surface (0.3 m) during high water campaigns. Our data show a well-mixed water column without stratification close to the Pimental dam in the Xingu Reservoir (suplemmentary material figure S1). In addition to surface, correlation results of near-bottom depth were added (L292-294) (table 3).*

*Nevertheless, its intake is at the bottom, where even with high $O_2$, the $pCO_2$ is higher than at the surface. In Belo Monte dam $pCO_2$ follows the same pattern, although the $O_2$ decrease drastically at approximately 50 m (as mentioned above). As such, Belo Monte intake is in the $O_2$ rich zone. The depth profiles can be seen in the supplementary material, Figure S1 .*

L292. This is not one of your results.

*Removed.*

L296. The Discussion should start with your most important finding, not with citing other studies.

*Updated as suggested (L297-309).*

L303. This seems to be an important finding. Could you make a figure that illustrates this finding, to make it visible and convincing?

*Thank you for the suggestion. The Figure 3 was updated , and this findings has been highlighted in panels (e) and (f).*

L309-326. This discussion is very hypothetical and not much related to your data.

*Thank you for this comment. We have deleted this paragraph.*

L327. Not really. In your own data, there is an example of differences in k producing very different emission fluxes in spite of similar $pCO_2$ (see my comment above).

*We have added the clarifying statement "…although we did observe some specific examples of differences in k producing different emission fluxes even when $pCO_2$ was similar" (L297-298).*

L328-334. This may be the main message of this paper. It would be good if you produced a Figure that illustrates this finding.

*We have made a table summarizing these results (Table 5).*

L340-341. The Methods need to describe explicitly which areas were flooded with intact biomass, or after biomass harvesting.

*This information was already presented in table 1, however it was not as explicit as it could be. More information about the vegetation removal in each reservoir was included in the text (L125-129).*

L350. Could you actually observe increased water clarity in your data / samplings? If not, this discussion is not helpful to explain your data.

*No. Our turbidity data was not reliable due to poor calibration, therefore, we removed it from the paper. We have removed this sentence.*

L355. $pCO_2$ were only lower during low water compared to high water in the downstream and dam categories. For flooded and river channel, they were similar (Fig.3). So it is not warranted to speak about a "drastic decrease".

*We have deleted the word "drastic" from the text. The statistical test showed difference among seasons and to environment categories, which is corroborated by the lower $pCO_2$ averages during low water both to flooded areas and river channel (as shown in table 2). This sentence was rewritten in the new version of the manuscript (L326-329).*

L375-383. Could the difference between Belo Monte and Petit Saut be explained by different water intake depths? Do you have water profile data?

*After further observation, the near bottom anoxia in IR could be related to contamination (probe in contact with sediment). The most reasonable assumption is that any site had a stratified pattern (Supplementary material Figure S1). As Petit Saut has hypolimnetic water intake and lack of vegetation clearing the downstream emission is higher than Belo Monte, a ROR complex with well mixed waters and vegetal cover removed in most of the flooded areas. This information was updated in the manuscript (L346-353).*

L391. It seems not warranted to assume that any site or time point should serve as a "reference" for river $pCO_2$, since it varies in time and space.

*We have deleted this sentence.*

L395. What is meant by "turbine activity"?

*Since it was a cloudy discussion it was deleted.*

L398-406. I think you could further explore the patterns in k, e.g. between environments, and between reservoirs. Were the values in these reservoirs rather similar to other reservoirs or lakes, or rather to rivers?

*The $k_{600}$ in the XR (22.99 ± 8.00 and 22.89 ± 21.40 cm $h^{-1}$ on high and low water, respectively) were in the same range of the Furnas  reservoir (19.58 ± 2.5 cm $h^{-1}$) installed on Grande River located in Cerrado region (Paranaíba et al., 2018). The IR presented a wider range of values among seasons (7.13 ± 1.59 and 60.80 ± 18.02 cm $h^{-1}$ on high and low water, respectively), but the $k_{600}$ observed at the high water season was similar to values observed in the Javaes River (8.22 ± 3.80 cm $h^{-1}$) (Rasera et al 2013) (L388-393).*

L403. There is no strong positive correlation between wind speed and $FCO_2$ in your data. Fig 5 shows weak relationships, at best.

*Thank you for the highlight, we have updated the text as suggested (L393-401).*

L423-425. This sounds like the main result of this study. Make a figure to show and highlight this result, and discuss it in terms of reservoir properties and operation type.

*We have included a new figure that shows this difference, and the suggested points were included in the discussion (Figure 3).*

Figure 3. In panels c and d, I would suggest you order the environments in flow direction. That is, upstream first then XR environments, then IR environemnts, then downstream. If it gets too crowded, make two separate panels for high and low water. And the same for $pCO_2$ and $FCO_2$ and $k_{600}$, i.e. you may end up in 6 panels instead of 2. Together with panels a and b, it would be 8 panels.

*We have reordered and changed the categories. We added the categories in the following order: "unaffected river upstream", "XR", "IR", "downstream of the dams" and "unaffected river downstream". Panels were also separated by season and new panels $k_{600}$ were added.*

Figure 4. When seeing this figure, I wonder how much of this spatial variability is driven by differences in $pCO_2$, and how much by differences in k.

*To make the spatial variability more visible, we have added one more panel related to $k_{600}$ to figure 3.*

Table 2. What are the values, mean ± standard deviation? How many measurements are behind each of these averages? Could you introduce a column with "n"? The k values are high and resemble rather riverine systems than lakes or reservoirs, I guess an effect of the fast water flow. The comparison with literature values would be better and more visible in a graph than in a table.

*The values are averages ± standard deviation, except for Sawakuchi et al. 2017. During high water $FCO_2$ was measured three times (L177), and during low water two $FCO_2$ measurements were made simultaneously (L180) and headspace was sampled on triplicates (L164). It is feasible that the turbulence in both reservoirs is mostly related to water flow. A compilation of the literature information of $k_{600}$ were made and is presented in Table 5.*

Referee 2 comments

Please find our answers bellow after each referee comment in *italic* font.

Review of Araujo et al. This manuscript describes the results from a 2-yr study during high and low water seasons on the Belo Monte hydropower complex that consists of two main reservoirs, one of which is defined as a run-of-river and the other as storage. The authors aimed to contrast the impact of these two reservoir types on the $CO_2$ dynamics of the entire complex. Additionally, they contrasted $CO_2$ dynamics across various flooded environments within the complex. The manuscript has some nice data but is predominantly descriptive. Regardless, data in tropical reservoirs is currently necessary and it is interesting to contrast these two types of system. Not to mention the huge dispute over this massive Amazonian project. I have many suggestions for how to improve this manuscript before this paper is ready for publication.

*We appreciate your suggestions, which have greatly enhanced the manuscript and its potential impact.*

General comments

1. Be careful with the word 'traits' in the title. It implies features that do not vary in time. Is that the focus here? Do you mean ROR vs storage, plus flooded landscapes? That would be okay then. But if that was the case then I did not get the impression enough from your discussion that that was your focus. You need to bring out your main points much more. Try focusing the research questions or objectives more narrowly. This will help you throughout the entire publication.

*Thank you and that was the point. Our initial goal was to define the group of characteristics that classify each reservoir as 'ROR' or 'storage'. However, after restructuring the hypothesis and re-evaluating the reservoir's characteristics we removed those classifications. Although a larger flooded area, deeper and a lake like aspect the Intermediate reservoir, it could not be classified as a storage reservoir due to its short water residence time (L142) (Faria et al., 2015). Both reservoirs were considered as ROR (L87) and they were now compared according to their extention of flooded area and water residence time (L84-91). Consequently , we have changed the tittle accordingly to the Referee 1 suggestion ("$CO_2$ concentrations and emission in the newly constructed Belo Monte hydropower complex in the Xingu River, Amazonia").*

2. Language overall needs improvement. Too many commas used. Too many sentences that are confusing (many are mentioned in specific comments below).

*The language and style of the whole text was revised and improved.*

3. Abstract needs more quantitative results in it

*Thank you, this section was revised. Some unnecessary information was removed (L2-9) to make space adding a better description of our results findings (L9-24).*

4. Introduction does not discuss the importance of this particular reservoir more.

*More details concerning Belo Monte and its controversy were added to the text as suggested by both reviewers (L63-72).*

5. Methods – description of how reservoirs are connected is not clear. In the map figure there appears to be a channel connecting them too. Please improve the description of how the reservoirs interact, including flow directions, which should be on your Figure 2, and individual surface areas.

*The Pimental dam in the XR, regulates the water flow towards the IR through a 28 km artificial channel, constructed at the left margin of the XR, to feed the IR where the main powerhouse is located (Brasil, 2009) (L122). The channel description and reservoir interaction were clarified (L119-124). Also, Figure 2 was updated as suggested.*

6. I find section 3.1 of the results very confusing to read and absorb fully. There are a lot of numbers that are perhaps not necessary and very distracting from understanding what you are trying to describe. I would suggest a schematic to help describe the temporal (high vs low water) variability you see that also includes the spatial variability (across environments). You can use weighted markers for the various fluxes and concentrations that correspond to high and low values, if not the real values.

*This section was revised for conciseness and clarity. After these changes the text became clearer and we believe that a schematic figure is not needed.*

7. Figure 2 – needs arrows for direction of flow.

*Done.*

8. Figure 3 – You can make these 4 plots into just 2 in the following manner: put the white boxplots from (a) and (b) that are $pCO_2$ in the beginning of (c) labeled 'High water' and 'Low water', and the gray boxplots that are for $FCO_2$ in the beginning of (d) with the same labels. Also, are the environments in c and d labeled in the proper order – from one are to another? Or does it not work like that because of the reservoir geomorphology? Either way, I would put downstream the dams on the right side since most people read left to right and you naturally think downstream to the right.

*Thank you for the interesting suggestion. The environments were previously organized in alphabetical order on the plots. However, we agree that it will be easier for the reader to follow the downstream orientation. Therefore, it was corrected to flow order. We added more plots to this image according to referee 1 suggestion, with season separately and an additional variable ($k_{600}$). The categories were also changed to "unaffected river upstream", "XR", "IR", "downstream the dams" and "unaffected river downstream". In the last version there was an error in the $FCO_2$ unity. Fluxes were in $\mu mol\ CO_2\ m^2\ s^{-1}$, but instead $s^{-1}$ in the legend was written $d^{-1}$. This error was corrected.*

9. Figure 4 – you need units listed for the values; direction of flow arrows would be good; and mention in caption that (a) includes 2 years of data while (b) only has one year (and list which years).

*Updated as suggested. Figure 4 had the error on $s^{-1}$ unity replaced by $d^{-1}$, that was already corrected. Also colors were updated.*

10. Figure 5 – you mention these figures in terms of stats but there are no lines on it and no equations or states in the figure caption.

*These figures show the correlation between $k_{600}$ and $FCO_2$ with wind speed, We have used the Spearman correlation test, which is ranked test, and do not have mathematical model or equation. The Rho values were reported for each comparison. In this new version, figure 3 includes the $k_{600}$ results, as suggested by referee 1 and the figure 5 is now included in the supplement material.*

11. The discussion seems like a bunch of descriptive paragraphs thrown together. It is lacking some cohesive red line to follow and it is hard to locate your main points. Perhaps you can start to fix this by using subsections. Looks like you broke it down into the following: Seasonal variability; Vertical heterogeneity; $FCO_2$; Spatial variability;

Comparison to other reservoirs; k600; Operation. These are all just descriptions of data in reality. You want to discuss the most interesting findings of your study and then compare them with other studies. Figure out your few most important findings and try to arrange the discussion around those first. You also measured the system right after flooding, which is when emissions should be highest. This needs to be addressed in your conclusions.

*The discussion was rewritten in order to address the main points of the work (L297-401). The text was rearranged and divided into subsections that we believe are now more connected with our main findings and hypothesis, as nicely suggested. The high water season reservoirs comparison was removed since they were not significantly different. We have also highlighted in the manuscript that measurements were made during the first years after impounding (L406-408).*

Specific comments

Line 16-17 – did you measure clearwater rivers yourself ? if not, then either change or delete this sentence because it makes it sound like you.

*Our measurements were done only on the Xingu River; therefore, we altered the sentence to clarify this issue (Lines14 and 17).*

Line 41 – You mention that 'inland waters' have an area of '624,000 km2' and cite who with regards to this number? This number is very small compared to the 2.5 – 5 million km2 range that actually exists for all inland waters surface area coverage. I think you mean to cite only rivers surface area with your 0.624 million km2 value so you need to be specific when you say 'inland waters' and you need a specific reference for this river surface area number. But then you cite the 1.8 – 3.8 Pg values, presumably from Drake et al. 2018 and those values are for all inland waters specifically. If you want to discuss inland waters surface area coverage total then you need to use either Downing et al. 2006, Verpoorter et al. 2014 or Messager et al. 2016 or Feng et al. 2016.

*The area previously mentioned was for rivers and streams according to Raymond et al. (2013). We have changed this paragraph to discuss inland waters as a whole and on a global scale (L40- 44).*

Line 45 – clean up language (e.g., don't need 'water' so many times

*Done (L45-46).*

Line 50 – should be: 'to the autochthonous respiration of OM deposited'

*Thank you, we have updated this sentence as suggested (L50).*

Line 54 – should explain more how the stimulation of OM decomposition via those two processes actually effects $CO_2$ – similar to how you did in the first half of the sentence saying higher $CO_2$ uptake

*Agreed, we have divided this sentence in two and rewritten it to clarify the text. Those processes were added to the text and briefly explained as suggested (L52-56).*

Line 66 – I believe it was actually DelSontro et al. 2010 and not 2016

*Absolutely, thank you. This citation was corrected (L77).*

Line 69-70 – Start a new sentence with 'Newly flooded reservoirs...' and then give examples/references of the few poorly studied reservoirs.

*We have rewritten and added more information about GHG emissions from newly built and future tropical reservoirs estimates from the literature (L73-83).*

Line 73 – should be 'variability' and not 'variation

*Updated as suggested (L93).*

Line 73 – give the abbreviation for fluxes here '($FCO_2$)' that you will use the rest of the paper, and delete 'and its relevance for GHG fluxes'

*Thank you, corrected accordingly (L93).*

Line 75 – end this sentence with '..complex in eastern Amazon, a tropical region poised to gain XXX more hydropower projects in the coming decades (REF).' This puts your work into a bigger perspective at the end of your intro.

*Thank you, this fragment was included, but converted in another sentence (L94-95).*

Line 83 – the 1984 study is quite old… Is there nothing newer?

*It is related to a classical study that classifies Amazonian rivers according to physicochemical characteristics. Although relatively old, it is still largely used for the classification of large Amazonian river.*

Line 98-100 – in this sentence give the names of the two reservoirs after you mention them.

*This sentence has been removed, but new information about the reservoirs and the Belo Monte complex were added in the introduction, including reservoir's names (Lines 84 and 85-86).*

Line 101 – give more details about these calculates from Faria et al. 2015

*The residence time was calculated by the equation RT= V/ Q, where RT is the residence time in seconds, V is the reservoir volume in $m^3$ and Q is the volumetric discharge in $m^3$/s. To convert RT in days the value was divided by the number of seconds in a day. We altered this sentence and added this information in the text (L135-143).*

Line 104 – once you have given the XR abbreviation for Xingu Reservoir then use it for the rest of the paper, and do you mean 'as islands' instead of 'in islands'?

*This sentence was removed.*

Line 107 – 'classified' instead of 'denominated' – and this paragraph should contain the surface area of these reservoirs already

*This sentence was removed. Reservoir areas were added to the text on lines 125 and 128.*

Line 115 – the residence time of the IR reservoir is still ridiculously short (1.57 days). How do you call that a storage reservoir? Still want to know the surface area of these reservoirs already

There was an error in our RT calculations due to the discharge data used. The previous RT values were based on an environmental impact study (EIA) that estimated the highest discharge values of each reservoir. We performed new calculations using the average historic discharge series from Water Agency of Brazil database. The corrected RT of 20.2 (IR) and 3.4 days (XR) were updated in the manuscript. In

addition, we added the surface areas of the IR (154 km²) and XR (342 km², including the 228 km² originally occupied by the river channel). Due to the low RT of the IR we are not considering it as a storage reservoir. However, both reservoirs are still compared (*L125-134*).

Line 116 – should give maximum depths of the reservoirs

*Thank you. Maximum depths were added (L120).*

Line 117 – why did you give the total surface area of the 2 reservoirs together? You should provide values for the two different reservoirs. If this is difficult because of the difference between rainy and dry season then state this but still give approximate values for the individual reservoirs since you are evaluating them separately.

*The reservoirs areas were added as mentioned above.*

Line 121 – what is the 25.4 km2/MW? Why should I care about this value? Give some explanation behind your reporting of this value (or don't report it).

*It was removed.*

Line 131-132 – I really do not understand your description of water depth sampling. You classified the sampling sites based on their maximum depths? Where did you measure in the water column? If a site was 10 m deep, did you sample at 3 depths? Did you sample 0.3 m, 6 m, and 9 m? Be more explicit with your description here. Why did you pick 60% of max total depth for sampling?

*The water column sampling method was as you have described. The sampled depths were related to the total depth of each site. These depths were chosen based on the variation of water speed and transport of suspended particles in the water column of rivers. Our 60% depth was a mid-depth sampling point to compare to surface and bottom waters. The three depths were sampled only in deeper sites where higher water velocity variation occurs. Since water flow and topography drives pressure gradients on sediment interface that affect particulate matter transport (Huettel et al. 1996). In shallow sites (depth <7,5 m) samples were only taken at 60% the depth. We revised this sentence to clarify the text and moved to headspace sampling description (L162-164).*

Line 136 – state that the flooded areas sampled were in both reservoirs if that is the case.

*Thank you, updated as suggested (L152-154).*

Line 143 – 'according' not 'accordingly'

*We corrected this word (L161).*

Line 148 – what did you collect the headspace air in?

*The samples from the headspace and atmospheric air were collected using 60 ml syringes and immediately transferred to evacuated glass vials closed with butyl rubber stoppers and sealed with aluminum crimps. The vials were evacuated immediately before transferring samples using a needle. We have updated this information (L169-171).*

Line 150 – how were the gas samples transferred? Via needle and syringe because the vials were pre-capped, I presume.

*Please see the above comment.*

Line 154-156 – combine these two sentences into one

*Thank you, we have made this change (L173-175).*

Line 158 – if you made measurements from a drifting boat in a river, I presume you drifted quite a bit. Did you consider this drifting distance in your measurements of flux? This is an important point. How far did you drift? You need more details regarding this sampling approach.

*Drifting distance was not measured during deployments. Based on visualization in Google Earth we estimate that the maximum distance drifted may be approximately 1 km for measurements in the river channel up and downstream of the reservoirs. In sheltered areas located in bays and over islands with standing trees, where the water flow was low, drifting was very short and caused by wind. An estimate of the drifting distance in the natural river channel and in the main channel of the Xingu Reservoir was obtained by using the average water velocity measured by the National Water Agency of Brazil at the Altamira station. We separated the historical values into before and after 2016, when the dams was completed. Therefore, representing estimates of water velocity in the natural river (between 2005 and 2016), and in the Xingu Reservoir main channel (after 2016). The average water velocities at Altamira are 0.74 and 0.24 $m\ s^{-1}$ for before and after the dam, respectively. Assuming that there is no resistance of the boat with the water or air, drifting speed is similar to the water velocity. The total time of deployment was up to 30 minutes for the three consecutive measurements. Based on these we found that in the main channel of the Xingu Reservoir the drifting distance would be 432 m, and 1332 m for the natural river channel up and downstream the reservoirs. These details were added to supplementary material.*

Line 161 – 'calculated' instead of 'done' and delete 'the eq. (1)'

*Thank you, done (L182).*

Line 168 – use 'erroneous' instead of 'same sampling site'

*Thank you, the sentence was altered (L187-189).*

Line 171 and eq. 2 – you say that k was based on the flux measurements but I do not see them in equation 2. I guess it is somehow in the partial pressure measurements since some are in the chamber but I think this needs a better explanation. You didn't find k using $FCO_2$, but rather using the concentrations in the chamber? That is how I perceive this equation.

*Thank you, that is correct. The calculations were not made with fluxes, but with the $CO_2$ partial pressures inside the chamber. We corrected this sentence in the manuscript (L193-194).*

Line 176 – need 'respectively' at the end of the sentence

*We have altered this sentence (L198).*

Line 177 – grammar is poor here

*Thank you, sentence rewritten (L199).*

Line 184 – give a bit more detail here about how the gas transfer velocities were not calculated from 2016 data. I am guessing it is because the other loggers did not allow it

somehow, but I don't see why you couldn't perform the calculations using concentrations from those loggers too.

*This sentence was removed, but the lack of gas transfer velocities for the high water season of 2016 is due to the lack of water $pCO_2$ data in this campaign.*

Line 187-188 – I do not understand why or how these measurements were made according to the water depth classes. Do you just mean depths? And did you do this at each sampling site?

*This part was changed. Depth profiles were done along the whole water column at each sampling site (see suplementary Figure S1). However, for testing the relationship between the physicochemical parameters and the $pCO_2$ we have selected only the physicochemical data for the same depths where $pCO_2$ was measured. The depth profiles for $O_2$ and temperature for each environment are presented in the supplementary material.*

Line 199 – what does 'assessed separately by season' mean?

*Thank you for the observation. That means that the statistical test was done using results for each season separately, since there was no inter-calibration among the different sampling method on each season.*

Line 208 – you should restate here specifically that you are comparing high and low water from 2017 only.

*We have added this statement as suggested (L230).*

Line 208 – replace 'presented a significant variation' with 'varied significantly'

*Thank you, we have altered this sentence (L231).*

Line 221 – it gets confusing a bit when you go between comparing seasons to looking at the whole dataset so be specific when you can. For example, I would add 'From the overall dataset,' before 'Higher $pCO_2$ was registered..'

*Thank you, we have re-evaluated this section to clarify the manuscript. This sentence was altered as suggested (L238). Whole paragraph was rewritten and reorganized.*

Line 223 – I am confused by this sentence and what is respective to each other. Rewrite this one.

*This sentences was rewritten as suggested (L238-240).*

Line 228 – Because you only had $pCO_2$ data for 2017 then I guess you couldn't find a correlation between $pCO_2$ and $FCO_2$ in the 2016 data, correct? You need to specific again here and state that the correlation was only for the one method.

*That is correct, we could not do this test for 2016 due to the lack of water $pCO_2$ data. The correlation found between $pCO_2$ and $FCO_2$ corresponded to 2017. This sentence was rewritten, near-bottom results were added (L241-243) (Table 3).*

Line 232-234 – does it really matter if the two sensors were not cross calibrated in terms of absolute concentrations if it is just the slope of the increase of concentration over time that you need for flux calculations? If it is merely slope then you should be able to estimate and then compare the rates of flux, no?

*This is correct, but because we were not able to do any intercomparison we chose to be conservative and evaluate them separately to avoid any source of error in our interpretations.*

Line 235-237 – how is it that that the low water season had the highest and lowest $FCO_2$ values but was also homogeneous? This is very confusing.

*The homogeneity in the $FCO_2$ occurred when both reservoirs were evaluated together, however when each reservoir is considered separately the fluxes differed. Therefore, the pattern observed in low water season is driven by the reservoirs characteristics, not the spatial heterogeneity. In the low water season, the IR presented the highest $FCO_2$ that may be attributed to the presence of remainings of plant-derived material left from vegetation clearing. On the other hand, the $FCO_2$ in the XR decrease may be related to the natural seasonal pattern of $FCO_2$ observed in undisturbed rivers in the Amazon. The area of the XR is in its majority the original river channel where rocky and sandy substrates predominate (L372-385).*

Line 242-243 – this sentence is kind of just hanging here by itself. Shouldn't it belong somewhere in a paragraph.

*We have removed this sentence.*

Line 244 – I would rename this section a bit more specific to what you are doing: 'pCO$_2$ and FCO$_2$ in ROR versus storage reservoir'

*Thank you, we have altered the section names properly since ROR and storage classifications are not used anymore.*

Line 245-246 – if you consider the standard deviation of your measurements then I would say the differences are not so significant between seasons as they then overlap, especially for IR

*Actually there was no statistical difference among reservoirs' $pCO_2$ according season, therefore we removed this sentence. Conclusions had to be updated accordingly (L404-405).*

Line 249 – the difference in IR is much more significant than XR. I would point that out here.

*Thank you, we have altered this sentence and removed those average values from the text. A summary of the average values is now presented in Table 2 (L263-264). The difference observed in the reservoirs are now pointed out by our statistical analysis (L269).*

Line 250-252 – I don't understand what you mean here. You did a spatial analysis but lumped all spatially different environments together? I think you mean to say that you compared the total emission from XR to the total emission of IR despite the emitting environment. Is that right?

*Thank you, this was exactly what we meant. We have better addressed this in the text (L261-263).*

Line 252-255 – I don't understand how you see no significant difference between $pCO_2$ of XR and IR but then suddenly find that XR had $pCO_2$ 721 uatm lower. And lower than what? I guess IR. These few sentences are very confusing.

*The T-Test analysis was removed since it is related to a descriptive result.*

Line 256 – You cannot just present an idea like 'Standing vegetation type in XR flooded areas influenced $pCO_2$' without explaining the data that led you to that conclusion.

*Thank you. This sentence was modified to explain the data better (L254-255).*

Line 264 – use 'especially' instead of 'specifically'

*This sentence was removed in results section rewriting.*

Line 266 – what is a 'gradient pattern downstream'??

*We refer to the pattern of both $pCO_2$ and $FCO_2$ that are higher directly downstream the dam and decreases on the sites most distant from the reservoir. We have rewritten this sentence and removed this term (L257-259).*

Line 272 – again with this 'separately to each season' – I still do not understand what this means. You have to come up with a better way of describing this.

*This sentence was removed following referee 1 suggestion. But to explain, the $FCO_2$ data was measured using different equipment in 2016 (high water) and 2017 (high and low water), and we chose to evaluate them separately to avoid any potential source of error in the comparisons due to the lack of cross calibration. Thus the seasonal comparison was done using only data from 2017, when only the $CO_2$ loggers were used to measure the $CO_2$ fluxes.*

Line 274 – use 'without significant spatial heterogeneity across environments'

*Thank you, we have modified this sentence as suggested (L274-276).*

Line 275 – use '$k_{600}$ strongly correlated with wind…' and does this relate to Fig 5b? Should you reference this?

*Yes, this sentence is related with fig 5b. The sentence was changed and table mentioned (L276-277).*

Line 280 – there is not environmental breakdown in the data in Figure 5

*Thank you, this reference was removed.*

Line 287 – so you have water column data? Where is this data?

*Depth profiles were made in 2016 and at the high water of 2017. You can find in the supplementary information a description about how it was done and Figure S1 showing the $O_2$ and temperature variation accordingly to type of environment*

Line 303 – decrease in what?

*This sentence seemed unnecessary and it was removed after posterior revision.But we meant that the $pCO_2$ decreased was due to the transition from high to low water caused by a lower organic matter input.*

Line 344 – what is 'vegetal suppression'? I figured out that it is when you remove vegetation prior to flooding but is this the correct term for this? It sounds very strange.

*Vegetation clearing is the most adequate term. This was altered through whole text.*

Line 344-345 – this sentence is too long with poor grammar

*Thank you, this sentence was rewritten (L315-318).*

Line 354-356 – combine those sentences

*Done (L322-324).*

Line 356 – how many of the environments? Do you mean all except IR? This is confusing. If it is just IR that ithe exception then you need to state it as 'all except IR'

*Exactly, we observed an increase in $FCO_2$ and $pCO_2$ at the low wate season for the IR only. The sentence was altered as suggested (L324-326).*

Line 357-358 – negative fluxes can be replaced with 'observed $CO_2$ uptake'

*Thank you, done (L327).*

Line 358 – 'light penetration and low suspended sediment'

*Thank you, updated as suggested (L328).*

Line 363-365 – you already spoke about this earlier. Try not to be redundant

*We have altered this sentence detailing the influence of vegetation prior to flooding on the $FCO_2$ (L379-380).*

Line 370 – need 'which' before 'would'

*Done (L385).*

Line 372-373 – I don't think you need these values here in the discussion.

*We agreed, this sentence was removed.*

Line 387 – can you give a site number for the 'site downstream IR'?

*Absolutely, this site is P21. This information was updated in the manuscript (L338).*

Line 391 – I don't think this true and I don't think you need this sentence about a reference for natural $FCO_2$ values

*Agreed. We have removed this sentence.*

Line 397-398 – do you mean that the downstream sites resembled river channel sites in terms of $pCO_2$ and $FCO_2$ values? Don't use 'traits' to describe this. Traits more refers to features that don't vary.

*Yes, that was what we meant. However, we removed this sentence due discussion rewriting.*

Line 408-409 – are you saying that the old reservoir you are using for comparison is Tucurui? The grammar here is confusing.

*Exactly, Tucuruí reservoir was compared to both XR and IR. The sentence  was rewritten (L360-362).*

Line 412 – what do you mean by hypolimentical waters? It should be 'hypolimnetic' by the way. But this just means bottom waters with an implication of stratification, but what specifically do you want to express here?

*We have removed this sentence.*

Line 419 – bad grammar in last sentence

*This sentence was unnecessary and removed.*

Additional References

*Brasil: Aproveitamento Hidrelétrico Belo Monte, Environmental Impact Study, Eletrobrás, Rio de Janeiro, 426pp, 2009.*

*Downing, J. A., Middelburg, J. J. and Melack, J.: Plumbing the global carbon cycle: Integrating inland waters into the terrestrial carbon budget, Ecosystems, 10, 171–184, doi:10.1007/s10021-006-9013-8, 2007.*

*Drake, T. W., Raymond, P. A. and Spencer, R. G. M.: Terrestrial carbon inputs to inland waters: A current synthesis of estimates and uncertainty, Limnol. Oceanogr. Lett., 3, doi:10.1002/lol2.10055, 2018.*

*Faria, F. A. M., Jaramillo, P., Sawakuchi, H. O., Richey, J. E. and Barros, N.: Estimating greenhouse gas emissions from future Amazonian hydroelectric reservoirs, Environ. Res. Lett., 10, 124019, doi:10.1088/1748-9326/10/12/124019, 2015.*

*Huettel, M., Ziebis, W. and Forster, S. Flow-induced uptake of particulate matter in permeable sediments. Limnology and Oceanography, 41, 309-322, doi: 10.4319/lo.1996.41.2.0309, 1996.*

*Paranaíba, J. R., Barros, N., Mendonça, R., Linkhorst, A., Isidorova, A., Roland, F., Almeida, R. M. and Sobek, S.: Spatially resolved measurements of $CO_2$ and $CH_4$ concentration and gas-exchange velocity highly influence carbon-emission estimates of reservoirs, Environ. Sci. Technol., 52, 607–615, doi:10.1021/acs.est.7b05138, 2018.*

*Rasera, M. de F. F. L., Krusche, A. V., Richey, J. E., Ballester, M. V. R. and Victória, R. L.: Spatial and temporal variability of $pCO_2$ and $CO_2$ efflux in seven Amazonian Rivers, Biogeochemistry, 116, 241–259, doi:10.1007/s10533-013-9854-0, 2013.*

*Raymond, P. A. and Cole, J. J.: Gas Exchange in Rivers and Estuaries: Choosing a Gas Transfer Velocity, Estuaries, 24, 312, doi:10.2307/1352954, 2001.*

*Verpoorter, C., Kutser, T., Seekell, D. A. and Tranvik, L. J.: A global inventory of lakes based on high-resolution satellite imagery, Geophys. Res. Lett., 41, 6396–6402, doi:10.1002/2014GL060641., 2014.*

Relevant changes list

*Referee 1 comments*

Title, pg. 1.

Introduction:

Belo Monte controversies, pg. 3, lines 63-68.

Inland waters information updated, pg. 3, lines 40-44.

Reservoir carbon balance, pg. 3-4, lines 73-84.

Study hypothesis, pg. 4, lines 84-99.

Material and methods:

Reservoirs characteristics, pg. 4-5, lines 119-143.

Residence time corrections, pg. 5, lines 135-143.

ROR and storage classes removed.

pCO2 380 ppm base altered, pg. 6 lines 187-189.

Depth profiles, pg. 7, line 207.

PERMANOVA description, pg. 7, lines 219-221.

T-test analysis removed.

Results:

Unnecessary average values and comparisons removed.

Statistical results to near-bottom and surface depths, pg. 8-9, lines 241-243, lines 292-294.

Rewritten and reorganized, pg. 7-8, lines 230-246, lines 254-259.

Downstream site error corrected, pg.8, lines 257-259.

Rewritten and reorganized, pg. 8, lines 261-272.

Rewritten and reorganized, pg. 8, lines 274-278.

Conductivity minimum value corrected, pg. 9, line 290.

Discussion:

Seasonal discussion, pg. 9, lines 297-309.

Petit Saut comparison rewritten, pg. 10, lines 346-353.

Turbine activity discussion deleted.

K600 comparisons added, pg. 11, lines 388-393.

Discussion updated, pg. 11, lines 393-401.

Figure 3 updated, new panels added, pg. 18-19.

Table 2 literature values removed, pg. 23.

Table 3 added, pg. 24-25.

Table 5 added, pg. 25.

Supplement material added.

*Referee 2 comments*

Abstract:

Rewriting, pg. 2, lines 9-24.

Introduction:

Inland waters information updated, pg. 3, lines 40-44.

Belo Monte controversy added, pg. 3, lines 63-72.

Material and methods:

Reservoirs characteristics, pg. 4-5, lines 119-143.

Residence time corrections, pg. 5, lines 135-143.

ROR and storage classifications removed.

T-test analysis removed.

Sampling methods better detailed, pg. 6, lines 169-171.

Results:

Sub-section names altered.

Statistical results to near-bottom and surface depths, pg. 8-9, lines 241-243, lines 292-294.

Discussion:

Section divided in sub-sections.

Discussion rewritten and reorganized, pg. 9-11, lines 297-401.

Conclusions:

Updated according manuscript chances, pg. 11-12, lines 404-405.

Updated according manuscript chances, pg. 12, lines 406-408.

Figure 3 updated, new panels added, pg. 18-19.

Figure 5, added to supplement material.

Marked manuscript:

*Rewriting on results and discussion sections were not marked to avoid confusion, since it was too extensive (lines 230-246, 254-259, 261-272, 274-278 and 297-401).

[revised manuscript text omitted]

---

## Author Response (AR2)

*Dear editor,*

*We are grateful for all comments and suggestions.*

*Please find our answers in italic following Editor Comments (EC) and Reviewer Comments (RC):*

EC1: Both reviewers had suggested a thorough English editing (a reviewer comment and your response shown below). Despite your efforts, there are still many sentences that need to be checked by a native speaker. Just to name a few examples here (pay attention to the parts in "…"):

*Author Response (AR): Thank you for the constructive comments. Co-author Ward, a native English speaker, has thoroughly reviewed the manuscript for grammar and typos including the sections highlighted below.*

Abstract - The Belo Monte hydropower complex located in the Xingu River is "one of the largest in the world" in terms of energy production capacity, and "the largest operating" as a run-of-the-river (ROR) hydroelectric system…

*AR: We modified this sentence as follows: "The Belo Monte hydropower complex located in the Xingu River is the largest run-of-the-river (ROR) hydroelectric system in the world and has one of the highest energy production capacities among dams".*

The $FCO_2$ (0.90 ± 0.47 and 1.08 ± 0.62 µmol $m^2$ $d^{-1}$ "to XR and IR respectively")…

*AR: We modified this sentence as follows: "$FCO_2$ (0.90 ± 0.47 and 1.08 ± 0.62 µmol $m^2$ $d^{-1}$ for XR and IR, respectively) and $pCO_2$ (1,647 ± 698 and 1,676 ± 323 µatm for XR and IR, respectively) measured during the high water season were on the same order of magnitude as previous observations in other Amazonian clearwater rivers unaffected by impoundment for the same season".*

However, the associated intermediate reservoir may "overcome" these emissions due to altered riverine characteristics….

*AR: We modified this sentence as follows: "However, the associated Intermediate reservoir (IR) may exceed natural river emission rates due to pre-impounding vegetation influence".*

Introduction – …which "has" 5.1 Pg y-1 of carbon "terrestrially delivered" (Drake et al. 2018) and "???" about 2.1 Pg C annually "emitted" to the atmosphere (Raymond et al., 2013)…

*AR: We modified this sentence as follows: "Inland waters cover an approximate area of 4.6 to 5 million km² or about 3% of Earth's land surface (Downing et al., 2006; Verpoorter et al. 2014). Roughly 5.1 Pg C $y^{-1}$ of carbon is mobilized into inland waters from the terrestrial biosphere (Drake et al. 2018), of which about 2.1 Pg C $y^{-1}$ is emitted to the atmosphere as $CO_2$ (Raymond et al., 2013)".*

Great debate emerged "from" the Belo Monte hydropower project…

*AR: We modified this sentence as follows: "Significant debate has surrounded the Belo Monte hydropower project since its initial survey in the 1980's due to the magnitude of the environmental impact and threat to local indigenous people".*

The aim of this study is to "characterize" the $CO_2$ emissions from the Belo Monte reservoirs in the first two years post-impounding by assessing the spatial and temporal variability of $CO_2$ partial pressure ($pCO_2$) and carbon dioxide fluxes ($FCO_2$) in the XR and IR.

*AR: We modified this sentence as follows: "The aim of this study is to evaluate $CO_2$ emissions from the Belo Monte hydropower complex during the first two years post-impoundment by assessing the spatial and temporal variability of $CO_2$ partial pressure ($pCO_2$) and carbon dioxide fluxes ($FCO_2$) in the XR and IR".*

Discussion - Although $pCO_2$ and $FCO_2$ are correlated (Rasera, et al., 2013), "in this study was observed" some specific examples where k produces different fluxes even when $pCO_2$ was similar.

*AR: We modified this sentence as follows: "Although $pCO_2$ and $FCO_2$ are typically correlated (Rasera, et al., 2013), in this study we observed several examples where variability in gas transfer velocities drive variable fluxes even when $pCO_2$ was fairly constant".*

RC (reviewer comment; reviewer 2): 2. Language overall needs improvement. Too many commas used. Too many sentences that are confusing (many are mentioned in specific comments below).

*AR: The language and style of the whole text was revised and improved.*

EC2: Please define ROR at its first use in the introduction (…run-of-the-river (ROR) hydropower systems have smaller reservoirs…).

*AR2: We have updated this section with a brief description of ROR dams and their main characteristics (L59-61).*

EC3: As the two reviewers had commented earlier, specifying your research objective or hypotheses would help you transform the descriptive manuscript into a more focused one. There is still no clear goal statement in the abstract. Please consider specifying the major objectives (and hypotheses) in the following sentence: "We evaluated spatiotemporal variations of surface water $CO_2$ partial pressure (pCO_2), water-atmosphere $CO_2$ fluxes (FCO_2), and gas exchange coefficients (k600) in the XR and IR during the first two years after the impoundment of the Xingu River."

*AR3: Thank you for the constructive comment. We have rewritten this sentence and added hypotheses both in the abstract (L7-9) and introduction (L93-95 and 97-100).*

EC4: Please specify what "altered riverine characteristics" are responsible for the observed change in $CO_2$ emissions from IR (Abstract – "However, the associated intermediate reservoir may overcome these emissions due to "altered riverine characteristics".). You could also make your point clearer in the concluding remark on ROR ("$CO_2$ emissions from ROR reservoirs to the atmosphere are in the range of natural Amazon rivers."), particularly in relation to your hypotheses. Do you mean that ROR does not alter $CO_2$ emissions?

*AR4: We have rewritten this sentence for clarity (L22-23). We have also added some additional concluding remarks to reinforce our hypotheses (L423-425).*

EC5: The technical detail questioned by the first reviewer (as shown below) is important, so please provide in Methods how you handled any dilution effect and checked the potential contamination by the remaining air in the evacuated vial. As the reviewer commented, typical vial evacuation procedures cannot remove the air inside 100%. Transferring a similar volume of air sample to the evacuated vial cannot guarantee that the vial has been removed of air 100%. You could easily test (and correct, if required) any potential left-over gas concentration by measuring CO2 concentration after filling the evacuated vial with the same volume of N2 gas as the typical volume of samples.

RC (reviewer 1): L150. How good was the evacuation? In my experience, it"s very difficult to get a good vaccum, but probably 10% or more atmosphere will remain, which may dilute or contaminate your samples. Was this checked?

*AR: We are confident in our vial evacuation procedures and sample storage methods, which our team has extensive experience with. A vacuum pump was used to create a vacuum, which was confirmed since the volume of gas pulled from the syringe into the vial was similar to the vial volume without the needing to manually depress the syringe's plunger. We have modified the sentence for clarity (L171-172) and in addition evaluated our vial evacuation as requested. Vials were evacuated in two ways, 1) with an electric pump and 2) manually using a syringe and a three-way valve. In both methods we observed minimal contamination. With the vacuum pump method, the vials with $N_2$ added contained ~3% contamination by atmospheric air (410 ppm). With the syringe method there was ~2% contamination (Table 1). We do not feel that corrections to the data are needed with these minimal amounts of contamination.*

Table 1: Summary of $CO_2$ contamination test with pump and syringe evacuations.

|  | Pump | Syringe |
| --- | --- | --- |
| **Avg CO$_2$ conc (ppm)** | 14.04 | 8.2 |
| **Std Dv** | 3.17 | 1.58 |
| **CO$_2$ percent relative to atmosphere (%)** | 3.4 | 2 |
| **Sample n =** | 5 | 5 |

EC6: Spatial and temporal variations in pCO_2 and FCO_2 might be one of your key findings, but their descriptions are inconsistent or inaccurate, as sampled below. Please note that FCO_2 in the low-flow period was distinctively higher in IR. And differences in pCO_2 between XR and IR are not clear enough to say the difference definitely. Please double check and correct all the related descriptions throughout the manuscript.

Abstract – "Spatial heterogeneity was observed for pCO_2 during both low and high water seasons while FCO_2 showed significant spatial heterogeneity only during the high water period."

Results - Significant difference in $FCO_2$ was observed among environments sampled during high water season ($F_{3:28}= 7.94$, $R^2= 0.43$, $p= 0.0089$) while the low water season had relatively homogeneous $FCO_2$ values ($F_{3:17}= 2.67$, $R^2= 0.14$, $p= 0.08$) (Fig.4 and Table 3).

Discussion - The increase in $pCO_2$ during the high water season and can be related with the increased input of terrestrial organic and inorganic carbon into the rivers by surface run-off and subsurface flow of water…

During the low water season most of $pCO_2$ and $FCO_2$ decreased, especially in the river channel environment…

*AR6: The manuscript has been revised to address these inconsistencies (L10-12; L18-19; L249-250; L263-265; L302-304; L322-324).*

EC7: Adding to the previous comment on the key findings on spatiotemporal variations in $pCO_2$ and $FCO_2$, Fig 3 needs to be improved in presentation quality. It is a very big and busy figure and there are also formatting errors (e.g., units not put in parentheses) and lacking information about the labels (e.g., UD, DD?). Please think about how to present all the plots in a more efficient way, by e.g. putting "High water" and "Low water" labels on top of all the plots and sharing "Environment" category labels only once at the bottom of all the plots. If you cannot show all results on a page, splitting them into two sub-figures or using a horizontal page format could be an alternative idea. Plotting over the same range on the vertical axis would make it easier to compare the high- and low-flow results. (e.g., -1000 to 3500 for the plots c and d).

*AR7: All suggestions were added to the image and labels were made more detailed (L627-630).*

EC8: In section 4.2., your explanations about the higher $FCO_2$ from IR during low water than in high water season were focused on the flooded OM as a potential source. What about the counteracting effect of phytoplankton uptake of $CO_2$, which might have increased in the low water season, due to reduced turbidity and enhanced phytoplankton uptake of $CO_2$? Please refer to other studies that compared seasonal differences in reservoir primary production and $CO_2$ emissions; and provide a more balanced discussion of potential contributing factors for seasonal variations in the source and sink of $CO_2$ in reservoirs.

*AC8: Thank you for the suggestion. We have added a new paragraph related to phytoplankton and IR $CO_2$ during the low water season as requested (L375-387).*

EC9: Please provide more details on the several same variables in Table 3. It is difficult to understand how the same variables of $pCO_2$ and $FCO_2$ differ across the rows.

*AC9: Updated as requested.*

Relevant changes list:

Abstract:

Hypotheses added, pg. 2, lines 7-9.

Inconsistency review, pg.2, lines 10-12.

Inconsistency review, pg.2, lines 18-19.

Introduction:

Run-of-the-river reservoirs description, pg. 3, lines 59-61.

Results:

Inconsistency review, pg.8, lines 249-250.

Inconsistency review, pg.8, lines 263-265.

Discussion:

Inconsistency review, pg.9, lines 302-304.

Inconsistency review, pg.9, lines 322-324.

Discussion added, pg. 11, lines 375-387.

Conclusions:

Hypotheses concluding remark added, pg. 12, lines 423-425.

Figures:

Fig 3 description alteration, pg.16, lines 627-630.

Fig 3 alterations, pg.18.

Tables:

Table 3 alterations, pg. 23.

Marked-up manuscript:

[revised manuscript text omitted]

---

## Author Response (AR3)

*Dear editor,*

*We are grateful for the manuscript acceptance and appreciate your handling of our submission and your very useful feedback.*

*Please find our answers in italic following Editor Comments (EC):*

EC: Line 17: please correct "for, XR and IR respectively) in IR" to "for XR and IR, respectively)".
*AR: Updated as suggested (L17).*

EC: Line 95: please use $CO_2$ instead of "carbon dioxide"
*AR: The sentence was updated as requested (L94).*

EC: Lines 98-99 (and throughout the manuscript): "$FCO_2$" and "$pCO_2$" have already been defined in the previous sentences.
*AR: We revised whole text and altered as suggested (L25; L98; L145; L181-182; L185; L228; L252; L330; L375; L414; L422; L426).*

EC: Lines 423-424: The sentence ("ROR reservoirs did not appear to alter $CO_2$ emissions compared to naturally flowing Amazonian clearwater rivers.") is contradictory to your key conclusion, so you need to rewrite the sentence or could add certain conditions under which ROR reservoirs did not alter $CO_2$ emissions substantially.
*AR: Thank you for the suggestion. We have rewritten this sentence (L422-424).*

EC: References with the author name "Brasil": It appears that you have used some reports published by the government of "Brazil". Please use English translation and indicate that the reports are written in Portuguese. If available, names of ministries or government institutes would be better the country name "Brazil".

*AR: Thank you for the constructive comment. We have added the responsible government agencies in the author's name and report's titles were translated. Portuguese indications were also added as requested (L65; L104; L116; L124-125; L126; L133; L137; L494-500).*

EC: Figures: correct "u" in unit "umol" to the true micro.
*AR: updated as suggested.*

Relevant changes list:

Abstract:
Sentence altered, pg. 2, line 17.

Introduction:
Sentence altered, pg. 4, line 94.

Term correction and review, pg. 2, 4, 5, 6, 7, 8, 10, 11, 12, lines *25; 98; 145; 181-182; 185; 228; 252; 330; 375; 414; 422; 426.*

Conclusion:
Concluding remarks altered, pg. 12, lines 422-424.

References:
Brazilian report references correction, pg. 13, lines 494-500.

Figures:
Unit corrections, pg. 18, 19.

Marked manuscript:

[revised manuscript text omitted]